# Uncertainty growth and forecast reliability during extratropical cyclogenesis

Mark J. Rodwell[1] and Heini Wernli[2]

[1]European Centre for Medium-Range Weather Forecasts, Reading, UK
[2]Institute for Atmospheric and Climate Science, ETH Zürich, Switzerland

**Correspondence:** Mark Rodwell (mark.rodwell@ecmwf.int)

**Abstract.**

In global numerical weather prediction, the strongest contribution to ensemble variance growth over the first few days is at synoptic scales. Hence it is particularly important to ensure that this synoptic scale variance is reliable. Here we focus on wintertime synoptic scale growth in the North Atlantic storm track. In the 12 h background forecasts of the Ensemble of Data Assimilations (EDA) from the European Centre for Medium-Range Weather Forecasts (ECMWF), we find that initial variance growth at synoptic scales tends to be organised in particular flow situations, such as during the deepening of cyclones (cyclogenesis). Both baroclinic and diabatic aspects may be involved in the overall growth rate. However, evaluation of reliability through use of an extended error–spread equation indicates that the ECMWF ensemble forecast, which is initialised from the EDA but with additional singular vector perturbations, appears to have too much variance at a lead time of 2 days, and that this over-spread is associated with cyclogenesis situations. Comparison of variance growth rates and reliability with other forecast systems within the TIGGE archive, indicates some sensitivity to the model or its initialisation. For the ECMWF ensemble forecast, sensitivity experiments suggest that a large part of the total day–2 spread in cyclogenesis cases is associated with the growth of EDA uncertainty, but up to 25% can be associated with the additional singular vector perturbations to the initial conditions, and up to 25% with the representation of model uncertainty. The sensitivities of spread to resolution, the explicit representation of convection, and the assimilation of local observations are also considered. The study raises the question whether the EDA now successfully represents initial uncertainty (and the enhanced growth rates associated with cyclogenesis) to the extent that singular vector perturbations could be reduced in magnitude to improve storm track reliability. This would leave a more seamless forecast system, allowing short-range diagnostics to better help improve the model and model-uncertainty representation, which could be beneficial throughout the forecast range.

## 1 Introduction

The chaotic nature of the atmosphere, with its large sensitivity to small perturbations (Sutton, 1954; Lorenz, 1963, 1972) has necessitated the development of probabilistic approaches to forecasting. In a "reliable" probabilistic forecast system, when aggregated over many forecasts, the issued probabilities for a given weather event should match the outcome frequencies (Sanders, 1958). The event could be, for example, that the temperature will be below 0°C at a given location. Forecast reliability

implies that, of all the times that a forecast probability $p$ (e.g. $p = 0.5$) is issued for the given event, the event happens a fraction $p$ of the time (to within sampling uncertainty). Reliability is clearly important for forecast users because it means that they can make unbiased decisions (e.g. Rodwell et al., 2020). In addition to reliability, a probabilistic forecast system needs to be "sharp" — issuing probabilities as close to either 0 or 1 as possible. A forecast system which always issues the (sample) climatological probability for an event will be perfectly reliable but, in general, it will not be sharp. Conversely, a single deterministic forecast can only issue probabilities of 0 or 1 so will be completely sharp but cannot be perfectly reliable in the face of chaotic error growth. The goal of probabilistic forecasting is, thus, to maximise the sharpness of the predictive distributions subject to reliability (Gneiting and Raftery, 2007). A key question of practical importance is how can this goal be achieved, or worked towards? An issue here is that the same forecast probability $p$ can arise from very different courses of events. For example, a 6 day forecast probability for cold weather over Europe might be dependent on the decay of an existing blocking situation or might be dependent on the development of mesoscale convection over North America earlier in the forecast (Rodwell et al., 2013). The same probabilities can thus rely on the abilities of the forecast model to represent very different phenomena, and their associated uncertainties. Rodwell et al. (2018) argued that it does not make sense to group such forecasts together. Instead, they suggest that a focus on short range flow-dependent reliability (and a reduction of initial condition uncertainty) could represent a practical path to more skillful (more reliable and sharp) probabilistic forecasts.

This study focuses on extratropical cyclonic flow-types and their intensification within the North Atlantic storm track. An example based on European Centre for Medium-Range Forecasts (ECMWF) Re-Analysis version 5 (ERA5; Hersbach et al., 2020) is given in Fig. 1. A little earlier than shown in Figure 1, on 26 November 2019, the cyclone had begun to develop over the Midwestern U.S. A day later, it had reached the Great Lakes with a central pressure at mean sea level (PMSL) of about 990 hPa. At 12 UTC on 28 November (Fig. 1a), the cyclone reaches the North American east coast with a similar PMSL intensity (grey contours). There is a cutoff in potential vorticity on the $\theta = 315$ K isentrope ($P315$, thick black contour) aloft, and intense surface rainfall (shading) in the region of ascent identified as a warm conveyor belt (WCB; red hatching, following the approach of Wernli and Davies, 1997; Madonna et al., 2014). One day later (Fig. 1b), the cyclone has strongly deepened to below 974 hPa as it moves slowly out into the Atlantic. Intense precipitation continues in the WCB ascent regions along the cold and bent-back fronts. In the next 24 h, the cyclone remains stationary, and it deepens further to below 970 hPa (Fig. 1c). The WCB and associated band of intense precipitation now extend from about 25 to 55°N, and the $P315$ pattern attains the classical structure of a mature cyclone, with a large-amplitude trough-ridge dipole up- and downstream of the surface cyclone, respectively. The heterogeneity of the precipitation rate along the WCB is reminiscent of the occurrence of embedded convection (Oertel et al., 2020). The example emphasizes the strong deepening (cyclogenesis) of the system off the North American east coast and its combined baroclinic and diabatic character. This case, and another which is not shown here, were chosen for sensitivity studies because they were quite clean, without being strongly affected by other flow perturbations in their environment, and because they involved both baroclinic and WCB aspects. However, the rate of deepening and the growth of uncertainty during the forecast were not considered in the choice. While these case studies are useful in several ways, the evaluation of forecast reliability is based here on cyclogenesis events over a whole season.

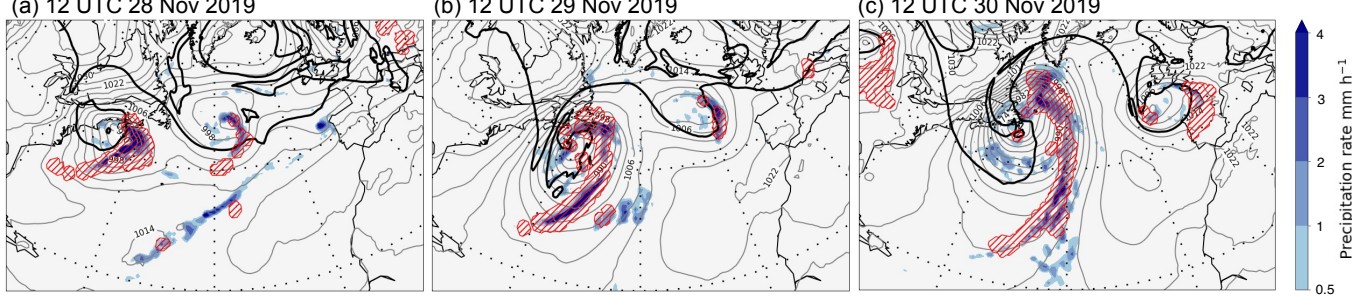

**Figure 1.** Synoptic overview, based on ERA5 reanalyses, of a North Atlantic cyclogenesis event at 12 UTC on (a) 28 November, (b) 29 November, and (c) 30 November 2019. Shown are PMSL (grey contours at intervals of 4 hPa), surface precipitation accumulated over the previous hour (colour shading, in mm h$^{-1}$), the $P315 = 2$ PVU contour indicating the tropopause on 315 K (thick black contour), and the WCB ascent region (red hatching). WCBs are identified as 48 h trajectories that ascend from the lower troposphere by more than 600 hPa in 48 h (e.g., Wernli and Davies, 1997; Madonna et al., 2014), and the ascent region corresponds to the envelope of the horizontal positions of all WCB trajectories, initialized every 6 h, that are located between 800 and 500 hPa at the indicated time (corresponding approximately to their mid-ascent time).

It has been shown that dry low-resolution singular vectors identify baroclinic structures as the flow features leading to the largest day–2 uncertainty growth in the linear regime (Molteni and Palmer, 1993). The first objective for the current study is to investigate whether cyclogenesis events can coordinate strong growth of forecast uncertainty when moist processes are included and the non-linear forecast model is run at much higher resolution. For example, does cyclogenesis act to reduce sharpness, and thus markedly shift forecast probabilities away from the desired values of 0 and 1? To explore this possibility, 12 h exponential growth rates are quantified following Rodwell et al. (2018) and related to dynamical and diabatic sources. The possibility that there is synoptic-scale coordination of uncertainty growth is interesting. Whether this hints at an intrinsic property of the atmosphere, or is dependent on the formulation of the forecast system, is explored by comparing models within "The International Grand Global Ensemble" (TIGGE, Swinbank et al., 2016) archive.

Probabilistic weather forecasts are generally based on ensemble techniques, whereby a set (or ensemble) of initial conditions, representing the uncertainty in the current state, is used to initialise an ensemble of numerical model forecasts (Palmer et al., 1992; Molteni et al., 1996). The variance of the ensemble, which generally grows with lead time, is then a measure of forecast uncertainty. Under various assumptions (Stephenson and Doblas-Reyes, 2000), a consequence of reliability is that the truth should be statistically indistinguishable from any ensemble member (Hamill, 2001; Saetra et al., 2004), and hence the squared difference between the truth $T$ and the forecast ensemble mean $\widehat{\mu}_F$ should match the ensemble variance $\widehat{\sigma}_F^2$, when averaged over a sufficiently large set of forecasts: $\overline{(T - \widehat{\mu}_F)^2} \approx \overline{\widehat{\sigma}_F^2}$ (Leutbecher and Palmer, 2008). This is often known as the "error-spread equation". Note that a circumflex $\widehat{\phantom{x}}$ is used to indicate an estimator throughout this study, and a thick overline $\overline{\phantom{x}}$ indicates a mean over forecast initial times.

The maintenance of reliability during synoptic-scale uncertainty growth is particularly important in weather prediction because these scales become the largest contributor to global ensemble variance over the first few days of the forecast (Tribbia and Baumhefner, 2004, and also in the current study). The second objective for this study is thus to evaluate how well the ensemble system maintains short range (2 day) reliability within the North Atlantic storm track and, with the help of a flow clustering technique, specifically during cyclogenesis events. To do this, an extended version of the error-spread equation, which takes account of bias and uncertainties in our estimation of $T$, is evaluated. Again the TIGGE archive is used to investigate sensitivity to forecast system formulation.

The final objective of this study is to quantify the impacts of individual aspects of the ECMWF ensemble configuration on forecast uncertainty during cyclogenesis. The aspects include the initial uncertainty, singular vector perturbations, model uncertainty, convective parametrisation, increased model resolution, and the assimilation of observational data within the vicinity of the cyclogenesis event.

The study is structured as follows. Section 2 details the models, data, and methodologies used. Section 3 quantifies uncertainty growth rates, with particular application to cyclogenesis (objective 1). Section 4 evaluates forecast reliability, with application to seasonal means in the North Atlantic storm track region, and when composited on cyclogenesis cases (objective 2). Section 5 investigates the sensitivity of forecast uncertainty to aspects of the ECMWF forecast system during cyclogenesis (objective 3). A summary, conclusions and discussion about prospective research are given in Sect. 6. Supplementary material includes animations of uncertainty growth rates.

## 2   Models, data and methods

This study makes use of several models, data sets and methodologies. These are described here for reference.

### 2.1   The ECMWF ensemble assimilation and forecast system

The underlying earth system model for the ECMWF Ensemble of Data Assimilations (EDA; Isaksen et al., 2010; Lang et al., 2019), and Ensemble forecast (ENS; Palmer et al., 1992; Molteni et al., 1996) uses spherical harmonics to compute much of the dynamics, with physical parametrizations computed in grid-point space. Of particular interest here is the parametrization of convection, which was originally based on Tiedtke (1989), but includes revisions to entrainment and coupling with the large scale (Bechtold et al., 2008; Hirons et al., 2013), and improvements in the diurnal cycle of convection through use of a modified convective available potential energy (CAPE) closure (Bechtold et al., 2014).

The 50-member EDA used here has a nominal horizontal grid resolution of $\sim 16$ km. More specifically, in the final EDA iteration, the underlying non-linear model retains a Triangular configuration of spherical harmonics with total wavenumbers $\leqslant 639$ and uses a Cubic octahedral grid so that 4 grid points represent the smallest waves (the resolution is thus summarised as TCo639). It has 137 levels in the vertical (L137) and uses a 12 min timestep. 4D Variational data assimilation (4DVar, Rabier et al., 2000) uses the full non-linear model, together with tangent-linear and adjoint versions, and observation operators to extract the information content from many millions of conventional and satellite observations during each 12 h analysis

cycle. This is done by first screening observations and correcting them using Variational Bias Correction (VarBC, Dee, 2004). For each EDA member, the observations are then randomly perturbed to simulate observation uncertainty (Isaksen et al., 2010). The background (or first-guess) for a given EDA member is the non-linear forecast initialised from the same member's previous analysis. The estimated uncertainty in this forecast is based on the variances and correlations between the ensemble of background forecasts, with a climatological contribution to correlations for improved stability. For each EDA member, 4DVar combines its background forecast and perturbed observation set in a way that is consistent with the estimated uncertainties in the background (Bonavita et al., 2016) and observations (Geer et al., 2018). An additional unperturbed EDA member, with no perturbations to the observations, is also made.

The EDA is used to initiate a 50-member ENS, with resolution TCo639, L91 and a 12 min timestep. Initial conditions are re-centred on a more recent ("Early Delivery") unperturbed high-resolution (HRES) 4DVar analysis. Singular vector (Molteni and Palmer, 1993; Leutbecher and Lang, 2014) perturbations are added to the initial conditions as a pragmatic means of boosting ENS spread over the first 2 days. A model uncertainty parametrization, which partly aims to represent scale interactions with (missing) sub-grid-scale variations, is important for the general growth of ENS spread into the medium-range (e.g., to day 10). Here, this model uncertainty representation is based on "Stochastic Perturbation to Physical Tendencies" (SPPT, Buizza et al., 1999), which represents a perturbation to the total physical tendency, and is applied throughout the forecast range. Note that SPPT is also applied to the non-linear model in the perturbed members of the EDA, but there are no singular vector perturbations in the EDA.

The sensitivity experiments discussed in Sect. 5 are based on cycle 46r1 of the ECMWF Integrated Forecasting System (IFS). This cycle was operational from 11 June 2019 to 20 June 2020, and prior information for the experiments comes from the operational EDA. In the higher-resolution $\sim 4\,\mathrm{km}$ ENS experiment, the model is run in single numerical precision (Váňa et al., 2017; Lang et al., 2021) with resolution TCo2559 L91 and a 4 min timestep.

## 2.2 Other ensemble forecast systems in the TIGGE archive

In comparisons with ensemble forecasts from other centres, all data are retrieved from the TIGGE archive, which currently contains global ensemble forecasts from about a dozen of the world's operational forecasting centres (12 centres were present on 1 September 2021). These forecasts are available a few days behind real-time and are a valuable resource for diagnostic studies. Three other TIGGE models are compared with the ECMWF model. These are from the Japan Meteorological Agency (JMA), the United States' National Centers for Environmental Prediction (NCEP), and the United Kingdom's Met Office (UKMO). Salient details of these four ensemble forecast systems are presented in Table 1 and further documentation is available from the TIGGE website (https://confluence.ecmwf.int/display/TIGGE/Models). Here, comparisons are based on the common run times of 00 and 12 UTC.

## 2.3 The extended error-spread equation

In Sect. 1, the error-spread equation was discussed. This equation is not precise enough for application at the short lead times of interest here because uncertainty in our knowledge of the verifying truth is not negligible in the calculation of forecast

**Table 1.** Details of the four TIGGE models used in this study, valid during the period of investigation. EDA=Ensemble of 4DVar (Isaksen et al., 2010), 4DVar=4D Variational data assimilation (Rabier et al., 2000), EnKF=Ensemble Kalman Filter (Evensen, 1994), ETKF=Ensemble Transform Kalman Filter (Bishop et al., 2001), LETKF=Local Ensemble Transform Kalman Filter (Hunt et al., 2007), SPPT=Stochastic Perturbation to Physical Tendencies (Buizza et al., 1999), SV=Singular Vector (Molteni and Palmer, 1993), RP=Random Parameters (McCabe et al., 2016), SKEB=Stochastic Kinetic Energy Backscatter (Shutts, 2004).

| Centre | ECMWF | JMA | NCEP | UKMO |
|---|---|---|---|---|
| Resolution (nominal) | 16 km | 42 km | 25 km | 21 km |
| Vertical levels | 91 | 100 | 64 | 70 |
| Number of perturbed members | 50 | 26 | 30 | 17 |
| Run times (UTC) | 0,12 | 00,12 | 00,06,12,18 | 00,06,12,18 |
| Initial perturbation strategy | EDA, SV | LETKF, SV | EnKF | ETKF |
| Model uncertainty representation | SPPT | SPPT | SPPT, SKEB | RP, SKEB |

error. It is also important to consider forecast and analysis bias (arguably this can be important at all lead times). To account for these aspects, the observation-space reliability assessment of Rodwell et al. (2016) employed an extended error–spread equation. Here we modify their equation to reflect the fact that uncertainty in the truth is estimated by the variance of the

145 verifying analysis, rather than the estimated variance of the observation error. Let $\widetilde{F}_j$ and $\widetilde{A}_j$ be the (imperfect) forecast and verifying analysis distributions, respectively, for forecast initial time $t_j$, $j = 1, \cdots, n$. The ensembles give us $m$ samplings of these distributions. Let $d_j = \widehat{\mu}_{\widetilde{F}j} - \widehat{\mu}_{\widetilde{A}j}$ be the difference between the ensemble-mean of the forecast $\widehat{\mu}_{\widetilde{F}j}$ and the ensemble-mean of the analysis $\widehat{\mu}_{\widetilde{A}j}$, and write $\widehat{\sigma}^2_{\widetilde{F}}$ and $\widehat{\sigma}^2_{\widetilde{A}}$ for the forecast and verifying analysis ensemble variances, respectively. Then, using a thick overline $^-$ to indicate a mean over the $n$ forecast times, the extended error–spread equation for $n$ forecasts of an

150 ensemble with $m$ members can be written as

$$\frac{n}{n-1}\overline{d^2} = \frac{m+1}{m-1}\overline{\left(\widehat{\sigma}^2_{\widetilde{F}} + \widehat{\sigma}^2_{\widetilde{A}}\right)} + \frac{n}{n-1}\overline{d}^2 + \overline{R} \qquad .$$

(1)

$$\underbrace{\qquad}_{\text{Error}^2} \quad \underbrace{\qquad}_{\text{Spread}^2} \quad \underbrace{\qquad}_{\text{AnUnc}^2} \quad \underbrace{\qquad}_{\text{Bias}^2} \quad \underbrace{\qquad}_{\text{Residual}^{(2)}}$$

where the various terms have been named for future reference (superscript 2 indicates that these are in squared units). In addition to the error and spread terms, there are now terms reflecting analysis uncertainty, squared bias (associated with mean differences between the forecast and analysis), and a residual $\overline{R}$, which closes the budget (this can be negative or positive, and

155 written as the mean of the $R_j$ which close the budget for each value of $j$, for given mean bias estimate $\overline{d}$). The contents of the residual are discussed more fully in Appendix A, following a full and consistent derivation of Eq. (1). The residual is seen to represent the mean deficit in the combined forecast and analysis variance and, potentially, the flow-dependent variance in forecast bias. As lead time increases beyond 12 h, one might expect the bias and residual to become dominated by deficiencies in the forecast. Hence this equation is useful for monitoring progress towards the goal of probabilistic forecasting: the Spread$^2$

relates to ensemble sharpness, and the Residual$^{(2)}$ and Bias$^2$ relate to the ensemble reliability. A flow-dependent application of

Eq. (1), which is possible because it is valid at short lead times, should also diminish the impact of flow-dependent variations in forecast bias. Here, the equation is applied to 2 day forecasts of geopotential height at 250 hPa ($Z250$), and temperatures at 500 hPa ($T500$), started at 00 and 12 UTC, and valid during the December–February 2020/21 season (DJF 2020/21).

For display purposes the terms in Eq. (1) can be put into more understandable units by taking their square-roots, signified by removing the superscript $^2$ label. Note that the square-root of Bias$^2$ is Bias (with its correct sign). Since $\overline{R}$ can be positive or negative, Residual $\equiv \sqrt{|\overline{R}|} \cdot \mathrm{SGN}(\overline{R})$ is plotted to retain the sign. While smaller terms will look more important than they are in the squared budget, the residual still correctly indicates spread deficiencies. Statistical significance is determined (at the 5% level with a t-test) from the un-rooted terms (except for Bias$^2$, which must be determined in its non-squared form).

Note that the observation-space version of the extended error–spread equation (Rodwell et al., 2016) would probably represent a more fundamental evaluation of the forecast system, if assigned observation error variances aimed to reflect their true values. In reality, some observation error variances can be inflated to account for representativeness error (Rennie et al., 2021), observation error correlations, or when associated with non-linear observation operators. This can lead to a large residual term for a given observation type, even if the resulting analyses and forecasts are reliable in model-space. Hence the current model-space application represents a complementary approach, which can be more readily applied to a range of models and lead times, and which can be used to assess ensemble initialisation aspects.

## 2.4 K-means clustering

This study aims to evaluate ensemble reliability during cyclogenesis events, and so a means of focusing on such events is required. The method used is *K*-means clustering (Hartigan and Wong, 1979), which seeks to minimise the sum of squared deviations from the relevant cluster-mean. To identify cyclogenesis events in the analyses, the clustering is applied in $30°$lon × $20°$lat boxes at 00 and 12 UTC during DJF 2020/21 jointly to the fields of $Z250$ and zonal and meridional wind at 850 hPa ($u850$, $v850$) at step 0 from the control forecast, and ensemble mean 12 h accumulated total precipitation from the previous forecast (so the end of the accumulation period corresponds to the time of the circulation fields). It is thought that these fields should be able to capture upper-tropospheric Rossby waves, baroclinic structures and associated diabatic processes. It is the ability to cluster on structures which motivated the choice of the *K*-means approach. The choice was made to find three clusters, since it was thought that these would provide sufficient degrees of freedom to differentiate the local synoptic-scale structures, while giving large-enough clusters to obtain statistical significance. Each field is on a regular F32 ($\sim 300$ km) Gaussian grid, standardised (about its area- and temporal-mean) and root-cos-latitude-weighted prior to application of the clustering algorithm (total precipitation is standardised by dividing by the square root of its area- and temporal-mean-squared value). This is to give approximately equal weight to each field and sub-region. Three random date/times (out of the 180 date/times available within the season) are used to initialise the clusters. Since there is no guarantee that the algorithm will identify the optimal solution, it was initialised 100 times. No reduction in the sum of squared deviations from the cluster means was found after the first $\sim 5$ such initialisations — indicating that the final solution is optimal.

## 2.5 Potential vorticity

A useful quantity for this study is isentropic potential vorticity (IPV, Hoskins et al., 1985), $P = -g(f + \zeta_\theta)\frac{\partial\theta}{\partial p}$. Here, $g$ is the gravitational acceleration, $f$ is the Coriolis parameter, $p$ is pressure, $\theta$ is potential temperature, and $\zeta_\theta = \boldsymbol{k} \cdot \boldsymbol{\nabla}_\theta \times \boldsymbol{v}$ is the isentropic vorticity, where $\boldsymbol{k}$ is the local unit vertical vector, $\boldsymbol{\nabla}_\theta$ is the horizontal gradient operator on an isentropic surface, and $\boldsymbol{v}$ is the horizontal wind vector. IPV is usually measured in PV units (PVU) with $1\,\mathrm{PVU} = 10^{-6}\,\mathrm{m}^2\,\mathrm{s}^{-1}\mathrm{K}\,\mathrm{kg}^{-1}$. A key advantage of using IPV here is that its tendencies due to dynamic and diabatic effects can be readily separated:

$$\frac{\partial P}{\partial t} + \boldsymbol{v} \cdot \boldsymbol{\nabla}_\theta P = \mathcal{D} \qquad , \tag{2}$$

where $\mathcal{D}$ represents the effects of non-conservative (diabatic and frictional) processes (Holton, 2004, Eq. 4.36). IPV is thus conserved following the horizontal flow on an isentrope in the absence of such processes. This study will use $P315$, the IPV on the 315 K isentrope, which typically intersects the dynamical tropopause ($P$=2 PVU) during winter in the midlatitudes.

## 2.6 The Lagrangian growth rate of forecast uncertainty

It is useful to quantify the rate of growth of uncertainty, to relate this to local and remote sources, and investigate its flow-dependence. The local exponential growth rate of ensemble spread can be estimated as $\widehat{\sigma}_X^{-1}\,\partial\widehat{\sigma}_X/\partial t$ where $\widehat{\sigma}_X$ is the ensemble standard deviation of some atmospheric parameter field $X$ and $t$ is the forecast lead time. Rodwell et al. (2018) observed that, for $X = P315$ in the North Atlantic storm track region, a large component of the local growth rate over the first 12 h appeared to be associated with the advection of uncertainty. This led them to calculate a material "Lagrangian growth rate" following the ensemble-mean wind:

$$\mathrm{LGR}_P \equiv \frac{1}{\widehat{\sigma}_P}\left\{\frac{\partial\widehat{\sigma}_P}{\partial t} + \overline{\boldsymbol{v}} \cdot \boldsymbol{\nabla}_\theta\widehat{\sigma}_P\right\} = \frac{1}{\widehat{\sigma}_P^2}\overline{P'\mathcal{D}'} - \frac{1}{\widehat{\sigma}_P^2}\overline{P'\boldsymbol{v}' \cdot \boldsymbol{\nabla}_\theta P} \quad . \tag{3}$$

The left hand side of Eq. (3) follows Rodwell et al. (2018), the right hand side is deduced in this study, using Eq. (2) and taking inspiration from Baumgart and Riemer (2019). A derivation is given in Appendix B, although the equation is quite self-explanatory. A thin overline $^-$ represents a mean over the ensemble members, and a prime denotes a deviation from the ensemble-mean. The variance estimator for $P$ is then given by $\widehat{\sigma}_P^2 = \overline{P'^2}$. The use of the ensemble-mean horizontal wind $\overline{\boldsymbol{v}}$ in the definition of $\mathrm{LGR}_P$ makes most intuitive sense when very short lead times are considered, so that all ensemble members are representing essentially the same synoptic flow situation. The two covariance terms on the right hand side provide a useful glimpse at the processes driving the material growth rate. The first is proportional to the covariance of PV deviations $P'$ with the deviations in local diabatic and frictional PV sources $\mathcal{D}'$. The second is proportional to the covariance of PV deviations with the advection of PV by the deviation winds $\boldsymbol{v}'$.

The normalisation in Eq. (3) by $\widehat{\sigma}_P$, to give the exponential form of the growth rate, allows comparison of ensembles with differing magnitudes of initial uncertainty. Note that the corresponding exponential growth rate of the ensemble variance is simply twice that of the standard deviation. Equation (3) will be discussed in the context of cyclogenesis in Sect. 3.1.

Baumgart and Riemer (2019) investigated the remote and dynamical sources of variance growth over a 10 day forecast using a slightly different equation — their Eq. (8). To deal with the large redistribution of variance, they choose to disregard the flux divergence term. This leaves an additional source of variance associated with wind divergence. In view of the potential for local cancellations, the choice of strategy may depend on the application. Here, the divergence of ensemble mean winds $\boldsymbol{\nabla}_\theta \cdot \overline{\boldsymbol{v}}$ at a 12 h lead time is likely to be well constrained by the observations, and of less concern for our evaluation goal.

To concentrate on growth rates at synoptic scales (but due to interactions at all scales), the plotted Lagrangian growth rates are smoothed with a synoptic spatio-temporal filter. Details of this filter and other technical information are given below.

LGR$_P$ is constructed using the 12 h non-linear background forecasts from the EDA (so no lead times are greater than 12 h), started at 06 and 18 UTC. Calculations use centred-means and differences between consecutive hourly lead times. For the upper tropospheric fields, $P315$ and winds at $\theta = 315$ K ($\boldsymbol{v}315$) are first interpolated to an N32 reduced Gaussian grid (with 32 latitudes between the pole and equator). The spatial derivatives within the advection term in LGR$_P$ are calculated using spectral transforms to and from a T42 spherical harmonic representation. Note that N32 is sufficient to avoid aliasing of higher harmonics of the quadratic advection term into the T42 representation. The 12 fields of $P315$ and LGR$_P$ are then concatenated over EDA cycles and smoothed with a synoptic spatio-temporal filter. This multiplies spectral coefficients with total wavenumber $n > n_s = 21$ by $\{n_s(n_s + 1)\}/\{n(n+1)\}$ so that scales larger than $\sim 700$ km are retained. The filter also includes a 24 h running-mean. The nominal validity time is at the centre of the running-mean window - placing the final fields back on the full hours. The resulting timeseries of fields can be used to produce animations of $P315$ which "shadow" (remain within the background uncertainty of) the true synoptic evolution of the flow, with LGR$_P$ highlighting the initial ($\sim 12$ h) rate of divergence of the ensemble about the synoptic state. For the lower-tropospheric fields shown in the same plots, winds and specific humidities at 850 hPa ($\boldsymbol{v}850$, $q850$) and surface pressure $p_*$, all from the background forecasts of the control EDA member, are first interpolated to an O32 octahedral reduced Gaussian grid (with 32 latitudes between the pole and equator). Values are set to "missing" where the 850 hPa surface is below the land surface (where $p_* < 850$ hPa) and the moisture flux is calculated as $q850|\boldsymbol{v}850|$. The ensemble mean total precipitation rate is used to indicate where precipitation is likely to occur. This is obtained on a higher resolution O80 octahedral grid so that grid points give a good symbolic representation (stippling) of rainfall when plotted. After similar concatenation of EDA cycles, the lower-tropospheric fields are smoothed with a 24 h running mean.

Since the required fields are not available in TIGGE, when comparing with other models in Section 3.2, the pragmatic decision is made to calculate the Lagrangian growth rate LGR$_Z = \widehat{\sigma}_z^{-1}(\partial \widehat{\sigma}_z/\partial t + \overline{\boldsymbol{v}} \cdot \boldsymbol{\nabla}_p \widehat{\sigma}_z)$ for $Z = Z250$, where $\boldsymbol{\nabla}_p$ is the horizontal gradient operator on the pressure surface. LGR$_Z$ is constructed using the first 12 h of each model's ensemble forecasts started at 00 and 12 UTC. Calculations use centred-means and differences between consecutive 6-hourly lead times (0 h,6 h,12 h). All other details are the same as for LGR$_P$, except that the plotted $Z250$ field is not spatially smoothed (it is considered already a synoptic scale field). Note that humidity fluxes could not be calculated for the UKMO model as specific humidity data was not available from TIGGE. The resulting timeseries of fields can again be used to produce animations (hourly frames being derived using linear interpolation between the 6 hourly fields)

## 3 Uncertainty growth rates during cyclogenesis

This study starts with an investigation of the uncertainty growth during cyclogenesis. It makes use of the Lagrangian growth rate discussed in Sect. 2.6. First, initial growth rates within the background forecasts of the EDA are discussed. After this, a comparison is made with other models in the TIGGE archive.

### 3.1 Uncertainty growth in the EDA

Figure 2 is a synoptically filtered analysis of the event shown in Fig. 1, valid at the mid-point of the cyclogenesis (Fig. 1b). The baroclinic westward tilt with height is evident from the positioning of the upper tropospheric $P315 = 2\,\text{PVU}$ contour (red) and the lower tropospheric $v850$ wind vectors. Ahead of the trough, there are strong moisture fluxes (where vectors are coloured blue) and strong precipitation (black dots). The $P315 = 2\,\text{PVU}$ contour also indicates upper tropospheric ridge development downstream of the trough.

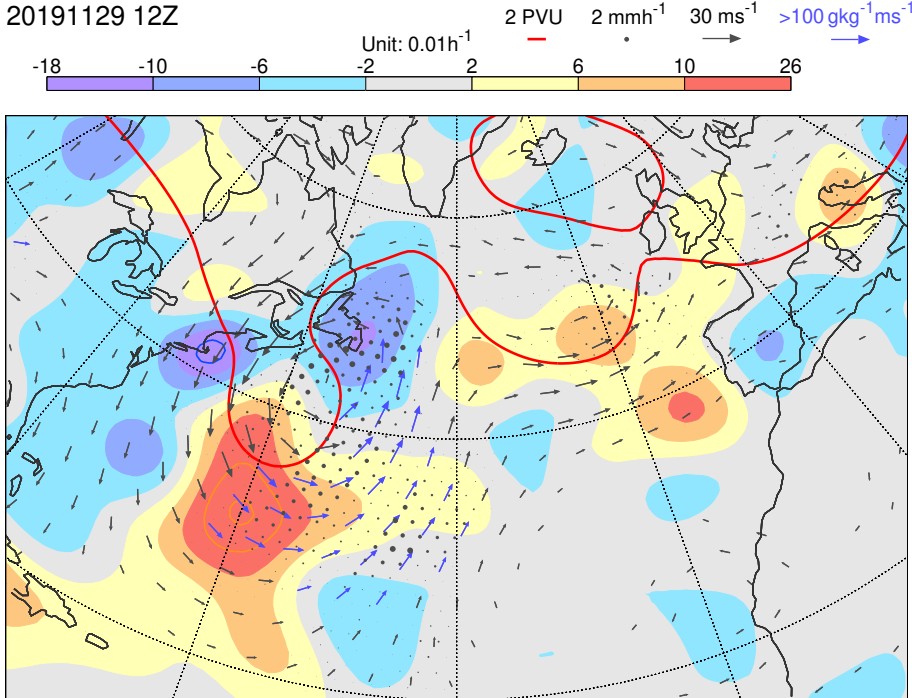

**Figure 2.** Growth rate $\text{LGR}_P$ for $P315$ in EDA background forecasts (shaded), centred at 12 UTC on 29 November 2019. Note that orange and blue contours extend the shading scheme, with the same interval. In these cases, the most extreme values are indicated at the ends of the colour bar. Also shown, for the unperturbed EDA member, are $P315 = 2\,\text{PVU}$ (red contour) and $v850$ (vectors, which are plotted blue if the moisture flux at 850 hPa exceeds $100\ \text{g}\,\text{kg}^{-1}\text{m}\,\text{s}^{-1}$). The ensemble mean precipitation is indicated with black dots (with radius proportional to the precipitation rate up to the maximum size at $2\,\text{mm}\,\text{h}^{-1}$).

The Lagrangian growth rate LGR$_P$ (shaded) highlights uncertainty growth in excess of $0.18\,\mathrm{h^{-1}}$ at the southern extent of the upper-level trough. It is tempting to speculate that this strong growth rate corresponds to uncertainties in baroclinic development, as depicted in the image of Hoskins et al. (1985) — their Fig. 21b. That figure highlights the strengthening of upper tropospheric PV anomalies due to advection by the equatorward winds induced from lower tropospheric PV anomalies. The variance growth would thus be associated with the covariance of ensemble deviations in this advection with the upper tropospheric PV deviations themselves $-\overline{P'\boldsymbol{v}' \cdot \boldsymbol{\nabla}_\theta P}$. This process would thus be represented in the second LGR$_P$ source term on the right hand side of Eq. (3). Interactions within and between scales, represented in this non-linear source term, can produce initial uncertainty growth at synoptic scales because the EDA contains considerable variance power at spatial scales between about 100 and 2000 km (illustrative power spectra of $T500$ are presented in Fig. C1 in Appendix C). This may not be the case in predictability studies, where initial uncertainty is restricted to grid-point noise (Judt, 2018). See also Durran and Gingrich (2014); Selz et al. (2022) for relevant discussions. Note that baroclinic development was not seen as the dominant process driving initial variance growth by Baumgart and Riemer (2019). They suggested that the strongest contribution to this source term is associated with local dynamical interactions between upper tropospheric PV deviations and the winds that they induce.

Also evident in Fig. 2 are weaker negative LGR$_P$ values (where the ensemble is tending to converge in this Lagrangian sense) particularly in the ridge-building region above the northern part of the WCB (see Fig. 1b). It is possible that this could be associated with uncertainties in the erosion of the eastern edge of the upper tropospheric trough, as depicted in the image of Ahmadi-Givi et al. (2004) — their their Fig. 14b. That figure highlights the effect of mid-tropospheric latent heating on the leading edge of the upper tropospheric PV anomaly. The variance reduction would thus be associated with the covariance of the strength of this erosion of PV with the upper tropospheric PV deviations themselves $\overline{P'\mathcal{D}'}$. This process would be represented in the first LGR$_P$ source term on the right hand side of Eq. (3). Note that such an effect might be quite sensitive to the magnitude of diabatic perturbations introduced by the representation of model uncertainty.

The supplementary material includes animations of similar plots to Fig. 2. They show fields of $P315$, $\boldsymbol{v}850$ and precipitation which effectively shadow the true synoptic evolution of the flow, with LGR$_P$ highlighting the initial ($\sim 12\,\mathrm{h}$) rate of divergence of the EDA background ensemble about the synoptic state. Synoptic features associated with large uncertainty growth rates are evident. The results highlighted in Fig. 2 are typical of many cases seen within the animations, and often include strongly precipitating WCBs (with embedded convection, Grams et al., 2018; Oertel et al., 2020). Other common situations for strong growth rates (not investigated here, but consistent with previous studies) are during the extratropical transitions of tropical cyclones (Riemer and Jones, 2014), and within high CAPE situations over North America where mesoscale convection is likely to develop (Rodwell et al., 2013; Sun and Zhang, 2016; Rodwell et al., 2018). All these situations can lead over Europe to extreme precipitation (Grams and Blumer, 2015), blocking events (Rodwell et al., 2013) and, in deterministic forecasts (which can be thought of a single sampling of an ensemble distribution), "busts" or "dropouts" (Lillo and Parsons, 2017).

## 3.2 Uncertainty growth in TIGGE forecasts

The question arises as to how well the ECMWF growth rates discussed in Sect. 3.1 agree with the growth rates derived from the ensembles of other operational forecast centres? This question is explored using the first 12 h of ensemble forecasts within

the TIGGE archive, for the models summarised in Sect. 2.2. As discussed in Sect. 2.6, growth rates are based on $Z250$ because PV is not available in the TIGGE archive. Figure 3 shows filtered $\mathrm{LGR}_Z$ (shaded) for the TIGGE models centred on the same time as that shown in Fig. 2. Other fields shown are the same as in Fig. 2 except that $Z250$ is contoured in green. For ECMWF in the region of cyclogenesis, filtered $\mathrm{LGR}_Z$ from the ENS (Fig. 3a) agrees quantitatively quite well with filtered $\mathrm{LGR}_P$ from the EDA (Fig. 2), despite being growth rates of different fields, and despite the use of additional singular vector perturbations in the ENS. One difference of note for later might be that the maximum ENS growth rate is placed a little more towards the western side of the upper level trough than is the case for the EDA growth rate (cf Fig. 2, Fig. 3a).

Comparison amongst the centres in Fig. 3 indicates strong agreement in the analysis of the synoptic situation, as displayed in the fields of $Z250$, $v850$, moisture fluxes and, to some extent, precipitation. Although there are commonalities, such as in the observational information available to each centre's data assimilation, this agreement suggests that the analyses shadow well the true synoptic evolution. Despite this agreement, the filtered $\mathrm{LGR}_Z$ differs widely amongst the models in this example. Assuming this result caries over to $\mathrm{LGR}_P$, it suggests that there are differences in the source terms on the right of Eq. (3). Looking over many examples (within the TIGGE animation for the DJF 2020/21 season in the supplementary material), the agreement can be better (see example in Fig. 4). Nevertheless, differences between the models' growth rates can be striking, with the ECMWF model tending to display the strongest values. The question arises as to whether the ECMWF growth rates are too strong in the vicinity of cyclogenesis? Is the ECMWF ensemble spread being over-inflated in these situations, with the consequent impact on forecast reliability and sharpness?

The differences in the models' filtered $\mathrm{LGR}_Z$ reflect differences in initialisation procedures, differences in the representation of model uncertainty, and differences in the deterministic models themselves (summarised in Table 1). While it is difficult to evaluate these growth rates per se, it is possible to assess how well each ensemble system maintains short-range statistical reliability within the North Atlantic storm track. This is done in Sect. 4.1.

## 4 Forecast reliability

The concept of forecast reliability, and the need to improve this in a flow-dependent sense was discussed in Sect. 1. As discussed in Sect. 2.3, the precision and short forecast ranges required for flow-dependent evaluation necessitate the use of an extended error-spread equation, Eq. (1), which accounts for bias and uncertainties in the verifying analysis. The residual in this equation estimates the sum of the deficits in forecast and analysis variance, plus a potentially non-negligible term representing flow-dependent variations in forecast bias. Hence a negative residual indicates a surplus in variance, while a positive residual *can* indicate a deficit in variance provided the flow-dependent variations in forecast bias are negligible or accounted for. In Sect.4.1, the equation is applied to day–2 TIGGE forecasts verifying within the season DJF 2020/21. In Sect. 4.2 and Sect. 4.3, flow-dependence is considered by compositing on cyclogenesis cases.

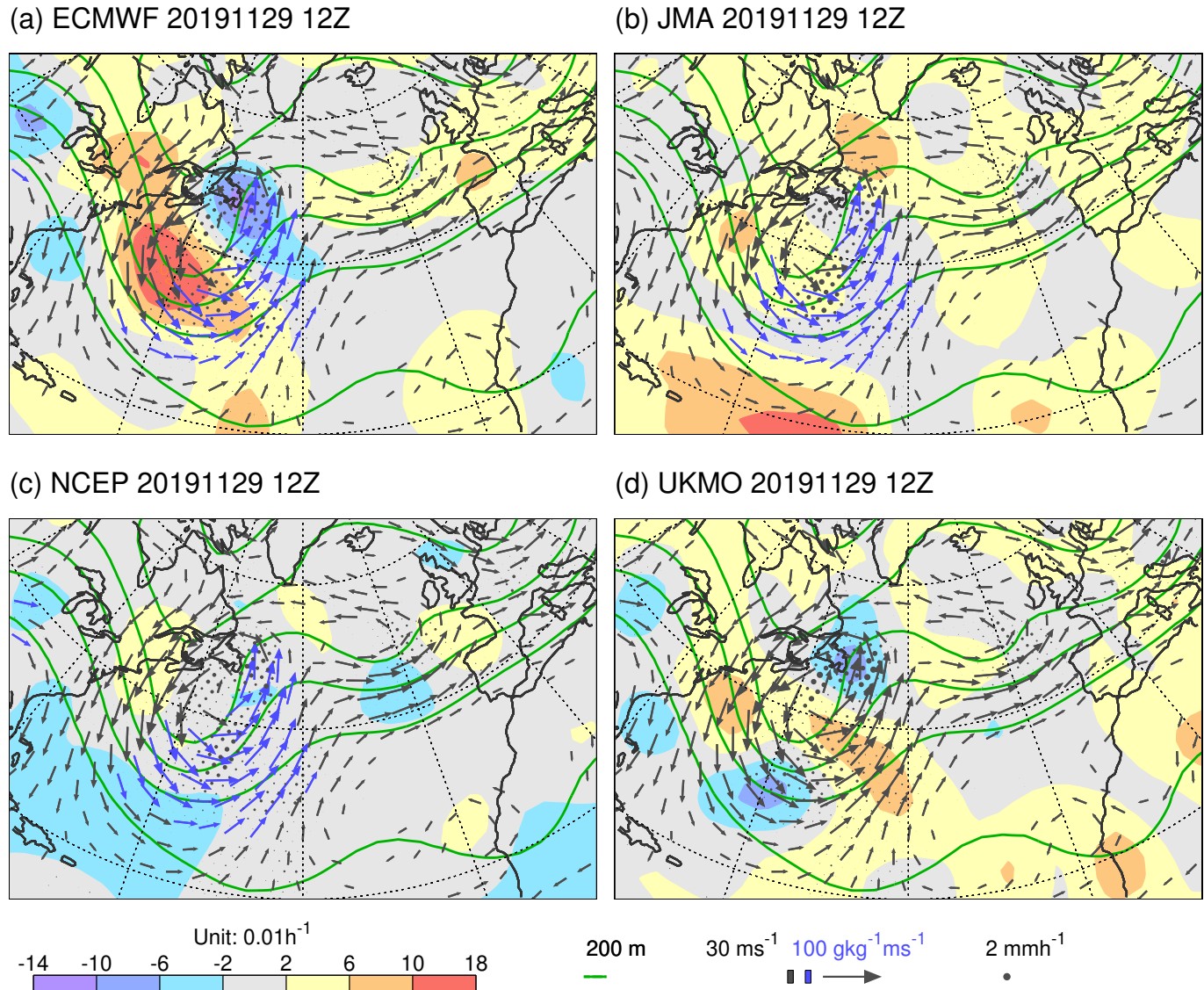

**Figure 3.** Growth rate LGR$_Z$ for $Z250$ for TIGGE models (shaded), centred at 12 UTC on 29 November 2019. Note that orange contours extend the shading scheme with the same interval, where required. In these cases, the most extreme values are indicated at the ends of the colour bar. Also shown, for the unperturbed ensemble members, are $Z250$ (green contours) and $v850$ (vectors, which are plotted blue if the moisture flux at 850 hPa exceeds $100 \, \mathrm{g \, kg^{-1} m \, s^{-1}}$). Ensemble mean precipitation is indicated with (black dots, with radius proportional to the precipitation rate up to the maximum size at $2 \, \mathrm{mm \, h^{-1}}$). The models shown are from (a) ECMWF, (b) JMA, (c) NCEP, and (d) UKMO as discussed in Sect. 2.2.

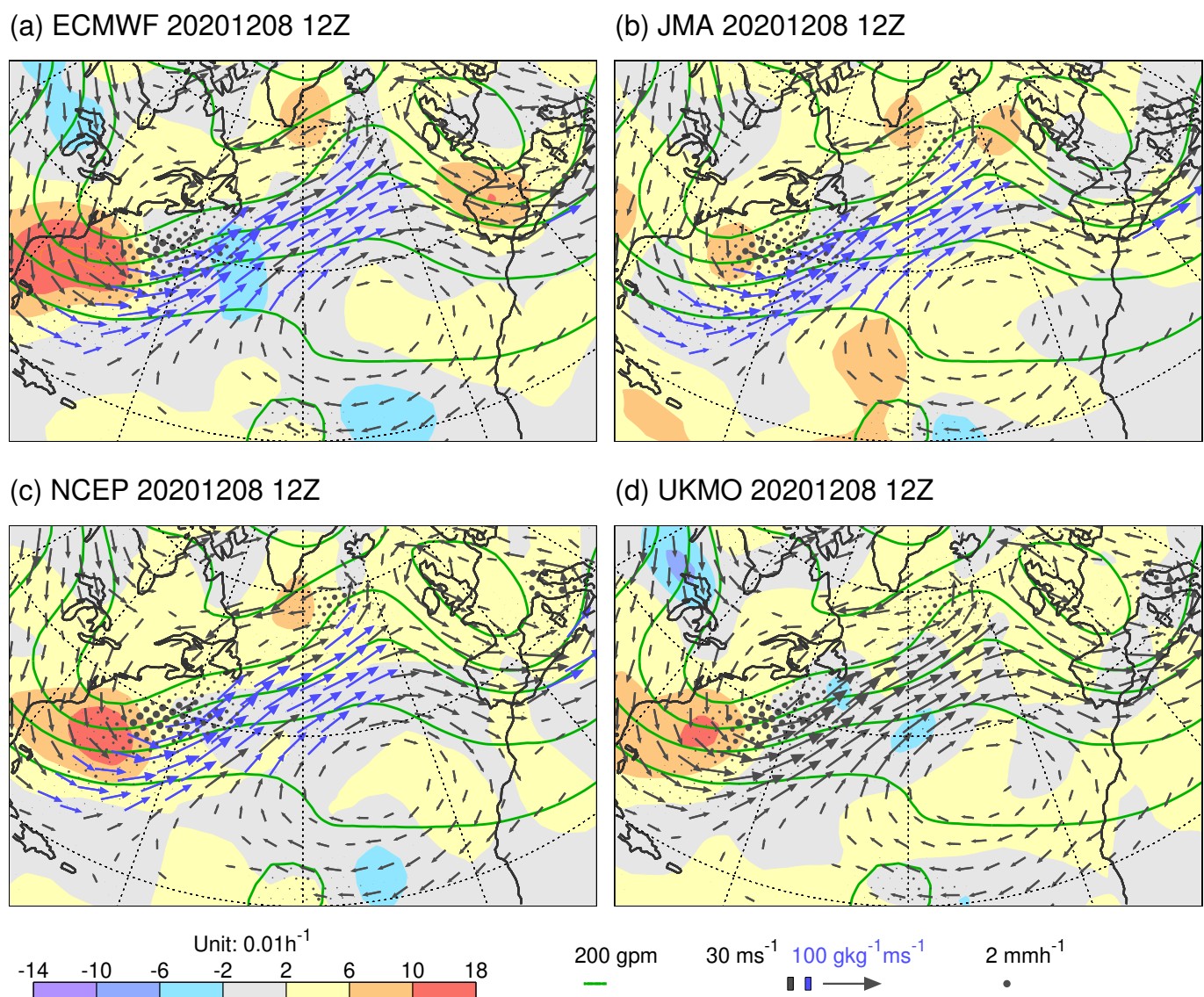

(a) ECMWF 20201208 12Z

(b) JMA 20201208 12Z

(c) NCEP 20201208 12Z

(d) UKMO 20201208 12Z

Unit: 0.01h$^{-1}$

-14   -10   -6   -2   2   6   10   18

gpm    30 ms$^{-1}$  100 gkg$^{-1}$ms$^{-1}$    2 mmh$^{-1}$

**Figure 4.** As Fig. 3, but centred at 12 UTC on 8 December 2020. This situation also corresponds to an event of cyclone intensification off the North American east coast associated with a WCB.

### 4.1 Seasonal mean forecast reliability in the TIGGE ensembles

Figure 5 shows the (square-rooted) terms of the extended error–spread Eq. (1) for $Z250$ based on all day–2 ensemble forecasts verifying in DJF 2020/21 from the ECMWF (top row), JMA (second row), NCEP (third row) and the UK Met Office (bottom row) ensembles. Focusing first on ECMWF (top), the North Atlantic winter storm track is evident as a region of enhanced

ensemble spread (Fig. 5b). Even without the AnUnc and Bias contributions, the Spread is larger than required to balance the

Error (Fig. 5a) — signifying over-spread in the storm track at day 2. Analysis uncertainty is also enhanced in the storm track region (Fig. 5c) and a statistically significant Bias is seen over the east coast of North America. Confirmation of the over-spread is seen with the large negative Residual (Fig. 5e). Note the different colour-bar convention for the Bias and Residual, which can take positive and negative values. Over the North American east coast, the squared budget Eq. (1) is roughly $15^2 = 18^2 + 3^2 + 6^2 - 12^2 \, \text{m}^2$. Because AnUnc and Bias are not negligible here, the "reliable Spread" (i.e., the Spread required for a zero Residual) in this region would actually be $\sim 13.4^2 \, \text{m}^2$, somewhat less than the $15^2 \, \text{m}^2$ which might be inferred from the standard error–spread relationship. Accounting for the variance of forecast bias would reduce further the reliable Spread a little (see Sect. 4.3). In contrast, there is a positive Residual over the subtropical North Atlantic (Fig. 5e). Whether this indicates insufficient Spread to account for the elevated Errors in this region (Fig. 5a) or is associated with variance in forecast bias is also discussed in Sect. 4.3. Note that Rodwell et al. (2018) indicated better reliability for the ECMWF ensemble, when averaged over a year and over the Northern Hemisphere. As part of the current study, but not shown here, this reflects compensating deficiencies elsewhere, a recent deterioration in ensemble reliability in the storm track, and the importance of accounting for bias and analysis uncertainty. These differences underscore the need for increased granularity (such as associated with flow-dependence) in the evaluation of reliability.

For the JMA ensemble during this season, day–2 Error is larger than for ECMWF (cf. Fig. 5a,f), and the Spread is increased by a larger amount (cf. Fig. 5b,g). Note that mean values and root-mean-square (RMS) values, integrated over the area shown, are indicated above each panel in Fig. 5 — it is the RMS values which are most appropriate for comparison. AnUnc and Bias are also larger for JMA (cf. Fig. 5c,h and d,i). Consequently, the residual is more strongly negative (cf. Fig. 5e,j) — indicating more severe over-spread in this ensemble at this lead time. For the NCEP ensemble relative to ECMWF, a smaller increase in Spread (cf Fig. 5b,l) than in Error (cf. Fig. 5a,k) leads to better variance reliability (cf. Fig. 5e,o), despite having larger AnUnc and Bias (cf Fig. 5c,m and d,n). For the UKMO ensemble relative to ECMWF, Error is larger (cf Fig. 5a,p) and Spread is reduced (cf. Fig. 5b,q) — leading to the best variance reliability of the four models (Fig. 5t); again despite having larger AnUnc and Bias (cf Fig. 5c,r and d,s) than for ECMWF. Note that conclusions drawn in this section appear to generalise to other parameters (such as geopotential heights and temperatures at $500 \, \text{hPa}$), other seasons, other storm tracks, and continue until the most recent check for the March–May season 2022 (not shown). With reference to Table 1, an interesting commonality of the two most over-spread systems (ECMWF and JMA) is the use of singular vector perturbations in their initial conditions. Puzzlingly, however, JMA appears to show weaker initial growth rates than ECMWF (see, e.g., Fig. 3, Fig. 4). Differences could be associated with the use here of an exponential growth rate, and the larger initial uncertainty in the JMA forecast. (Variance in initial conditions $\approx$ Variance in verifying analysis, so panel titles for Fig. 5c,f show that area-integrated initial variance is $1 - (3.69/2.81)^2 = 72\%$ larger for JMA than for ECMWF).

To answer the question in Sect. 3.2, whether the ECMWF growth rates are too strong in the vicinity of cyclogenesis, it needs to be determined whether the negative Residual in Fig. 5e is associated with a general level of over-spread or whether it can be linked to cyclogenesis events per se. In Sect. 4.2, cyclogenesis events are objectively identified and then, in Sect. 4.3, the reliability assessment is repeated for these events.

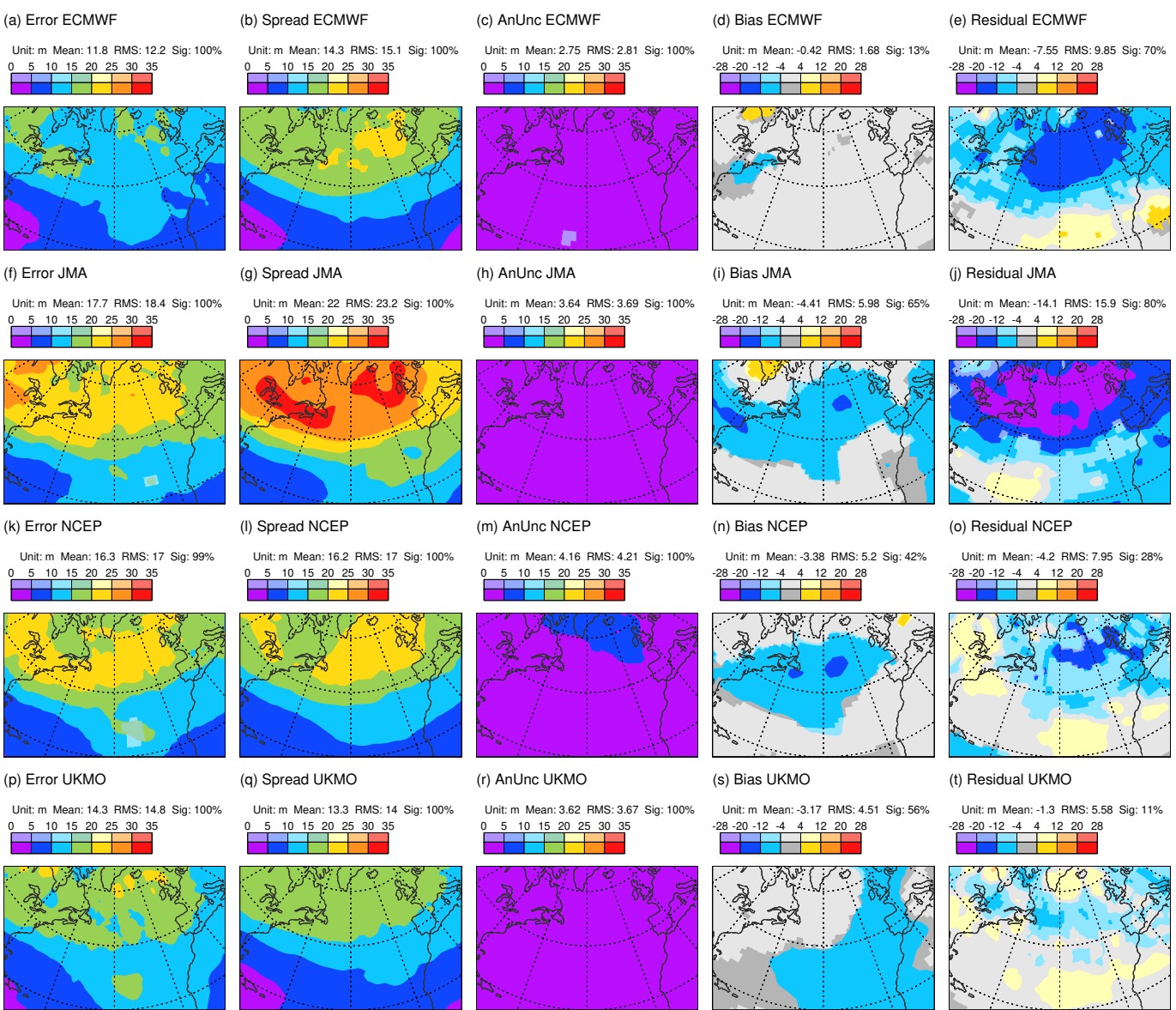

**Figure 5.** Square-roots of terms in the extended error–spread budget Eq. (1) for $Z250$ based on all day–2 forecasts verifying in DJF 2020/21. Data comes from the TIGGE archive for ECMWF (top row), JMA (second row), NCEP (third row) and UKMO (bottom row) ensembles. Note that the Bias and Residual, which can take positive and negative values, are shaded with a different interval to the other terms. Statistically significant values are shown with more saturated colours. Area means (for the area displayed) are indicated at the top of each panel (e.g. "Sig: 85%" means that 85% of the area shown is significant at the 5% significance level, using an auto-regressive AR(1) model to take account of serial correlation).

## 4.2 Compositing cases of cyclogenesis

Here, the *K*-means clustering methodology (Sect. 2.4) is used to objectively identify cyclone events within the DJF 2020/21 season, based on forecast step 0 fields of $Z250$, zonal and meridional components of $v850$ and ensemble-mean precipitation accumulated over 12 h periods ending at the times of the circulation fields. The first clustering region is located at the head of the North Atlantic storm track [80°–50°W, 30°–50°N] and contains 11×7 data points on the F32 grid. The rationale for choosing this region is that it corresponds to the North Atlantic hot spot for cyclone intensification (e.g. Wernli and Schwierz, 2006) and WCB activity (Madonna et al., 2014), and its size corresponds to a half-wavelength of a typical baroclinic wave.

Figure 6 (top row) shows the same fields as in Fig. 3 but averaged over the three sets of date/times obtained from the *K*-means clustering in this (indicated) region. Cluster 1 (Fig. 6a, 32 date/times) appears to capture a partially-evolved cyclogenesis flow-type off the east coast of North America, with a baroclinic westward tilt with height, intense horizontal moisture flux and precipitation ahead. Despite being a somewhat diffuse average of events, the mean drop in analysed PMSL over the previous 2 days attains 14 hPa (not shown), which emphasises the cyclogenesis interpretation. A general tendency for strong uncertainty growth rate at the southern extent of the upper-level trough is also evident. (Note that the growth rate is not used within the clustering algorithm since this could potentially bias the reliability assessment). Cluster 2 (Fig. 6b, 75 date/times) shows a broader trough, weaker moisture flux, and possible cyclogenesis, displaced further downstream. Cluster 3 (Fig. 6c, 73 date/times) shows a diffuse ridge with a trough even further downstream, and a surface anticyclone in the subtropical western North Atlantic.

To identify cyclone events further downstream in the storm track, a second clustering region is displaced north-eastward to [65°–35°W, 35°–55°N]. This region also contains 11×7 data points. Clustering results for this region are shown in Fig. 6 (bottom row). Cluster 1 (Fig. 6d, 62 date/times) highlights further cyclogenesis with a closed cluster-mean circulation at 850 hPa over Newfoundland and with strong growth rates, and (not shown) a mean drop in PMSL of 9 hPa. Note that, as might be expected, 44 of these 62 date/times (71%) were in cluster 2 for region 1 (Fig. 6b).

Further east, towards the end of the storm track, processes get more variable. There can be propagation of cyclones from upstream at different latitudes, the formation of secondary cyclones along fronts, and the formation of cutoff lows, for example. This makes it more difficult to meaningfully cluster this region into cyclogenesis and non-cyclogenesis cases.

By combining the two cyclogenesis clusters over the western North Atlantic (Fig. 6a and Fig. 6d), a total of 91 date/times were identified as cyclogenesis flow-types (32+62 minus 3 duplicates). For each of these date/times (and their 180-91=89 counterpart date/times), visual inspection of plots similar to those in Fig. 1 suggests that the objective clustering has been successful in partitioning the date/times into cyclogenesis and non-cyclogenesis flow-types. This then allows the evaluation of the extended error–spread budget for a large set of cyclogenesis events and of the counterpart set.

## 4.3 Forecast reliability during cyclogenesis in the ECMWF ensemble

Because Fig. 6a and Fig. 6d show partially evolved cyclogenesis flow-types, it is necessary to wind-back the date/times a little to evaluate the day–2 extended error–spread budget during cyclogenesis. Winding back by one and two days gives very similar

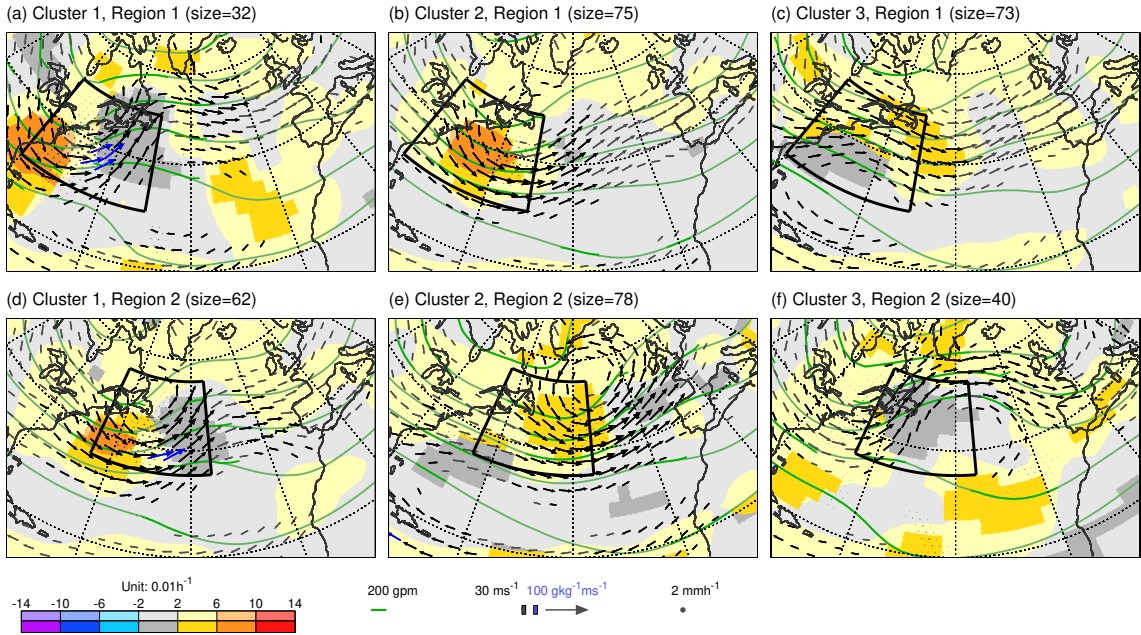

**Figure 6.** Means over the date/times for the three clusters obtained from *K*-means clustering in the first (top) and second (bottom) region on the fields of $Z250$ (contours), $u850$ and $v850$ (vectors) from ECMWF control (unperturbed) forecasts at step 0, and ensemble mean 12 h accumulated precipitation (dots) valid (or end of accumulation) at 00 and 12 UTC within the period DJF 2020/21. The two clustering regions are indicated by the black borders in each panel. Although not used in the clustering, shading shows the corresponding mean 12 h uncertainty growth rate LGR$_Z$ for $Z250$. Vectors are coloured blue when the humidity flux at 850 hPa exceeds $100\,\mathrm{g\,kg^{-1}\,m\,s^{-1}}$. More saturated shading, contours, vectors and dots indicate statistical significance at the 5% level (not accounting for autocorrelation due to the discontinuous nature of date/times in each cluster).

results (not shown). Also, conclusions are very similar for the evaluation based on the date/times obtained for the first region alone; albeit with the impact more confined to the western end of the storm track. Here, results are shown for a 2 day wind-back

(consistent with the time for moderate deepening of a low-pressure system; Wernli and Davies, 1997) and for both regions together. This means that the cyclogenesis and counterpart composites represent a 91:89 partition (nearly 50:50) of the data used in Fig. 5 (top row).

Figure 7 shows the (square-roots of) the terms in the extended error–spread equation Eq. (1) for $Z250$ at day 2 in the ECMWF ensemble, separately for cyclogenesis (top) and counterpart (middle) composites, and their difference (bottom). The

black border indicates the union of the two clustering regions. Comparison shows that ensemble spread for the cyclogenesis composite (Fig. 7b) is enhanced in the western part of the North Atlantic while the spread for the counterpart composite (Fig. 7g) is centred more downstream. There are also corresponding differences in analysis uncertainty (Fig. 7c and Fig. 7h), possibly due to differing uncertainty growth rates in the background forecasts used within the ensemble data assimilation process.

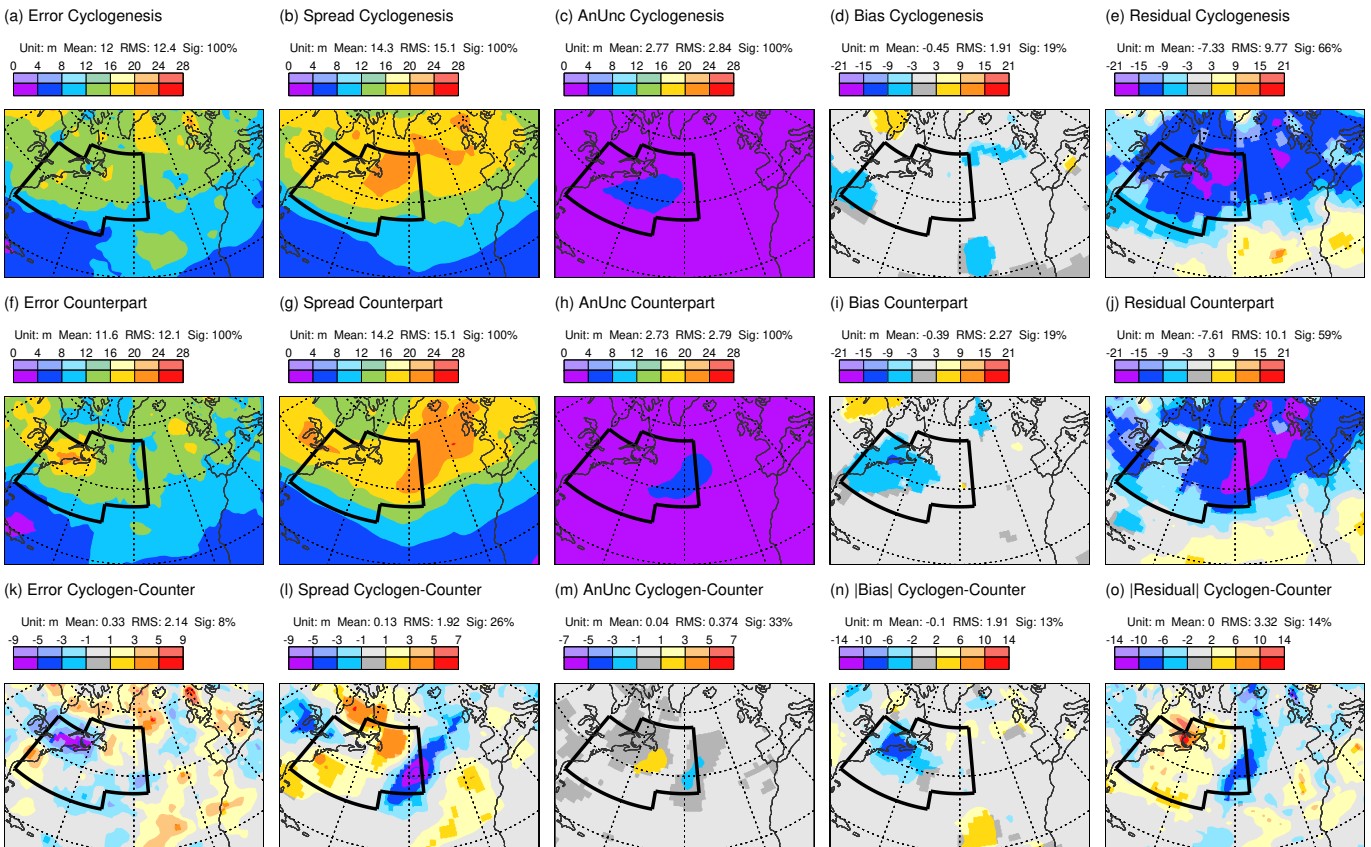

**Figure 7.** As Fig. 5 but separately for the cyclogenesis cluster date/times (top) and the non-cyclogenesis counterpart cluster date/times (middle), for forecasts that start two days prior to the cluster classifications. The bottom row shows cyclogenesis minus counterpart. Note that differences in absolute Bias and Residual are shown, so that blue (red) colours anywhere on the bottom row show where results for the cyclogenesis composite are better (worse) than for the counterpart. The union of the two clustering regions is indicated by the black border.

Figure 7n indicates a significant flow-dependent difference in absolute forecast bias (of about 6 m) along the east coast of North America. This equates to a variance in forecast bias of $\sim 3^2\,\mathrm{m}^2$ — suggesting that this term is as important in (the Residual term of) the day–2 extended error–spread budget for the whole of DJF 2020/21 as the explicitly represented analysis uncertainty (Fig. 5c), and would suggest an even stronger over-spread issue (cf. term estimates in Section 4.1, with a revised reliable spread of $\sim 13.1^2\,\mathrm{m}^2$). In contrast, the positive Residual term over the subtropical North Atlantic (Fig. 5e) could reflect
variance in forecast bias (implied in Fig. 7n) rather than simply indicating ensemble under-spread. Here, in this flow-dependent evaluation, the forecast bias is explicitly represented for each composite in Fig. 7d and Fig. 7i. Notice that the stronger bias along the east coast of North America for the counterpart composite (cf Fig. 7d,i) appears to account for the increased errors seen in this region (cf Fig. 7a,f).

The overall assessment of ensemble spread is seen in the residual terms (Fig. 7e and Fig. 7j). Here it is evident for the ECMWF ensemble that most of the over-spread during DJF 2020/21 in the western North Atlantic region of focus (Fig. 5e), is associated with the cyclogenesis composite — with statistically significant residuals in Fig. 7e and statistically insignificant residuals (indicated by the light blue and light grey colours) in Fig. 7j. Differences are shown in Fig. 7o. They are particularly strong and significant over Newfoundland (the 5% significance level is a stringent test for a season's worth of data, which might be all that is available for evaluation of a new forecast system, and for a diffuse set of cyclogenesis events over the western North Atlantic). Downstream, both composites individually display negative residuals. This is consistent with the above discussion since there has been no direct control for downstream cyclogenesis. Indirectly, it is likely that occurrences of cyclogenesis events over the western North Atlantic are anticorrelated with the occurrences immediately downstream (as seen in cluster patterns Fig. 6c,e) and this might explain the negative cluster differences in absolute residual for the downstream region (Fig. 7o).

Linking the day–2 storm track over-spread (in the region of focus) to cyclogenesis is a key conclusion of this study. It does appear, therefore, that ECMWF initial ENS growth rates (Fig. 3a) associated with cyclogenesis events are too strong. This issue could be associated with several different aspects of the forecast system. Through sensitivity experiments, Sect. 5 explores some of the potential causes.

## 5    Sensitivity experiments to quantify sources of uncertainty in the ECMWF ensemble

In Sect. 3, the initial growth rate of uncertainty (Eq. 3) was discussed. It has subsequently been demonstrated that these growth rates are likely to be too strong in the ECMWF ensemble during cases of extratropical cyclogenesis. Here, sensitivity experiments are used to investigate why this growth rate is too strong in the ECMWF ensemble during cyclogenesis, leading to over-spread at day 2.

Salient details of the ECMWF forecast system were presented in Sect. 2.1. Figure 8 shows the configuration of the sensitivity experiments. It refers to the base operational configuration as EDA=OP, ENS=OP. In the sensitivity experiments, this configuration is successively modified. Firstly, singular vector perturbations to the initial conditions of the ENS are turned off globally (OP-SV) and then model uncertainty in the ENS is turned off globally (OP-SV-MU). From this point, the parametrization of deep convection in the ENS is turned off in a local box (OP-SV-MU-DCP) or the ENS model horizontal grid resolution is increased to $\sim 4$ km (OP-SV-MU+4km). Finally, the assimilation of observations in a local box is turned off for a single EDA cycle (OP-Obs), and the ENS is run again in the OP-SV-MU configuration. Differences between these configurations allow the diagnosis of individual aspects. Vertical arrows in Fig. 8 indicate the sign convention of the difference to be plotted (end of arrow minus beginning of arrow). The conclusions are not thought to be sensitive to the ordering of the various modifications. For example, it will be seen that the impacts on *total* precipitation of DCP and +4km are small, and hence these impacts should be little changed in the presence of the SPPT form of MU. However, parametrized turbulent fluxes might be weakened with +4km, and hence this impact could be a somewhat different in the presence of MU.

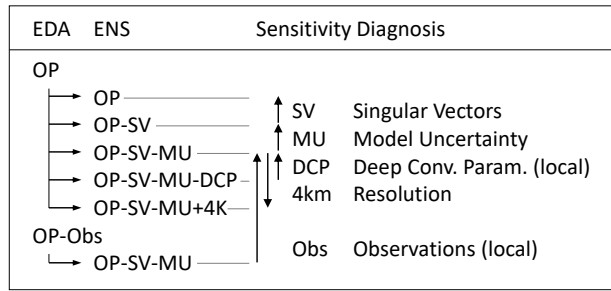

**Figure 8.** Configuration of IFS sensitivity experiments. These involve the Ensemble of Data Assimilations (EDA) and the Ensemble forecast (ENS). Differences between these configurations allow the diagnosis of sensitivity to individual aspects as indicated. See main text for further details.

Figure 9 shows day–2 results for the standard deviation (spread) in $Z250$ from sensitivity experiments for the cyclogenesis case initialised at 12 UTC on 28 November 2019 (i.e., the validity time is one day later than that where the growth rate fields were centred in Fig. 2 and Fig. 3a in order to see the combined effect of the uncertainty source and the growth rate). Grey contours show PMSL, and the black contour shows the 2 PVU-tropopause on 315 K in the unperturbed EDA analysis. A
parallel set of results for spread in $P315$ is shown in Fig. 10. Note that shading intervals vary over the panels shown in these two figures, so that the structures of all impacts can be seen. Mean values and RMS values, integrated over the area shown, are indicated above each panel.

Figure 9a shows the OP configuration with a well-developed surface low pressure system, as discussed in relation to Fig. 1. The WCB associated with this cyclone is seen to lead to the development of a prominent downstream upper-level ridge and a
trough west of Europe. As might be expected, the maximum $Z250$ spread is located downstream of the maximum Lagrangian growth rates (cf. Fig. 3a).

The impact of the initial singular vector perturbations on $Z250$ spread (Fig. 9b) is particularly pronounced along the western flank of the trough over the western North Atlantic (and also of the trough west of Europe). This likely indicates the potential for dynamic growth along the intense jets in these regions, qualitatively in line with the idealized studies by Hakim (2000).
This singular vector impact might help explain the apparent slight westward shift of the centre of maximum growth in the ENS (Fig. 3a) relative to the EDA (Fig. 2). There are places where the singular vector impact on spread is half the total (so that the fraction of variance explained reaches 25%). In contrast to the singular vector impact, the impact of the model uncertainty representation (Fig. 9c) is particularly pronounced in the cyclone centre and in the region of the WCB ahead of the surface low, i.e., in regions where cloud-related physical processes are particularly active. The large signal along the western flank of
the ridge southwest of Greenland is consistent with the results of Joos and Forbes (2016), who found a large influence of cloud microphysical processes in the WCB on the tropopause structure in this part of the downstream ridge. Model uncertainty also explains up to 25% of the total variance. The remaining variance must be associated with the growth of initial EDA analysis uncertainty. This suggests that, even without singular vectors and model uncertainty, this cyclogenesis event acts as a strong magnifier of the initial uncertainty.

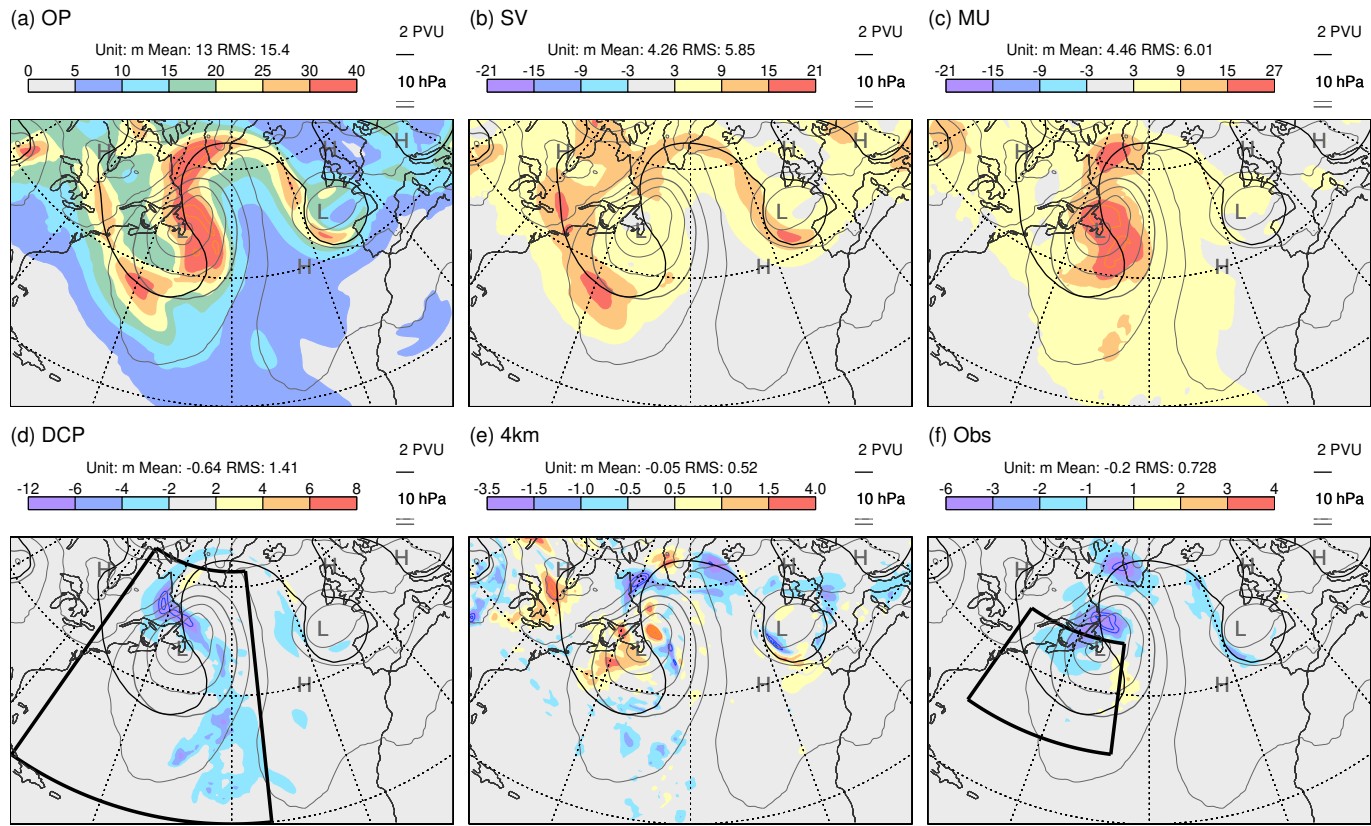

**Figure 9.** Sensitivity results showing day–2 $Z250$ spread (shaded) from ECMWF ensemble forecast experiments initialised at 12 UTC on 28 November 2019. (a) Total spread for the near-operational configuration OP. (b)–(f) Differences in spread between experiments which highlight the impacts of (b) including initial Singular Vector perturbations, (c) including the Model Uncertainty representation, (d) including the parametrization of Deep Convection in the indicated region [75°W–34°W, 20°N–63°N], (e) an increase in model grid resolution to $\sim 4$ km, and (f) the assimilation of observations in the indicated region [75°W–47°W, 30°N–49°N]. Note that the shading interval varies across the panels. Contours extend the shading scheme, with the same interval. In these cases, the most extreme values are indicated at the ends of the colour bar. Mean values and RMS values, integrated over the area shown, are indicated above each panel. Also shown in each panel are the PMSL (grey contours, with interval 10 hPa and the 1000 hPa contour shown more thickly) and PV=2 PVU on the 315 K isentrope (black contour) from the unperturbed EDA analysis.

Figure 9d shows the impact of including the deep convection parametrization (DCP) in the indicated region (note the smaller contour interval). There is a reduction in the spread, particularly in the WCB region, that would otherwise be created when the model is forced to represent this convection on its 16 km grid. Interestingly ensemble mean total precipitation (parametrized plus resolved) is little changed when turning off parametrized deep convection, both in location and amount (not shown).

  The impact of increasing the model grid resolution to $\sim 4$ km is mixed. For $Z250$ spread (Fig. 9e; note the smaller contour
interval), the impact is generally weak. In contrast the impact on $P315$ spread (Fig. 10e) is strong, particularly within the

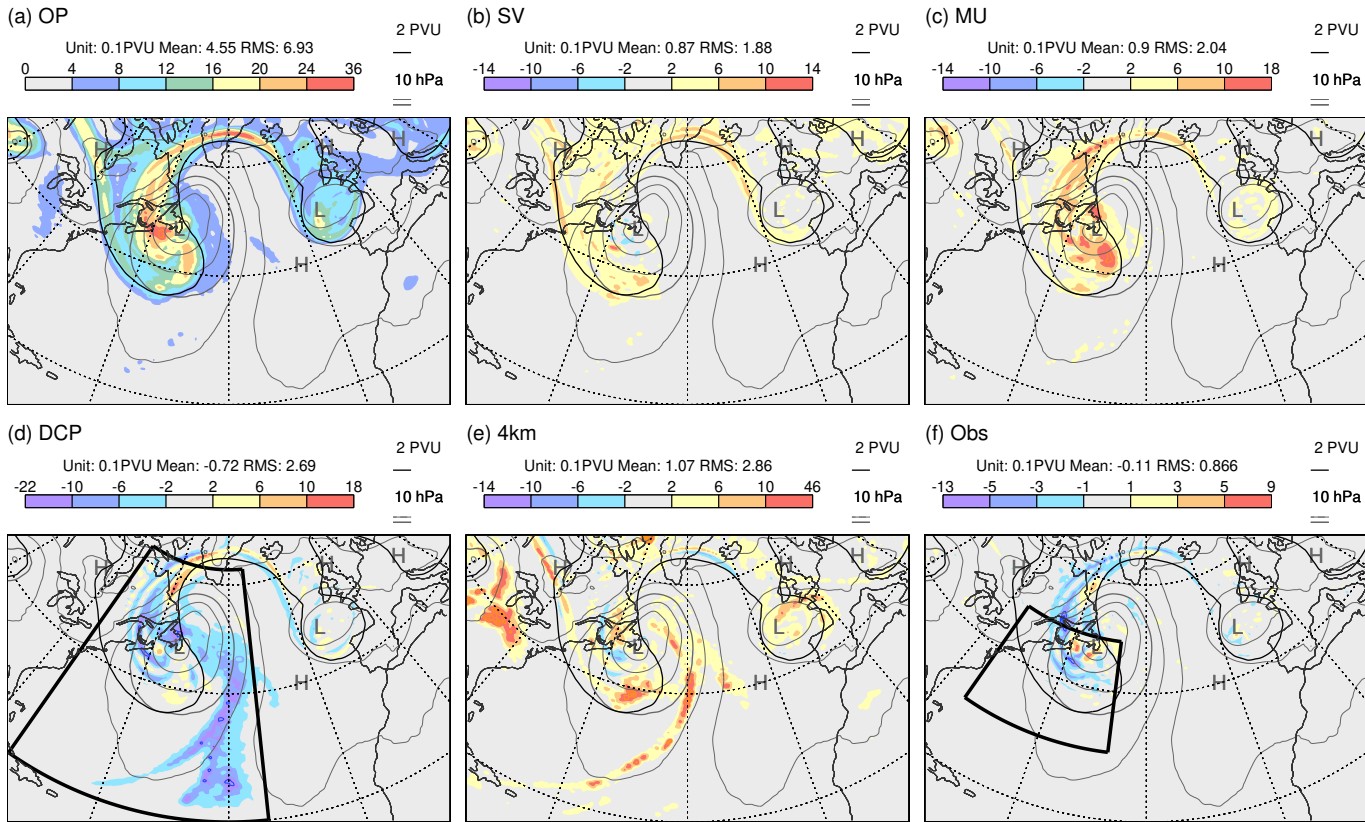

**Figure 10.** As Fig. 9, but shading shows the spread in $P315$.

WCB region. At 4 km, the model attempts to resolve more of the convection. The resolved convection can be associated with stronger updrafts, which might perturb the tropopause more vigorously, where PV gradients are particularly strong. Given that model uncertainty representation is thought to partly account for the impact of sub-grid-scale uncertainty, perhaps the most interesting aspect here is the lack of agreement with the model uncertainty impact (cf. Fig. 10e and Fig. 10c). The impact on
$P315$ uncertainty of allowing the model to resolve more of the convection at the 4 km resolution (Fig. 10e) appears to be in closer agreement with the response to turning off the deep convection parametrisation (minus Fig. 10d), when the model is forced to represent the convection on the 16 km grid. The increase in resolution also results in a small shift of ensemble mean precipitation from parametrized to resolved, with little change in the total (not shown).

The impact of assimilating local observations is obtained using an EDA experiment (OP-Obs, see Fig. 8) where all observa-
tions were denied to the EDA in the region [75°W–47°W, 30°N–49°N] for the single data assimilation cycle that generated the initial conditions; observations outside this region are used in both EDA=OP and EDA=OP-Obs. It is not so easy to anticipate the impact of local observations, particularly with 4DVar, when remote observational information can propagate into the region. Results demonstrate that the assimilation of observations in this baroclinic region does reduce initial uncertainty (not shown). Figure 9f and Fig. 10f show the impact at day 2 (note the smaller contour intervals compared to those for the singular vector

and model uncertainty impacts). There is a reduction in spread in the developing low, which suggests that the assimilation of local observations is beneficial in this case.

Very similar results to those above were obtained for a second set of experiments initialised at 00 UTC on 17 January 2020 — indicating that the conclusions about ensemble spread sensitivity (based on 50 members) may be robust even with only two cases (the same could not be said for error sensitivity with just two cases). In particular, the contributions to the total variance from singular vectors and model uncertainty, with the singular vector impact largely along the western flanks of the trough, and the model uncertainty impact in the region of the WCB. Again, the parametrization of deep convection acts to reduce spread in the WCB. For $P315$, the impact of increased resolution is again more similar to the effect of turning-off the deep convective parametrization than to that of model uncertainty. The main difference of note is that, while there is a reduction in initial spread from the assimilation of local observations, it is weaker than in the case shown above, and its impact at day 2 is marginal.

The above results suggest that singular vectors, model uncertainty, and the deterministic model itself all play important roles in the early development of operational ensemble spread during cyclogenesis. (This is true globally, as can be seen in the variance spectra in Fig. C1 in Appendix C). Since the motivation for the use of singular vector perturbations (Magnusson et al., 2009) and the initial reason for the development of model uncertainty representations (Buizza et al., 1999) was to increase ensemble spread, it makes sense to investigate how these techniques might be modified to reduce the growth of spread during cyclogenesis without negatively impacting the overall performance of the ensemble.

## 6   Summary and Conclusions

The goal of probabilistic forecasting is to maximise the sharpness of the predictive distributions subject to reliability (Gneiting and Raftery, 2007). In ensemble forecasting, better sharpness is associated with reduced ensemble variance, and reliability is associated with the mean agreement between ensemble and outcome distributions. As argued by Rodwell et al. (2018), a focus on short-range flow-dependent reliability (and reduced initial uncertainty) may represent a practical path towards this goal. Synoptic scales are particularly important in global numerical weather prdiction because these are the scales that contribute most to ensemble variance over the first few days (Tribbia and Baumhefner, 2004, see, also, Fig. C1 in Appendix C). It is important to understand what processes drive this growth, and how well reliability is maintained.

Here, a "Lagrangian growth rate" diagnostic $LGR_P$, Eq. (3) left hand side, has been calculated for $12\,\mathrm{h}$ forecasts of upper tropospheric potential vorticity $P315$ in the background ensemble of the ECMWF Ensemble of Data Assimilations (EDA). The supplementary animation of $LGR_P$ for the DJF 2020/21 season highlights a few flow situations over the North American / North Atlantic / European region where ensemble variance growth at synoptic scales is particularly strong and concentrated. Note that this concentration of growth-rates is not dependent on singular vectors perturbations, since these perturbations are not included in the EDA. One such situation is associated with the deepening of extratropical cyclones (cyclogenesis). The material growth identified by $LGR_P$ has two source terms, Eq. (3) right hand side. In situations of cyclogenesis, it is speculated here that uncertainties in baroclinic growth (Hoskins et al., 1985) might be important for enhancing variance growth at the southern extent of upper tropospheric troughs (Fig. 2). Uncertainties in the erosion of the leading edge of the upper tropospheric PV

anomaly by latent heating (Ahmadi-Givi et al., 2004) may be acting to weakly decrease variance in the downstream ridge building region. However, Baumgart and Riemer (2019) suggested that the strongest contribution to ensemble variance is associated with local dynamical interactions between upper tropospheric PV deviations and the winds that they induce. One issue is how to diagnose sources of ensemble variance growth in the face of considerable redistribution of existing variance. A better understanding of the dominant processes involved could help motivate modelling developments, and help data assimilation efforts focus on initialising the most important fields.

The multi-model TIGGE archive (Swinbank et al., 2016) has been used to calculate Lagrangian growth rates in 12 h forecasts of upper-tropospheric geopotential heights $Z250$ for several ensemble systems. Results (Fig. 3 and 4) suggest high sensitivity of the growth rate to initialisation and/or modelling aspects (Table 1). Based on the animations available in the supplementary material, all models show a general enhancement of growth rates in cyclogenesis situations, although often not to the same extent seen in the ECMWF system. Does background continuity within a given EDA member, from one analysis cycle to the next, facilitate the growth of differences (e.g. in baroclinic structures) between EDA members? Does this suggest that singular vector perturbations, applied during the initialisation of the medium-range ensemble forecast, are less needed following the introduction of the EDA? More generally, a better understanding of the differences in initial growth rates between TIGGE models would be useful. For example, what are there sensitivities to the deterministic models (Magnusson et al., 2022), to the representations of model uncertainty, and to the structures represented in the initial uncertainty?

The reliability of each TIGGE ensemble system was evaluated over the DJF 2020/21 season by assessing the consistency between day–2 forecast error and spread in $Z250$. This was done taking into account bias and uncertainty in the verifying analyses through use of the extended error–spread Eq. (1). The ECMWF ensemble displays the smallest error, analysis uncertainty and bias (Fig. 5 columns 1,3,4) in the North Atlantic storm track, but it is over-spread (Fig. 5e). The UK Met Office ensemble showed the best reliability (Fig. 5 column 5). There is some consistency, therefore, between the Lagrangian growth rate results and the reliability evaluation. The conclusions on reliability appear to generalise to other parameters (such as geopotential heights and temperatures at 500 hPa), other seasons, other storm tracks, and continue until the most recent check for the March–May season 2022 (not shown). We would argue that balance in the extended error-spread equation provides a superior reliability target for system development. For example, in the ECMWF ensemble, the standard deviation in day–2 $Z250$ over the east coast of North America during DJF 2020/21 was 18 m. The standard error-spread equation suggests a target of 15 m, while the extended error-spread equation suggests a target of 13.4 m. Including an estimate of the variance of forecast bias reduces this target further to 13.1 m. Moreover, at short forecast ranges, the extended equation can provide a better target for improvements in flow-dependent reliability.

Clustering on flow-types in the western part of the North Atlantic storm track (Fig. 6) demonstrated that the ECMWF over-spread in this region is associated with cyclogenesis events (Fig. 7). Sensitivity studies (Fig. 8) with the ECMWF ensemble reveal some information on the sources of uncertainty during cyclogenesis (Fig. 9 and Fig. 10). A large part is associated with the chaotic growth of the initial (EDA) uncertainty in the deterministic model. Singular vector perturbations to the initial conditions and the model uncertainty representation are also important for the day–2 spread. We speculate here that, when other factors permit, a reduction in the magnitude of the dry singular vector perturbations could improve both sharpness and

reliability within the storm track regions, and make the ensemble forecast more consistent over lead times (more seamless). This would then permit a better use of initial growth rates in the evaluation and improvement of the model and model-uncertainty

— something that would be beneficial throughout the forecast range.

Sensitivities to switching-off the parametrization of deep convection and to increasing model resolution (see, also, Wedi et al., 2020) suggest that the model uncertainty representation should be more strongly focused on convective instabilities (e.g., Christensen et al., 2017). It is possible that such a focus on instabilities (rather than the effects of already-triggered instabilities) might be better explored within the future "Stochastically Perturbed Parameter" (SPP) framework for model uncertainty —

575 perturbing triggering thresholds for example.

In addition to cyclogenesis situations, initial growth rates tend to support the idea that uncertainty can be concentrated in other flow situations, including those prone to mesoscale convection over North America (Palmer et al., 2014) and during the extratropical transition of tropical cyclones. Similar investigations of these initial growth rates, and how reliably the forecast predicts their evolution, could also lead to better flow-dependent reliability and improved overall forecast performance.

*Code and data availability.* The key conclusions in this study are derived from data in the TIGGE archive, which is freely accessible. The ERA5 reanalysis data are also freely available. Other data and diagnostic code are available from the authors upon request.

*Video supplement.* Growth rate animations are available as supplementary material.

## Appendix A: Forecast reliability

Figure A1 shows the key concepts involved in the evaluation of ensemble error–spread reliability. For clarity, the figure is

585 shown in two dimensions and could relate, for example, to the prediction of the location of a storm. For a reliable forecast system (Hamill, 2001; Saetra et al., 2004), the verifying truth $T$ should be statistically indistinguishable from any random sampling of the forecast distribution $F$, indicated by the grey concentric circles. This distribution, which need not be Gaussian or even unimodal, has mean $\mu_F$ and standard deviation of distances from the mean $\sigma_F$ (grey dashed line). Introducing a suffix $j$ to indicate the forecast initiated at time $t_j$ with $j \in \{1, \ldots, n\}$ then, since $F_j$ is reliable (and assumed to be the only

590 information available at time $t_j$ about $T_j$), the expectation $\mathbb{E}_j[\cdot]$ at time $t_j$ is that $\mathbb{E}_j[T_j] = \mu_{Fj}$ and $\mathbb{E}_j[(T_j - \mu_{Fj})^2] = \sigma_{Fj}^2$. The latter condition means that, in Fig. A1, the expected squared length of the top blue line should match the squared length of the grey dashed line. This leads to the well known error–spread relation in ensemble forecasting — that reliability requires $\overline{(T - \widehat{\mu}_F)^2} \approx \overline{\widehat{\sigma}_F^2}$, where $\widehat{\mu}$ and $\widehat{\sigma}$ are the mean and standard deviation estimators, respectively, based on $m$ ensemble members, and the thick overline indicates a mean over the $n$ forecasts. Equality here can be improved by accounting for the finiteness of

595 $m$ and increasing $n$ (Leutbecher and Palmer, 2008). However, for the short forecast lead times of interest here, the error–spread relation is not adequate in general. This is because uncertainties in the knowledge of the verifying truth can be non-negligible

compared to forecast variances at short lead times. Below, Appendix Sect. A1 gives a derivation of an extended error–spread equation, which accounts for bias, analysis uncertainty and sampling. The components of the "residual", which closes the budget, are discussed further in Appendix Sect. A2.

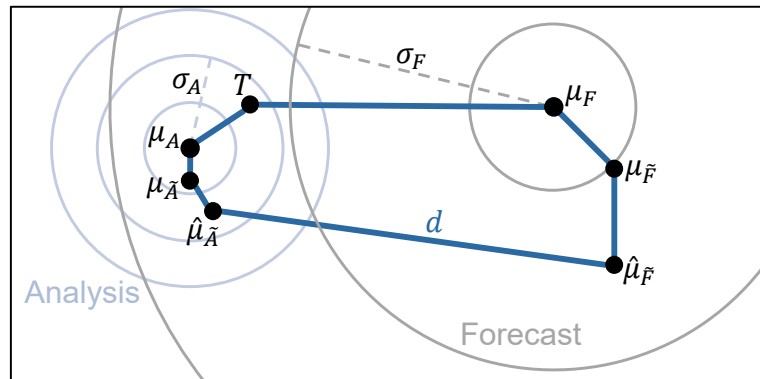

**Figure A1.** Key concepts in the evaluation of ensemble error–spread reliability. The departure $d$ as the difference between the ensemble mean of the forecast $\widehat{\mu}_{\widetilde{F}}$ and the ensemble mean of the verifying analysis $\widehat{\mu}_{\widetilde{A}}$. The forecast and analysis ensembles can be considered as finite samplings of underlying distributions with (mean,variance) $= (\mu_{\widetilde{F}}, \sigma_{\widetilde{F}}^2)$ and $(\mu_{\widetilde{A}}, \sigma_{\widetilde{A}}^2)$, respectively. Circles depict the hypothetical forecast and analysis distributions with (mean,variance) $= (\mu_F, \sigma_F^2)$ and $(\mu_A, \sigma_A^2)$, respectively, that would be created with a perfect model (but with imperfect/incomplete observational information). The truth is indicated with a $T$. See the main text for further discussion.

## A1  Derivation of the extended error–spread equation

The extended error–spread equation takes account of analysis uncertainty. A reliable analysis distribution $A$, which is consistent with the indicated truth $T$, is shown in Fig. A1 using blue concentric circles, with mean $\mu_A$ and standard deviation of distances from the mean $\sigma_A$ (blue dashed line). Let $\widetilde{F}$ and $\widetilde{A}$ be underlying distributions associated with an unreliable forecast system. These distributions could be biased — their means $\mu_{\widetilde{F}}$ and $\mu_{\widetilde{A}}$, respectively, are shown offset from those of the reliable distributions. The ensemble means, based on samplings of these distributions, are indicated by $\widehat{\mu}_{\widetilde{F}}$ and $\widehat{\mu}_{\widetilde{A}}$. These latter two parameters, along with their difference (or departure) $d$, are the only parameters shown in Fig. A1 that are available, being obtained from the ensemble forecast system. Perhaps more importantly for reliability, the distributions $\widetilde{F}$ and $\widetilde{A}$ could have deficiencies in their variances $\sigma_{\widetilde{F}}^2$ and $\sigma_{\widetilde{A}}^2$, respectively. Are they under-spread or over-spread with respect to the variances of the reliable distributions, for example? Again, it is only the ensemble sample estimators $\widehat{\sigma}_{\widetilde{F}}^2$ and $\widehat{\sigma}_{\widetilde{A}}$ that are available.

For initial time $t_j$, the departure of the ensemble mean forecast $\widehat{\mu}_{\widetilde{F}j}$ from the ensemble mean analysis $\widehat{\mu}_{\widetilde{A}j}$ can be written (by following the other solid blue lines in Fig. A1) as:

$$d_j = (\widehat{\mu}_{\widetilde{F}j} - \widehat{\mu}_{\widetilde{A}j})$$

$$= (\widehat{\mu}_{\widetilde{F}j} - \mu_{\widetilde{F}j}) + (\mu_{\widetilde{F}j} - \mu_{Fj}) + (\mu_{Fj} - T_j) + (T_j - \mu_{Aj}) + (\mu_{Aj} - \mu_{\widetilde{A}j}) + (\mu_{\widetilde{A}j} - \widehat{\mu}_{\widetilde{A}j}) \qquad \text{(A 1)}$$

$$= \underbrace{(\mu_{Fj} - T_j)}_{1} + \underbrace{(\widehat{\mu}_{\widetilde{F}j} - \mu_{\widetilde{F}j})}_{2} + \underbrace{(T_j - \mu_{Aj})}_{3} + \underbrace{(\mu_{\widetilde{A}j} - \widehat{\mu}_{\widetilde{A}j})}_{4} + \underbrace{(\mu_{\widetilde{F}j} - \mu_{Fj})}_{5} + \underbrace{(\mu_{Aj} - \mu_{\widetilde{A}j})}_{6} \quad ,$$

where the last line is just a convenient re-arrangement of the terms on the second line.

Although only $d_j$, $\widehat{\mu}_{\widetilde{F}j}$ and $\widehat{\mu}_{\widetilde{A}j}$ are quantifiable, it is possible to discuss the expected values of the terms **1–6** in Eq. (A 1). The expectation operator $\mathbb{E}_j[\cdot]$, introduced above, is the expectation for a given initial time $t_j$. The forecast distributions $F_j$ and $\widetilde{F}_j$ are fixed for given initial time, so the expectation $\mathbb{E}_j$ is over the potential $T_j$ (with distribution $F_j$) and the potential analysis distributions $A_j$ and $\widetilde{A}_j$ (which are dependent on $T_j$). The expectation is also over the finite ensemble samplings $\widehat{F}_j$ of $\widetilde{F}_j$ and $\widehat{A}_j$ of $\widetilde{A}_j$. The reliability of $A_j(T_j)$ implies that $\mathbb{E}_j[T_j] = \mathbb{E}_j[\mu_{Aj}] (= \mu_{Fj})$ and $\mathbb{E}_j[(T_j - \mu_{Aj})^2] = \mathbb{E}_j[\sigma_{Aj}^2]$. Taking the expectation of the last line in Eq. (A 1) we have:

$$\mathbb{E}_j[d_j] = \mathbb{E}_j[\mathbf{5} + \mathbf{6}] = \mathbb{E}_j\left[(\mu_{\widetilde{F}j} - \mu_{Fj}) - (\mu_{\widetilde{A}j} - \mu_{Aj})\right] \equiv \beta_{\widetilde{F}j} - \beta_{\widetilde{A}j} \equiv \beta_j \quad , \qquad \text{(A 2)}$$

since terms **1–4** in Eq. (A 1) have zero expectation. Hence $\mathbb{E}_j[d_j] = \beta_j$ : the expected bias of the unreliable forecast $\beta_{\widetilde{F}j}$ minus the expected bias of the unreliable analysis $\beta_{\widetilde{A}j}$. As lead time increases, one might expect that the forecast bias will become the dominant component of $\beta_j$.

The extended error–spread equation is based on the expected square of Eq. (A 1). This involves squared terms, such as $\mathbb{E}_j[\mathbf{1} \cdot \mathbf{1}]$, and cross terms, such as $2\mathbb{E}_j[\mathbf{1} \cdot \mathbf{6}]$. With the squared terms presented in the same order as in the last line in Eq. (A 1) the expected square of Eq. (A 1) can be written as:

$$\mathbb{E}_j[d_j^2] = \sigma_{Fj}^2 + \tfrac{1}{m}\sigma_{\widetilde{F}j}^2 + \sigma_{Aj}^2 + \tfrac{1}{m}\sigma_{\widetilde{A}j}^2 + (\beta_{\widetilde{F}j} - \beta_{\widetilde{A}j})^2 + \mathcal{E}_j$$

$$= \tfrac{m+1}{m}(\sigma_{\widetilde{F}j}^2 + \sigma_{\widetilde{A}j}^2) + \beta_j^2 + \Big\{ \underbrace{(\sigma_{Fj}^2 - \sigma_{\widetilde{F}j}^2) + (\sigma_{Aj}^2 - \sigma_{\widetilde{A}j}^2)}_{\text{Variance deficit}} + \mathcal{E}_j \Big\} \quad . \qquad \text{(A 3)}$$

The term in braces { } comprises the "variance deficit" (which compares the reliable and unreliable variances of the ensemble forecast and analysis) and $\mathcal{E}_j$, which collects any potentially non-zero cross terms and the (expected) variance in analysis bias (discussed further in Appendix Sect. A2).

From Eq. (A 3), the expected squared departure (over an infinite set of initial forecast times $t_j$) can then be written as

$$\mathbb{E}[d^2] = \tfrac{m+1}{m}\mathbb{E}[\sigma_{\widetilde{F}}^2 + \sigma_{\widetilde{A}}^2] + \mathbb{E}[\beta]^2 + \mathbb{E}[R] \quad , \qquad \text{(A 4)}$$

with expected residual:

$$\mathbb{E}[R] = \mathbb{E}\left[(\sigma_F^2 - \sigma_{\widetilde{F}}^2) + (\sigma_A^2 - \sigma_{\widetilde{A}}^2)\right] \;+\; \mathbb{E}[\mathcal{E}] \;+\; \sigma_\beta^2 \quad, \tag{A 5}$$

where the variance in forecast bias $\sigma_\beta^2 = \mathbb{V}[\beta]$ accounts for the explicit replacement of $\mathbb{E}[\beta^2]$ with $\mathbb{E}[\beta]^2$ in Eq. (A 4). It can be seen that all terms that involve variations in bias have been moved into the residual.

For an unbiased estimator of Eq. (A 4), we note that $\left(\mathbb{E}[d^2] - \mathbb{E}[\beta]^2\right) = \left(\mathbb{E}[d^2] - \mathbb{E}[d]^2\right) = \mathbb{V}[d]$ which has unbiased estimator $\frac{n}{n-1}\left(\overline{d^2} - \overline{d}^2\right)$, and obtain the extended error–spread equation:

$$\underset{\text{Error}^2}{\frac{n}{n-1}\,\overline{d^2}} = \underset{\text{Spread}^2 \;\; \text{AnUnc}^2}{\frac{m+1}{m-1}\,\overline{\left(\widehat{\sigma}_{\widetilde{F}}^2 + \widehat{\sigma}_{\widetilde{A}}^2\right)}} + \underset{\text{Bias}^2}{\frac{n}{n-1}\,\overline{d}^2} + \underset{\text{Residual}^{(2)}}{\overline{R}} \quad . \tag{A 6}$$

where the various terms have been named for future reference. For the purposes of calculating statistical significance, $\overline{R}$ is written as the mean of the residuals $R_j$:

$$R_j \equiv \frac{n}{n-1}\left(d_j^2 - \overline{d}^2\right) - \frac{m+1}{m-1}\left(\widehat{\sigma}_{\widetilde{F}j}^2 + \widehat{\sigma}_{\widetilde{A}j}^2\right) \quad, \tag{A 7}$$

which close the budget for each initial time (for given, constant $\overline{d}$). The extent to which $\overline{R}$ is a good estimate for the expected variance deficit in Eq. (A 5) will depend on the magnitude of the other terms $\mathbb{E}[\mathcal{E}] + \sigma_\beta^2$ in Eq. (A 5). This is investigated in Appendix Sect. A2. It is suggested that the main term which could be non-negligible is the variance in forecast bias, $\sigma_\beta^2$.

## A2  Estimating contributions to the residual term

It is not trivial to estimate the contributions to the Residual$^{(2)}$ term in Eq. (A 6), but a rough attempt is made here. It is useful to write $b_j = \mu_{\widetilde{A}j} - \mu_{Aj}$ (= minus term **6** in Eq. (A 1)) for the bias in the verifying analysis distribution associated with a given realisation of the truth, and subsequent assimilation of observations. Then, from Eq. (A 2), the expected bias in the verifying analysis is given by $\mathbb{E}_j[b_j] = \beta_{\widetilde{A}_j}$.

With reference to the numbered terms in Eq. (A 1), one potential cross term contained in $\mathcal{E}_j$ of Eq. (A 3) is $2\mathbb{E}_j[\mathbf{1}\cdot\mathbf{6}] = 2\sigma_{T_j b_j}$. The covariance $\sigma_{T_j b_j}$, when divided by $\sigma_{T_j}^2 (= \sigma_{F_j}^2)$, measures the linear dependence of the analysis bias on the truth, and this could be non-zero. In the ECMWF EDA, each analysis member is an innovation of that member's background forecast, and this lack of independence might imply that the cross-term $2\mathbb{E}_j[\mathbf{2}\cdot\mathbf{4}] = -\frac{2}{m}\sigma_{\widetilde{F}j\widetilde{A}j}$ is non-zero, particularly at very short lead times and for small ensemble size $m$. (Note that the division by $m$ here is because the covariance of ensemble means is being estimated by the covariance of the two distributions). All other cross-terms are likely to have zero expectation with the exception of $2\mathbb{E}_j[\mathbf{5}\cdot\mathbf{6}] = -2\beta_{\widetilde{F}j}\beta_{\widetilde{A}j}$, which is explicitly represented in the $\beta_j^2$ term in Eq. (A 3). Finally note that $\mathbb{E}_j[\mathbf{6}^2] = \mathbb{V}_j[\mathbf{6}] + \beta_{\widetilde{A}j}^2$ and so $\mathcal{E}_j$ should also include $\mathbb{V}_j[\mathbf{6}] \equiv \sigma_{b_j}^2$, the (expected) variance of the analysis bias. Hence

$$\mathcal{E}_j \approx 2\sigma_{T_j b_j} - \tfrac{2}{m}\sigma_{\widetilde{F}_j \widetilde{A}_j} + \sigma_{b_j}^2 \quad . \tag{A 8}$$

The expected residual in Eq. (A 5) can then be written as

$$\mathbb{E}[R] = \mathbb{E}\left[(\sigma_F^2 - \sigma_{\widetilde{F}}^2) + (\sigma_A^2 - \sigma_{\widetilde{A}}^2)\right] \;+\; \mathbb{E}\left[2\sigma_{Tb} - \tfrac{2}{m}\sigma_{\widetilde{F}\widetilde{A}} + \sigma_b^2\right] \;+\; \sigma_{\widetilde{\beta}}^2 \quad , \tag{A 9}$$

As noted in the main text, all terms that involve variations in bias have been moved into the residual. Here, an attempt is made to estimate these terms for day–2 forecasts of $T500$ during the December–February (DJF) season. Mid-tropospheric temperatures are choose here because they are relatable, through vertical integration of the hydrostatic equation, to the upper-tropospheric

height field discussed in the main text, and because relevant temperature observations are assimilated into the EDA. Here, "AMSUA" channel 5 satellite brightness temperature observations are used. These measure mid-tropospheric temperatures with a maximum weighting at $\sim 500\,\mathrm{hPa}$, and have had variational bias correction, VarBC, applied. For reference, Fig. A2 shows the terms of the extended error–spread Eq. (A 6) in squared units. Consistent with the upper-tropospheric height field results in the main text, this also highlights a potential over-spread in the storm track. The following list outlines the approach

taken here to estimate the right-most four terms in Eq. (A 9).

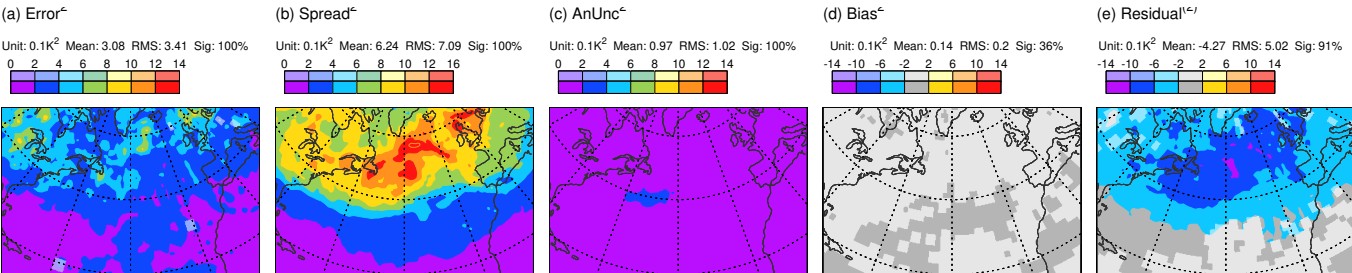

**Figure A2.** Extended error–spread budget (squared terms) for DJF 2020/21 for $T500$ on day 2 from ECMWF ensemble forecasts. Statistically significant values are shown with more saturated colours. Area means (for the area displayed) are indicated at the top of each panel (e.g. "Sig: 85%" means that 85% of the area is significant at the 5% significance level).

– $2\mathbb{E}[\sigma_{Tb}]$ in Eq. (A 9) is tricky to estimate as it requires the state-dependent estimation of analysis bias. One approach is to utilise inter-annual variability over the $n_y$ (=10) years 2012–2021, and to regress the DJF-mean analysis bias (relative to a set of observations) against the DJF-mean observations themselves. The so-called "analysis departure" is the observation minus the analysis-equivalent of the observation (after applying the relevant observation operator).

Assuming that the analysis bias (here for $T500$) closely matches minus the mean analysis departure from the AMSUA brightness observations, we can make the estimate

$$2\mathbb{E}\left[\sigma_{Tb}\right] \approx -\frac{2m}{m-1}\frac{\widehat{\sigma}_{o(o-a)}}{\widehat{\sigma}_o^2}\overline{\widehat{\sigma}_F^2} \quad , \tag{A 10}$$

where $o$ refers to the seasonal-means of the observations, and $a$ refers to the seasonal-means of the analyses. Notice that the bias variations are scaled to reflect the synoptic variations in truth during DJF 2020/21. Figure A3(a) shows that, over the North Atlantic, this estimated covariance is $\sim 0.025\,\mathrm{K}^2$. This is generally smaller in magnitude than the analysis variance $\sim 0.2\,\mathrm{K}^2$ (Fig. A2c), and clearly smaller in magnitude than the full residual $\sim 0.6\,\mathrm{K}^2$ (Fig. A2e).

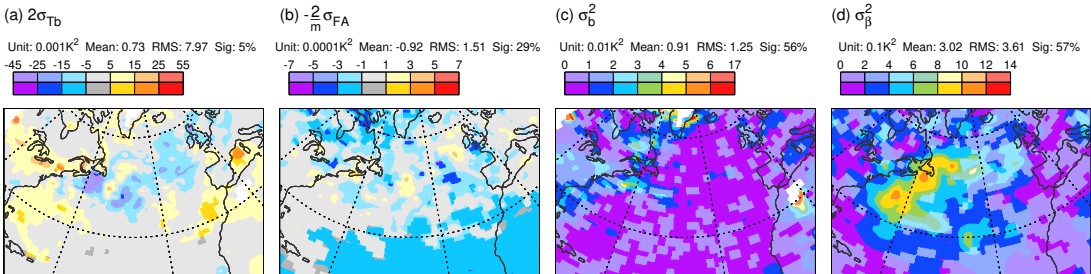

**Figure A3.** Estimates of the magnitudes of the other terms included in the Residual for $T500$ during the period DJF 2020/21. (a) Covariance of the analysis bias with the atmospheric state ($\times 2$). (b) Covariance of the ensemble-mean day–2 forecast and ensemble-mean analysis ($\times -2$). (c) Variance of the analysis bias. (d) Variance of the day–2 forecast bias (relative to analysis). Note that the units differ widely between panels. Statistically significant values are shown with more saturated colours. Area means (for the area displayed) are indicated at the top of each panel (e.g. "Sig: 29%" means that 29% of the area is significant at the 5% significance level). Please see main text for further details.

– $\sigma_b^2$, in Eq. (A 9) can also be estimated by utilising inter-annual variability of seasonal-means:

$$\mathbb{E}\left[\sigma_b^2\right] \approx \frac{n' n_y}{n_y - 1}\widehat{\sigma}_{(o-a)}^2 \quad , \tag{A 11}$$

where the multiplier $n'$ is the number of synoptic degrees of freedom in a season (here $n' = 30$ assuming a 3 day synoptic decorrelation timescale). Figure A3(c) shows that this term is $\sim 0.02\,\mathrm{K}^2$ in the North Atlantic region, so again smaller than the analysis variance and residual terms.

– $\sigma_\beta^2$, in Eq. (A 9) can be estimated by trying a similar approach:

$$\mathbb{E}\left[\sigma_\beta^2\right] \approx \frac{n' n_y}{n_y - 1}\widehat{\sigma}_{(f-a)}^2 \quad , \tag{A 12}$$

where $f$ refers to the day–2 seasonal-means of the (deterministic HRES) forecasts. Figure A3(d) shows that this term is $\sim 1\,\mathrm{K}^2$ over the western North Atlantic region. This may be an over-estimate since reducing predictive skill with lead time means that $(f - a)$ begins to reflect (minus) the observed anomaly from climate, and thus becomes less useful at indicating forecast bias as the forecast lead time increases (e.g. by day 10). There is also some uncertainty in the value

chosen for $n'$. Nevertheless, variations in forecast bias may well provide a non-negligible contribution to the residual term — implying here that the over-spread is even larger over the western North Atlantic. Flow-dependent evaluation of the extended error–spread budget, discussed in Sect. 4.3, represents a means of reducing issues associated with flow-dependent variations in forecast bias.

– $-\frac{2}{m}\sigma_{\widetilde{F}\widetilde{A}}$ in Eq. (A 9) does not involve bias and can be estimated in-sample as

$$\mathbb{E}\left[-\frac{2}{m}\sigma_{\widetilde{F}\widetilde{A}}\right] \approx -\frac{2}{m-1}\widehat{\overline{\sigma_{\widetilde{F}\widetilde{A}}}} \quad . \tag{A 13}$$

The scale to the colour bar in Fig. A3(b) shows that this term is several orders of magnitude smaller than the analysis variance and residual terms.

Considering all these terms, it appears that the residual in the day–2 $T500$ budget largely represents the ensemble variance deficit and, possibly, the variance in forecast bias $\sigma_\beta^2$ - which is appreciable in the cyclogenesis region off the east coast of North America. With the relationship between mid-tropospheric temperatures and upper-tropospheric heights discussed above, it is thought that this conclusion will carry over to the upper-tropospheric height field budget discussed in the main text.

## Appendix B: Derivation of the Lagrangian growth rate equation

For any parameter $Y$, write $Y_i$ for its value in ensemble forecast member $i \in \{1,\ldots,m\}$. Using a thin overline to denote a mean over ensemble members $\overline{Y} = m^{-1}\sum_i Y_i$ (even for non-linear terms: $\overline{YZ} = m^{-1}\sum_i Y_i Z_i$) and a prime to denote deviations from the mean $Y_i' = Y_i - \overline{Y}$, then the variance estimator for potential vorticity $P$ can be written as $\widehat{\sigma}_P^2 = \overline{P'^2}$. Using this notation and the results that $\overline{Y'} = 0$ and $\overline{Y'\overline{Z}} = 0$, the local exponential growth rate for $P$ can be written as

$$
\begin{aligned}
\frac{1}{\widehat{\sigma}_P}\frac{\partial\widehat{\sigma}_P}{\partial t} &= \frac{1}{2\widehat{\sigma}_P^2}\frac{\partial\widehat{\sigma}_P^2}{\partial t} \\
&= \frac{1}{2\widehat{\sigma}_P^2}\frac{\partial\overline{P'^2}}{\partial t} \\
&= \frac{1}{\widehat{\sigma}_P^2}\overline{P'\left(\frac{\partial P}{\partial t} - \frac{\partial\overline{P}}{\partial t}\right)} \\
&= \frac{1}{\widehat{\sigma}_P^2}\overline{P'\left(\mathcal{D} - \boldsymbol{v}\cdot\boldsymbol{\nabla}_\theta P\right)} \\
&= \frac{1}{\widehat{\sigma}_P^2}\overline{P'\left(\mathcal{D}' - \overline{\boldsymbol{v}}\cdot\boldsymbol{\nabla}_\theta P' - \boldsymbol{v}'\cdot\boldsymbol{\nabla}_\theta P\right)} \\
&= -\frac{1}{2\widehat{\sigma}_P^2}\overline{\boldsymbol{v}}\cdot\boldsymbol{\nabla}_\theta\overline{P'^2} + \frac{1}{\widehat{\sigma}_P^2}\overline{P'\left(\mathcal{D}' - \boldsymbol{v}'\cdot\boldsymbol{\nabla}_\theta P\right)} \\
&= -\frac{1}{\widehat{\sigma}_P}\overline{\boldsymbol{v}}\cdot\boldsymbol{\nabla}_\theta\widehat{\sigma}_P + \frac{1}{\widehat{\sigma}_P^2}\overline{P'\mathcal{D}'} - \frac{1}{\widehat{\sigma}_P^2}\overline{P'\boldsymbol{v}'\cdot\boldsymbol{\nabla}_\theta P} \quad ,
\end{aligned}
\tag{B 1}
$$

The first term on the last line is the advection of uncertainty by the ensemble mean wind. Equation (B 1) can be rearranged to give Eq. (3).

## Appendix C: Variance spectra for the sensitivity experiments

Figure C1 shows the $T500$ variance spectra of global spherical harmonics against total wavenumber for the experiments discussed in Sect. 5. The approximate scale is indicated on the top x-axis. Initial uncertainty from the EDA (Fig. C1, thin green curve) occurs at all scales, with the largest variance contributions at scales $\sim 400\,$km. The thin black curve shows the initial spectrum after addition of the singular vector perturbations (at scales $\geqslant 1000\,$km). The contributions to day–2 variance associated with singular vector perturbations and model uncertainty representation are indicated by the red and green shaded regions, respectively. Singular vectors contribute strongly at synoptic scales while model uncertainty contributes at synoptic and planetary scales. The thick green curve indicates the impact of pure chaotic growth of EDA uncertainty without singular vectors or model uncertainty in the forecast). Notice that the day–2 and EDA variance spectra coincide at scales smaller than $\sim 100\,$km — indicating that the EDA variance is already saturated at these scales. The thick blue curve shows the day–2 spectrum when the gridpoint resolution of the forecast model is increased to $\sim 4\,$km. The spectrum is seen to shallow, even at scales already represented by the original $\sim 16\,$km model (as indicated by the blue shaded region). A similar plot based on the same set of experiments for the second initial date (00 UTC on 17 January 2020) is almost indistinguishable from Fig. C1.

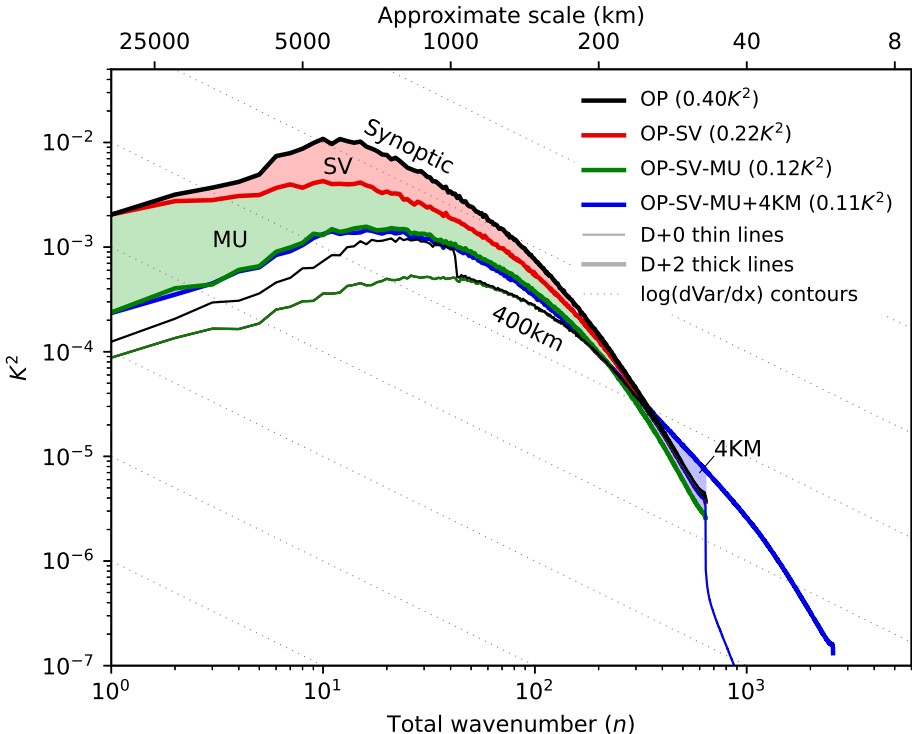

**Figure C1.** Spectra of ensemble $T500$ variance for the indicated experiments, summarised in Fig. 8. Spectra show the variance of the ensemble filtered on total wavenumber $n$ for global spherical harmonics. The approximate scale of each harmonic is indicated on the top x-axis (this relates to a half wavelength assuming equal zonal and meridional wavenumbers). Thin lines show initial time and thick lines show day 2. The red, green and blue shaded regions indicate the impacts at day 2 of including initial singular vector (SV) perturbations, model uncertainty (MU) representation and increasing the gridpoint resolution of the forecast model to $\sim 4\,\mathrm{km}$. Diagonal dotted lines are contours of variance contribution per linear unit length on the x-axis, with contour value indicated where they intersect the y-axis. Total variance for each experiment is indicated in parenthesis in the key.

*Author contributions.* This study arose from close collaboration between the authors during HW's period as an ECMWF Fellow, and from ideas generated in the Warm Conveyor Belt workshop, which they co-organised in 2020. The mathematical content and much of the diagnostic work for this study was completed by MJR, with HW providing important guidance throughout.

*Competing interests.* Heini Wernli is a member of the editorial board of Weather and Climate Dynamics. No other competing interests.

*Acknowledgements.* For the sensitivity experiments, the authors would like to thank Peter Bechtold, Andrew Dawson, Richard Forbes, Alan Geer, Elias Hólm, Bruce Ingleby, Simon Lang, Inna Polichtchouk, and Gabor Radnoti for their considerable technical assistance. The authors would also like to thank Katharina Heitmann (ETH Zurich) for preparing Fig. 1 and Jonathan Day, Rebecca Emerton, David Lavers, Martin Leutbecher, Linus Magnusson, Florian Pappenberger, and David Richardson for useful discussions about this work. Finally, the authors would like to express their great appreciation to Ron McTaggart-Cowan and another anonymous reviewer for their time and insightful
remarks, which have led to improvements in this manuscript.

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
