# Peer review of "Uncertainty growth and forecast reliability during extratropical cyclogenesis"

_Weather and Climate Dynamics, 2022_

## Referee Comment (RC1)

Notes on "The Cyclogenesis Butterfly: Uncertainty growth and forecast reliability during extratropical cyclogenesis"

**Overview**

This manuscript addresses the very interesting problem of flow-dependent variability in ensemble reliability. Such an analysis is of significant practical utility because it gives ensemble designers important robust insights into their system's behaviour under identifiable meteorological conditions. Specifically for ensemble applications, such information is an essential replacement for the case study approach – although arguably conditional evaluations should also be preferred for deterministic systems.

It is clear from the breadth of the analysis that an impressive amount of work has gone into this investigation. However, the manuscripts suffers from the lack of a clearly stated objective for the complex diagnostics employed. As a result, the text gets mired in technical discussions rather than focusing on interpretations and discussions that support the objectives of the work and advance the main narrative of the manuscript. Similarly, many of the novel diagnostics themselves (for example Eqs. 1 and 2) seem to be overly complex for a study that arrives at relatively straight-forward – though very useful – conclusions regarding conditional overdispersion in the ECMWF ensemble.

As described in General Comments #1 and #2 below, I think that this work is interesting and important enough to be split into two separate manuscripts. The result will be two independent but complimentary studies that better motivate and demonstrate the utility of the proposed techniques. Such a reshaping of the investigation will also permit the introduction of more synthesis and interpretation of the results, resulting in a pair of papers that will have a larger impact on the field.

Recommendation: Resubmit after splitting the study into two separate manuscripts.

Reviewer: Ron McTaggart-Cowan

**General Comments**

1. This manuscript presents a huge amount of material and it is clear that an awful lot of work has gone into this analysis. However, I think that the vast array of content actually reduces the potential impact of the study. Stronger curation of the information would focus the manuscript – and the reader – on the truly important elements of the work that lead directly to the conclusions. One way to start improving the focus of the study will be to identify and clearly state the objective of the work. That could effectively be done at the start of the last paragraph of the introduction. I encourage the authors then to take a serious look at each element of content and decide whether or not it is essential to advancing the manuscript towards this objective. Components that do not fit into this focus should be removed and could probably form the basis for a separate submission.
2. In the end, I think that this is really two papers. The first paper is about ensemble-estimated uncertainty growth rates and their relationship to cyclone intensification and/or trough amplification over the western North Atlantic. The second paper is about documenting and

identifying the source of overdispersion in the ECMWF ensemble in the North Atlantic storm track.  Although the second is clearly motivated by the first, these topics are separate enough that they would not even need to be a two-part submission:  they could be treated entirely separately.  Having two separate papers would allow for an expansion of discussions and dynamical interpretations, in addition to the introduction of important material into the main text that is currently relegated to the multiple appendices.  I really think that the prodigious amount of effort that clearly went into this analysis would be much better served by two independent submissions.

3. [This comment is only directly relevant if the current submission is not split into two separate manuscripts.]  Organizing the paper into 11 sections is highly unusual.  Although I appreciate the use of sections and subsections as important tools for organizing content, I think that in this case there are so many sections that readers will lose the "big picture" of the manuscript's organization.  To a certain extent, the excessive number of sections appears to be a symptom of a stream-of-consciousness design.  Rather than presenting the work in the order that it was executed, consider reorganizing it into larger logical chunks for the reader.  For example, the extremely short Data section (2) should be augmented to include the methods currently described in sections 4 and 6, and part of section 8.  It seems like sections 3, 5 and 10 would be more logically grouped as a single (case study) section with appropriate subsections.  Sections 7, 8 and 9 should also be considered subsections of a "full-season" analysis section.  The result would be a 5-section paper:  (1) introduction, (2) data and methods, (3) case studies and sensitivity tests, (4) full-season analysis and model intercomparison and (5) conclusions.  I believe that such a reorganization would really help to increase the potential impact of this study on the field.

4. The two case studies appear to yield similar results.  If the current document is to be revised as a single submission, one of the two case studies could be relegated to supplemental material.  The main text could then claim demonstrable robustness with reference to the results shown in the supplement.  If the material will be split into two independent studies (General Comment #2), then the two case studies could be retained in the first paper, along with augmented evaluation and interpretation.

5. I think that a study of finite perturbation growth rates that cites the "butterfly effect" should mention Durran and Gingrich (2014), although I understand that the perturbation scales discussed here are much larger than the near-truncation scales found to be "unimportant" in the 2014 study (indeed, you mention this in your 2018 BAMS article).  Perhaps this suggests that the "cyclogenesis butterfly" is a bit of a misnomer and (although catchy) might introduce some confusion:  these are **very** big butterflies.

6. Based on the time periods discussed in the case studies, I think that "rapid cyclone deepening" would be a better description of the uncertainty precursor than "cyclogenesis".  Both cyclones form 1-2 days before the period of interest, but intensify rapidly over the Gulf Stream.  I think that the distinction is important particularly in this region, where secondary cyclogenesis (i.e. the formation of a cyclonic circulation where none existed previously) is common and could easily be misunderstood to be the "butterfly".  Clarifying the focus on rapid deepening of preexisting cyclones (if I am right about that) further emphasizes the fact that this study is looking at synoptic-scale uncertainty seeds, rather than the potentially mesoscale cyclone development precursors.

7.  Although the breakdown of the Lagrangian growth rate into "non-conservative" and "advective" components (Eq. 1) is interesting, it does not seem to have any impact on this work. The analysis appears to proceed to look at only the Lagrangian growth rate itself, i.e. the l. h. s. of Eq. 1 rather than the forcing terms. If this is true, then the focus of the manuscript can be tightened by removing Eq. 1 and associated discussions, including most of appendix B (the remainder should be included in the augmented "Data and Methods" section, particularly if Z250 is adopted throughout as recommended in General Comment #13).

8.  The study references animations periodically. This means that readers will need to interrupt their progress to look at animations available in supplemental material. As far as I can tell, most of the relevant information could be presented as additional panels in the existing figures. For example, Figs. 2 and 3 are both single panel, but could be augmented to show other lead times to avoid the need for references to separate animations in the text.

9.  Differences in the ensemble perturbation techniques between the different modelling systems investigated here seem potentially important, particularly given the short lead time. The use of SV perturbations in ECMWF ENS distinguishes it from most other systems in the TIGGE database, other than perhaps JMA. A discussion of these differences (or at least their itemization in an introductory table) would be very useful.

10. This study looks at uncertainty (ensemble spread) growth rates from the perspective of synoptic cyclone dynamics. To make a convincing connection between the uncertainty growth and cyclone development it would be very useful to compare the former to something like the moist baroclinic growth rate (e.g. Booth et al. 2015; ASL). A high degree of correlation between the two would be good evidence of the importance of rapid cyclone deepening to spread growth in the ensemble. Even something relatively simple like comparing the time series of area-averaged (over the Gulf Stream region) ensemble growth rates and moist baroclinic growth rates (with rapid deepening events identified) would provide a really nice dynamically based assessment of the importance of cyclone development to uncertainty.

11. The maximum uncertainty growth region in Fig. 2 is upshear of the trough axis, where vorticity advection is negative aloft. Why is this? In both cases (Figs. 2 and 3) the cyclone is located between the dipole in growth rates, not at all within the peak growth rate south of the trough. This is not "ahead of the base of the upper-level trough" or "preceeding cyclogenesis" (line 142). I understand that some amount of spatial smearing arises from the use of 12-h differences to compute the growth rates, but the cyclones do not even appear to move through the maximum growth rate region. So then would it be more accurate to link large spread growth rates to amplifying upper-level troughs rather than cyclones per se? For example, perhaps uncertainties in the strength of the jet streak on the upshear flank of the trough (associated with its meridional extension) are more important than the lower-level cyclone itself.

12. The bulk of discussions around the spread-error relationship appear to focus on the Spread and Residual terms of Eq. 2, leading to conclusions about overdispersion in the North Atlantic storm track. Is there no simpler way to arrive at the important conclusions of the study without going through this rather complicated derivation and analysis? The interesting flow-dependent aspect of the spread-error relationship is achieved through independent stratification (currently via cluster analysis), so I think the only thing that might be lost would be the conditional bias shown in Fig. 10i. However, this bias could be evaluated directly and shown to contribute significantly to the increased RMSE in the "counterpart" cluster without resorting to Eq. 2. The apparent

ambiguity of the Residual term makes the discussions surrounding Eq. 2 quite difficult to follow and appears to make it difficult to make definitive statements about sources of problems within the ensemble. If the important message to be delivered by this work relates to the flow-dependent overdispersion in the ensemble, then a simpler analysis (perhaps including regional and/or flow-stratified spread-reliability diagrams) might be a more effective vehicle. However, if the current investigation is just a showcase for the analytic technique itself then (a) that should be clarified and (b) the advantages of this technique over a simpler analysis should be emphasized.

13. The lack of PV in the TIGGE database requires the use of Z250, which appears to produce similar results (Figs. 2-5). Although I can completely understand the appeal of starting with PV in this discussion, I think that for pragmatic reasons the entire study should focus on Z250. In the Data and Methods section the rationale for this can be very clearly explained. This would only really affect current sections 4 and 5. The PV 315 diagnostics in (current) section 10 could still be used because they are separate from the growth rate discussion.

14. Why was the clustering approach (current section 8) preferred over a much simpler cyclone identification approach? It seems as though clusters 2 and 3 for both domains are lumped into the "non-cyclogenesis" category when the results from the two domains were aggregated. As such, this seems like a very complicated way to identify dates with cyclones in the western North Atlantic.

15. I am not sure grammatically why "growth-rate" is hyphenated throughout. This does not seem to be a common construction.

16. I do not believe that forecast "lead-time" is usually hyphenated. More generally, there appears to be over-hyphenation throughout the text. Please limit the use of hyphens and ensure that they are represented using hyphen characters rather than the current em-dashes.

17. Please confirm that date/time formatting conforms with WCD standards.

**Specific Comments**

18. [L45] Distinguish between the true unstable modes of the flow and the computed singular vectors (optimal tangent linear growth with limited moist physics). The note about the "linear regime" points in this direction, but it would be useful to make this distinction right off the bat.

19. [L48] It would be useful to itemize some of these approximations here because the difference between ensemble spread growth and error growth rate is fundamental to this study.

20. [L52-54] The punctuation of this sentence makes it difficult to follow: consider rewording.

21. [L54] Remove hyphen from "ensemble-mean".

22. [L55-56] Replace "Jetstream" with "jet stream", "wave-guide" with "waveguide", and "down-stream" with "downstream".

23. [L57-72] This "outline" paragraph is overly long and complex because it strays into "abstract" territory by summarizing results. Consider shortening this paragraph by restricting its content to section descriptions only.

24. [L58] Provide a reference for TIGGE if it is to be mentioned here. Also confirm that this acronym can be used without definition in WCD, or define it.

25. [L73] Suggest dropping the first two sentences of this section and including all dataset descriptions here so that the flow of the remainder of the text is not interrupted by them. As

noted in General Comment #1, this section should be rewritten to include information about the datasets and methods used throughout the study.

26. [L73] I believe that "re-analysis" is more usually "reanalysis", including in Hersbach et al. (2020).
27. [L75] The forecast range of the background does not seem to be identified here or in Appendix E. It seems to be 12 h (line 136), but that should be clarified here.
28. [L77] TIGGE stands for the "THORPEX Interactive Grand Global Ensemble".
29. [L80-83] This information would probably be better displayed as a table for easier reference in later sections.
30. [L84] Suggest, "These data are used …".
31. [Fig. 1] Are the trajectories that are used to identify the WCB region extending from -24h to +24h from the analysis valid time (i.e. these are the trajectory midpoints)? Suggest using the "red hatching" term consistently in the caption, rather than "shown in red".
32. [Fig. 1] Should the mks form of PVU be provided in the caption?
33. [L104-106] Are these the forecast experiments discussed in section 10? If so, then this is additional motivation to move that section up as a "case study" subsection.
34. [L108] Suggest "… uncertainty grow-rate estimate …" because the ensemble provides only an estimate of the true forecast uncertainty.
35. [L109] What does the "1-dimensional" restriction mean here? Would this be better identified as "scalar", or can multiple state variables be included in a 1D state vector? This is obviously important because it reappears elsewhere in the text.
36. [L114] The phrase "but with a different formulation" is too vague.
37. [L118-124] This is a very complex sentence mixes conservative and non-conservative forcings in Eq. 1. It would be more useful to split this sentence to describe the physical relevance of the terms on the r.h.s of Eq. 1 individually.
38. [L126] Should "Equation" be capitalized here? It wasn't in section 1. I do not think that the back-reference to section 1 is very useful here because the introduction did not go into much additional detail about the Liouville equation. A citation to relevant literature would be more useful here.
39. [L125-130] I think that this discussion is fine, but it does not seem to advance the main thread of the study. It could be dropped to reduce the length of the manuscript.
40. [L132-140] This information should be contained in the captions (most of it is) and/or left for supplemental material because it disrupts the flow of the main text.
41. [Fig. 2] What is the contour interval for the contours showing extreme values?
42. [Fig. 3] Should this read "Case 2"?
43. [L145] Is this a third case study being introduced? I think that discussion of the full-season perspective should be left for the subsequent section (in the reorganized paper).
44. [L145-150] These seem like "future work" suggestions that would be better left for the concluding discussion.
45. [L152-155] The 12-h forecasts from the TIGGE database are for ENS rather than EDA, is that correct? If so, then is it true that Figs. 4a and 5a look different from Figs. 2 and 3 not only because the field is different but also because the perturbations are different? If I understand the ECMWF system correctly, SV perturbations are not added within the EDA cycle, but are added before ENS initialization. In that case, Figs. 4a and 5a have an additional source of optimized growth. That seems to make the comparison interesting, although it is complicated

by the change in diagnostic field. Would it not be surprising if the SV perturbations have little impact on growth rates in these cases? Perhaps the Z250 growth rates could be shown for Figs. 2 and 3 to make this comparison possible.

46. [L155-156] So are these case studies (particularly Fig. 4) not representative of the general behaviour of these models? If so, perhaps another case study should be chosen for this comparison.

47. [L172-174] The source of Eq. 2 (appendix C) should be cited at the beginning of this discussion.

48. [L181-182] I have a hard time understanding a lot of this discussion and how it relates to Fig. 6. It would be great to label the lines in Fig. 6 with the names of the terms in Eq. 2 that they relate to. The lines seem to be more directly related to the discussion in Appendix C, so perhaps Fig. 6 would be more appropriate in the appendix.

49. [L192] Does this "main additional term in the Residual" refer to Eq. C5? If so, it would be useful to cite that equation here.

50. [Fig. 7] The change in colour scale range for panels (n) and (o) make comparison of the plots on the bottom row difficult. With the current plotting scheme, it looks like the difference in residual is almost entirely explicable by the difference in spread, but that is not really the case (is it)? The contour intervals for values beyond the standard colour bars should be noted in the caption.

51. [L219-223] It is challenging to follow this discussion because of two forward-references to a description of the variance of forecast biases. It seems like that aspect of the discussion should be introduced before this text appears. In fact, it is not clear what discussion the forward-references here are actually describing (the section 9 discussion seems to take an understanding of the forecast bias variance's impact on the Residual for granted).

52. [L233] It was not obvious that this is a "key question", so hopefully a clear statement of the study's objective(s) in the introduction will help to make that link more direct.

53. [L233-235] Does this "either-or" statement arise from the form of the Residual term (Eq. C5)? If so, then it seems like it would be useful to put this equation in the main text, hopefully as part of a discussion on the meaning of "variance in forecast bias", which I think might be related to the "difficulties" proposed here (?).

54. [L242-243] This region is quite complex: why would three clusters necessarily "provide sufficient degrees of freedom"? The optimal number of clusters is difficult to determine, but usually drop-offs in quantities like the AIC or BIC serve as some sort of semi-quantifiable justification for the number of clusters.

55. [L256-258] This is the only discussion of the uncertainty growth rate in this section, and it does not seem to lead to any particular conclusion. Is there a good reason to include it here and in the Fig. 8 and 9 plots? (It does not seem to be discussed in the subsequent section either.)

56. [L294] The phrase "almost the entire over-spread" seems like a bit of an overstatement. It is probably more defensible in terms of variance, but could perhaps be softened to "much of the overdispersion" or similar.

57. [L297-301] I am afraid that I do not fully understand this discussion. How would the stratification of the groups (cyclone vs. non-cyclone) be done differently with multiple seasons or an independent assessment? Could this "regression to the mean" alternatively be considered a sampling bias?

58. [L302.5] Consider simplifying the section title to "Sensitivity experiments to quantify uncertainty sources".

59. [L315] I understand that resource constraints likely make additional tests difficult or impossible, but is it not conceivable that the ordering of MU and 4K is important? Systematic changes in the physics tendencies should be expected between 18 km and 4 km grid spacing (for example as more turbulent fluxes are represented by the dynamics), which will impact SPPT directly. This might mean that the impact of switching MU on and off at 4 km is different from what is observed in the 18 km configuration. I do not think that this is a big enough deal (or close enough to the focus of the paper) to justify additional simulations; however, you may want to put a bit more nuance in the wording of this statement.

60. [L318] Why not show results from the 1200 UTC 27 November 2019 initialization so that the day-2 forecast aligns with the panels shown in Figs. 1, 2 and 4?

61. [L326] Does the upshear maximum in the SV plot (Fig. 12b) really very well described as being in the "cold sector" of the cyclone? The cold sector is defined based on low-level airstreams but here the plot is showing spread differences in Z250. I think that this is much more related to the growth of perturbations in the jet streak on the upshear side of the trough, which is contributing to the "digging" of the trough / meridional amplification. Could the upper-level jet-front structure not an ideal place to have rapid SV growth (e.g. Hakim 2000; JAS)? By increasing vorticity at the base of the trough this feature will indirectly impact troposphere-deep cyclogenesis, but I think it is possible that the origins of the spread are more local. (The same is true for the second trough over the eastern North Atlantic that appears to be approximately equivalent barotropic.)

62. [L327-329] The spatial separation of the SV and MU contributions is beautiful. I think that it is very understandable based on the previous comment and the fact that model physics is largely inactive in upper-level jet-fronts, other than perhaps some turbulence. The MU is focusing on the regions where the physics is active (lower-level cyclone and WCB) while the SV is picking up dynamic growth along the jet streak on the waveguide. If you agree with this assessment, it could be a useful inference to add to the text.

63. [L328] Missing closing parenthesis for figure reference.

64. [L334-336] Discussion of total precipitation seems tangential to this study (also L344-345).

65. [L346-351] This is the first time that observation location is discussed. The Obs experiment seems largely unrelated to the other experiments and should be eliminated to focus the study on the "controllable" sources of spread quantified in the other experiments.

66. [L343] Suggest changing to "… appears to yield a better depiction of uncertainty than that generated by …".

67. [L343] Remove extra "km".

68. [L343] This seems like a really important statement because it suggests that the huge computational cost of a 4 km ensemble is not justifiable from this perspective.

69. [L374] Although they can likely be inferred, neither baroclinic nor convective instabilities were demonstrated in the analysis.

70. [L382] This conclusion does not seem as direct as it ought to be. Perhaps "could" should be replaced with "should"?

71. [L382-383] This seems like a fairly weak and somewhat confusing statement on which to end the manuscript. Moist singular vectors would be implemented in the TL/AD forms of the model, and

as far as I know are quite independent of the SPPT-based model uncertainty estimate. Perhaps this discussion could instead be extended to consider the SPP-based uncertainty formulation as a look into the future ECMWF system.

72. [L392] Is a ^2 missing on the l.h.s of definition of the variance?
73. [L444] Why would the squared terms necessarily dominate, particularly if there are correlations between the constituents of the cross terms?
74. [L465-469] Providing a quantitative assessment of the relative size of each of these terms seems like it would be useful, particularly because the Residual is one of the (two) leading terms assessed in the text is key to conclusions regarding overdisperison.
75. [Appendix D] Why is a new field (500 hPa height) and season (JJA) introduced just for this appendix? I guess it might be to show the robustness of the analysis, but I think that the text on L228-232 distracts from the main message of the study. In a two-paper solution (General Comment #2), this figure and discussion could form the basis for a short subsection instead.

---

## Author Comment (AC1)

Notes on "The Cyclogenesis Butterfly: Uncertainty growth and forecast reliability during extratropical cyclogenesis"

**Overview**

This manuscript addresses the very interesting problem of flow-dependent variability in ensemble reliability.  Such an analysis is of significant practical utility because it gives ensemble designers important robust insights into their system's behaviour under identifiable meteorological conditions. Specifically for ensemble applications, such information is an essential replacement for the case study approach – although arguably conditional evaluations should also be preferred for deterministic systems.

It is clear from the breadth of the analysis that an impressive amount of work has gone into this investigation.  However, the manuscripts suffers from the lack of a clearly stated objective for the complex diagnostics employed.  As a result, the text gets mired in technical discussions rather than focusing on interpretations and discussions that support the objectives of the work and advance the main narrative of the manuscript.  Similarly, many of the novel diagnostics themselves (for example Eqs. 1 and 2) seem to be overly complex for a study that arrives at relatively straight-forward – though very useful – conclusions regarding conditional overdispersion in the ECMWF ensemble.

As described in General Comments #1 and #2 below, I think that this work is interesting and important enough to be split into two separate manuscripts.  The result will be two independent but complimentary studies that better motivate and demonstrate the utility of the proposed techniques. Such a reshaping of the investigation will also permit the introduction of more synthesis and interpretation of the results, resulting in a pair of papers that will have a larger impact on the field.

Recommendation:  Resubmit after splitting the study into two separate manuscripts. Reviewer:

Ron McTaggart-Cowan

The authors would like to thank the reviewer for the time and care that they put into reviewing this manuscript, and for their insightful comments. We have tried to answer their comments below – in particular with a better statement of the objective and less distractions. We will argue that both equations are important for this study. We also hope that this work motivates further flow-dependent evaluations, where these equations could be a useful reference.

**General Comments**

1. This manuscript presents a huge amount of material and it is clear that an awful lot of work has gone into this analysis.  However, I think that the vast array of content actually reduces the potential impact of the study.  Stronger curation of the information would focus the manuscript – and the reader – on the truly important elements of the work that lead directly to the conclusions.  One way to start improving the focus of the study will be to identify and clearly state the objective of the work.  That could effectively be done at the start of the last paragraph of the introduction.  I encourage the authors then to take a serious look at each element of content and decide whether or not it is essential to advancing the manuscript towards this

objective.  Components that do not fit into this focus should be removed and could probably form the basis for a separate submission.

The broad objective of the work is in the title of the manuscript and this was outlined further on lines 49-52. However, both reviewers made this point, so we have worked to make the statement of objectives clearer. We did give careful thought to the structure of the manuscript. In particular trying to cater for specialists and non-specialists by putting technical details in the appendices. This has also clearly failed, and we have tried to improve the structure by using sign-posts in the main text. In the light of the reviewers' comments, we have removed discussion of the connection to the Liouville equation, and now only show one of the case studies. The appendices have also been reduced.

2. In the end, I think that this is really two papers.  The first paper is about ensemble-estimated uncertainty growth rates and their relationship to cyclone intensification and/or trough amplification over the western North Atlantic.  The second paper is about documenting and identifying the source of overdispersion in the ECMWF ensemble in the North Atlantic storm track.  Although the second is clearly motivated by the first, these topics are separate enough that they would not even need to be a two-part submission:  they could be treated entirely separately.  Having two separate papers would allow for an expansion of discussions and dynamical interpretations, in addition to the introduction of important material into the main text that is currently relegated to the multiple appendices.  I really think that the prodigious amount of effort that clearly went into this analysis would be much better served by two independent submissions.

Clearly there are several parts to this study, but they really are strongly connected. Equation 1 (left hand side) is used to produce maps (and animations) of growth-rates. We focus on synoptic scales because these show the largest contribution to the overall variance growth over the first two days – as shown in the attached variance power-spectrum plot. Note that this growth involves diabatic effects and interactions between all scales – as demonstrated by equation 1 right-hand-side. These maps demonstrate that growth-rates are concentrated into specific synoptic flow situations. This motivates the investigation into cyclogenesis and the title of the paper "the cyclogenesis butterfly". The question then arises as to whether other models display the same growth-rates – we show that they are not that similar. To get a view of which might be best, we compare the models in terms of an extended spread-error equation. Following the other reviewer's advice, we now include all four models in this comparison. After showing that the ECMWF ensemble is over-spread in cyclogenesis events, we go on to investigate what might be done to improve this with a set of sensitivity experiments. These experiments suggest we could reduce the use of singular vectors. This would give us a more seamless system, and thus allow a better evaluation of model uncertainty and the key physical/dynamical processes driving the growth-rates. We have tried to strengthen the motivation and links between the section to justify keeping this as a single study.

[Figure]

3. [This comment is only directly relevant if the current submission is not split into two separate manuscripts.] Organizing the paper into 11 sections is highly unusual. Although I appreciate the use of sections and subsections as important tools for organizing content, I think that in this case there are so many sections that readers will lose the "big picture" of the manuscript's organization. To a certain extent, the excessive number of sections appears to be a symptom of a stream-of-consciousness design. Rather than presenting the work in the order that it was executed, consider reorganizing it into larger logical chunks for the reader. For example, the extremely short Data section (2) should be augmented to include the methods currently described in sections 4 and 6, and part of section 8. It seems like sections 3, 5 and 10 would be more logically grouped as a single (case study) section with appropriate subsections. Sections 7, 8 and 9 should also be considered subsections of a "full-season" analysis section. The result would be a 5-section paper: (1) introduction, (2) data and methods, (3) case studies and sensitivity tests, (4) full-season analysis and model intercomparison and (5) conclusions. I believe that such a reorganization would really help to increase the potential impact of this study on the field.

We apologise if the study comes across as a stream of consciousness. We thought hard about the structure, and it is not presented in the order it was done. With several methodologies, we did not

think it appropriate to place these all into a single section near the beginning, but rather discuss these only once they had been motivated. It also did not seem appropriate to place the sensitivity studies before the over-spread had been identified, and thus solutions need to be considered. We have tried again to improve the flow of the paper by grouping subsections and including better motivation, and we hope this is acceptable to the reviewer.

4. The two case studies appear to yield similar results. If the current document is to be revised as a single submission, one of the two case studies could be relegated to supplemental material. The main text could then claim demonstrable robustness with reference to the results shown in the supplement. If the material will be split into two independent studies (General Comment #2), then the two case studies could be retained in the first paper, along with augmented evaluation and interpretation.

We have followed the first suggested course of action here, and think this does lighten the manuscript, thank you.

5. I think that a study of finite perturbation growth rates that cites the "butterfly effect" should mention Durran and Gingrich (2014), although I understand that the perturbation scales discussed here are much larger than the near-truncation scales found to be "unimportant" in the 2014 study (indeed, you mention this in your 2018 BAMS article). Perhaps this suggests that the "cyclogenesis butterfly" is a bit of a misnomer and (although catchy) might introduce some confusion: these are **very** big butterflies.

Yes these are big butterflies. As discussed above, our justification comes from the observation that growth-rates in the first few days are largest at synoptic scales, and these growth-rates appear to be orchestrated by specific synoptic flow-types. We are not discussing the Butterfly Effect (interpreted as an intrinsic predictability limit) and we make this clearer now. We did/do cite the Durran and Gingrich paper.

6. Based on the time periods discussed in the case studies, I think that "rapid cyclone deepening" would be a better description of the uncertainty precursor than "cyclogenesis". Both cyclones form 1-2 days before the period of interest, but intensify rapidly over the Gulf Stream. I think that the distinction is important particularly in this region, where secondary cyclogenesis (i.e. the formation of a cyclonic circulation where none existed previously) is common and could easily be misunderstood to be the "butterfly". Clarifying the focus on rapid deepening of preexisting cyclones (if I am right about that) further emphasizes the fact that this study is looking at synoptic-scale uncertainty seeds, rather than the potentially mesoscale cyclone development precursors.

The attached variance power-spectrum indicated that the largest contribution to ensemble variance at initial time is from scales around 400km. Is likely that interactions between these mesoscale uncertainties, along with interactions at larger scales, do play a role in the synoptic uncertainty that develops by day 2. We have added more discussion of this.

7. Although the breakdown of the Lagrangian growth rate into "non-conservative" and "advective" components (Eq. 1) is interesting, it does not seem to have any impact on this work. The analysis appears to proceed to look at only the Lagrangian growth rate itself, i.e. the l. h. s. of Eq. 1 rather than the forcing terms. If this is true, then the focus of the manuscript can be tightened by removing Eq. 1 and associated discussions, including most of appendix B (the remainder

should be included in the augmented "Data and Methods" section, particularly if Z250 is adopted throughout as recommended in General Comment #13).

*We agree that the right hand side of equation 1 was only briefly discussed before. We have added further discussion of the right hand side of equation 1 – in particular we emphasise how it represents multi-scale interactions.*

8. The study references animations periodically. This means that readers will need to interrupt their progress to look at animations available in supplemental material. As far as I can tell, most of the relevant information could be presented as additional panels in the existing figures. For example, Figs. 2 and 3 are both single panel, but could be augmented to show other lead times to avoid the need for references to separate animations in the text.

*We now only show one case study, but include an extra panel for a different lead-time. This should avoid the need to refer to supplementary material.*

9. Differences in the ensemble perturbation techniques between the different modelling systems investigated here seem potentially important, particularly given the short lead time. The use of SV perturbations in ECMWF ENS distinguishes it from most other systems in the TIGGE database, other than perhaps JMA. A discussion of these differences (or at least their itemization in an introductory table) would be very useful.

*We will include these are cite the TIGGE archive if available there.*

10. This study looks at uncertainty (ensemble spread) growth rates from the perspective of synoptic cyclone dynamics. To make a convincing connection between the uncertainty growth and cyclone development it would be very useful to compare the former to something like the moist baroclinic growth rate (e.g. Booth et al. 2015; ASL). A high degree of correlation between the two would be good evidence of the importance of rapid cyclone deepening to spread growth in the ensemble. Even something relatively simple like comparing the time series of area-averaged (over the Gulf Stream region) ensemble growth rates and moist baroclinic growth rates (with rapid deepening events identified) would provide a really nice dynamically based assessment of the importance of cyclone development to uncertainty.

*This is an important idea. The authors are aiming to address this more fully in a future study focused on equation 1.*

11. The maximum uncertainty growth region in Fig. 2 is upshear of the trough axis, where vorticity advection is negative aloft. Why is this? In both cases (Figs. 2 and 3) the cyclone is located between the dipole in growth rates, not at all within the peak growth rate south of the trough. This is not "ahead of the base of the upper-level trough" or "preceeding cyclogenesis" (line 142). I understand that some amount of spatial smearing arises from the use of 12-h differences to compute the growth rates, but the cyclones do not even appear to move through the maximum growth rate region. So then would it be more accurate to link large spread growth rates to amplifying upper-level troughs rather than cyclones per se? For example, perhaps uncertainties in the strength of the jet streak on the upshear flank of the trough (associated with its meridional extension) are more important than the lower-level cyclone itself.

*The 'Lagrangian' growth-rate plotted does not include the advection of ensemble variance by the ensemble-mean flow. This advection is a major term and would immediately lead to Eulerian growth-rates more aligned with the cyclone. We now make this clearer in the text. It is interesting*

that the Lagrangian growth-rate highlights the upper-level trough region. We briefly discussed this in the manuscript. It is intended to be the focus of future work.

12. The bulk of discussions around the spread-error relationship appear to focus on the Spread and Residual terms of Eq. 2, leading to conclusions about overdispersion in the North Atlantic storm track. Is there no simpler way to arrive at the important conclusions of the study without going through this rather complicated derivation and analysis? The interesting flow-dependent aspect of the spread-error relationship is achieved through independent stratification (currently via cluster analysis), so I think the only thing that might be lost would be the conditional bias shown in Fig. 10i. However, this bias could be evaluated directly and shown to contribute significantly to the increased RMSE in the "counterpart" cluster without resorting to Eq. 2. The apparent ambiguity of the Residual term makes the discussions surrounding Eq. 2 quite difficult to follow and appears to make it difficult to make definitive statements about sources of problems within the ensemble. If the important message to be delivered by this work relates to the flowdependent overdispersion in the ensemble, then a simpler analysis (perhaps including regional and/or flow-stratified spread-reliability diagrams) might be a more effective vehicle. However, if the current investigation is just a showcase for the analytic technique itself then (a) that should be clarified and (b) the advantages of this technique over a simpler analysis should be emphasized.

The intention is to showcase the technique and to consider carefully the assumptions made, so that it can be used in future (flow-dependent) evaluations of ensemble forecasts – when systems may have smaller deficiencies. While the stratification would highlight the importance of bias in the non-cyclogenesis cluster, it is not immediately obvious that variance in bias would be important in the non-stratified case. If we understand correctly, we would argue that spread-reliability diagrams make the same set of assumptions.

13. The lack of PV in the TIGGE database requires the use of Z250, which appears to produce similar results (Figs. 2-5). Although I can completely understand the appeal of starting with PV in this discussion, I think that for pragmatic reasons the entire study should focus on Z250. In the Data and Methods section the rationale for this can be very clearly explained. This would only really affect current sections 4 and 5. The PV 315 diagnostics in (current) section 10 could still be used because they are separate from the growth rate discussion.

We agree that changing between PV315 and Z250 is not ideal. Nevertheless, we consider that it is important in order to discuss equation 1, where the right-hand-side requires PV. We have simplified the case studies, which should help a little. Emphasizing the lack of PV in the TIGGE archive is also considered important to motivate its inclusion at some point.

14. Why was the clustering approach (current section 8) preferred over a much simpler cyclone identification approach? It seems as though clusters 2 and 3 for both domains are lumped into the "non-cyclogenesis" category when the results from the two domains were aggregated. As such, this seems like a very complicated way to identify dates with cyclones in the western North Atlantic.

It is important to have an objective approach to classification, and we wanted the data to 'speak for itself' by showing the structures that emerged. In the end, the results are probably very similar to cyclone identification (as was alluded to in the manuscript). Many of the events in cluster 2 for region 1 do find their way into cluster 1 for region 2.

15. I am not sure grammatically why "growth-rate" is hyphenated throughout. This does not seem to be a common construction.

We will consult on this.

16. I do not believe that forecast "lead-time" is usually hyphenated. More generally, there appears to be over-hyphenation throughout the text. Please limit the use of hyphens and ensure that they are represented using hyphen characters rather than the current em-dashes.

We will consult on this.

17. Please confirm that date/time formatting conforms with WCD standards.

We will consult on this.

**Specific Comments**

18. [L45] Distinguish between the true unstable modes of the flow and the computed singular vectors (optimal tangent linear growth with limited moist physics). The note about the "linear regime" points in this direction, but it would be useful to make this distinction right off the bat.

We will do.

19. [L48] It would be useful to itemize some of these approximations here because the difference between ensemble spread growth and error growth rate is fundamental to this study.

We will do.

20. [L52-54] The punctuation of this sentence makes it difficult to follow: consider rewording.

21. [L54] Remove hyphen from "ensemble-mean".

22. [L55-56] Replace "Jetstream" with "jet stream", "wave-guide" with "waveguide", and "downstream" with "downstream".

23. [L57-72] This "outline" paragraph is overly long and complex because it strays into "abstract" territory by summarizing results. Consider shortening this paragraph by restricting its content to section descriptions only.

This has been shortened as suggested.

24. [L58] Provide a reference for TIGGE if it is to be mentioned here. Also confirm that this acronym can be used without definition in WCD, or define it.

The Swinbank et al paper was referenced at L77. We will bring this forward.

25. [L73] Suggest dropping the first two sentences of this section and including all dataset descriptions here so that the flow of the remainder of the text is not interrupted by them. As noted in General Comment #1, this section should be rewritten to include information about the datasets and methods used throughout the study.

Both reviewers have mentioned including the methods here. The problem is that it would require so much motivation up-front, or otherwise the methods would seem somewhat arbitrary. We have emphasized this argument in the text, and signpost appropriately. We hope this satisfies both reviewers.

26. [L73] I believe that "re-analysis" is more usually "reanalysis", including in Hersbach et al. (2020).

27. [L75] The forecast range of the background does not seem to be identified here or in Appendix E. It seems to be 12 h (line 136), but that should be clarified here.

28. [L77] TIGGE stands for the "THORPEX Interactive Grand Global Ensemble".

The words that TIGGE stands for have changed over the years. The "I" now stands for "international" (please see: https://www.ecmwf.int/en/research/projects/tigge)

29. [L80-83] This information would probably be better displayed as a table for easier reference in later sections.

We will do this.

30. [L84] Suggest, "These data are used …".

Done thanks

31. [Fig. 1] Are the trajectories that are used to identify the WCB region extending from -24h to +24h from the analysis valid time (i.e. these are the trajectory midpoints)? Suggest using the "red hatching" term consistently in the caption, rather than "shown in red".

32. [Fig. 1] Should the mks form of PVU be provided in the caption?

33. [L104-106] Are these the forecast experiments discussed in section 10? If so, then this is additional motivation to move that section up as a "case study" subsection.

34. [L108] Suggest "… uncertainty grow-rate estimate …" because the ensemble provides only an estimate of the true forecast uncertainty.

35. [L109] What does the "1-dimensional" restriction mean here? Would this be better identified as "scalar", or can multiple state variables be included in a 1D state vector? This is obviously important because it reappears elsewhere in the text.

36. [L114] The phrase "but with a different formulation" is too vague.

37. [L118-124] This is a very complex sentence mixes conservative and non-conservative forcings in Eq. 1. It would be more useful to split this sentence to describe the physical relevance of the terms on the r.h.s of Eq. 1 individually.

38. [L126] Should "Equation" be capitalized here? It wasn't in section 1. I do not think that the back-reference to section 1 is very useful here because the introduction did not go into much additional detail about the Liouville equation. A citation to relevant literature would be more useful here.

39. [L125-130] I think that this discussion is fine, but it does not seem to advance the main thread of the study. It could be dropped to reduce the length of the manuscript.

40. [L132-140] This information should be contained in the captions (most of it is) and/or left for supplemental material because it disrupts the flow of the main text.

41. [Fig. 2] What is the contour interval for the contours showing extreme values?

42. [Fig. 3] Should this read "Case 2"?

43. [L145] Is this a third case study being introduced? I think that discussion of the full-season perspective should be left for the subsequent section (in the reorganized paper).

44. [L145-150] These seem like "future work" suggestions that would be better left for the concluding discussion.

45. [L152-155] The 12-h forecasts from the TIGGE database are for ENS rather than EDA, is that correct? If so, then is it true that Figs. 4a and 5a look different from Figs. 2 and 3 not only because the field is different but also because the perturbations are different? If I understand the ECMWF system correctly, SV perturbations are not added within the EDA cycle, but are

added before ENS initialization.  In that case, Figs. 4a and 5a have an additional source of optimized growth.  That seems to make the comparison interesting, although it is complicated by the change in diagnostic field.  Would it not be surprising if the SV perturbations have little impact on growth rates in these cases?  Perhaps the Z250 growth rates could be shown for Figs. 2 and 3 to make this comparison possible.

The reviewer is correct that the ENS is used from TIGGE and that, for ECMWF this includes SVs (and SPPT) while the EDA does not include SVs (but does include SPPT). We did discuss this in the context of improving seamlessness in order to be better able to diagnose the (2-day) growth-rates of the model (and its uncertainty representation) from short-range forecasts. We will make this distinction clearer. Similarities between the EDA PV315 growth-rates and the ENS Z250 growth-rates suggest that there are strong growth-rates even without the explicit inclusion of SVs. While EDA Z250 growth-rates would be interesting, we feel that they would complicate the manuscript unnecessarily.

46. [L155-156] So are these case studies (particularly Fig. 4) not representative of the general behaviour of these models?  If so, perhaps another case study should be chosen for this comparison.

The main point being made is that they disagree in terms of growth-rates. We mentioned the slightly better agreement for some other cases for completeness. The full range of agreement/disagreement can be seen in the animations that we will provide with the revised paper.

47. [L172-174] The source of Eq. 2 (appendix C) should be cited at the beginning of this discussion.

We will bring this forward in the text.

48. [L181-182] I have a hard time understanding a lot of this discussion and how it relates to Fig. 6. It would be great to label the lines in Fig. 6 with the names of the terms in Eq. 2 that they relate to.  The lines seem to be more directly related to the discussion in Appendix C, so perhaps Fig. 6 would be more appropriate in the appendix.

We did experiment with this, but it is difficult to label all the lines in Fig. 6. For example, there are two lines which contribute to the forecast variance term, and two which contribute to the analysis variance term. Lines are also conditional on the truth. We will try to improve the text.

49. [L192] Does this "main additional term in the Residual" refer to Eq. C5?  If so, it would be useful to cite that equation here.

50. [Fig. 7] The change in colour scale range for panels (n) and (o) make comparison of the plots on the bottom row difficult.  With the current plotting scheme, it looks like the difference in residual is almost entirely explicable by the difference in spread, but that is not really the case (is it)?  The contour intervals for values beyond the standard colour bars should be noted in the caption.

51. [L219-223] It is challenging to follow this discussion because of two forward-references to a description of the variance of forecast biases.  It seems like that aspect of the discussion should be introduced before this text appears.  In fact, it is not clear what discussion the forwardreferences here are actually describing (the section 9 discussion seems to take an understanding of the forecast bias variance's impact on the Residual for granted).

52. [L233] It was not obvious that this is a "key question", so hopefully a clear statement of the study's objective(s) in the introduction will help to make that link more direct.

53. [L233-235] Does this "either-or" statement arise from the form of the Residual term (Eq. C5)? If so, then it seems like it would be useful to put this equation in the main text, hopefully as part of a discussion on the meaning of "variance in forecast bias", which I think might be related to the "difficulties" proposed here (?).

54. [L242-243] This region is quite complex: why would three clusters necessarily "provide sufficient degrees of freedom"? The optimal number of clusters is difficult to determine, but usually dropoffs in quantities like the AIC or BIC serve as some sort of semi-quantifiable justification for the number of clusters.

55. [L256-258] This is the only discussion of the uncertainty growth rate in this section, and it does not seem to lead to any particular conclusion. Is there a good reason to include it here and in the Fig. 8 and 9 plots? (It does not seem to be discussed in the subsequent section either.)

56. [L294] The phrase "almost the entire over-spread" seems like a bit of an overstatement. It is probably more defensible in terms of variance, but could perhaps be softened to "much of the overdispersion" or similar.

57. [L297-301] I am afraid that I do not fully understand this discussion. How would the stratification of the groups (cyclone vs. non-cyclone) be done differently with multiple seasons or an independent assessment? Could this "regression to the mean" alternatively be considered a sampling bias?

58. [L302.5] Consider simplifying the section title to "Sensitivity experiments to quantify uncertainty sources".

59. [L315] I understand that resource constraints likely make additional tests difficult or impossible, but is it not conceivable that the ordering of MU and 4K is important? Systematic changes in the physics tendencies should be expected between 18 km and 4 km grid spacing (for example as more turbulent fluxes are represented by the dynamics), which will impact SPPT directly. This might mean that the impact of switching MU on and off at 4 km is different from what is observed in the 18 km configuration. I do not think that this is a big enough deal (or close enough to the focus of the paper) to justify additional simulations; however, you may want to put a bit more nuance in the wording of this statement.

60. [L318] Why not show results from the 1200 UTC 27 November 2019 initialization so that the day-2 forecast aligns with the panels shown in Figs. 1, 2 and 4?

61. [L326] Does the upshear maximum in the SV plot (Fig. 12b) really very well described as being in the "cold sector" of the cyclone? The cold sector is defined based on low-level airstreams but here the plot is showing spread differences in Z250. I think that this is much more related to the growth of perturbations in the jet streak on the upshear side of the trough, which is contributing to the "digging" of the trough / meridional amplification. Could the upper-level jet-front structure not an ideal place to have rapid SV growth (e.g. Hakim 2000; JAS)? By increasing vorticity at the base of the trough this feature will indirectly impact troposphere-deep cyclogenesis, but I think it is possible that the origins of the spread are more local. (The same is true for the second trough over the eastern North Atlantic that appears to be approximately equivalent barotropic.)

62. [L327-329] The spatial separation of the SV and MU contributions is beautiful. I think that it is very understandable based on the previous comment and the fact that model physics is largely inactive in upper-level jet-fronts, other than perhaps some turbulence. The MU is focusing on

the regions where the physics is active (lower-level cyclone and WCB) while the SV is picking up dynamic growth along the jet streak on the waveguide. If you agree with this assessment, it could be a useful inference to add to the text.

63. [L328] Missing closing parenthesis for figure reference.

64. [L334-336] Discussion of total precipitation seems tangential to this study (also L344-345).

65. [L346-351] This is the first time that observation location is discussed. The Obs experiment seems largely unrelated to the other experiments and should be eliminated to focus the study on the "controllable" sources of spread quantified in the other experiments.

66. [L343] Suggest changing to "… appears to yield a better depiction of uncertainty than that generated by …".

67. [L343] Remove extra "km".

68. [L343] This seems like a really important statement because it suggests that the huge computational cost of a 4 km ensemble is not justifiable from this perspective.

69. [L374] Although they can likely be inferred, neither baroclinic nor convective instabilities were demonstrated in the analysis.

70. [L382] This conclusion does not seem as direct as it ought to be. Perhaps "could" should be replaced with "should"?

71. [L382-383] This seems like a fairly weak and somewhat confusing statement on which to end the manuscript. Moist singular vectors would be implemented in the TL/AD forms of the model, and as far as I know are quite independent of the SPPT-based model uncertainty estimate. Perhaps this discussion could instead be extended to consider the SPP-based uncertainty formulation as a look into the future ECMWF system.

72. [L392] Is a ^2 missing on the l.h.s of definition of the variance?

73. [L444] Why would the squared terms necessarily dominate, particularly if there are correlations between the constituents of the cross terms?

74. [L465-469] Providing a quantitative assessment of the relative size of each of these terms seems like it would be useful, particularly because the Residual is one of the (two) leading terms assessed in the text is key to conclusions regarding overdisperison.

75. [Appendix D] Why is a new field (500 hPa height) and season (JJA) introduced just for this appendix? I guess it might be to show the robustness of the analysis, but I think that the text on L228-232 distracts from the main message of the study. In a two-paper solution (General Comment #2), this figure and discussion could form the basis for a short subsection instead.

---

## Author Comment (AC2)

Review of "The Cyclogenesis Butterfly: Uncertainty growth and forecast reliability during extratropical cyclogenesis" by Mark John Rodwell and Heini Wernli

The paper investigates ensemble forecast reliability at 48h lead time, mainly in the ECMWF system with some comparison to other centers. To do so, a new spread-error budget is derived. It is found that the ECMWF ensemble is overspread in stormtracks in the winter season and it is argued that this is related to cyclogenesis events. In my opinion the core topic of this work is interesting and worth being published since it contradicts the intuitive expectation that cyclogenesis is associated with bad forecasts and low predictability. However, the paper requires a substantial revision. It is too long, difficult to read and not well structured. It spends a lot of time on tangent discussions and detailed case analyses, which I find distracting and misleading. Especially the discussion of the butterfly effect is imprecise, irrelevant and in part incorrect. On the other hand, topics more directly related to reliability and ensemble forecasting systems are very little discussed, if at all.

We appreciate the reviewer's comments and their time taken to review this manuscript. We attempt to address their specific comments below.

Major comments:

"The cyclogenesis butterfly"

This term, given already in the title, is never properly defined. It is introduced by phrases like "here we think of it as...". I am still not clear what is actually meant by this. The paper investigates reliability, which is an aspect of practical predictability, while the "real" butterfly effect refers to the existence of an intrinsic predictability limit caused by scale interaction in a multi-scale system (see Lorenz, 1969 and Palmer etal, 2014). Current weather prediction system are started from initial condition uncertainties that are much larger than butterflies and are on average far away from hitting the intrinsic limit (e.g. Zhang etal, 2019). The existence of singular vectors are not a manifestation of the butterfly effect, since they are still consistent with infinite predictability due to their constant growth rate. Judt, 2018 (Fig. 8b, day 0-2) for example has demonstrated the extreme increase in the error growth rate if the atmosphere really is perturbed with "butterflies" only. I am however not recommending to discuss the butterfly effect more precisely in this paper but rather to remove this discussion and to focus on much more relevant aspects with respect to (practical) reliability, like the long-standing underdispersive problem of ensemble forecasts and various methods that have been used to mitigate it (e.g. EDA, singular vectors, breading vectors, SPPT, SPP, etc.). It is probably a shortcoming in one or several of those methods that leads to the reliability problem that the paper investigates.

Sorry for the confusion here. We are not talking about the "real butterfly effect", intrinsic predictability limits, or Lorenz, 1969. To be fair, we did not mention the butterfly effect, but rather stated that our butterflies were defined as "local flow configurations where the chaotic and exponential growth–rate of uncertainty is particularly strong" (line 32). We also discussed SVs as indicating that divergence of trajectories within state–space is not uniform over the attractor (line 31), rather than as any manifestation of the butterfly effect. Our reference to Lorenz was to his 1963 paper – discussing sensitivity to initial conditions but not any intrinsic limit to predictability. Our use of the term "Cyclogenesis butterfly" is, rather, an attempt to encourage more flow-dependent thinking in the evaluation and development of forecast models. However, the potential for confusion is recognised, and we now ensure the reader does not get the wrong idea from the outset.

The attached figure shows power spectra at day 0 and day 2 for the ECMWF ensemble. The maximum initial (EDA) variance contribution is from waves around 400km, while the maximum D+2 variance contribution is from waves around 1500km – i.e. synoptic scales. Notice that the initial variance is saturated at scales smaller than about 100km. This plot motivates/justifies our interest in the growth of variance at the synoptic scales to D+2 – due to physics and dynamical interactions at all scales.

[Figure]

Incidentally, the Palmer et al paper uses the current lead-author's previous example to speculate that intrinsic predictability limits might be longer due to the potential confinement of error-growth to intermittent flow regimes. While we do not discuss intrinsic predictability limits, the idea that certain flow-types can organise multi-scale interactions and focus error-growth is very much in the spirit of the current study.

While lack of reliability might be explainable by shortcomings in the mitigation methods mentioned by the reviewer, this is not obviously the case. For example, sensitivity of parametrized convection to uncertainties in the resolved flow might be important.

To improve brevity, we have removed discussion of links to the Liouville equation. The thought was that the use of model uncertainty represents a route by which NWP could converge on the true dispersion-rates on the real-world attractor but, reflecting on the reviewer's comments, it is probably true to say that NWP will never be able to say anything definitive about the real butterfly effect, as it pertains to the real world.

Reliability in a larger context

The specific overspread that is found over the Northern Atlantic stormtrack in winter (Fig. 7) has not been put into a wider context. If the system is reliable on average there must be compensating underspread somewhere, e.g. over the continents, outside the midlatitudes or in the other seasons. This should also be discussed, as well as the question if the system really is reliable on average at the considered 48h lead time. Some information can be found in appendix D, but I think this discussion should be central in the paper. Furthermore I am wondering what the downstream consequences of the stormtrack-overspread are. Does the overspread persist beyond the end of the stormtrack into the continent in a lagrangian sense, e.g. is the 5 day forecast for Europe in the winter season also overspread? Finally, what is the relation to forecast busts? According to Lillo and Parsons, 2017, East coast cyclogenesis has the potential to generate particularly bad forecasts over Europe. This kind of contradicts the (average) results from this paper. Possibly one season of data is not enough to investigate this but some discussion here would be helpful.

There is some compensation elsewhere. The authors' previous paper on flow-dependent reliability shows good agreement in the annual-mean hemispheric-mean sense between RMSE and ensemble standard deviations. The current work (e.g. Figure D1) demonstrates that this agreement may not have been so good (at day 2) if bias and analysis uncertainty had been factored-in. The lead author has also found that things have deteriorated (in the stormtracks) since then. For T850, some of the compensation in the operationally calculated scores actually occurs under Tibet, and this is screened-out in the diagnostics developed here. These details seemed to the authors to be of little long-term practical use to readers on this study. At day 5, the over-spread is still evident but general interactions and the loss of any continued SV contribution make this less clear. The emphasis of this study is on the short timescales because these are the only timescales where agreement between ensemble members is sufficient to be able to make meaningful calculations of their dispersion rates. We do not see any contradiction with the Lillo and Parsons paper – cyclogenesis clearly results in large spread and deterministic forecast busts. The Lillo and Parsons paper is a major reason why we investigated cyclogenesis.

More use of TIGGE

While for the case studies 4 centers have been compared, the spread-error budget comparison is only done between the ECMWF and the UKMO system and finally the clustering analysis is only done for the ECMWF system. A reason for this is not given. I think the paper may miss an opportunity here to investigate possible reasons for the (ECMWF) overspread since the different centers use different methods to generate their ensemble. Hence I recommend to include more centers throughout the paper, particularly in the clustering analysis to see if the cyclogenesis overspread is a more general problem or specific to ECMWF.

We avoided a detailed 'beauty contest' because this can be problematic in manuscripts. For example, making sure that the details of the ensemble initialisation over the period of interest are documented correctly, and checking whether there are other factors to consider. Nevertheless, on the reviewer's advice, the variance budget has been calculated for the four models discussed, and this figure will replace the previous one. Applying the clustering analysis to all the models would include a lot more plots and require even more discussion. The original work was, of course, aimed at evaluating and understanding uncertainty growth-rates in the ECMWF model.

More focus

The paper spends a lot of time with a detailed discussion of two cases which in my point of view gives little insight. Furthermore, the paper oscillates between analyzing the cases and the entire winter and also between theta- and pressure-level analysis or squared and non-squared metrics, which I find confusing. With Eq. 1 the paper introduces a rather sophisticated diagnostic which later

is not used at all. I think this is not a good use of the time of potential readers. The information given in this paper is distributed over 8 sections, 5 appendices, 17 figures and additional supplementary material. I suggest the authors should consider condensing the paper to the essential parts and keeping analysis and methods consistent across the paper.

Equation 1 is central to the study. Plotting it, demonstrates that strong growth-rates are confined to particular flow features. The right-hand-side shows that the growth-rate (for the PV variable) represents the effects of uncertainties in diabatic processes and in non-linear dynamical scale-interactions. This is now revisited in the conclusions. It is hoped that equation 1 will feed into subsequent work examining these aspects in more detail. For the growth-rate plots, only one case is now presented – with one PV growth-rate example for ECMWF (which relates directly to the equation) and one comparison of TIGGE model growth-rates (which highlight similarities and differences). The sensitivity study is considered important as it highlights the relative roles of model uncertainty and singular vectors, and points to possibilities for future development. The lack of a clear signal associated with assimilation of local observations is consistent with the fact that the variance at small scales is already saturated in the initial conditions – the skill is coming from the larger scales which are adequately constrained in the 4D Var without the local observations. This is now better discussed.

Specific comments:

L1:

This statement is incorrect (see major comment about the butterfly effect and comment below).

We know state that "Global numerical weather prediction is often limited by particular flow features which are associated with pronounced uncertainty growth–rates".

L21:

The Liouville equation as formulated in Ehrendorfer, 1994 assumes that the propagation operator is known and constant. Hence it cannot describe growth due to model uncertainty.

As discussed above, we have removed reference to the Liouville equation.

L26:

I would not say that EDA represents model uncertainty. The model uncertainty is rather part of the assimilation process to generate the initial condition ensemble.

Yes, we are saying that model uncertainty representation is included in the EDA system. The missing piece of information seems to be that we do not mention we are referring to the ECMWF EDA system. We now say "… in particular, at ECMWF, the ensemble data assimilation system (Isaksen et al., 2010) aims to represent flow–dependent error covariances in the background or "first–guess" (Bonavita et al., 2016) and in the observations (Geer et al., 2018), as well as model uncertainty (MU; Buizza et al., 1999) and a model grid's lack of representativity of point observations (Janjić et al., 2018)".

L31:

This statement is incorrect. Lorenz-type butterflies, i.e. small-scale and small-amplitude perturbations limit predictability via scale interactions and not only due to chaos and strong sensitivity to initial conditions (see Lorenz, 1969 and especially Palmer etal., 2014). Hence the constantly growing singular vectors do not represent this "real" butterfly effect. Furthermore, current errors and uncertainties in forecasting system are neither small in scale nor small in

amplitude and cannot be regarded as butterflies. If they were this would mean that current systems operate already now at the intrinsic limit, which is not true (e.g. Zhang etal. 2019). I agree that in some situations error growth in current forecasts is worse than average but if this might be related to the butterfly effect in rare cases is an open question.

We have changed the text "predictability is often limited by the Lorenz–type butterflies in the flow (Lorenz, 1972). Here we think of these as local" to "prediction is often limited by specific"

L51:

I am not sure if I understand correctly what you mean with "cyclogenesis butterfly". The term is never clearly defined.

We now make this clearer, as discussed above.

L51:

"The key question is..."

This is a big gap in the line of argument and comes as a surprise to me. Please consider rewriting the introduction to focus on this question and the importance and flow-dependence of reliability and the need to extend the "spread-error" relationship.

The question is in the title. The re-wording of the introduction also makes this clearer now.

L57:

The paper outline contains to many details in my opinion. Some should have been mentioned and discussed in the introduction, some are results.

We have shortened the outline to remove details and results.

Sec. 2, Data:

Since an entire section is dedicated to describe the data I would prefer that all the relevant details are given here rather than being distributed over the rest of the paper and the appendix. With respect to the other centers, only resolution and ensemble size are given but potentially interesting differences in the ensemble design are not mentioned and later not investigated (see major comment above).

L74, caption Fig. 2:

What do you mean by "background ensemble/forecast"?

L90:

The arguments given about the case selection are very vague. How are they related to the key question? Are these cases in which the forecast was particularly unreliable? Or in which the cyclogenesis was very rapid?

Fig. 2-4:

It is unclear, which forecast lead time you show and why you chose to focus on this particular forecast lead time.

L109:

I don't understand what you mean be 1-dimensional state-measure. Substitute 4-dim atmospheric field?

Eq. 1:

sigma_hat is now the standard deviation of the PV, right? Use P_hat instead of sigma? The hat is not explained, same meaning as in eq. 2?

I suggest to add an index i to P, P' and NC to indicate that these are quantities from individual members.

L120-L130:

Needs more introduction and explanation. However, this diagnostic is not used in the paper anyway. Consider removing it (see below).

L142:

"often preceding cyclogenesis", "occur within strongly precipitating WCBs":

These statements are rather vague and are either obvious or seem speculative. Do you mean that the the growth rate is correlated with the amount of precipitation in the cyclone? And with cyclogenesis do you mean a depending of the trough where the growth rate peak occurs or does it lead to a cyclogenesis downstream?

L145:

"Further investigation..."

I find the following statements distracting. But more importantly, the reader might have invested some time to understand eq. 1, the relevant papers and the appendix only to find out now that you are not investigating the right hand side of eq. 1 at all and leave this for future work. Hence I recommend to remove section 4 and appendix A and just state here that you are plotting a lagrangian growth rate.

L152:

It is very confusing that you switch now to geopotential growth rates. I understand that PV is not in TIGGE. But what is the point of showing the PV growth rates first, especially since you did not explore the right-hand side of eq. 1, which for me is the main purpose of using PV? I suggest to stick with Z250 then and to omit the PV-plots.

L157:

"... are very evident"

I actually was surprised to see how bad the agreement is. Not much is said however about where these discrepancies come from, L160 makes a very general statement.

L161:

"It would be useful..."

Please clarify what you mean. Uncertainty growth rates cannot be close to the truth since the truth is not uncertain.

L167:

You state the essential information as e.g. in brackets. I suggest to change that and maybe write down an equation. Also I suggest to be more precise what average means (case average, area average, ensemble average).

L174 (also L168):

I suggest to replace "ensemble forecast start times" with "(large number of) cases". Also please explain the symbols first and discuss the visualization afterwards. The notation is inconsistent with eq. 1, maybe express the ensemble mean with <..>.

L188:

Is mu_A equal to the truth? And is mu_F also equal to the truth? I suggest to not discuss this in the figure caption.

L196:

I find this square-root operation just for "more understandable units" confusing, especially since it introduces the complication with the residual and you admit that small contributions look larger than they actually are. Moreover, in the supplementary figure you switch back to square units. I suggest to keep the squares in every plot.

Sec. 7:

I wonder why you switched from comparing 4 centers to now only 2. Is there a reason for that? And why did you choose to compare with the UKMO?

L233:

In Rodwell etal, 2018 you showed (Fig. 1) that the (traditional) spread-error relation is perfectly matched for the Northern Hemisphere at any forecast lead time over an entire year (2014). Hence (if this is still true, is it?) the overspread you now show for the stormtracks in the winter must be compensated by an underspread at some other location or some other season. It would be interesting to investigate this (see major comment above).

L238:

I suggest to explain the K-mean clustering method with a couple of sentences.

L245:

Why do you weight with the root? Isn't the grid cell area scaling with cos(lat)?

L264:

Does this mean you are combining the cluster1 cases from both clustering areas? Could you further justify this approach? The shift of the region doesn't seem that large. Would one clustering analysis based on a combined region lead to similar results? What about separating clusters by the surface pressure tendency in the region? Wouldn't this be a simpler method more directly related to cyclogenesis?

L293:

To me this statement seems a bit exaggerated. I would say that the overspread is reduced in the cyclogenesis composite. Also if I read the colors correctly, the residual difference does not reach statistical significance. Why is the overspread enhanced in the counterpart over the central/east Atlantic. Is it because cyclogenesis is shifted downstream in the counterpart cases? Is this spread reduction in cyclogenesis events also happening at other centers (see major comment above)?

L296:

I did not understand this paragraph.

L313:

I notice you leave the convection scheme on at 4km resolution. Could you explain why? Usually only shallow convection is used at such high resolutions (e.g. Judt, 2018).

L324:

I don't understand this sentence.

L333:

"attempts to resolve". This is misleading since the resolution is still 18km, right?

L338:

I am not sure about the relevance of this increased spread. It could just be a consequence of slightly displaced and explicitly resolved updrafts. Would there also be any enhanced spread in e.g. the precipitation averaged over the front?

L342:

Possibly there are now explicitly simulated updrafts which are slightly displaced among the ensemble members and generate grid-pointwise spread. Again I am not sure how relevant this is. I don't think this is the kind of uncertainty SPPT was designed to account for. So I am not surprised to see less effect from SPPT in this region.

L379:

If a reduction of singular vectors would make forecasts more consistent then why are they still used? I suspect they do show a benefit at a different location, flow regime, lead time, etc. I think you should discuss these aspects in more detail and also possible alternatives (e.g. inflating SPPT or EDA, using SPP, higher resolution, etc). See also major comment above.

L381:

This would only make sense if the observed increased spread with resolved convection does not mainly result from rather small displacements of individual updrafts. But this has not been investigated (see related comments above).

Minor comments:

Fig. 1:

The figure is hard to read and evaluate. I suggest to use a color for the >2PVU regions and to omit the red hatching, since it is kind of obvious and does not add any extra information. As labels of the panels I suggest "Case 1", "Case 1, +24h" or something like this for easier identification.

Fig. 2 and others:

There is a lot of doubling between the figure caption and the text. I suggest to not repeat details in the text that are already included in the caption.

L132:

"Single frame of animation" is not a good description of the plot.

Fig. 3 caption:

I think you meant case 2.

L175:

Better "sampling from a population"?

L178:

Any reason why you call this "departure"? It is just a difference, isn't it?

L180:

"number of forecasts" is ambiguous. I suggest "cases".

L182:

I suggest to remove the 2-superscript (looks like a footnote). It is clear from the context that you are considering squared quantities.

L191:

Remove (.

Fig. 7 (and others):

The color bars are misleading to me. First I would appreciate if the color bars in panels a-j and k-o were identical. The main point of the figure is the comparison and identical color bars will help with that. Also it is misleading to color small positive value with saturated dark colors (e.g. panel c, it looks like a massive AnUnc). I suggest to use reddish colors for positive values, blueish colors for negative values and gray/neutral around zero (e.g. like you did in panel d).

Fig. 8:

The green geopotential lines are very hard to see. Please make them more prominent.

L273:

"Head of the stormtrack". This term is not clear to me. From the context I guess you mean the start/beginning (west).

Fig. 11:

I find the arrows more confusing than helpful. Also: CP->DP

Fig. 12:

I suggest to also revise the color bar. Panel a) shows a positive variable and should use neutral to reddish colors. Panels b)-f) should have the same color bar since this makes it much easier to assess the individual contributions. I cannot really distinguish gray from black contours.

L328:

) missing.

References:

Edward N. Lorenz (1969) The predictability of a flow which possesses manyscales of motion, Tellus, 21:3, 289-307

T. Palmer et al, 2014: The real butterfly effect. Nonlinearity 27 R123

Judt, F. (2018). Insights into Atmospheric Predictability through Global Convection-Permitting Model Simulations, Journal of the Atmospheric Sciences, 75(5), 1477-1497.

Zhang, F. et al, 2019: What is the predictability limit of midlatitude weather?. Journal of the Atmospheric Sciences, 76(4), 1077-1091.

Citation: https://doi.org/10.5194/wcd-2022-6-RC2

---

## Author Comment (AC3)

Notes on "The Cyclogenesis Butterfly: Uncertainty growth and forecast reliability during extratropical cyclogenesis"

**Overview**

This manuscript addresses the very interesting problem of flow-dependent variability in ensemble reliability. Such an analysis is of significant practical utility because it gives ensemble designers important robust insights into their system's behaviour under identifiable meteorological conditions. Specifically for ensemble applications, such information is an essential replacement for the case study approach – although arguably conditional evaluations should also be preferred for deterministic systems.

It is clear from the breadth of the analysis that an impressive amount of work has gone into this investigation. However, the manuscripts suffers from the lack of a clearly stated objective for the complex diagnostics employed. As a result, the text gets mired in technical discussions rather than focusing on interpretations and discussions that support the objectives of the work and advance the main narrative of the manuscript. Similarly, many of the novel diagnostics themselves (for example Eqs. 1 and 2) seem to be overly complex for a study that arrives at relatively straight-forward – though very useful – conclusions regarding conditional overdispersion in the ECMWF ensemble.

As described in General Comments #1 and #2 below, I think that this work is interesting and important enough to be split into two separate manuscripts. The result will be two independent but complimentary studies that better motivate and demonstrate the utility of the proposed techniques. Such a reshaping of the investigation will also permit the introduction of more synthesis and interpretation of the results, resulting in a pair of papers that will have a larger impact on the field.

Recommendation: Resubmit after splitting the study into two separate manuscripts. Reviewer:

Ron McTaggart-Cowan

The authors would like to thank the reviewer for the time and care that they put into reviewing this manuscript, and for their insightful comments. We have tried to answer their comments below – in particular with a better statement of the objective and less distractions. We will argue that both equations are important for this study. We also hope that this work motivates further flow-dependent evaluations, where these equations could be a useful reference. In our replies below, all line numbers refer to the originally submitted manuscript

**General Comments**

1. This manuscript presents a huge amount of material and it is clear that an awful lot of work has gone into this analysis. However, I think that the vast array of content actually reduces the potential impact of the study. Stronger curation of the information would focus the manuscript – and the reader – on the truly important elements of the work that lead directly to the conclusions. One way to start improving the focus of the study will be to identify and clearly state the objective of the work. That could effectively be done at the start of the last paragraph of the introduction. I encourage the authors then to take a serious look at each element of content and decide whether or not it is essential to advancing the manuscript towards this objective. Components that do not fit into this focus should be removed and could probably form the basis for a separate submission.

The broad objective of the work is in the title of the manuscript and this was outlined further on lines 49-52. However, both reviewers made this point, so we have worked to make the statement of objectives clearer. We did give careful thought to the structure of the manuscript. In particular we did try to cater for specialists and non-specialists by putting technical details in the appendices. This has also clearly failed, and we now have tried to improve the structure by using sign-posts in the main text. In the light of the reviewers' comments, we have removed discussion of the connection to the Liouville equation, and now only show one of the case studies. The appendices have also been reduced.

2. In the end, I think that this is really two papers. The first paper is about ensemble-estimated uncertainty growth rates and their relationship to cyclone intensification and/or trough amplification over the western North Atlantic. The second paper is about documenting and identifying the source of overdispersion in the ECMWF ensemble in the North Atlantic storm track. Although the second is clearly motivated by the first, these topics are separate enough that they would not even need to be a two-part submission: they could be treated entirely separately. Having two separate papers would allow for an expansion of discussions and dynamical interpretations, in addition to the introduction of important material into the main text that is currently relegated to the multiple appendices. I really think that the prodigious amount of effort that clearly went into this analysis would be much better served by two independent submissions.

Clearly there are several parts to this study, but they really are strongly connected. Equation 1 (left hand side) is used to produce maps (and animations) of growth-rates. We focus on synoptic scales because these show the largest contribution to the overall variance growth over the first two days – as shown in the attached variance power-spectrum plot. Note that this growth involves diabatic effects and interactions between all scales – as demonstrated by equation 1 right-hand-side. These maps demonstrate that growth-rates are concentrated into specific synoptic flow situations. This motivates the investigation into cyclogenesis and the title of the paper "the cyclogenesis butterfly". The question then arises as to whether other models display the same growth rates – we show that they are not that similar. To get a view of which might be best, we compare the models in terms of an extended spread-error equation. Following the other reviewer's advice, we now include all four models in this comparison. After showing that the ECMWF ensemble is over-spread in cyclogenesis events, we go on to investigate what might be done to improve this with a set of sensitivity experiments. These experiments suggest we could reduce the use of singular vectors. This would give us a more seamless system, and thus allow a better evaluation of model uncertainty and the key physical/dynamical processes driving the growth rates. We have tried to strengthen the motivation and links between the sections to justify keeping this as a single study.

[Figure]

3. [This comment is only directly relevant if the current submission is not split into two separate manuscripts.] Organizing the paper into 11 sections is highly unusual. Although I appreciate the use of sections and subsections as important tools for organizing content, I think that in this case there are so many sections that readers will lose the "big picture" of the manuscript's organization. To a certain extent, the excessive number of sections appears to be a symptom of a stream-of-consciousness design. Rather than presenting the work in the order that it was executed, consider reorganizing it into larger logical chunks for the reader. For example, the extremely short Data section (2) should be augmented to include the methods currently described in sections 4 and 6, and part of section 8. It seems like sections 3, 5 and 10 would be more logically grouped as a single (case study) section with appropriate subsections. Sections 7, 8 and 9 should also be considered subsections of a "full-season" analysis section. The result would be a 5-section paper: (1) introduction, (2) data and methods, (3) case studies and sensitivity tests, (4) full-season analysis and model intercomparison and (5) conclusions. I believe that such a reorganization would really help to increase the potential impact of this study on the field.

We apologise if the study comes across as a stream of consciousness. We thought hard about the structure, and it is not presented in the order it was done. With several methodologies used in the

study, we did not think it appropriate to place these all into a single section near the beginning, but rather discuss these only once they had been motivated. It is considered important to introduce the case studies at the beginning (now only one shown) because these give concrete examples of what we go on to aggregate. However, it does not seem appropriate to place the sensitivity studies before the over-spread had been identified, because only then do solutions to the over-spread need to be considered. Furthermore, the sensitivity studies point to future avenues of research. We have tried again to improve the flow of the paper by grouping subsections and including better motivation, and we hope this is acceptable to the reviewer.

4. The two case studies appear to yield similar results. If the current document is to be revised as a single submission, one of the two case studies could be relegated to supplemental material. The main text could then claim demonstrable robustness with reference to the results shown in the supplement. If the material will be split into two independent studies (General Comment #2), then the two case studies could be retained in the first paper, along with augmented evaluation and interpretation.

We have followed the first suggested course of action here, and think this does lighten the manuscript, thank you.

5. I think that a study of finite perturbation growth rates that cites the "butterfly effect" should mention Durran and Gingrich (2014), although I understand that the perturbation scales discussed here are much larger than the near-truncation scales found to be "unimportant" in the 2014 study (indeed, you mention this in your 2018 BAMS article). Perhaps this suggests that the "cyclogenesis butterfly" is a bit of a misnomer and (although catchy) might introduce some confusion: these are **very** big butterflies.

Yes, these are big butterflies. As discussed above, our justification comes from the observation that growth rates in the first few days are largest at synoptic scales, and these growth rates appear to be orchestrated by specific synoptic flow-types. We are not discussing the Butterfly Effect (interpreted as an intrinsic predictability limit) and we make this clearer now. We did/do cite the Durran and Gingrich paper.

6. Based on the time periods discussed in the case studies, I think that "rapid cyclone deepening" would be a better description of the uncertainty precursor than "cyclogenesis". Both cyclones form 1-2 days before the period of interest, but intensify rapidly over the Gulf Stream. I think that the distinction is important particularly in this region, where secondary cyclogenesis (i.e. the formation of a cyclonic circulation where none existed previously) is common and could easily be misunderstood to be the "butterfly". Clarifying the focus on rapid deepening of preexisting cyclones (if I am right about that) further emphasizes the fact that this study is looking at synoptic-scale uncertainty seeds, rather than the potentially mesoscale cyclone development precursors.

The variance power-spectrum shown in the figure above indicated that the largest contribution to ensemble variance at initial time is from scales around 400 km. It is likely that interactions between these mesoscale uncertainties, along with interactions at larger scales, do play a role in the synoptic uncertainty that develops by day 2. We have added more discussion of this.

7. Although the breakdown of the Lagrangian growth rate into "non-conservative" and "advective" components (Eq. 1) is interesting, it does not seem to have any impact on this work. The analysis appears to proceed to look at only the Lagrangian growth rate itself, i.e. the l. h. s. of Eq.

1 rather than the forcing terms. If this is true, then the focus of the manuscript can be tightened by removing Eq. 1 and associated discussions, including most of appendix B (the remainder should be included in the augmented "Data and Methods" section, particularly if Z250 is adopted throughout as recommended in General Comment #13).

We agree that the right-hand side of equation 1 was only briefly discussed before. We have added further discussion of the right-hand side of equation 1 – in particular we emphasise how it represents multi-scale interactions. The impact of multiscale interactions is evident in the blue shaded region of the figure shown above, where the representation of smaller scales in the 4 km experiments leads to increased ensemble variance at scales already represented in the 18 km model.

8. The study references animations periodically. This means that readers will need to interrupt their progress to look at animations available in supplemental material. As far as I can tell, most of the relevant information could be presented as additional panels in the existing figures. For example, Figs. 2 and 3 are both single panel, but could be augmented to show other lead times to avoid the need for references to separate animations in the text.

We now only show one case study, but include an extra panel for a different lead time. This should avoid the need to refer to supplementary material.

9. Differences in the ensemble perturbation techniques between the different modelling systems investigated here seem potentially important, particularly given the short lead time. The use of SV perturbations in ECMWF ENS distinguishes it from most other systems in the TIGGE database, other than perhaps JMA. A discussion of these differences (or at least their itemization in an introductory table) would be very useful.

We will include these and cite the TIGGE archive documentation.

10. This study looks at uncertainty (ensemble spread) growth rates from the perspective of synoptic cyclone dynamics. To make a convincing connection between the uncertainty growth and cyclone development it would be very useful to compare the former to something like the moist baroclinic growth rate (e.g. Booth et al. 2015; ASL). A high degree of correlation between the two would be good evidence of the importance of rapid cyclone deepening to spread growth in the ensemble. Even something relatively simple like comparing the time series of area-averaged (over the Gulf Stream region) ensemble growth rates and moist baroclinic growth rates (with rapid deepening events identified) would provide a really nice dynamically based assessment of the importance of cyclone development to uncertainty.

This is an important idea. The authors are aiming to address this more fully in a future study focused on equation 1, as alluded to in lines 145-149 of the original manuscript.

11. The maximum uncertainty growth region in Fig. 2 is upshear of the trough axis, where vorticity advection is negative aloft. Why is this? In both cases (Figs. 2 and 3) the cyclone is located between the dipole in growth rates, not at all within the peak growth rate south of the trough. This is not "ahead of the base of the upper-level trough" or "preceeding cyclogenesis" (line 142). I understand that some amount of spatial smearing arises from the use of 12-h differences to compute the growth rates, but the cyclones do not even appear to move through the maximum growth rate region. So then would it be more accurate to link large spread growth rates to amplifying upper-level troughs rather than cyclones per se? For example, perhaps uncertainties in the strength of the jet streak on the upshear flank of the trough (associated with its meridional extension) are more important than the lower-level cyclone itself.

The 'Lagrangian' growth rate plotted does not include the advection of ensemble variance by the ensemble-mean flow. This advection is a major term and would immediately lead to Eulerian growth rates more aligned with the cyclone. We now make this clearer in the text. It is interesting that the Lagrangian growth rate highlights the upper-level trough region. We briefly discussed this in the manuscript. It is intended to be the focus of future work.

12. The bulk of discussions around the spread-error relationship appear to focus on the Spread and Residual terms of Eq. 2, leading to conclusions about overdispersion in the North Atlantic storm track. Is there no simpler way to arrive at the important conclusions of the study without going through this rather complicated derivation and analysis? The interesting flow-dependent aspect of the spread-error relationship is achieved through independent stratification (currently via cluster analysis), so I think the only thing that might be lost would be the conditional bias shown in Fig. 10i. However, this bias could be evaluated directly and shown to contribute significantly to the increased RMSE in the "counterpart" cluster without resorting to Eq. 2. The apparent ambiguity of the Residual term makes the discussions surrounding Eq. 2 quite difficult to follow and appears to make it difficult to make definitive statements about sources of problems within the ensemble. If the important message to be delivered by this work relates to the flow dependent overdispersion in the ensemble, then a simpler analysis (perhaps including regional and/or flow-stratified spread-reliability diagrams) might be a more effective vehicle. However, if the current investigation is just a showcase for the analytic technique itself then (a) that should be clarified and (b) the advantages of this technique over a simpler analysis should be emphasized.

The intention is to showcase the technique and to consider carefully the assumptions made, so that it can be used in future (flow-dependent) evaluations of ensemble forecasts as well as here. Note that we are not just showing that there is a mismatch between MSE of the ensemble mean and variance of the ensemble; we are showing that the mean bias and analysis uncertainty cannot account for the mismatch. While the stratification would highlight the importance of bias in the non-cyclogenesis cluster, it is not immediately obvious to the reader that variance in bias would be important in the non-stratified budget. If we understand correctly, we would argue that spread-reliability diagrams make the same set of assumptions.

13. The lack of PV in the TIGGE database requires the use of Z250, which appears to produce similar results (Figs. 2-5). Although I can completely understand the appeal of starting with PV in this discussion, I think that for pragmatic reasons the entire study should focus on Z250. In the Data and Methods section the rationale for this can be very clearly explained. This would only really affect current sections 4 and 5. The PV 315 diagnostics in (current) section 10 could still be used because they are separate from the growth rate discussion.

We agree that changing between PV315 and Z250 is not ideal. Nevertheless, we consider that it is important in order to discuss equation 1, where the right-hand-side requires PV. We have simplified the case studies, which should help a little. Emphasizing the lack of PV in the TIGGE archive is also considered important to motivate its inclusion at some point.

14. Why was the clustering approach (current section 8) preferred over a much simpler cyclone identification approach? It seems as though clusters 2 and 3 for both domains are lumped into the "non-cyclogenesis" category when the results from the two domains were aggregated. As such, this seems like a very complicated way to identify dates with cyclones in the western North Atlantic.

It is important to have an objective approach to classification, and we wanted the data to 'speak for itself' by showing the structures that emerged. In the end, the results are probably very similar to cyclone identification (as was alluded to in the manuscript). Many of the events in cluster 2 (44 out of 75) for region 1 do find their way into cluster 1 for region 2.

15. I am not sure grammatically why "growth-rate" is hyphenated throughout. This does not seem to be a common construction.

We will consult on this.

16. I do not believe that forecast "lead-time" is usually hyphenated. More generally, there appears to be over-hyphenation throughout the text. Please limit the use of hyphens and ensure that they are represented using hyphen characters rather than the current em-dashes.

We will consult on this.

17. Please confirm that date/time formatting conforms with WCD standards.

We will consult on this.

**Specific Comments**

18. [L45] Distinguish between the true unstable modes of the flow and the computed singular vectors (optimal tangent linear growth with limited moist physics). The note about the "linear regime" points in this direction, but it would be useful to make this distinction right off the bat.

We will do.

19. [L48] It would be useful to itemize some of these approximations here because the difference between ensemble spread growth and error growth rate is fundamental to this study.

We will do.

20. [L52-54] The punctuation of this sentence makes it difficult to follow: consider rewording.

We will try to improve this.

21. [L54] Remove hyphen from "ensemble-mean".

The hyphen here has been fairly standard, but happy to go with the WCD style on this.

[L55-56] Replace "Jetstream" with "jet stream", "wave-guide" with "waveguide", and "down-stream" with "downstream".

We will do.

22. [L57-72] This "outline" paragraph is overly long and complex because it strays into "abstract" territory by summarizing results. Consider shortening this paragraph by restricting its content to section descriptions only.

This has been shortened as suggested.

23. [L58] Provide a reference for TIGGE if it is to be mentioned here. Also confirm that this acronym can be used without definition in WCD, or define it.

The Swinbank et al. paper was referenced at L77. We will bring this forward.

24. [L73] Suggest dropping the first two sentences of this section and including all dataset descriptions here so that the flow of the remainder of the text is not interrupted by them. As noted in General Comment #1, this section should be rewritten to include information about the datasets and methods used throughout the study.

Both reviewers have mentioned including the methods here. The problem is that it would require so much motivation up-front. We have emphasized this argument in the text, and signpost appropriately. We hope this satisfies both reviewers.

25. [L73] I believe that "re-analysis" is more usually "reanalysis", including in Hersbach et al. (2020).

We will change this.

26. [L75] The forecast range of the background does not seem to be identified here or in Appendix E. It seems to be 12 h (line 136), but that should be clarified here.

It is now clarified here (12 h)

27. [L77] TIGGE stands for the "THORPEX Interactive Grand Global Ensemble".

The words that TIGGE stands for have changed over the years. The "I" now stands for "international" (please see: https://www.ecmwf.int/en/research/projects/tigge)

28. [L80-83] This information would probably be better displayed as a table for easier reference in later sections.

We will do this.

29. [L84] Suggest, "These data are used …".

Done thanks.

30. [Fig. 1] Are the trajectories that are used to identify the WCB region extending from -24h to +24h from the analysis valid time (i.e. these are the trajectory midpoints)? Suggest using the "red hatching" term consistently in the caption, rather than "shown in red".

The WCB region shown at a particular time $t*$ is based on all WCB trajectories, which, according to the WCB identification method by Madonna et al. (2014), are within the layer from 800 and 500 hPa at $t*$. Since this method selects trajectories that ascend at least 600 hPa in 48 h, this means that the WCB region can be, in principle, based on trajectories calculated during all 48-h periods from [$t*$ - 48 h, $t*$], [$t*$ - 42 h, $t*$ + 6 h] to [$t*$, $t*$ + 48 h], but given the fact that most trajectories ascend from about 900 to 300 hPa, there will be most likely no contributions from the very early and late of these periods, and as suggested by the reviewer, the bulk of trajectories shown at $t*$ are expected to be from the periods [$t*$ - 24 h, $t*$ + 24 h] and 6 h earlier and later, respectively. This explanation may sound complicated, but the interpretation of the WCB region shown in the figures is not: it indicates the region where WCB trajectories, irrespective of their exact start and end time, are ascending across the mid-tropospheric layer when they produce most of the latent heating and precipitation formation.

We will change text to "red hatching"

31. [Fig. 1] Should the mks form of PVU be provided in the caption?

We will change "2 PVU" to "2 PVU (=2 x $10^{-6}$ $m^2$ $s^{-1}$ K $kg^{-1}$)". It probably only makes sense to do this if we also define isentropic PV as $-g\zeta \frac{\partial \theta}{\partial p}$ in the main text, so we will do this as well.

32. [L104-106] Are these the forecast experiments discussed in section 10? If so, then this is additional motivation to move that section up as a "case study" subsection.

We have argued, in response to point 3, that the case studies need to be up-front to give concrete examples, and then revisited once the result of the over-spread has been presented. Two examples are not enough to say anything definitive about reliability, but the similarity between the two cases suggests that two are sufficient to say something about the sensitivities of ensemble variance.

33. [L108] Suggest "… uncertainty grow-rate estimate …" because the ensemble provides only an estimate of the true forecast uncertainty.

Agreed, we change to "can be estimated as".

34. [L109] What does the "1-dimensional" restriction mean here?  Would this be better identified as "scalar", or can multiple state variables be included in a 1D state vector?  This is obviously important because it reappears elsewhere in the text.

This was a mistake and both reviewers suggested different solutions. We have changed "some 1–dimensional state–measure (of the atmosphere)" to "some atmospheric parameter field". Thank you.

35. [L114] The phrase "but with a different formulation" is too vague.

We now say "Following, initially, the derivation of Baumgart and Riemer (2019), the growth rate of this measure can be related to sources of uncertainty growth via the equation".

36. [L118-124] This is a very complex sentence mixes conservative and non-conservative forcings in Eq. 1.  It would be more useful to split this sentence to describe the physical relevance of the terms on the r.h.s of Eq. 1 individually.

Without further study (which is anticipated) we cannot say much more about the relative importance of each of these aspects. We have given a citation for each aspect, but will also make the text more readable.

37. [L126] Should "Equation" be capitalized here?  It wasn't in section 1.  I do not think that the back-reference to section 1 is very useful here because the introduction did not go into much additional detail about the Liouville equation.  A citation to relevant literature would be more useful here.

We have removed the link to the Liouville equation after considering the comments of both reviewers on the topic of inherent predictability limits.

38. [L125-130] I think that this discussion is fine, but it does not seem to advance the main thread of the study.  It could be dropped to reduce the length of the manuscript.

It has been dropped, as discussed above.

39. [L132-140] This information should be contained in the captions (most of it is) and/or left for supplemental material because it disrupts the flow of the main text.

Much of this detail is required to explain Figs. 2 and 3. This is because the figures show the growth rate after the spatio-temporal filter has been applied. We have placed the text in a methodology sub-section.

40. [Fig. 2] What is the contour interval for the contours showing extreme values?

We stated that "Contours extend the shading scheme to the most extreme values, which are indicated at the ends of the colour bar". We now clarify a little more with "Contours extend the shading scheme (with the same interval) to the most extreme values, which are indicated at the ends of the colour bar".

41. [Fig. 3] Should this read "Case 2"?

Yes, it should – thank you. Note that we are now only showing one case study in the main text.

42. [L145] Is this a third case study being introduced?  I think that discussion of the full-season perspective should be left for the subsequent section (in the reorganized paper).

This text refers to the animation in the supplementary material, which was mentioned on line 139. We now re-emphasise that this is in the supplementary material. We would be happy to share this animation with the reviewer.

43. [L145-150] These seem like "future work" suggestions that would be better left for the concluding discussion.

Yes, agreed. This will be placed in the discussion, and we will simply say here that the growth rates could be model-dependent and then ask the final question "How well do growth rates agree amongst the TIGGE models?"

44. [L152-155] The 12-h forecasts from the TIGGE database are for ENS rather than EDA, is that correct? If so, then is it true that Figs. 4a and 5a look different from Figs. 2 and 3 not only because the field is different but also because the perturbations are different? If I understand the ECMWF system correctly, SV perturbations are not added within the EDA cycle, but are added before ENS initialization. In that case, Figs. 4a and 5a have an additional source of optimized growth. That seems to make the comparison interesting, although it is complicated by the change in diagnostic field. Would it not be surprising if the SV perturbations have little impact on growth rates in these cases? Perhaps the Z250 growth rates could be shown for Figs. 2 and 3 to make this comparison possible.

The reviewer is correct that the ENS is used from TIGGE and that, for ECMWF this includes SVs (and SPPT) while the EDA does not include SVs (but does include SPPT). We did discuss this in the context of improving seamlessness in order to be better able to diagnose the (2-day) growth rates of the model (and its uncertainty representation) from short-range forecasts. We will make this distinction clearer. Similarities between the EDA PV315 growth rates and the ENS Z250 growth rates suggest that there are strong growth rates even without the explicit inclusion of SVs. While EDA Z250 growth rates would be interesting, we feel that they would complicate the manuscript unnecessarily.

45. [L155-156] So are these case studies (particularly Fig. 4) not representative of the general behaviour of these models? If so, perhaps another case study should be chosen for this comparison.

The main point being made is that they disagree in terms of growth rates. We mentioned the slightly better agreement for some other cases for completeness. The full range of agreement/disagreement can be seen in the animations that we will provide with the revised paper.

46. [L172-174] The source of Eq. 2 (appendix C) should be cited at the beginning of this discussion.

We will bring this forward in the text.

47. [L181-182] I have a hard time understanding a lot of this discussion and how it relates to Fig. 6. It would be great to label the lines in Fig. 6 with the names of the terms in Eq. 2 that they relate to. The lines seem to be more directly related to the discussion in Appendix C, so perhaps Fig. 6 would be more appropriate in the appendix.

We did experiment, but it is difficult to label all the lines in Fig. 6. For example, there are two lines which contribute to the forecast variance term, and two which contribute to the analysis variance term. Lines are also conditional on the truth. The reviewer is correct that the figure relates more directly with the variables in Appendix equation (C1). We have moved the derivation to the main text, alongside the figure, and made it possible for readers to skip this if they choose.

48. [L192] Does this "main additional term in the Residual" refer to Eq. C5? If so, it would be useful to cite that equation here.

Yes, it is the last term in (C5). This has now been brought into the main text, as discussed above.

49. [Fig. 7] The change in colour scale range for panels (n) and (o) make comparison of the plots on the bottom row difficult. With the current plotting scheme, it looks like the difference in residual is almost entirely explicable by the difference in spread, but that is not really the case (is it)? The contour intervals for values beyond the standard colour bars should be noted in the caption.

The reviewer is correct in their interpretation. In response to the other reviewer's recommendation, this figure now includes the four TIGGE models investigated, with no differences plotted.

50. [L219-223] It is challenging to follow this discussion because of two forward-references to a description of the variance of forecast biases. It seems like that aspect of the discussion should be introduced before this text appears. In fact, it is not clear what discussion the forward references here are actually describing (the section 9 discussion seems to take an understanding of the forecast bias variance's impact on the Residual for granted).

The forward reference is to the paragraph on lines 283-293 (in Section 9). In Section 9, the variance of the forecast bias (at least the "inter–cluster variability in forecast bias" – line 290) is explicitly represented (i.e. in the bias term, rather than in the residual). We have now changed the text of line 291 to say "explicitly represented in the bias terms shown in Fig. 10d and Fig. 10i". This is a subtle but important point in motivating the use of flow-dependent evaluations. We hope, with the derivation now in the main text, and further text refinements, that this will be clearer to the general reader.

51. [L233] It was not obvious that this is a "key question", so hopefully a clear statement of the study's objective(s) in the introduction will help to make that link more direct.

Hopefully so, motivation at the beginning seems to have been a key issue.

52. [L233-235] Does this "either-or" statement arise from the form of the Residual term (Eq. C5)? If so, then it seems like it would be useful to put this equation in the main text, hopefully as part of a discussion on the meaning of "variance in forecast bias", which I think might be related to the "difficulties" proposed here (?).

The either-or statement is about whether the residual is associated with specific synoptic flow types (such as those with high growth-rates as in Figs. 2 and 3) or a more general issue. For example associated with scale interactions with planetary wave uncertainties, or other scale interactions which might be more ubiquitous. We have tried to make this clearer.

53. [L242-243] This region is quite complex: why would three clusters necessarily "provide sufficient degrees of freedom"? The optimal number of clusters is difficult to determine, but usually dropoffs in quantities like the AIC or BIC serve as some sort of semi-quantifiable justification for the number of clusters.

The aim was to balance realism of clusters with the need to obtain a sufficient sample size. This will be a compromise, but a broad ridge, a broad trough, and a tighter cyclogenesis flow-type seem to cover most eventualities. We try to make this clearer now. Note that later on, on line 265, we stated that "visual inspection of plots similar to those for the two case studies (Fig. 1) suggests that the objective clustering has been successful in partitioning the date/times into cyclogenesis and non–cyclogenesis flow types". In addition, the clustering was successful-enough to provide statistically significant differences between the partitioned date/times. We now make this point too.

54. [L256-258] This is the only discussion of the uncertainty growth rate in this section, and it does not seem to lead to any particular conclusion. Is there a good reason to include it here and in the Fig. 8 and 9 plots? (It does not seem to be discussed in the subsequent section either.)

This is useful for two reasons: firstly it again demonstrates that the cyclogenesis cluster is associated with the strongest growth rates (note that most of the date/times for cluster 1, area 2 come from the date/times in cluster 2, area 1) and hence consistent with Figs. 2 and 3, and secondly it provides the opportunity to note that these growth rates are not used in the clustering (that could potentially bias the reliability assessment). We change "since this is what will be evaluated" to "since this could potentially bias the reliability assessment".

55. [L294] The phrase "almost the entire over-spread" seems like a bit of an overstatement. It is probably more defensible in terms of variance, but could perhaps be softened to "much of the overdispersion" or similar.

Since this is one of the major conclusions of the paper, we have followed the reviewer's suggestion, changing the text to "much of the over-spread in the region of focus, at the head of the North Atlantic winter storm-track (Fig. 7e), is associated with the cyclogenesis composite – with statistically insignificant residuals (largely light blue and light grey) in Fig. 10j and statistically significant residuals (largely dark blue and dark purple) in Fig. 10e. Differences are shown in Fig. 10o. Over Newfoundland in particular, they are comparable in magnitude to the full departures in Fig. 10a and statistically significant. Downstream, differences in the residual have the opposite sign – possibly associated with differences in downstream cyclogenesis, and consistent with the increased spread noted above."

56. [L297-301] I am afraid that I do not fully understand this discussion. How would the stratification of the groups (cyclone vs. non-cyclone) be done differently with multiple seasons or an independent assessment? Could this "regression to the mean" alternatively be considered a sampling bias?

One approach could be to deduce a composite of the (local) initial conditions leading to cyclogenesis within one season, and to then pick date/times from the same season in a different year which project strongly onto the initial condition composite. Since this approach has not been tested, it is difficult to predict how successful it would be. Regression to the mean refers to the fact that, if one sampling of a random variable is extreme, the next sampling is likely to be less extreme. In the interests of brevity, we have removed this paragraph.

57. [L302.5] Consider simplifying the section title to "Sensitivity experiments to quantify uncertainty sources".

We have changed to the suggested title – thank you.

58. [L315] I understand that resource constraints likely make additional tests difficult or impossible, but is it not conceivable that the ordering of MU and 4K is important? Systematic changes in the physics tendencies should be expected between 18 km and 4 km grid spacing (for example as more turbulent fluxes are represented by the dynamics), which will impact SPPT directly. This might mean that the impact of switching MU on and off at 4 km is different from what is observed in the 18 km configuration. I do not think that this is a big enough deal (or close enough to the focus of the paper) to justify additional simulations; however, you may want to put a bit more nuance in the wording of this statement.

Thank you for pointing this out. We had been thinking more about SPPT's action on diabatic processes (there is a shift from 'convective' to 'large-scale' precipitation at 4 km, but the total precipitation – and thus diabatic tendency which SPPT works on – is largely unaffected). The power spectra in the figure on page 3 of this document do highlight more resolved variance in the 4 km experiments. We will add the words "(the impact of increased resolution might be somewhat different in the presence of model uncertainty – because the parametrized turbulent fluxes could be weaker)".

59. [L318] Why not show results from the 1200 UTC 27 November 2019 initialization so that the day-2 forecast aligns with the panels shown in Figs. 1, 2 and 4?

The sensitivity plots show the impacts at day 2 of the growth rates that occurred previously – hence the difference in time.

60. [L326] Does the upshear maximum in the SV plot (Fig. 12b) really very well described as being in the "cold sector" of the cyclone? The cold sector is defined based on low-level airstreams but here the plot is showing spread differences in Z250. I think that this is much more related to the growth of perturbations in the jet streak on the upshear side of the trough, which is contributing to the "digging" of the trough / meridional amplification. Could the upper-level jet-front structure not an ideal place to have rapid SV growth (e.g. Hakim 2000; JAS)? By increasing vorticity at the base of the trough this feature will indirectly impact troposphere-deep cyclogenesis, but I think it is possible that the origins of the spread are more local. (The same is true for the second trough over the eastern North Atlantic that appears to be approximately equivalent barotropic.)

Thank you for these interesting suggestions. See the reply below to your comment 61.

61. [L327-329] The spatial separation of the SV and MU contributions is beautiful. I think that it is very understandable based on the previous comment and the fact that model physics is largely inactive in upper-level jet-fronts, other than perhaps some turbulence. The MU is focusing on the regions where the physics is active (lower-level cyclone and WCB) while the SV is picking up dynamic growth along the jet streak on the waveguide. If you agree with this assessment, it could be a useful inference to add to the text.

Indeed, these are interesting inferences. They are difficult to prove, but we agree with the reviewer that we should include these considerations in the description of Fig. 12 b,c. We therefore will change the paragraph (L322-330) as follows:

"Figure 12a shows the OP configuration with a well–developed surface low pressure system, as discussed in relation to Fig. 1. The warm conveyor belt (WCB) associated with this cyclone is seen to lead to the development of a prominent downstream upper–level ridge and a downstream trough west of Europe. As one might expect, the maximum spread is not co-located with the maximum 'Lagrangian' growth rates (cf. Fig. 4a). The impact of the initial SV perturbations on Z250 spread (Fig. 12b) is particularly pronounced along the western flanks of the prominent troughs over the western and eastern North Atlantic, respectively. This likely indicates the potential for dynamic growth along the intense jets in these regions, qualitatively in line with the idealized studies by Hakim (2000). There are places where the SV impact on spread is half the total (so that the fraction of variance explained reaches 25%). In contrast to the SV impact, the impact of the model uncertainty (MU) representation (Fig. 12c) is particularly pronounced in the cyclone centre and in the region of the

WCB ahead of the surface low, i.e., in regions where cloud-related physical processes are particularly active. The large signal along the western flank of the ridge southwest of Greenland is consistent with the results of Joos and Forbes (2016), who found a large influence of cloud microphysical processes in the WCB on the tropopause structure in this part of the downstream ridge. MU also explains up to 25% of the total variance. The remaining variance must be associated with the (deterministic) growth of initial EDA analysis uncertainty."

Hakim, G. J., 2000. Role of nonmodal growth and nonlinearity in cyclogenesis initial-value problems. J. Atmos. Sci., 57, 2951-2967.

Joos, H., and R. Forbes, 2016. Impact of different IFS microphysics on a warm conveyor belt and the downstream flow evolution. Quart. J. Roy. Meteor. Soc., 142, 2727-2739.

62. [L328] Missing closing parenthesis for figure reference.

This has been added, thanks.

63. [L334-336] Discussion of total precipitation seems tangential to this study (also L344-345).

The observation (L334-336) that the total precipitation stays the same but the spread is altered seems interesting. As discussed above, L344-345 is useful in the argument about the ordering of the experiments not being too important. We will add the words to the effect "and hence the impact of model uncertainty (in the form of SPPT) should be less sensitive to this aspect of increased resolution".

64. [L346-351] This is the first time that observation location is discussed. The Obs experiment seems largely unrelated to the other experiments and should be eliminated to focus the study on the "controllable" sources of spread quantified in the other experiments.

The reason that the observational experiments are included is that they help us quantify what extra predictability is currently achieved through the assimilation of local observations, such as cloud-affected radiances. While they do not directly impact reliability estimates, they allow us to gauge the relative importance of working towards reducing the use of SVs. The juxtaposition here is also useful because it motivates a future more comprehensive study of the impacts of assimilating observations in cyclogenesis flow situations.

65. [L343] Suggest changing to "… appears to yield a better depiction of uncertainty than that generated by …".

We have changed to "The impact on PV315 uncertainty of allowing the model to resolve more of the convection at the 4km resolution (Fig. 13e) appears to be in closer agreement with the response to turning off the deep convection parametrization (minus Fig. 13d), when the model is forced to represent the convection on the 18km grid".

66. [L343] Remove extra "km".

Thank you.

67. [L343] This seems like a really important statement because it suggests that the huge computational cost of a 4 km ensemble is not justifiable from this perspective.

From this perspective, agreed. As noted, 18 km is not the scale to resolve convection.

68. [L374] Although they can likely be inferred, neither baroclinic nor convective instabilities were demonstrated in the analysis.

Agreed, but we feel that the inference from baroclinic development and convection back to their respective instabilities can be assumed.

69. [L382] This conclusion does not seem as direct as it ought to be. Perhaps "could" should be replaced with "should"?

We have changed to "should".

70. [L382-383] This seems like a fairly weak and somewhat confusing statement on which to end the manuscript. Moist singular vectors would be implemented in the TL/AD forms of the model, and as far as I know are quite independent of the SPPT-based model uncertainty estimate. Perhaps this discussion could instead be extended to consider the SPP-based uncertainty formulation as a look into the future ECMWF system.

We have made a more direct statement now. The point we were trying to make was that it would be good to explore the idea of focussing model uncertainty on potential instabilities, rather than the effects of already-triggered instabilities. The former is more like what SVs are doing, although SVs would be costly to calculate at every timestep. It is possible that SPP could evolve into targeting these potential instabilities if the perturbed parameters were the triggering thresholds, for example. Since submitting the original manuscript, SPP has become more competitive with SPPT, and looks likely to be implemented at ECMWF in the near future. We will change the wording to the effect "It is possible that such a focus on instabilities (rather than the effects of already-triggered instabilities) might be better explored within the future 'stochastically perturbed parameter' (SPP) framework for model uncertainty – perturbing triggering thresholds for example."

71. [L392] Is a ^2 missing on the l.h.s of definition of the variance?

Yes, thank you for spotting this.

72. [L444] Why would the squared terms necessarily dominate, particularly if there are correlations between the constituents of the cross terms?

In general, the terms are uncorrelated. We have reinstated our estimation of the cross-terms which might be non-zero (please see below).

73. [L465-469] Providing a quantitative assessment of the relative size of each of these terms seems like it would be useful, particularly because the Residual is one of the (two) leading terms assessed in the text is key to conclusions regarding overdisperison.

These terms were estimated in an earlier draft of this manuscript, but this was removed for brevity. A shortened version will be included in the appendix of the re-submitted manuscript.

74. [Appendix D] Why is a new field (500 hPa height) and season (JJA) introduced just for this appendix? I guess it might be to show the robustness of the analysis, but I think that the text on L228-232 distracts from the main message of the study. In a two-paper solution (General Comment #2), this figure and discussion could form the basis for a short subsection instead.

The figures were meant for the supplementary material. We have removed them now, and simply state "Note that the over–spread is not specific to Z250 or the North Atlantic stormtrack (not shown)".

The authors would sincerely like to thank the reviewer for their insight and diligence in reviewing this manuscript. We feel that changes made have led to useful improvements.

---

## Author Comment (AC4)

Review of "The Cyclogenesis Butterfly: Uncertainty growth and forecast reliability during extratropical cyclogenesis" by Mark John Rodwell and Heini Wernli

The paper investigates ensemble forecast reliability at 48h lead time, mainly in the ECMWF system with some comparison to other centers. To do so, a new spread-error budget is derived. It is found that the ECMWF ensemble is overspread in stormtracks in the winter season and it is argued that this is related to cyclogenesis events. In my opinion the core topic of this work is interesting and worth being published since it contradicts the intuitive expectation that cyclogenesis is associated with bad forecasts and low predictability. However, the paper requires a substantial revision. It is too long, difficult to read and not well structured. It spends a lot of time on tangent discussions and detailed case analyses, which I find distracting and misleading. Especially the discussion of the butterfly effect is imprecise, irrelevant and in part incorrect. On the other hand, topics more directly related to reliability and ensemble forecasting systems are very little discussed, if at all.

We appreciate the reviewer's comments and their time taken to review this manuscript. We attempt to address their specific comments below. In our replies, all line numbers refer to the originally submitted manuscript

Major comments:

"The cyclogenesis butterfly"

This term, given already in the title, is never properly defined. It is introduced by phrases like "here we think of it as...". I am still not clear what is actually meant by this. The paper investigates reliability, which is an aspect of practical predictability, while the "real" butterfly effect refers to the existence of an intrinsic predictability limit caused by scale interaction in a multi-scale system (see Lorenz, 1969 and Palmer etal, 2014). Current weather prediction system are started from initial condition uncertainties that are much larger than butterflies and are on average far away from hitting the intrinsic limit (e.g. Zhang etal, 2019). The existence of singular vectors are not a manifestation of the butterfly effect, since they are still consistent with infinite predictability due to their constant growth rate. Judt, 2018 (Fig. 8b, day 0-2) for example has demonstrated the extreme increase in the error growth rate if the atmosphere really is perturbed with "butterflies" only. I am however not recommending to discuss the butterfly effect more precisely in this paper but rather to remove this discussion and to focus on much more relevant aspects with respect to (practical) reliability, like the long-standing underdispersive problem of ensemble forecasts and various methods that have been used to mitigate it (e.g. EDA, singular vectors, breading vectors, SPPT, SPP, etc.). It is probably a shortcoming in one or several of those methods that leads to the reliability problem that the paper investigates.

Sorry for the confusion here. We are not talking about the "real butterfly effect", intrinsic predictability limits, or Lorenz, 1969. To be fair, we did not mention the butterfly effect, but rather stated that our butterflies were defined as "local flow configurations where the chaotic and exponential growth–rate of uncertainty is particularly strong" (line 32). We also discussed SVs as indicating that divergence of trajectories within state–space is not uniform over the attractor (line 31), rather than as any manifestation of the butterfly effect. Our reference to Lorenz was to his 1963 paper – discussing sensitivity to initial conditions but not any intrinsic limit to predictability. Our use of the term "Cyclogenesis butterfly" is, rather, an attempt to encourage more flow-dependent thinking in the evaluation and development of forecast models. However, the potential for confusion is recognised, and we now ensure the reader does not get the wrong idea from the outset.

The figure below shows power spectra at day 0 and day 2 for the ECMWF ensemble. The maximum initial (EDA) variance contribution is from waves around 400 km, while the maximum D+2 variance

contribution is from waves around 1500 km – i.e. synoptic scales. Notice that the initial variance is saturated at scales smaller than about 100 km. This plot motivates/justifies our interest in the growth of variance at the synoptic scales to D+2 – due to physics and dynamical interactions at all scales.

[Figure]

Incidentally, the Palmer et al. paper uses the current lead-author's previous example to speculate that intrinsic predictability limits might be longer due to the potential confinement of error-growth to intermittent flow regimes. While we do not discuss intrinsic predictability limits, the idea that certain flow-types can organise multi-scale interactions and focus error growth is very much in the spirit of the current study.

While lack of reliability might be explainable by shortcomings in the mitigation methods mentioned by the reviewer, this is not obviously the case. For example, sensitivity of parametrized convection to uncertainties in the resolved flow might be important.

To improve brevity, we have removed discussion of links to the Liouville equation. The thought was that the use of model uncertainty represents a route by which NWP could converge on the true dispersion rates on the real-world attractor but, reflecting on the reviewer's comments, it is probably true to say that NWP will never be able to say anything definitive about the real butterfly effect, as it pertains to the real world.

Reliability in a larger context

The specific overspread that is found over the Northern Atlantic stormtrack in winter (Fig. 7) has not been put into a wider context. If the system is reliable on average there must be compensating

underspread somewhere, e.g. over the continents, outside the midlatitudes or in the other seasons. This should also be discussed, as well as the question if the system really is reliable on average at the considered 48h lead time. Some information can be found in appendix D, but I think this discussion should be central in the paper. Furthermore I am wondering what the downstream consequences of the stormtrack-overspread are. Does the overspread persist beyond the end of the stormtrack into the continent in a lagrangian sense, e.g. is the 5 day forecast for Europe in the winter season also overspread? Finally, what is the relation to forecast busts? According to Lillo and Parsons, 2017, East coast cyclogenesis has the potential to generate particularly bad forecasts over Europe. This kind of contradicts the (average) results from this paper. Possibly one season of data is not enough to investigate this but some discussion here would be helpful.

There is some compensation elsewhere. The authors' previous paper on flow-dependent reliability shows good agreement in the annual-mean hemispheric-mean sense between RMSE and ensemble standard deviations. The current work (e.g. Fig. D1) demonstrates that this agreement may not have been so good (at day 2) if bias and analysis uncertainty had been factored-in. The lead author has also found that things have deteriorated (in the stormtracks) since then. For T850, some of the compensation in the operationally calculated scores actually occurs under Tibet, and this is screened-out in the diagnostics developed here. These details seemed to the authors to be of little long-term practical use to readers on this study. At day 5, the over-spread is still evident but general interactions and the loss of any continued SV contribution make this less clear. The emphasis of this study is on the short timescales because these are the only timescales where agreement between ensemble members is sufficient to be able to make meaningful calculations of their dispersion rates. We do not see any contradiction with the Lillo and Parsons paper – cyclogenesis clearly results in large spread and deterministic forecast busts. The Lillo and Parsons paper is a major reason why we investigated cyclogenesis.

More use of TIGGE

While for the case studies 4 centers have been compared, the spread-error budget comparison is only done between the ECMWF and the UKMO system and finally the clustering analysis is only done for the ECMWF system. A reason for this is not given. I think the paper may miss an opportunity here to investigate possible reasons for the (ECMWF) overspread since the different centers use different methods to generate their ensemble. Hence I recommend to include more centers throughout the paper, particularly in the clustering analysis to see if the cyclogenesis overspread is a more general problem or specific to ECMWF.

We avoided a detailed 'beauty contest' because this can be problematic in manuscripts. For example, making sure that the details of the ensemble initialisation over the period of interest are documented correctly, and checking whether there are other factors to consider. Nevertheless, on the reviewer's advice, the variance budget has been calculated for the four models discussed, and this figure will replace the previous one. Applying the clustering analysis to all the models would include a lot more plots and require even more discussion. The primary aim of the study was the evaluation and understanding of uncertainty growth rates in the ECMWF model.

More focus

The paper spends a lot of time with a detailed discussion of two cases which in my point of view gives little insight. Furthermore, the paper oscillates between analyzing the cases and the entire winter and also between theta- and pressure-level analysis or squared and non-squared metrics, which I find confusing. With Eq. 1 the paper introduces a rather sophisticated diagnostic which later is not used at all. I think this is not a good use of the time of potential readers. The information given in this paper is distributed over 8 sections, 5 appendices, 17 figures and additional supplementary

material. I suggest the authors should consider condensing the paper to the essential parts and keeping analysis and methods consistent across the paper.

Equation 1 is central to the study. Plotting it demonstrates that large growth rates are confined to particular flow features. The right hand-side shows that the growth rate (for the PV variable) represents the effects of uncertainties in diabatic processes and in non-linear dynamical scale-interactions. This is now revisited in the conclusions (thank you for pointing this omission). It is hoped that equation 1 will feed into subsequent work examining these aspects in more detail. For the growth rate plots, only one case is now presented – with one PV growth rate example for ECMWF (which relates directly to the equation) and one comparison of TIGGE model growth rates (which highlight similarities and differences). The sensitivity study is considered important as it highlights the relative roles of model uncertainty and singular vectors, and points to possibilities for future development. The lack of a clear signal associated with assimilation of local observations is consistent with the fact that the variance at small scales is largely saturated in the initial conditions – the grid-point skill is coming from the larger scales which are adequately constrained in the 4D Var without the local observations. This is now better discussed.

Specific comments:

L1:

This statement is incorrect (see major comment about the butterfly effect and comment below).

We now state that "Global numerical weather prediction is often limited by particular flow features which are associated with pronounced uncertainty growth rates".

L21:

The Liouville equation as formulated in Ehrendorfer, 1994 assumes that the propagation operator is known and constant. Hence it cannot describe growth due to model uncertainty.

As discussed above, we have removed reference to the Liouville equation.

L26:

I would not say that EDA represents model uncertainty. The model uncertainty is rather part of the assimilation process to generate the initial condition ensemble.

Yes, we are saying that model uncertainty representation is included in the EDA system. The missing piece of information seems to be that we do not mention that we are referring to the ECMWF EDA system. We now say "… in particular, at ECMWF, the ensemble data assimilation system (Isaksen et al., 2010) aims to represent flow–dependent error covariances in the background or "first–guess" (Bonavita et al., 2016) and in the observations (Geer et al., 2018), as well as model uncertainty (MU; Buizza et al., 1999) and a model grid's lack of representativity of point observations (Janjić et al., 2018)".

L31:

This statement is incorrect. Lorenz-type butterflies, i.e. small-scale and small-amplitude perturbations limit predictability via scale interactions and not only due to chaos and strong sensitivity to initial conditions (see Lorenz, 1969 and especially Palmer etal., 2014). Hence the constantly growing singular vectors do not represent this "real" butterfly effect. Furthermore, current errors and uncertainties in forecasting system are neither small in scale nor small in amplitude and cannot be regarded as butterflies. If they were this would mean that current systems operate already now at the intrinsic limit, which is not true (e.g. Zhang etal. 2019). I agree that in

some situations error growth in current forecasts is worse than average but if this might be related to the butterfly effect in rare cases is an open question.

We have changed the text "predictability is often limited by the Lorenz–type butterflies in the flow (Lorenz, 1972). Here we think of these as local" to "prediction is often limited by specific".

L51:

I am not sure if I understand correctly what you mean with "cyclogenesis butterfly". The term is never clearly defined.

We now make this clearer, as discussed above.

L51:

"The key question is..."

This is a big gap in the line of argument and comes as a surprise to me. Please consider rewriting the introduction to focus on this question and the importance and flow-dependence of reliability and the need to extend the "spread-error" relationship.

The question is in the title. Clearly, we needed to work harder on motivating this in the introduction, and we have done this now.

L57:

The paper outline contains to many details in my opinion. Some should have been mentioned and discussed in the introduction, some are results.

We have shortened the outline to remove details and results.

Sec. 2, Data:

Since an entire section is dedicated to describe the data I would prefer that all the relevant details are given here rather than being distributed over the rest of the paper and the appendix. With respect to the other centers, only resolution and ensemble size are given but potentially interesting differences in the ensemble design are not mentioned and later not investigated (see major comment above).

We now increase the discussion of the ensembles from the other centers and include the initialisation details available within the TIGGE archive. This section has been re-titled "data sources". We feel that it does not make sense to describe in detail how this data is used at this point (the parameters, lead-times etc.) – before the methodologies have been motivated. We have tried to indicate this structure better now, with better sub-sectioning throughout the manuscript.

L74, caption Fig. 2:

What do you mean by "background ensemble/forecast"?

We now say "Depiction of cyclogenesis Case 1, as represented in the background forecasts of the EDA". In response to the other reviewer's comments, we have dropped most of the explicit references to the animations here. We still consider these animations to be very useful, and we would still like to provide them as supplementary material – we would be happy to share these with the reviewer if a means can be found (they are quite big files).

L90:

The arguments given about the case selection are very vague. How are they related to the key question? Are these cases in which the forecast was particularly unreliable? Or in which the cyclogenesis was very rapid?

This is a good point. We stated why the cases were chosen but did not make it clear that other attributes had not been a factor. The cases were not chosen because they were unreliable (difficult to establish for a single case) or that cyclogenesis was particularly rapid, or that uncertainty growth rates were unusually large. They were simply chosen to motivate to the reader the kinds of events we are considering, and for their suitability for the sensitivity experiments ("without being strongly affected by other flow perturbations in their environment"). We now make this clearer.

Fig. 2-4:

It is unclear, which forecast lead time you show and why you chose to focus on this particular forecast lead time.

The information for these growth rate figures was given in Appendix B (Further details on the growth rate plots). We appreciate that this is not ideal and have now brought Appendix B into a subsection in the main text (with a note to say that this is for the interested reader and could be skipped).

To answer the reviewer's question, Figs 2 and 3 are constructed using the 12 h background forecasts from the EDA (so no lead-times are greater than 12 h), started at 6 and 18 UTC. The fields shown are based on centred-means and differences between consecutive hourly lead times. The 24 h running-mean temporal filter then places the smoothed fields back on the whole hours. More specifically, Fig. 2 shows fields centred at 12 UTC on 29 November 2019. For the winds (including humidity fluxes), PV315 contour, the standard deviation in the growth rate parameter (PV315) and its advection, the lead times used are thus {28 Nov 18 UTC + 6,7,8,9,10,11,12 h}, {29 Nov 06 UTC + 0,1,2,3,4,5,6,7,8,9,10,11,12 h}, and {20 Nov 18 UTC + 0,1,2,3,4,5,6 h}. The ensemble-mean precipitation in a given hour and the time derivative in the standard deviation of PV315 are based on the differences (in precipitation accumulations and PV315 standard deviations) between these lead times {28 Nov 18 UTC+ (7-6),…,(12-11) h}, {29 Nov 06 UTC+ (1-0),…,(12-11) h}, and {20 Nov 18 UTC+ (1-0),…,(6-5) h}.

For Figs. 4 and 5, again the lead times used are no greater than 12 h. The differences are that the forecasts are started at 00 and 12 UTC and data are only available at 6 h intervals. Hence for Fig. 4, the lead times used are thus {29 Nov 00 UTC + 0,6,12 h} and {29 Nov 12 UTC + 0,6,12 h} and the running mean has length 4 rather than 24.

We focus on these shortest lead times possible so that the ensemble members are as close as possible to each other, in particular representing the same synoptic systems, and hence the growth rates are the best flow-specific growth rates we can calculate. We now make this motivation more clearly. For example, at L138, we change "and highlight the features associated with enhanced growth rates" to ". Because ensemble members are synoptically very close to each other at short lead times, we can identify the synoptic features associated with enhanced uncertainty growth rates".

L109:

I don't understand what you mean be 1-dimensional state-measure. Substitute 4-dim atmospheric field?

This was a mistake and both reviewers suggested different solutions. We have changed "some 1–dimensional state–measure (of the atmosphere)" to "some atmospheric parameter field". Thank you.

Eq. 1:

sigma_hat is now the standard deviation of the PV, right? Use P_hat instead of sigma? The hat is not explained, same meaning as in eq. 2?

Sigma is the standard deviation, and the hat signifies that this is an estimator. This is now made clear in the text. We do not use P_hat since we use the same formulation of the left-hand side of (1) for other quantities (in particular, Z250). Yes it is the same meaning as in (2).

I suggest to add an index i to P, P' and NC to indicate that these are quantities from individual members.

The derivation of (1), which was in Appendix A, has been brought into the main text. Here subscripts are initially used (e.g. L389-392). We feel that the current approach of using an overbar and no subscripts to signify a mean is neater and consistent with the standard approach for signifying the mean of a linear quantity. We use this approach throughout the paper. We appreciate that this can initially cause confusion and so where the variance of the $P_i$ is defined on L392 (there is a superscript 2 missing here), we spell-out more clearly that the subscripts are dropped even for non-linear terms: "Note that we drop the subscripts below the overbar for non-linear as well as linear quantities so, e.g., $\overline{ab} \equiv \frac{1}{m}\sum_{i=1}^{m} a_i b_i$.

L120-L130:

Needs more introduction and explanation. However, this diagnostic is not used in the paper anyway. Consider removing it (see below).

Following the reviewer's first major comment, we have dropped the discussion of the Liouville equation (L125-130). We feel that L120-124 are important in the discussion of the processes that can lead to the enhanced growth rates – these would be the (non-linear, scale interactive) processes that act on the initial uncertainty (including applied SVs) and the perturbations introduced by the model uncertainty. We did not discuss this fully enough previously, and have rectified this in the new Conclusions and Discussion section, with a pointer from L120-124 to this discussion.

L142:

"often preceding cyclogenesis", "occur within strongly precipitating WCBs":

These statements are rather vague and are either obvious or seem speculative. Do you mean that the growth rate is correlated with the amount of precipitation in the cyclone? And with cyclogenesis do you mean a depending of the trough where the growth rate peak occurs or does it lead to a cyclogenesis downstream?

This statement is an observation about the animation. We did not intend to make any link to a deepening of the trough or downstream cyclogenesis – simply the colocation of the growth rate with the base of the upper-level trough.  The results (Fig. 8 and 9) tend to confirm that there is enhanced growth at the base of the trough. We now make this link more clearly at L257. We haven't focussed on the WCBs so cannot confirm any correlation with precipitation amount – we change the text slightly to say "They also seem to occur within strongly precipitating WCBs".

L145:

"Further investigation..."

I find the following statements distracting. But more importantly, the reader might have invested some time to understand eq. 1, the relevant papers and the appendix only to find out now that you are not investigating the right hand side of eq. 1 at all and leave this for future work. Hence I recommend to remove section 4 and appendix A and just state here that you are plotting a lagrangian growth rate.

We do now discuss the right-hand side of (1) more fully. We direct the reader to more discussion in the Conclusions and Discussion section (so that it does not distract so much here).

L152:

It is very confusing that you switch now to geopotential growth rates. I understand that PV is not in TIGGE. But what is the point of showing the PV growth rates first, especially since you did not explore the right-hand side of eq. 1, which for me is the main purpose of using PV? I suggest to stick with Z250 then and to omit the PV-plots.

We agree that changing between PV315 and Z250 is not ideal. However, we do not wish to leave the reader under the impression that it is all about SVs and model uncertainty – both these aspects will be acting through the right-hand side of (1), in the PV context, to generate the spread. Emphasizing the lack of PV in the TIGGE archive is also considered important to motivate its inclusion at some point.

L157:

"... are very evident"

I actually was surprised to see how bad the agreement is. Not much is said however about where these discrepancies come from, L160 makes a very general statement.

We agree that the differences in growth rates are surprisingly large. Following a recommendation from the other reviewer, we now include a table summarising the initialisation procedures of the other models and show the extended spread error equation for these models.

L161:

"It would be useful..."

Please clarify what you mean. Uncertainty growth rates cannot be close to the truth since the truth is not uncertain.

We argue that there is a true growth rate associated with the real-world's attractor – this was why we discussed the Liouville equation. As discussed above, we agree that this might not be an attainable goal for NWP – or indeed if we can actually define such a goal. We have changed the wording from "which is closest to the truth" to "which system best maintains short-range reliability".

L167:

You state the essential information as e.g. in brackets. I suggest to change that and maybe write down an equation. Also I suggest to be more precise what average means (case average, area average, ensemble average).

We have re-worded and re-arranged the text to say "when averaged over a sufficient number of ensemble forecasts, the average difference between the truth and the ensemble mean should match the average difference between an ensemble member and the ensemble mean (so, for

example, the mean–squared–error of the ensemble–mean should match the mean ensemble variance). Adding an equation here would introduce extra notation that we do not consider helpful in the derivation of the extended equation.

L174 (also L168):

I suggest to replace "ensemble forecast start times" with "(large number of) cases". Also please explain the symbols first and discuss the visualization afterwards. The notation is inconsistent with eq. 1, maybe express the ensemble mean with <..>.

The key issue here seems to be the definition of ensemble members and ensemble forecasts. We have now made this much clearer, and now simply replace with "ensemble forecasts". By bringing the derivation from the Appendix into the main text, the symbols are explained beforehand. If one looks through the literature, an overbar is generally used to denote a mean, regardless of what it is a mean over. Conversely, "<..>" is often used to denote an inner product. Hence, we would prefer to use an overbar in both (1) and (2), but we now make it clearer that the overbar in (2) is an average over ensemble forecasts.

L188:

Is mu_A equal to the truth? And is mu_F also equal to the truth? I suggest to not discuss this in the figure caption.

These parameters are displayed in the figure and are different from the truth. We hope that by bringing the derivation together with the figure, this will become a lot clearer – since (C1) is the equation which relates directly to the figure.

L196:

I find this square-root operation just for "more understandable units" confusing, especially since it introduces the complication with the residual and you admit that small contributions look larger than they actually are. Moreover, in the supplementary figure you switch back to square units. I suggest to keep the squares in every plot.

Reliability is about bias as well as spread (and all other moments too of course). Hence having bias in its correct units is valuable. We have made this point now. We have also dropped the squared figures from the supplementary material.

Sec. 7:

I wonder why you switched from comparing 4 centers to now only 2. Is there a reason for that? And why did you choose to compare with the UKMO?

We were trying to avoid a beauty contest, together with the difficulties in getting all the historical initialisation details correct. Since both reviewers have asked, we do now include all 4 models, and have discovered that the TIGGE archive provides the necessary initialisation information.

L233:

In Rodwell et al, 2018 you showed (Fig. 1) that the (traditional) spread-error relation is perfectly matched for the Northern Hemisphere at any forecast lead time over an entire year (2014). Hence (if this is still true, is it?) the overspread you now show for the stormtracks in the winter must be compensated by an underspread at some other location or some other season. It would be interesting to investigate this (see major comment above).

We have replied to the major comment above. We will discuss this in the revised manuscript.

L238:

I suggest to explain the K-mean clustering method with a couple of sentences.

We will include this explanation up-front. It was alluded to on L248, but not sufficiently.

L245:

Why do you weight with the root? Isn't the grid cell area scaling with cos(lat)?

Because the clustering method is on the variance, which re-instates the square.

L264:

Does this mean you are combining the cluster1 cases from both clustering areas? Could you further justify this approach? The shift of the region doesn't seem that large. Would one clustering analysis based on a combined region lead to similar results? What about separating clusters by the surface pressure tendency in the region? Wouldn't this be a simpler method more directly related to cyclogenesis?

Yes, we are combining cluster 1 cases from both clustering areas to produce a 'cyclogenesis composite'. We have made this clearer now. We used the clustering approach (and the smaller regions) to give some coherence in flow structures, and to be consistent with the fields displayed in Figs. 2 and 3 (and the animations). To a large extent, the results justify the approach – we see the structures in the clusters and obtain statistically significant differences between the two composites. Other approaches, such as the one suggested by the reviewer, could also be successful and might be amenable to the use of a single combined region (indeed the whole storm-track). We suggest that this could be tested in future studies.

L293:

To me this statement seems a bit exaggerated. I would say that the overspread is reduced in the cyclogenesis composite. Also if I read the colors correctly, the residual difference does not reach statistical significance. Why is the overspread enhanced in the counterpart over the central/east Atlantic. Is it because cyclogenesis is shifted downstream in the counterpart cases? Is this spread reduction in cyclogenesis events also happening at other centers (see major comment above)?

We have moderated and extended the text to say "much of the over–spread in the region of focus, at the head of the North Atlantic winter storm-track (Fig. 7e), is associated with the cyclogenesis composite – with statistically insignificant residuals (largely light blue and grey) in Fig. 9j and statistically significant residuals (largely dark blue and purple) in Fig. 10e. Differences are shown in Fig. 10o. Over Newfoundland in particular, they are comparable in magnitude to the full departures in Fig. 10a and statistically significant. Downstream, differences in the residual have the opposite sign – possibly associated with differences in downstream cyclogenesis, and consistent with the increased spread noted above."

We have not examined the other models with this composite approach. This would require more work and a lot more explanation.

L296:

I did not understand this paragraph.

This paragraph was a note about conditional sampling. In the interests of brevity, we have removed the paragraph. The key text remains at L258: "(Note that the growth rate is not used within the clustering algorithm since this is what will be evaluated)".

L313:

I notice you leave the convection scheme on at 4km resolution. Could you explain why? Usually only shallow convection is used at such high resolutions (e.g. Judt, 2018).

The 4 km experiment with the parametrization of deep convection turned off was also run. Turning off this parametrization at 4 km leads to a further enhancement in D+2 uncertainty in PV315 along the southern extreme of the cold front. It was concluded that, even at 4 km, forcing the ECMWF model to resolve convection can be unrealistic. This is an area of active research by others.

L324:

I don't understand this sentence.

We have changed the sentence 'The Z250 spread represents the temporal integral of the local tendency, and hence the effects of "material" generation and advection.' to "As one might expect, the maximum spread is not co-located with the maximum 'Lagrangian' growth rates (cf. Fig. 4a)."

L333:

"attempts to resolve". This is misleading since the resolution is still 18km, right?

There is an understandable misunderstanding in our use of "resolve". We have changed this to "that would otherwise be created when the model is forced to represent this convection on its 18 km grid". Thank you.

L338:

I am not sure about the relevance of this increased spread. It could just be a consequence of slightly displaced and explicitly resolved updrafts. Would there also be any enhanced spread in e.g. the precipitation averaged over the front?

We agree with the reviewer here, and have modified our tentative explanation on L339-340 from "The smaller scales associated with PV315, and its sensitivity to vertical gradients in diabatic heating in the upper troposphere might help explain this sensitivity to the increase in resolution" to "At 4 km, the model attempts to resolve more of the convection. The resolved convection can be associated with stronger updrafts, which might perturb the tropopause more vigorously, where PV gradients are particularly strong". There is likely to be enhanced spread in precipitation within the frontal region, although we have not looked at this. It is difficult to say whether the precipitation averaged over the front will be more or less uncertain.

L342:

Possibly there are now explicitly simulated updrafts which are slightly displaced among the ensemble members and generate grid-pointwise spread. Again I am not sure how relevant this is. I don't think this is the kind of uncertainty SPPT was designed to account for. So I am not surprised to see less effect from SPPT in this region.

This comment is very thought provoking. We have changed the text on L342-343 from "Forcing the model to explicitly resolve this convection (even if at the wrong ~18 km scale) appears to better locate the uncertainty with that generated by the ~4 km model" to "The impact on PV315 uncertainty of allowing the model to resolve more of the convection at the 4km resolution (Fig. 13e)

appears to be in closer agreement with the response to turning off the deep convection parametrisation (minus Fig. 13d), when the model is forced to represent the convection on the 18km grid".

L379:

If a reduction of singular vectors would make forecasts more consistent then why are they still used? I suspect they do show a benefit at a different location, flow regime, lead time, etc. I think you should discuss these aspects in more detail and also possible alternatives (e.g. inflating SPPT or EDA, using SPP, higher resolution, etc). See also major comment above.

The SVs are particularly efficient at generating spread (by day 2) at synoptic scales (see the power spectra plot on p. 2 of this document). Recent spread-error spectra produced by the lead author suggest that this spread is now somewhat too large. The MU also produces spread at synoptic and planetary scales – at the planetary scales the spread-error agreement is much better. Hence this is evidence that a reduction in the magnitude of the SVs would be useful. We do now discuss the advantages of the SPP framework in the modified final sentence: "It is possible that such a focus on instabilities (rather than the effects of already-triggered instabilities) might be better explored within the future 'stochastically perturbed parameter' (SPP) framework for model uncertainty – perturbing triggering thresholds for example."

L381:

This would only make sense if the observed increased spread with resolved convection does not mainly result from rather small displacements of individual updrafts. But this has not been investigated (see related comments above).

Please see our reply to the reviewer's comment on L342.

Minor comments:

Fig. 1:

The figure is hard to read and evaluate. I suggest to use a color for the >2PVU regions and to omit the red hatching, since it is kind of obvious and does not add any extra information. As labels of the panels I suggest "Case 1", "Case 1, +24h" or something like this for easier identification.

We only show one case now. We will experiment with the colouring suggestions.

Fig. 2 and others:

There is a lot of doubling between the figure caption and the text. I suggest to not repeat details in the text that are already included in the caption.

We will try to eliminate duplication – this might be easier now because some of the explanations are brought from the appendices into the main text.

L132:

"Single frame of animation" is not a good description of the plot.

This has been changed, thanks

Fig. 3 caption:

I think you meant case 2.

Yes thank you.

L175:

Better "sampling from a population"?

We prefer "underlying distribution", as that is what is drawn in the figure. However, they mean the same.

L178:

Any reason why you call this "departure"? It is just a difference, isn't it?

The word departure comes from the world of data assimilation. It is used instead of "error" because the analysis (or observation) does not represent the truth. All the lines are differences.

L180:

"number of forecasts" is ambiguous. I suggest "cases".

We have better defined what we mean by "number of forecasts" now.

L182:

I suggest to remove the 2-superscript (looks like a footnote). It is clear from the context that you are considering squared quantities.

These superscripts are used to distinguish with the rooted terms.

L191:

Remove (.

Done, thank you

Fig. 7 (and others):

The color bars are misleading to me. First I would appreciate if the color bars in panels a-j and k-o were identical. The main point of the figure is the comparison and identical color bars will help with that. Also it is misleading to color small positive value with saturated dark colors (e.g. panel c, it looks like a massive AnUnc). I suggest to use reddish colors for positive values, blueish colors for negative values and gray/neutral around zero (e.g. like you did in panel d).

The magnitude of these terms depends on the reliability of the forecast and the lead time. In some configurations, the departures and spread might be an order of magnitude larger than the bias and residual but, of course, the difference between the departures and the spread (the measure of unreliability) has the same order of magnitude as the bias and spread. Hence we really need to be able to see these latter terms in some detail. Moreover, the bias and residual can be of either sign, while the departures and spread are always positive. Hence, there is no clear reason to plot these terms with the same contour intervals.

Fig. 8:

The green geopotential lines are very hard to see. Please make them more prominent.

We will thicken these.

L273:

"Head of the stormtrack". This term is not clear to me. From the context I guess you mean the start/beginning (west).

We will make this clearer.

Fig. 11:

I find the arrows more confusing than helpful. Also: CP->DP

We will change to DCP! We have experimented and feel that the arrows help.

Fig. 12:

I suggest to also revise the color bar. Panel a) shows a positive variable and should use neutral to reddish colors. Panels b)-f) should have the same color bar since this makes it much easier to assess the individual contributions. I cannot really distinguish gray from black contours.

We will thicken the black PV contour. We make it clear throughout where the contour intervals change – we feel it is important to see where a particular change has an impact, even if it is small compared to that of another change.

L328:

) missing.

We will add this, thanks.

The authors would very much like to thank the reviewer for the time and insight that they have given to this review process. We feel that changes made have led to useful improvements.

References:

Edward N. Lorenz (1969) The predictability of a flow which possesses manyscales of motion, Tellus, 21:3, 289-307

T. Palmer et al, 2014: The real butterfly effect. Nonlinearity 27 R123

Judt, F. (2018). Insights into Atmospheric Predictability through Global Convection-Permitting Model Simulations, Journal of the Atmospheric Sciences, 75(5), 1477-1497.

Zhang, F. et al, 2019: What is the predictability limit of midlatitude weather?. Journal of the Atmospheric Sciences, 76(4), 1077-1091.

Citation: https://doi.org/10.5194/wcd-2022-6-RC2

---

## Author Response (AR1)

wcd-2022-6

**The Cyclogenesis Butterfly: Uncertainty growth and forecast reliability during extratropical cyclogenesis**

by Mark J. Rodwell and Heini Wernli

**Replies to the reviewers' comments**

The authors would like to thank both reviewers for the time and care that they put into reviewing this manuscript, and for their insightful comments, which helped to improve the clarity of the presentation of our results. Before addressing the individual comments in detail below, the main changes compared to the original submission are briefly summarized:

- Large parts of the text have been rewritten to better explain the objectives of the study, to streamline the logic of the paper, and to increase the clarity of the analyses.
- Parts of the text have been shortened or omitted (e.g., discussion of link to Liouville equation).
- Only one case study is shown, as suggested by the reviewers.
- Reliability has been evaluated for all four TIGGE models.
- Two dates are now shown for the TIGGE models to give a better reflection of the range of results.
- The mathematical parts have been better integrated in the paper.

**Reviewer 1**

Notes on "The Cyclogenesis Butterfly: Uncertainty growth and forecast reliability during extratropical cyclogenesis"

**Overview**

This manuscript addresses the very interesting problem of flow-dependent variability in ensemble reliability. Such an analysis is of significant practical utility because it gives ensemble designers important robust insights into their system's behaviour under identifiable meteorological conditions. Specifically for ensemble applications, such information is an essential replacement for the case study approach – although arguably conditional evaluations should also be preferred for deterministic systems.

It is clear from the breadth of the analysis that an impressive amount of work has gone into this investigation. However, the manuscripts suffers from the lack of a clearly stated objective for the complex diagnostics employed. As a result, the text gets mired in technical discussions rather than focusing on interpretations and discussions that support the objectives of the work and advance the main narrative of the manuscript. Similarly, many of the novel diagnostics themselves (for example Eqs. 1 and 2) seem to be overly complex for a study that arrives at relatively straight-forward – though very useful – conclusions regarding conditional overdispersion in the ECMWF ensemble.

As described in General Comments #1 and #2 below, I think that this work is interesting and important enough to be split into two separate manuscripts. The result will be two independent but complimentary studies that better motivate and demonstrate the utility of the proposed techniques. Such a reshaping of the investigation will also permit the introduction of more synthesis and interpretation of the results, resulting in a pair of papers that will have a larger impact on the field.

Recommendation: Resubmit after splitting the study into two separate manuscripts. Reviewer:

Ron McTaggart-Cowan

The authors would like to thank the reviewer for the time and care that they put into reviewing this manuscript, and for their insightful comments. We have tried to answer their comments below – with a better statement of the objective and less distractions. We will argue that both equations are important for this study. We also hope that this work motivates further flow-dependent evaluations, where these equations could be a useful reference. In our replies below, numbering for lines, figures, tables and equations refer to the revised manuscript unless otherwise stated.

**General Comments**

1.  This manuscript presents a huge amount of material and it is clear that an awful lot of work has gone into this analysis. However, I think that the vast array of content actually reduces the potential impact of the study. Stronger curation of the information would focus the manuscript – and the reader – on the truly important elements of the work that lead directly to the conclusions. One way to start improving the focus of the study will be to identify and clearly state the objective of the work. That could effectively be done at the start of the last paragraph of the introduction. I

encourage the authors then to take a serious look at each element of content and decide whether or not it is essential to advancing the manuscript towards this objective. Components that do not fit into this focus should be removed and could probably form the basis for a separate submission.

Both reviewers made this point, so we have worked to make the statement of objectives clearer in the revised Introduction. We did give careful thought to the structure of the manuscript. In particular, we did try to cater for specialists and non-specialists by putting technical details in the appendices. This has also clearly failed, and we now have tried to improve the structure by using sign-posts in the main text. In the light of the reviewers' comments, we have removed discussion of the connection to the Liouville equation, and now only show one of the case studies. The appendices have also been reduced.

2. In the end, I think that this is really two papers. The first paper is about ensemble-estimated uncertainty growth rates and their relationship to cyclone intensification and/or trough amplification over the western North Atlantic. The second paper is about documenting and identifying the source of overdispersion in the ECMWF ensemble in the North Atlantic storm track. Although the second is clearly motivated by the first, these topics are separate enough that they would not even need to be a two-part submission: they could be treated entirely separately. Having two separate papers would allow for an expansion of discussions and dynamical interpretations, in addition to the introduction of important material into the main text that is currently relegated to the multiple appendices. I really think that the prodigious amount of effort that clearly went into this analysis would be much better served by two independent submissions.

Clearly there are several parts to this study. We have thought hard about splitting these up, but they really are strongly connected. Equation 3 (left hand side) is used to produce maps (and animations) of growth-rates. We focus on synoptic scales because these show the largest contribution to the overall variance growth over the first two days – as shown in Fig. B1. The maps demonstrate that growth-rates are concentrated into specific synoptic flow situations – identifying the extratropical flow-types others have linked to poor forecast skill. Hence, we suggest that it is a useful diagnostic for the study of flow-dependent predictability. This motivates the investigation into cyclogenesis, the title of the paper "the cyclogenesis butterfly", and the evaluation of flow-dependent reliability. Note that the growth involves diabatic effects and interactions between all scales – as demonstrated by Eq. 3 right-hand-side. We now make it clearer that this will depend on the scales of initial uncertainty, which will be different for intrinsic predictability studies, the representation of model uncertainty, and any use of initial SV perturbations. The question then arises as to whether other models display the same growth rates – we show that they are not that similar. To get a view of which might be best, we compare the models in terms of an extended spread-error equation. Following the other reviewer's advice, we now include all four models in this comparison. After showing that the ECMWF ensemble is over-spread in cyclogenesis events, we go on to investigate what might be done to improve this with a set of sensitivity experiments. These experiments are discussed in terms of Eq. 3 and suggest we could reduce the use of singular vectors. This would give us a more seamless system, and thus allow a better evaluation of model uncertainty and the key physical/dynamical processes driving the growth rates. We have tried to strengthen the motivation and links between the sections to justify keeping this as a single study.

3. [This comment is only directly relevant if the current submission is not split into two separate manuscripts.] Organizing the paper into 11 sections is highly unusual. Although I appreciate the use of sections and subsections as important tools for organizing content, I think that in this case there

are so many sections that readers will lose the "big picture" of the manuscript's organization. To a certain extent, the excessive number of sections appears to be a symptom of a stream-of-consciousness design. Rather than presenting the work in the order that it was executed, consider reorganizing it into larger logical chunks for the reader. For example, the extremely short Data section (2) should be augmented to include the methods currently described in sections 4 and 6, and part of section 8. It seems like sections 3, 5 and 10 would be more logically grouped as a single (case study) section with appropriate subsections. Sections 7, 8 and 9 should also be considered subsections of a "full-season" analysis section. The result would be a 5-section paper: (1) introduction, (2) data and methods, (3) case studies and sensitivity tests, (4) full-season analysis and model intercomparison and (5) conclusions. I believe that such a reorganization would really help to increase the potential impact of this study on the field.

We apologise if the study comes across as a stream of consciousness. We thought hard about the structure, and it is not presented in the order it was done. With several methodologies used in the study, we did not think it appropriate to place these all into a single section near the beginning, but rather discuss these only once they had been motivated. It is considered important to introduce the case studies at the beginning (now only one shown) because this gives a concrete example of what we go on to aggregate. However, it does not seem appropriate to place the sensitivity studies before the over-spread had been identified, because only then do solutions to the over-spread need to be considered. Furthermore, the sensitivity studies point to future avenues of research. We have tried again to improve the flow of the paper by grouping subsections, so that there are now only 6 sections, and including better motivation. We hope this is acceptable to the reviewer.

4. The two case studies appear to yield similar results. If the current document is to be revised as a single submission, one of the two case studies could be relegated to supplemental material. The main text could then claim demonstrable robustness with reference to the results shown in the supplement. If the material will be split into two independent studies (General Comment #2), then the two case studies could be retained in the first paper, along with augmented evaluation and interpretation.

We have followed the first suggested course of action here, and think this does lighten the manuscript, thank you.

5. I think that a study of finite perturbation growth rates that cites the "butterfly effect" should mention Durran and Gingrich (2014), although I understand that the perturbation scales discussed here are much larger than the near-truncation scales found to be "unimportant" in the 2014 study (indeed, you mention this in your 2018 BAMS article). Perhaps this suggests that the "cyclogenesis butterfly" is a bit of a misnomer and (although catchy) might introduce some confusion: these are **very** big butterflies.

Yes, these are big butterflies. As discussed above, our justification comes from the observation that growth rates in the first few days are largest at synoptic scales, and these growth rates appear to be orchestrated by specific synoptic flow-types. We are not discussing the Butterfly Effect (interpreted as an intrinsic predictability limit) and we make this clearer now. We did cite the Durran and Gingrich paper in the original manuscript, but this paper is cited more extensively now.

6. Based on the time periods discussed in the case studies, I think that "rapid cyclone deepening" would be a better description of the uncertainty precursor than "cyclogenesis". Both cyclones form 1-2 days before the period of interest, but intensify rapidly over the Gulf Stream. I think that the

distinction is important particularly in this region, where secondary cyclogenesis (i.e. the formation of a cyclonic circulation where none existed previously) is common and could easily be misunderstood to be the "butterfly". Clarifying the focus on rapid deepening of preexisting cyclones (if I am right about that) further emphasizes the fact that this study is looking at synoptic-scale uncertainty seeds, rather than the potentially mesoscale cyclone development precursors.

It seems difficult to define where a wave becomes a pre-existing system. In the composites, we identified cyclone structures, but the initial conditions were taken two days prior to these. The variance power-spectrum shown in the figure above indicated that the largest contribution to ensemble variance at initial time is from scales around 400 km (the diagonal lines indicate variance power per linear-unit distance on the x axis). It is likely that interactions between these mesoscale uncertainties, along with interactions at larger scales, play a role in the synoptic uncertainty that develops by day 2. We have added more discussion of this (see also Appendix B).

7.  Although the breakdown of the Lagrangian growth rate into "non-conservative" and "advective" components (Eq. 1) is interesting, it does not seem to have any impact on this work. The analysis appears to proceed to look at only the Lagrangian growth rate itself, i.e. the l. h. s. of Eq. 1 rather than the forcing terms. If this is true, then the focus of the manuscript can be tightened by removing Eq. 1 and associated discussions, including most of appendix B (the remainder should be included in the augmented "Data and Methods" section, particularly if Z250 is adopted throughout as recommended in General Comment #13).

We agree that the right-hand side of Eq. 3 was only briefly discussed before. We have added further discussion of the right-hand side of Eq. 3 – in particular, we emphasise how it represents multi-scale interactions and discuss the relationship to PV advection and generation within cyclogenesis events. The impact of multiscale interactions is evident in the blue shaded region of the figure shown above, where the representation of smaller scales in the 4 km experiments leads to increased ensemble variance even at scales already represented in the 18 km model.

8.  The study references animations periodically. This means that readers will need to interrupt their progress to look at animations available in supplemental material. As far as I can tell, most of the relevant information could be presented as additional panels in the existing figures. For example, Figs. 2 and 3 are both single panel, but could be augmented to show other lead times to avoid the need for references to separate animations in the text.

We have dropped the second case study but, for the TIGGE comparison, we show an additional date (Fig. 4) where the models' growth rates agree better. This allows for a more balanced discussion of their comparison which better represents the many cases in the animation.

9.  Differences in the ensemble perturbation techniques between the different modelling systems investigated here seem potentially important, particularly given the short lead time. The use of SV perturbations in ECMWF ENS distinguishes it from most other systems in the TIGGE database, other than perhaps JMA. A discussion of these differences (or at least their itemization in an introductory table) would be very useful.

We have included a table (Table 1) of these details.

10. This study looks at uncertainty (ensemble spread) growth rates from the perspective of synoptic cyclone dynamics. To make a convincing connection between the uncertainty growth and cyclone development it would be very useful to compare the former to something like the moist baroclinic growth rate (e.g. Booth et al. 2015; ASL). A high degree of correlation between the two would be

good evidence of the importance of rapid cyclone deepening to spread growth in the ensemble. Even something relatively simple like comparing the time series of area-averaged (over the Gulf Stream region) ensemble growth rates and moist baroclinic growth rates (with rapid deepening events identified) would provide a really nice dynamically based assessment of the importance of cyclone development to uncertainty.

This is an important idea. The authors are aiming to address this more fully in a future study focused on Eq. 3. We have discussed how this relates to cyclone development from the PV perspective in Sect. 3.2 and discuss the potential for future work in Sect. 6.

11. The maximum uncertainty growth region in Fig. 2 is upshear of the trough axis, where vorticity advection is negative aloft. Why is this? In both cases (Figs. 2 and 3) the cyclone is located between the dipole in growth rates, not at all within the peak growth rate south of the trough. This is not "ahead of the base of the upper-level trough" or "preceeding cyclogenesis" (line 142). I understand that some amount of spatial smearing arises from the use of 12-h differences to compute the growth rates, but the cyclones do not even appear to move through the maximum growth rate region. So then would it be more accurate to link large spread growth rates to amplifying upper-level troughs rather than cyclones per se? For example, perhaps uncertainties in the strength of the jet streak on the upshear flank of the trough (associated with its meridional extension) are more important than the lower-level cyclone itself.

The 'Lagrangian' growth rate plotted does not include the advection of ensemble variance by the ensemble-mean flow. This advection is a major term and would immediately lead to Eulerian growth rates more aligned with the cyclone. We now make this clearer in the text. It is interesting that the Lagrangian growth rate highlights the upper-level trough region. Please see the previous response and new text in Sect. 3.2.

12. The bulk of discussions around the spread-error relationship appear to focus on the Spread and Residual terms of Eq. 2, leading to conclusions about overdispersion in the North Atlantic storm track. Is there no simpler way to arrive at the important conclusions of the study without going through this rather complicated derivation and analysis? The interesting flow-dependent aspect of the spread-error relationship is achieved through independent stratification (currently via cluster analysis), so I think the only thing that might be lost would be the conditional bias shown in Fig. 10i. However, this bias could be evaluated directly and shown to contribute significantly to the increased RMSE in the "counterpart" cluster without resorting to Eq. 2. The apparent ambiguity of the Residual term makes the discussions surrounding Eq. 2 quite difficult to follow and appears to make it difficult to make definitive statements about sources of problems within the ensemble. If the important message to be delivered by this work relates to the flow dependent overdispersion in the ensemble, then a simpler analysis (perhaps including regional and/or flow-stratified spread-reliability diagrams) might be a more effective vehicle. However, if the current investigation is just a showcase for the analytic technique itself then (a) that should be clarified and (b) the advantages of this technique over a simpler analysis should be emphasized.

The intention is to showcase the technique and to consider carefully the assumptions made, so that it can be used in future (flow-dependent) evaluations of ensemble forecasts as well as here. Note that we are not just showing that there is a mismatch between MSE of the ensemble mean and variance of the ensemble; we are showing that the mean bias and analysis uncertainty cannot account for the mismatch. While the stratification would highlight the importance of bias in the non-cyclogenesis cluster, it is not immediately obvious to the reader that variance in bias would be

important in the non-stratified budget. If we understand correctly, we would argue that spread-reliability diagrams make the same set of assumptions. What we have done is to highlight to the reader just before Sect. 4.1 that they do not need to go through the derivation, and that the essential summary of the equation is given in Sect. 4.2.

13. The lack of PV in the TIGGE database requires the use of Z250, which appears to produce similar results (Figs. 2-5). Although I can completely understand the appeal of starting with PV in this discussion, I think that for pragmatic reasons the entire study should focus on Z250. In the Data and Methods section the rationale for this can be very clearly explained. This would only really affect current sections 4 and 5. The PV 315 diagnostics in (current) section 10 could still be used because they are separate from the growth rate discussion.

    We agree that changing between PV315 and Z250 is not ideal. Nevertheless, we consider that it is important in order to discuss Eq. 3, where the right-hand-side requires PV. There is more extensive discussion of this equation now. We have simplified the case studies, which should help a little. Emphasizing the lack of PV in the TIGGE archive is also considered important to motivate its inclusion at some point.

14. Why was the clustering approach (current section 8) preferred over a much simpler cyclone identification approach? It seems as though clusters 2 and 3 for both domains are lumped into the "non-cyclogenesis" category when the results from the two domains were aggregated. As such, this seems like a very complicated way to identify dates with cyclones in the western North Atlantic.

    It is important to have an objective approach to classification, and we wanted the data to 'speak for itself' by showing the structures that emerged. In the end, the results are probably very similar to cyclone identification (as was alluded to in the manuscript). Many of the events in cluster 2 (44 out of 75) for region 1 do find their way into cluster 1 for region 2.

15. I am not sure grammatically why "growth-rate" is hyphenated throughout. This does not seem to be a common construction.

    We now use "growth rate".

16. I do not believe that forecast "lead-time" is usually hyphenated. More generally, there appears to be over-hyphenation throughout the text. Please limit the use of hyphens and ensure that they are represented using hyphen characters rather than the current em-dashes.

    We will consult on this.

17. Please confirm that date/time formatting conforms with WCD standards.

    Thanks, it now does.

**Specific Comments**

18. [L45] Distinguish between the true unstable modes of the flow and the computed singular vectors (optimal tangent linear growth with limited moist physics). The note about the "linear regime" points in this direction, but it would be useful to make this distinction right off the bat.

    We have removed this link to SVs in the introduction, and now distinguish SVs from intrinsic growth rates in Sect. 3.2.

19. [L48] It would be useful to itemize some of these approximations here because the difference between ensemble spread growth and error growth rate is fundamental to this study.

The differences are more clearly discussed in Sect. 3.2. Although not itemized in a list, they are separated into the two growth rate aspects (dynamical interactions and non-conservative aspects).

20. [L52-54] The punctuation of this sentence makes it difficult to follow: consider rewording.

This has been re-worded at L51-55, and is more clearly discussed now at the beginning of Sect. 4.

21. [L54] Remove hyphen from "ensemble-mean".

This has been removed throughout.

[L55-56] Replace "Jetstream" with "jet stream", "wave-guide" with "waveguide", and "down-stream" with "downstream".

We have done this.

22. [L57-72] This "outline" paragraph is overly long and complex because it strays into "abstract" territory by summarizing results. Consider shortening this paragraph by restricting its content to section descriptions only.

This has been shortened as suggested.

23. [L58] Provide a reference for TIGGE if it is to be mentioned here. Also confirm that this acronym can be used without definition in WCD, or define it.

The reference to the Swinbank et al. paper has been moved to L49 - the first place TIGGE is referred to (after the abstract).

24. [L73] Suggest dropping the first two sentences of this section and including all dataset descriptions here so that the flow of the remainder of the text is not interrupted by them. As noted in General Comment #1, this section should be rewritten to include information about the datasets and methods used throughout the study.

Section 2 on "Models, data sources and key parameters" has been extensively re-written to describe the datasets used and details of the TIGGE models. Both reviewers have mentioned including the methods here. The problem is that it would require so much motivation up-front. We hope that the revised approach, with better sub-sectioning later-on works better now.

25. [L73] I believe that "re-analysis" is more usually "reanalysis", including in Hersbach et al. (2020).

We have changed this.

26. [L75] The forecast range of the background does not seem to be identified here or in Appendix E. It seems to be 12 h (line 136), but that should be clarified here.

It is now clarified at L187 (12 h)

27. [L77] TIGGE stands for the "THORPEX Interactive Grand Global Ensemble".

The words that TIGGE stands for have changed over the years. The "I" now stands for "international" (please see: https://www.ecmwf.int/en/research/projects/tigge)

28. [L80-83] This information would probably be better displayed as a table for easier reference in later sections.

This has been done.

29. [L84] Suggest, "These data are used …".

The re-wording has removed this issue.

30. [Fig. 1] Are the trajectories that are used to identify the WCB region extending from -24h to +24h from the analysis valid time (i.e. these are the trajectory midpoints)? Suggest using the "red hatching" term consistently in the caption, rather than "shown in red".

The WCB region shown at a particular time $t*$ is based on all WCB trajectories, which, according to the WCB identification method by Madonna et al. (2014), are within the layer from 800 and 500 hPa at $t*$. Since this method selects trajectories that ascend at least 600 hPa in 48 h, this means that the WCB region can be, in principle, based on trajectories calculated during all 48-h periods from [$t*$ - 48 h, $t*$], [$t*$ - 42 h, $t*$ + 6 h] to [$t*$, $t* + 48 h$], but given the fact that most trajectories ascend from about 900 to 300 hPa, there will be most likely no contributions from the very early and late of these periods, and as suggested by the reviewer, the bulk of trajectories shown at $t*$ are expected to be from the periods [$t*$ - 24 h, $t*$ + 24 h] and 6 h earlier and later, respectively. This explanation may sound complicated, but the interpretation of the WCB region shown in the figures is not: it indicates the region where WCB trajectories, irrespective of their exact start and end time, are ascending across the mid-tropospheric layer when they produce most of the latent heating and precipitation formation.

We have changed the text to "red hatching".

31. [Fig. 1] Should the mks form of PVU be provided in the caption?

    PV is now better introduced in Sect. 2.3.

32. [L104-106] Are these the forecast experiments discussed in section 10? If so, then this is additional motivation to move that section up as a "case study" subsection.

    We have argued, in response to point 3, that the case studies (now one) need to be up-front to give concrete examples, and then revisited once the result of the over-spread has been presented. Two examples are not enough to say anything definitive about reliability, but the similarity between the two cases suggests that two are sufficient to say something about the sensitivities of ensemble variance.

33. [L108] Suggest "… uncertainty grow-rate estimate …" because the ensemble provides only an estimate of the true forecast uncertainty.

    Agreed, we change to "can be estimated as".

34. [L109] What does the "1-dimensional" restriction mean here? Would this be better identified as "scalar", or can multiple state variables be included in a 1D state vector? This is obviously important because it reappears elsewhere in the text.

    This was a mistake and both reviewers suggested different solutions. We have changed "some 1–dimensional state–measure (of the atmosphere)" to "some atmospheric parameter field". Thank you.

35. [L114] The phrase "but with a different formulation" is too vague.

    We now make this clearer on L154.

36. [L118-124] This is a very complex sentence mixes conservative and non-conservative forcings in Eq. 1. It would be more useful to split this sentence to describe the physical relevance of the terms on the r.h.s of Eq. 1 individually.

    Since this relates to potential future work, it is discussed in Sect. 6 L533-542. By this stage, the conservative and non-conservative aspects have (now) been discussed extensively.

37. [L126] Should "Equation" be capitalized here? It wasn't in section 1. I do not think that the back-reference to section 1 is very useful here because the introduction did not go into much additional detail about the Liouville equation. A citation to relevant literature would be more useful here.

We have removed the link to the Liouville equation after considering the comments of both reviewers on the topic of inherent predictability limits.

38. [L125-130] I think that this discussion is fine, but it does not seem to advance the main thread of the study. It could be dropped to reduce the length of the manuscript.

It has been dropped, as discussed above.

39. [L132-140] This information should be contained in the captions (most of it is) and/or left for supplemental material because it disrupts the flow of the main text.

We have included as much information as we can in the captions and point to this at L188. The discussion in the text hopefully now keeps the reader engaged where details (e.g. of filtering) are required.

40. [Fig. 2] What is the contour interval for the contours showing extreme values?

We stated that "Contours extend the shading scheme to the most extreme values, which are indicated at the ends of the colour bar". We now clarify a little more with "Note that orange and blue contours extend the shading scheme, with the same interval. The most extreme values are indicated at the ends of the colour bar".

41. [Fig. 3] Should this read "Case 2"?

Yes, it should – thank you. Note that we are now only showing one case study in the main text.

42. [L145] Is this a third case study being introduced? I think that discussion of the full-season perspective should be left for the subsequent section (in the reorganized paper).

This text referred to an example in the animation in the supplementary material, which was mentioned on old L139. We now do not mention the animation until later (L198-207) and it is not so necessary to the flow of the paper.

43. [L145-150] These seem like "future work" suggestions that would be better left for the concluding discussion.

Yes, agreed, it is now in the conclusions Sect. 6.

44. [L152-155] The 12-h forecasts from the TIGGE database are for ENS rather than EDA, is that correct? If so, then is it true that Figs. 4a and 5a look different from Figs. 2 and 3 not only because the field is different but also because the perturbations are different? If I understand the ECMWF system correctly, SV perturbations are not added within the EDA cycle, but are added before ENS initialization. In that case, Figs. 4a and 5a have an additional source of optimized growth. That seems to make the comparison interesting, although it is complicated

by the change in diagnostic field. Would it not be surprising if the SV perturbations have little impact on growth rates in these cases? Perhaps the Z250 growth rates could be shown for Figs. 2 and 3 to make this comparison possible.

The reviewer is correct that the ENS is used from TIGGE and that, for ECMWF this includes SVs (and SPPT) while the EDA does not include SVs (but does include SPPT). We did discuss this in the context of improving seamlessness in order to be better able to diagnose the (2-day) growth rates of the model (and its uncertainty representation) from short-range forecasts. We make this distinction clearer (L219-220). Similarities between the EDA PV315 growth rates and the ENS Z250 growth rates suggest that there are strong growth rates even without the explicit inclusion of SVs, but we note (L220-222 and L470-471) that there is a slight westward shift of the maximum growth rates in the

presence of SVs. While EDA Z250 growth rates would be interesting, we feel that they would complicate the manuscript unnecessarily.

45. [L155-156] So are these case studies (particularly Fig. 4) not representative of the general behaviour of these models?  If so, perhaps another case study should be chosen for this comparison.

The main point being made is that they disagree in terms of growth rates. We now also show a case (Fig. 4) where the agreement is better, to give a more balanced impression of the animation.

46. [L172-174] The source of Eq. 2 (appendix C) should be cited at the beginning of this discussion.

We have brought this Appendix content into the main text (as discussed above), tried to make it more accessible, and sign-posted that the reader can avoid if desired.

47. [L181-182] I have a hard time understanding a lot of this discussion and how it relates to Fig. 6.  It would be great to label the lines in Fig. 6 with the names of the terms in Eq. 2 that they relate to. The lines seem to be more directly related to the discussion in Appendix C, so perhaps Fig. 6 would be more appropriate in the appendix.

We did experiment, but it is difficult to label all the lines in Fig. 6. For example, there are two lines which contribute to the forecast variance term in the final equation, and two which contribute to the analysis variance term. We have added a couple of lines to indicate the spread of the two reliable distributions. The reviewer is correct that the figure relates more directly with the variables in Appendix Eq. C1. We have moved the derivation to the main text, alongside the figure, and made it possible for readers to skip this if they choose.

48. [L192] Does this "main additional term in the Residual" refer to Eq. C5?  If so, it would be useful to cite that equation here.

Yes, it is the last term in the old Eq. C5. This has now been brought into the main text, as discussed above.

49. [Fig. 7] The change in colour scale range for panels (n) and (o) make comparison of the plots on the bottom row difficult.  With the current plotting scheme, it looks like the difference in residual is almost entirely explicable by the difference in spread, but that is not really the case (is it)?  The contour intervals for values beyond the standard colour bars should be noted in the caption.

The reviewer is correct in their interpretation. In response to the other reviewer's recommendation, this figure now includes the four TIGGE models investigated, with no differences plotted.

50. [L219-223] It is challenging to follow this discussion because of two forward-references to a description of the variance of forecast biases.  It seems like that aspect of the discussion should be introduced before this text appears.  In fact, it is not clear what discussion the forward references here are actually describing (the section 9 discussion seems to take an understanding of the forecast bias variance's impact on the Residual for granted).

The variance in forecast bias, and its relevance are now discussed in Sect. 4.1, L294 and L303-304 (thank you).

51. [L233] It was not obvious that this is a "key question", so hopefully a clear statement of the study's objective(s) in the introduction will help to make that link more direct.

Hopefully so, motivation at the beginning seems to have been a key issue. We have tried to address this with changes to the Introduction.

52. [L233-235] Does this "either-or" statement arise from the form of the Residual term (Eq. C5)?  If so, then it seems like it would be useful to put this equation in the main text, hopefully as part of a

discussion on the meaning of "variance in forecast bias", which I think might be related to the "difficulties" proposed here (?).

The either-or statement is about whether the residual is associated with specific synoptic flow types (such as those with high growth-rates as in Figs. 2 and 3) or a more general issue. For example, associated with scale interactions with planetary wave uncertainties, or other scale interactions which might be more ubiquitous. We have tried to make this clearer from L364-367.

53. [L242-243] This region is quite complex: why would three clusters necessarily "provide sufficient degrees of freedom"? The optimal number of clusters is difficult to determine, but usually dropoffs in quantities like the AIC or BIC serve as some sort of semi-quantifiable justification for the number of clusters.

The aim was to balance realism of clusters with the need to obtain a sufficient sample size. This will be a compromise, but a broad ridge, a broad trough, and a tighter cyclogenesis flow-type seem to cover most eventualities. We try to make this balance of needs clearer now (L377-378). Note that later, on L402-404, we state that "visual inspection of plots similar to those for in Fig. 1 suggests that the objective clustering has been successful in partitioning the date/times into cyclogenesis and non–cyclogenesis flow types". In addition, the clustering was successful-enough to provide statistically significant differences between the partitioned date/times.

54. [L256-258] This is the only discussion of the uncertainty growth rate in this section, and it does not seem to lead to any particular conclusion. Is there a good reason to include it here and in the Fig. 8 and 9 plots? (It does not seem to be discussed in the subsequent section either.)

This is useful for two reasons: firstly it again demonstrates that the cyclogenesis cluster is associated with the strongest growth rates (note that most of the date/times for cluster 1, area 2 come from the date/times in cluster 2, area 1) and hence consistent with Figs. 2 and 3, and secondly it provides the opportunity to note that these growth rates are not used in the clustering (that could potentially bias the reliability assessment). We change "since this is what will be evaluated" to "since this could potentially bias the reliability assessment".

55. [L294] The phrase "almost the entire over-spread" seems like a bit of an overstatement. It is probably more defensible in terms of variance, but could perhaps be softened to "much of the overdispersion" or similar.

Since this is one of the major conclusions of the paper, we have followed the reviewer's suggestion, changing the text to "The overall assessment of ensemble spread is seen in the residual terms (Fig. 8e and Fig. 8j). Here it is evident for the ECMWF ensemble that most of the over–spread in the region of focus, at the western end of the North Atlantic winter storm-track (Fig. 6e), is associated with the cyclogenesis composite — with statistically significant residuals in Fig. 8e and statistically insignificant residuals (indicated by the light blue and light grey colours) in Fig. 8j. Differences are shown in Fig. 8o. They are particularly strong and significant over Newfoundland. Downstream, differences have the opposite sign — possibly associated with differences in downstream cyclogenesis, and consistent with the increased spread noted above. Linking the day 2 stormtrack over–spread (in the region of focus) to cyclogenesis is a key conclusion of this study. It does appear, therefore, that ECMWF initial growth rates (Fig. 2, Fig. 3a) associated with cyclogenesis events are too strong. The next section explores the root–causes for this problem."

56. [L297-301] I am afraid that I do not fully understand this discussion. How would the stratification of the groups (cyclone vs. non-cyclone) be done differently with multiple seasons or an independent assessment? Could this "regression to the mean" alternatively be considered a sampling bias?

One approach could be to deduce a composite of the (local) initial conditions leading to cyclogenesis within one season, and to then pick date/times from the same season in a different year which project strongly onto the initial condition composite. Since this approach has not been tested, it is difficult to predict how successful it would be. Regression to the mean refers to the fact that, if one sampling of a random variable is extreme, the next sampling is likely to be less extreme. In the interests of brevity, we have removed this paragraph.

57. [L302.5] Consider simplifying the section title to "Sensitivity experiments to quantify uncertainty sources".

We have changed to the suggested title – thank you.

58. [L315] I understand that resource constraints likely make additional tests difficult or impossible, but is it not conceivable that the ordering of MU and 4K is important? Systematic changes in the physics tendencies should be expected between 18 km and 4 km grid spacing (for example as more turbulent fluxes are represented by the dynamics), which will impact SPPT directly. This might mean that the impact of switching MU on and off at 4 km is different from what is observed in the 18 km configuration. I do not think that this is a big enough deal (or close enough to the focus of the paper) to justify additional simulations; however, you may want to put a bit more nuance in the wording of this statement.

Thank you for pointing this out. We had been thinking more about SPPT's action on diabatic processes (there is a shift from 'convective' to 'large-scale' precipitation at 4 km, but the total precipitation – and thus diabatic tendency which SPPT works on – is largely unaffected). The power spectra in Fig. B1 do highlight more resolved variance in the 4 km experiments. We have change the wording to (L452ff) "The conclusions are not thought to be sensitive to the ordering of the various modifications. For example, it will be seen that the impacts on total precipitation of DCP and +4km are small, and hence these impacts should be little changed in the presence of the SPPT form of MU. However, parametrized turbulent fluxes might be weakened with +4km, and hence this impact could be a somewhat different in the presence of MU".

59. [L318] Why not show results from the 1200 UTC 27 November 2019 initialization so that the day-2 forecast aligns with the panels shown in Figs. 1, 2 and 4?

The sensitivity plots show the impacts at day 2 of the growth rates that occurred previously – hence the difference in time.

60. [L326] Does the upshear maximum in the SV plot (Fig. 12b) really very well described as being in the "cold sector" of the cyclone? The cold sector is defined based on low-level airstreams but here the plot is showing spread differences in Z250. I think that this is much more related to the growth of perturbations in the jet streak on the upshear side of the trough, which is contributing to the "digging" of the trough / meridional amplification. Could the upper-level jet-front structure not an ideal place to have rapid SV growth (e.g. Hakim 2000; JAS)? By increasing vorticity at the base of the trough this feature will indirectly impact troposphere-deep cyclogenesis, but I think it is possible that the origins of the spread are more local. (The same is true for the second trough over the eastern North Atlantic that appears to be approximately equivalent barotropic.)

Thank you for these interesting suggestions. See the reply below to your comment 61.

61. [L327-329] The spatial separation of the SV and MU contributions is beautiful. I think that it is very understandable based on the previous comment and the fact that model physics is largely inactive in upper-level jet-fronts, other than perhaps some turbulence. The MU is focusing on the regions where the physics is active (lower-level cyclone and WCB) while the SV is picking up dynamic growth along the jet streak on the waveguide. If you agree with this assessment, it could be a useful inference to add to the text.

Indeed, these are interesting inferences. They are difficult to prove, but we agree with the reviewer that we should include these considerations in the description of Fig. 10 b,c. We therefore have changed the paragraph (L464-478) to:

"Figure 10a shows the OP configuration with a well–developed surface low pressure system, as discussed in relation to Fig. 1. The warm conveyor belt (WCB) associated with this cyclone is seen to lead to the development of a prominent downstream upper–level ridge and a downstream trough west of Europe. As might be expected, the maximum Z250 spread is located downstream of the maximum 'Lagrangian' growth rates (cf. Fig. 3a).

The impact of the initial SV perturbations on Z250 spread (Fig. 10b) is particularly pronounced along the western flanks of the two prominent troughs over the western and eastern North Atlantic, respectively. This likely indicates the potential for dynamic growth along the intense jets in these regions, qualitatively in line with the idealized studies by Hakim (2000). This SV impact might help explain the apparent slight westward shift of the centre of maximum growth in the ENS (Fig. 3a) relative to the EDA (Fig. 2). There are places where the SV impact on spread is half the total (so that the fraction of variance explained reaches 25%). In contrast to the SV impact, the impact of the model uncertainty (MU) representation (Fig. 10c) is particularly pronounced in the cyclone centre and in the region of the WCB ahead of the surface low, i.e., in regions where cloud-related physical processes are particularly active. The large signal along the western flank of the ridge southwest of Greenland is consistent with the results of Joos and Forbes (2016), who found a large influence of cloud microphysical processes in the WCB on the tropopause structure in this part of the downstream ridge. MU also explains up to 25% of the total variance. The remaining variance must be associated with the (deterministic) growth of initial EDA analysis uncertainty".

Hakim, G. J., 2000. Role of nonmodal growth and nonlinearity in cyclogenesis initial-value problems. J. Atmos. Sci., 57, 2951-2967.

Joos, H., and R. Forbes, 2016. Impact of different IFS microphysics on a warm conveyor belt and the downstream flow evolution. Quart. J. Roy. Meteorol. Soc., 142, 2727-2739.

62. [L328] Missing closing parenthesis for figure reference.

This has been added, thanks.

63. [L334-336] Discussion of total precipitation seems tangential to this study (also L344-345).

The observation that the total precipitation stays the same, but the spread is altered seems interesting. As discussed above, old L344-345 is useful in the argument about the ordering of the experiments not being too important.

64. [L346-351] This is the first time that observation location is discussed. The Obs experiment seems largely unrelated to the other experiments and should be eliminated to focus the study on the "controllable" sources of spread quantified in the other experiments.

The reason that the observational experiments are included is that they help us quantify what extra predictability is currently achieved through the assimilation of local observations, such as cloud-affected radiances. While they may not directly impact reliability estimates, they allow us to gauge the relative importance of working towards reducing the use of SVs. The juxtaposition here is also useful because it motivates a future more comprehensive study of the impacts of assimilating observations in cyclogenesis flow situations.

65. [L343] Suggest changing to "… appears to yield a better depiction of uncertainty than that generated by …".

In a combined response to both reviewers, we have changed the text L490-493 to: "The impact on P315 uncertainty of allowing the model to resolve more of the convection at the 4 km resolution (Fig. 11e) appears to be in closer agreement with the response to turning off the deep convection parametrisation (minus Fig. 11d), when the model is forced to represent the convection on the 16 km grid." Please note that the quoted ECMWF nominal resolution has been changed from 18km to 16km in line with the TIGGE documentation.

66. [L343] Remove extra "km".

Thank you.

67. [L343] This seems like a really important statement because it suggests that the huge computational cost of a 4 km ensemble is not justifiable from this perspective.

From this perspective, agreed. As noted, 16 km is not the scale to resolve convection.

68. [L374] Although they can likely be inferred, neither baroclinic nor convective instabilities were demonstrated in the analysis.

Agreed, but we feel that the inference from baroclinic development and convection back to their respective instabilities can be assumed.

69. [L382] This conclusion does not seem as direct as it ought to be. Perhaps "could" should be replaced with "should"?

We have changed to "should" at L568.

70. [L382-383] This seems like a fairly weak and somewhat confusing statement on which to end the manuscript. Moist singular vectors would be implemented in the TL/AD forms of the model, and as far as I know are quite independent of the SPPT-based model uncertainty estimate. Perhaps this discussion could instead be extended to consider the SPP-based uncertainty formulation as a look into the future ECMWF system.

We have made a more direct statement now. The point we were trying to make was that it would be good to explore the idea of focussing model uncertainty on potential instabilities, rather than the effects of already-triggered instabilities. The former is more like what SVs are doing, although SVs would be costly to calculate at every timestep. It is possible that SPP could evolve into targeting these potential instabilities if the perturbed parameters were the triggering thresholds, for example. Since submitting the original manuscript, SPP has become more competitive with SPPT, and looks likely to be implemented at ECMWF in the near future. We have changed the wording to "It is possible that such a focus on instabilities (rather than the effects of already-triggered instabilities) might be better explored within the future 'stochastically perturbed parameter' (SPP) framework for model uncertainty — perturbing triggering thresholds for example."

71. [L392] Is a ^2 missing on the l.h.s of definition of the variance?

72. [L444] Why would the squared terms necessarily dominate, particularly if there are correlations between the constituents of the cross terms?

    In general, the terms are uncorrelated. However, we no longer state that we would expect the squared terms to dominate, but rather say that the cross terms are included in the epsilon term, and then point to the Appendix where they are quantified.

73. [L465-469] Providing a quantitative assessment of the relative size of each of these terms seems like it would be useful, particularly because the Residual is one of the (two) leading terms assessed in the text is key to conclusions regarding overdisperison.

    These terms were estimated in an earlier draft of this manuscript, but this was removed for brevity. A shortened version is now included in the appendix of the re-submitted manuscript.

74. [Appendix D] Why is a new field (500 hPa height) and season (JJA) introduced just for this appendix? I guess it might be to show the robustness of the analysis, but I think that the text on L228-232 distracts from the main message of the study. In a two-paper solution (General Comment #2), this figure and discussion could form the basis for a short subsection instead.

    The figures were meant for the supplementary material. We have removed them now, and simply state L356-558 "Note that conclusions drawn in this section appear to generalise to other parameters (such as geopotential heights and temperatures at 500 hPa), other seasons, other stormtracks, and continue until the most recent check for the March — May season 2022 (not shown)."

    Please note that, for old Fig. 7, a small bug was identified. The diurnal averaging aspect (discussed in the old figure caption) was implemented in a way that the error terms in the budget were decreased slightly (error of the diurnal average rather than diurnal average of the error). This has been corrected by removing the diurnal averaging. The impact is so small that it does not affect the conclusions.

    The authors would sincerely like to thank the reviewer for their insight and diligence in reviewing this manuscript. We feel that changes made have led to very useful improvements.

**Reviewer 2**

Review of "The Cyclogenesis Butterfly: Uncertainty growth and forecast reliability during extratropical cyclogenesis" by Mark John Rodwell and Heini Wernli

The paper investigates ensemble forecast reliability at 48h lead time, mainly in the ECMWF system with some comparison to other centers. To do so, a new spread-error budget is derived. It is found that the ECMWF ensemble is overspread in stormtracks in the winter season and it is argued that this is related to cyclogenesis events. In my opinion the core topic of this work is interesting and worth being published since it contradicts the intuitive expectation that cyclogenesis is associated with bad forecasts and low predictability. However, the paper requires a substantial revision. It is too long, difficult to read and not well structured. It spends a lot of time on tangent discussions and detailed case analyses, which I find distracting and misleading. Especially the discussion of the butterfly effect is imprecise, irrelevant and in part incorrect. On the other hand, topics more directly related to reliability and ensemble forecasting systems are very little discussed, if at all.

We appreciate the reviewer's comments and their time taken to review this manuscript. We attempt to address their specific comments below. In our replies below, numbering for lines, figures, tables and equations refer to the revised manuscript unless otherwise stated.

Major comments:

"The cyclogenesis butterfly"

This term, given already in the title, is never properly defined. It is introduced by phrases like "here we think of it as...". I am still not clear what is actually meant by this. The paper investigates reliability, which is an aspect of practical predictability, while the "real" butterfly effect refers to the existence of an intrinsic predictability limit caused by scale interaction in a multi-scale system (see Lorenz, 1969 and Palmer etal, 2014). Current weather prediction system are started from initial condition uncertainties that are much larger than butterflies and are on average far away from hitting the intrinsic limit (e.g. Zhang etal, 2019). The existence of singular vectors are not a manifestation of the butterfly effect, since they are still consistent with infinite predictability due to their constant growth rate. Judt, 2018 (Fig. 8b, day 0-2) for example has demonstrated the extreme increase in the error growth rate if the atmosphere really is perturbed with "butterflies" only. I am however not recommending to discuss the butterfly effect more precisely in this paper but rather to remove this discussion and to focus on much more relevant aspects with respect to (practical) reliability, like the long-standing underdispersive problem of ensemble forecasts and various methods that have been used to mitigate it (e.g. EDA, singular vectors, breading vectors, SPPT, SPP, etc.). It is probably a shortcoming in one or several of those methods that leads to the reliability problem that the paper investigates.

Sorry for the confusion here. We are not talking about the "real butterfly effect", intrinsic predictability limits, or Lorenz, 1969. To be fair, we did not mention the butterfly effect, but rather stated that our butterflies were defined as "local flow configurations where the chaotic and exponential growth–rate of uncertainty is particularly strong" (old L32). We also discussed SVs as indicating that divergence of trajectories within state–space is not uniform over the attractor (old L31), rather than as any manifestation of the butterfly effect. Our reference to Lorenz was to his 1963 paper – discussing sensitivity to initial conditions but not any intrinsic limit to predictability. Our use of the term "Cyclogenesis butterfly" is, rather, an attempt to encourage more flow-dependent thinking in the

evaluation and development of forecast models. However, the potential for confusion is recognised, and we now ensure the reader does not get the wrong idea from the outset.

Figure B1 shows power spectra at day 0 and day 2 for the ECMWF ensemble. The maximum initial (EDA) variance contribution is from waves around 400 km, while the maximum D+2 variance contribution is from waves around 1500 km – i.e. synoptic scales. Notice that the initial variance is saturated at scales smaller than about 100 km. This plot motivates/justifies our interest in the growth of variance at the synoptic scales to D+2 – due to physics and dynamical interactions at all scales.

Incidentally, the Palmer et al. paper uses the current lead-author's previous example to speculate that intrinsic predictability limits might be longer due to the potential confinement of error-growth to intermittent flow regimes. While we did not discuss intrinsic predictability limits, the idea that certain flow-types can organise multi-scale interactions and focus error growth is very much in the spirit of the current study.

While lack of reliability might be explainable by shortcomings in the mitigation methods mentioned by the reviewer, this is not obviously the case. For example, sensitivity of parametrized convection to uncertainties in the resolved flow might be important.

To improve brevity, we have removed discussion of links to the Liouville equation. The thought was that the use of model uncertainty represents a route by which NWP could converge on the true dispersion rates on the real-world attractor (as initial uncertainty is reduced) but, reflecting on the reviewer's comments, it may be the case that NWP will never be able to say anything definitive about the real butterfly effect, as it pertains to the real world.

Reliability in a larger context

The specific overspread that is found over the Northern Atlantic stormtrack in winter (Fig. 7) has not been put into a wider context. If the system is reliable on average there must be compensating underspread somewhere, e.g. over the continents, outside the midlatitudes or in the other seasons. This should also be discussed, as well as the question if the system really is reliable on average at the considered 48h lead time. Some information can be found in appendix D, but I think this discussion should be central in the paper. Furthermore I am wondering what the downstream consequences of the stormtrack-overspread are. Does the overspread persist beyond the end of the stormtrack into the continent in a lagrangian sense, e.g. is the 5 day forecast for Europe in the winter season also overspread? Finally, what is the relation to forecast busts? According to Lillo and Parsons, 2017, East coast cyclogenesis has the potential to generate particularly bad forecasts over Europe. This kind of contradicts the (average) results from this paper. Possibly one season of data is not enough to investigate this but some discussion here would be helpful.

There is some compensation elsewhere. We now state L345-347 "Note that Rodwell et al. (2018) indicated better reliability for this model. Partly this reflects compensation in their annual and hemispheric means, partly it reflects the importance (here) of accounting for bias and analysis uncertainty, and partly it reflects a recent deterioration in stormtrack reliability".

At day 5, the over-spread is still evident but general interactions and the loss of any continued SV contribution make this less clear. The emphasis of this study is on the short timescales because these are the only timescales where agreement between ensemble members is sufficient to be able to make meaningful calculations of their dispersion rates. We do not see any contradiction with the Lillo and Parsons paper – cyclogenesis clearly results in large spread and deterministic forecast busts. The Lillo

and Parsons paper is a major reason why we investigated cyclogenesis. We are demonstrating that we are over-spread, but we do not dispute that the spread should be large in these situations.

More use of TIGGE

While for the case studies 4 centers have been compared, the spread-error budget comparison is only done between the ECMWF and the UKMO system and finally the clustering analysis is only done for the ECMWF system. A reason for this is not given. I think the paper may miss an opportunity here to investigate possible reasons for the (ECMWF) overspread since the different centers use different methods to generate their ensemble. Hence I recommend to include more centers throughout the paper, particularly in the clustering analysis to see if the cyclogenesis overspread is a more general problem or specific to ECMWF.

We avoided a detailed 'beauty contest' because this can be problematic in manuscripts. For example, making sure that the details of the ensemble initialisation over the period of interest are documented correctly, and checking whether there are other factors to consider. Nevertheless, on the reviewer's advice, the variance budget has been calculated for the four models discussed, and this figure replaces the previous one. It suggests that the JMA model is more over-spread. While this model also uses SV perturbations, we prefer to focus on the SV aspect in the ECMWF model alone. Applying the clustering analysis to all the models would include a lot more plots and require even more discussion. The primary aim of the study was the evaluation and understanding of uncertainty growth rates in the ECMWF model; the TIGGE models were there primarily for context.

More focus

The paper spends a lot of time with a detailed discussion of two cases which in my point of view gives little insight. Furthermore, the paper oscillates between analyzing the cases and the entire winter and also between theta- and pressure-level analysis or squared and non-squared metrics, which I find confusing. With Eq. 1 the paper introduces a rather sophisticated diagnostic which later is not used at all. I think this is not a good use of the time of potential readers. The information given in this paper is distributed over 8 sections, 5 appendices, 17 figures and additional supplementary material. I suggest the authors should consider condensing the paper to the essential parts and keeping analysis and methods consistent across the paper.

Equation 3 is central to the study. Plotting it demonstrates that large growth rates are confined to particular flow features. The right hand-side shows that the growth rate (for the PV variable) represents the effects of uncertainties in diabatic processes and in non-linear dynamical scale-interactions. This is now discussed more thoroughly in Sect. 3.2 and revisited in the conclusions (thank you for pointing out this omission). It is hoped that Eq. 3 will feed into subsequent work examining these aspects in more detail. For the growth rate plots, only one case is now presented – with one PV growth rate example for ECMWF (which relates directly to the equation). Two dates are shown for the TIGGE models to give a better reflection of the range of results seen in the animations (which highlight similarities and differences). The sensitivity study (only one shown now) is considered important as it highlights the relative roles of model uncertainty and singular vectors, and points to possibilities for future development. This is now better discussed.

Specific comments:

L1:

This statement is incorrect (see major comment about the butterfly effect and comment below).

The abstract has been re-written to emphasise the focus on NWP. The differences with intrinsic predictability are made clear.

L21:

The Liouville equation as formulated in Ehrendorfer, 1994 assumes that the propagation operator is known and constant. Hence it cannot describe growth due to model uncertainty.

As discussed above, we have removed reference to the Liouville equation.

L26:

I would not say that EDA represents model uncertainty. The model uncertainty is rather part of the assimilation process to generate the initial condition ensemble.

Yes, we are saying that model uncertainty representation is included in the EDA system. The EDA and ECMWF system in general are now described in Sect. 2.1 in the "Models, data sources and key parameters" section.

L31:

This statement is incorrect. Lorenz-type butterflies, i.e. small-scale and small-amplitude perturbations limit predictability via scale interactions and not only due to chaos and strong sensitivity to initial conditions (see Lorenz, 1969 and especially Palmer etal., 2014). Hence the constantly growing singular vectors do not represent this "real" butterfly effect. Furthermore, current errors and uncertainties in forecasting system are neither small in scale nor small in amplitude and cannot be regarded as butterflies. If they were this would mean that current systems operate already now at the intrinsic limit, which is not true (e.g. Zhang etal. 2019). I agree that in some situations error growth in current forecasts is worse than average but if this might be related to the butterfly effect in rare cases is an open question.

As discussed above, we were not implying that SVs represented the Butterfly Effect. We were saying that they demonstrated that growth rates are not uniform. The introduction has been re-written to make the topic of this work clearer, and the role of SVs is discussed in Sect. 3.2.

L51:

I am not sure if I understand correctly what you mean with "cyclogenesis butterfly". The term is never clearly defined.

We now make this clearer in the abstract, at L39-47, and in Sect. 3.2.

L51:

"The key question is..."

This is a big gap in the line of argument and comes as a surprise to me. Please consider rewriting the introduction to focus on this question and the importance and flow-dependence of reliability and the need to extend the "spread-error" relationship.

The question is in the title. Clearly, we needed to work harder on motivating this in the introduction, and we have done this now.

L57:

The paper outline contains to many details in my opinion. Some should have been mentioned and discussed in the introduction, some are results.

We have shortened the outline to remove details and results.

Sec. 2, Data:

Since an entire section is dedicated to describe the data I would prefer that all the relevant details are given here rather than being distributed over the rest of the paper and the appendix. With respect to the other centers, only resolution and ensemble size are given but potentially interesting differences in the ensemble design are not mentioned and later not investigated (see major comment above).

We now increase the discussion of the ensembles from the other centers and include the initialisation details available within the TIGGE archive in a table. This section has been re-titled "Models, data sources and key parameters". We feel that it does not make sense to describe methodologies until the need for them has been motivated. Hopefully, with the better sub-sectioning throughout the manuscript, this is acceptable.

L74, caption Fig. 2:

What do you mean by "background ensemble/forecast"?

This is described L82-83 in Sect. 2.1 on the ECMWF forecast system (previously it was in Appendix E).

L90:

The arguments given about the case selection are very vague. How are they related to the key question? Are these cases in which the forecast was particularly unreliable? Or in which the cyclogenesis was very rapid?

This is a good point. We stated why the cases were chosen but did not make it clear that other attributes had not been a factor. The cases were not chosen because they were unreliable (difficult to establish for a single case) or that cyclogenesis was particularly rapid, or that uncertainty growth rates were unusually large. They were simply chosen to motivate to the reader the kinds of events we are considering, and for their suitability for the sensitivity experiments ("without being strongly affected by other flow perturbations in their environment"). We now make this clearer from L128.

Fig. 2-4:

It is unclear, which forecast lead time you show and why you chose to focus on this particular forecast lead time.

The information for these growth rate figures was given in Appendix B (Further details on the growth rate plots). We appreciate that this is not ideal and have now brought Appendix B partly into the main text in Sect. 3.3, and partly in the captions to Fig. 2 and 3.

To answer the reviewer's question, Fig. 2 is constructed using the 12 h background forecasts from the EDA (so no lead-times are greater than 12 h), started at 6 and 18 UTC. The fields shown are based on centred-means and differences between consecutive hourly lead times. The 24 h running-mean temporal filter then places the smoothed fields back on the whole hours. More specifically, Fig. 2 shows fields centred at 12 UTC on 29 November 2019. For the winds (including humidity fluxes), PV315 contour, the standard deviation in the growth rate parameter (PV315) and its advection, the lead times used are thus {28 Nov 18 UTC + 6,7,8,9,10,11,12 h}, {29 Nov 06 UTC + 0,1,2,3,4,5,6,7,8,9,10,11,12 h},

and {20 Nov 18 UTC + 0,1,2,3,4,5,6 h}. The ensemble-mean precipitation in a given hour and the time derivative in the standard deviation of PV315 are based on the differences (in precipitation accumulations and PV315 standard deviations) between these lead times {28 Nov 18 UTC + (7-6),…,(12-11) h}, {29 Nov 06 UTC + (1-0),…,(12-11) h}, and {20 Nov 18 UTC + (1-0),…,(6-5) h}.

For Fig. 3, again the lead times used are no greater than 12 h. The differences are that the forecasts are started at 00 and 12 UTC and data are only available at 6 h intervals. Hence for Fig. 3, the lead times used are thus {29 Nov 00 UTC + 0,6,12 h} and {29 Nov 12 UTC + 0,6,12 h} and the running mean has length 4 rather than 24.

We focus on these shortest lead times possible so that the ensemble members are as close as possible to each other, in particular representing the same synoptic systems, and hence the growth rates are the best flow-specific growth rates we can calculate. We now make this motivation more clearly. On L184-186 we now state "To understand how growth rates depend on the synoptic flow situation, it is useful to consider very short leadtimes when all ensemble members are representing essentially the same synoptic flow situation. A natural choice is to use the short background forecasts from ensemble data assimilation".

L109:

I don't understand what you mean be 1-dimensional state-measure. Substitute 4-dim atmospheric field?

This was a mistake and both reviewers suggested different solutions. We have changed "some 1–dimensional state–measure (of the atmosphere)" to "some atmospheric parameter field". Thank you.

Eq. 1:

sigma_hat is now the standard deviation of the PV, right? Use P_hat instead of sigma? The hat is not explained, same meaning as in eq. 2?

Sigma is an accepted way of representing the standard deviation, and the hat signifies that this is an estimator. Yes, it has the same meaning throughout. This is now all made clear in the text (L144-145).

I suggest to add an index i to P, P' and NC to indicate that these are quantities from individual members.

The derivation of Eq. 2, which was in Appendix A, has been brought into the main text. The notation has been explained better now. Subscripts are initially used (L149-151) and the meaning of the overline is more fully explained, including for non-linear terms (L150-152). We feel that the current approach of using an overline and no subscripts to signify a mean is neater and consistent with the standard approach for signifying the mean of a linear quantity. We use this approach throughout the paper. The use of subscripts in non-linear terms can indicate "Einstein notation", when the summation over the subscript is implicit, and no overline is required. We do not want to cause this confusion for readers.

L120-L130:

Needs more introduction and explanation. However, this diagnostic is not used in the paper anyway. Consider removing it (see below).

Following the reviewer's first major comment, we have dropped the discussion of the Liouville equation (old L125-130). We feel that old L120-124 are important in the discussion of the processes that can lead to the enhanced growth rates. This discussion has been improved with an explanation of the terms in

Sect. 3.2. These would be the (non-linear, scale interactive) processes that act on the initial uncertainty (including applied SVs) and the perturbations introduced by the model uncertainty. The more prospective aspects discussed at old L120-124 have been moved to the conclusions in Sect. 6, with a pointer from the end of Sect. 3.2.

L142:

"often preceding cyclogenesis", "occur within strongly precipitating WCBs":

These statements are rather vague and are either obvious or seem speculative. Do you mean that the growth rate is correlated with the amount of precipitation in the cyclone? And with cyclogenesis do you mean a depending of the trough where the growth rate peak occurs or does it lead to a cyclogenesis downstream?

This statement is an observation about the animation. We did not intend to make any link to a deepening of the trough or downstream cyclogenesis – simply the colocation of the growth rate with the base of the upper-level trough (we didn't anticipate the spatial interpretation of the word 'preceding'). The cluster mean results (Fig. 7) tend to confirm that there is enhanced growth at the base of the trough. We now make this link more clearly at L391-392 associated with the clustering. Hopefully the paragraph at L198-207 is better written now.

L145:

"Further investigation..."

I find the following statements distracting. But more importantly, the reader might have invested some time to understand eq. 1, the relevant papers and the appendix only to find out now that you are not investigating the right hand side of eq. 1 at all and leave this for future work. Hence I recommend to remove section 4 and appendix A and just state here that you are plotting a lagrangian growth rate.

We do now discuss the right-hand side of Eq. 3 more fully, as explained above. We direct the reader to more discussion about future work in the Conclusions and Discussion section (so that it does not distract so much here).

L152:

It is very confusing that you switch now to geopotential growth rates. I understand that PV is not in TIGGE. But what is the point of showing the PV growth rates first, especially since you did not explore the right-hand side of eq. 1, which for me is the main purpose of using PV? I suggest to stick with Z250 then and to omit the PV-plots.

We agree that changing between PV315 and Z250 is not ideal. However, we do not wish to leave the reader under the impression that it is all about SVs and model uncertainty – other modelling aspects will also be acting through the right-hand side of Eq. 3, in the PV context, to generate the spread. Emphasizing the lack of PV in the TIGGE archive is also considered important to motivate its inclusion at some point.

L157:

"... are very evident"

I actually was surprised to see how bad the agreement is. Not much is said however about where these discrepancies come from, L160 makes a very general statement.

We agree that the differences in growth rates are surprisingly large. Following a recommendation from the other reviewer, we now include a table summarising the initialisation procedures of the other models and show the extended spread error equation for these models. We have also included another example (Fig. 4) which shows a case of better agreement. This helps give a more balanced impression of the possibilities in the full DJF 2020/21 season.

L161:

"It would be useful..."

Please clarify what you mean. Uncertainty growth rates cannot be close to the truth since the truth is not uncertain.

Our wording was quite vague. We were referring to our prior expectation of the truth. The revised text L232-236 avoids discussion of the truth here. In particular, we have changed the wording from "which is closest to the truth" to "While it is difficult to evaluate these growth rates per se, it is possible to assess how well each ensemble system maintains short–range statistical reliability within the North Atlantic stormtrack".

L167:

You state the essential information as e.g. in brackets. I suggest to change that and maybe write down an equation. Also I suggest to be more precise what average means (case average, area average, ensemble average).

We have worked hard to improve this introduction to reliability (the beginning of Sect. 4), including showing the equation suggested.

L174 (also L168):

I suggest to replace "ensemble forecast start times" with "(large number of) cases". Also please explain the symbols first and discuss the visualization afterwards. The notation is inconsistent with eq. 1, maybe express the ensemble mean with <..>.

The key issue here seems to be the definition of ensemble members and ensemble forecasts. We have now made this much clearer at the beginning of Sect. 4 (L246-249). If one looks through the literature, an overbar is generally used to denote a mean, regardless of what it is a mean over. Conversely, "<..>" is often used to denote an inner product. Hence, we would prefer to use an overbar throughout, including in both Eq. 3 and Eq. 9.

L188:

Is mu_A equal to the truth? And is mu_F also equal to the truth? I suggest to not discuss this in the figure caption.

These parameters are displayed in the figure and are different from the truth. We hope that by bringing the derivation together with the figure into the main text, this will become a lot clearer – since old Eq. C1 was the equation which related directly to the figure.

L196:

I find this square-root operation just for "more understandable units" confusing, especially since it introduces the complication with the residual and you admit that small contributions look larger than

they actually are. Moreover, in the supplementary figure you switch back to square units. I suggest to keep the squares in every plot.

Reliability is about bias as well as spread (and all other moments too of course). Hence having bias in its correct units is valuable. In addition, a panel showing spread in squared units seems pretty meaningless per se. At L321-322 we now say that "While smaller terms will look more important than they are in the squared budget, the residual still correctly indicates spread deficiencies".

Sec. 7:

I wonder why you switched from comparing 4 centers to now only 2. Is there a reason for that? And why did you choose to compare with the UKMO?

We were trying to avoid a beauty contest, together with the difficulties in getting all the historical initialisation details correct. Since both reviewers have asked, we do now include all 4 models in Fig. 6, and have discovered that the TIGGE archive provides the necessary initialisation information.

L233:

In Rodwell et al, 2018 you showed (Fig. 1) that the (traditional) spread-error relation is perfectly matched for the Northern Hemisphere at any forecast lead time over an entire year (2014). Hence (if this is still true, is it?) the overspread you now show for the stormtracks in the winter must be compensated by an underspread at some other location or some other season. It would be interesting to investigate this (see major comment above).

We have replied to the major comment above. We discuss this in the revised manuscript.

L238:

I suggest to explain the K-mean clustering method with a couple of sentences.

We include this explanation up-front, with the text L369-370 "which seeks to minimise the sum of squared deviations from the relevant cluster-mean".

L245:

Why do you weight with the root? Isn't the grid cell area scaling with cos(lat)?

Because the clustering method is on the variance, which re-instates the square.

L264:

Does this mean you are combining the cluster1 cases from both clustering areas? Could you further justify this approach? The shift of the region doesn't seem that large. Would one clustering analysis based on a combined region lead to similar results? What about separating clusters by the surface pressure tendency in the region? Wouldn't this be a simpler method more directly related to cyclogenesis?

Yes, we are combining cluster 1 cases from both clustering areas to produce a 'cyclogenesis composite'. We have made this clearer now. We used the clustering approach (and the smaller regions) to give some coherence in flow structures, and to be consistent with the fields displayed in Figs. 3 and 4 (and the animations). We state at L376-377 "It is the ability to cluster on structures which motivated the choice of the K–means approach". To a large extent, the results justify the approach – we see the structures in the clusters and obtain statistically significant differences between the two composites. Other

approaches, such as the one suggested by the reviewer, could also be successful and might be amenable to the use of a single combined region (indeed the whole storm-track), but the structure aspect might not be so well constrained. We suggest that this could be tested in future studies.

L293:

To me this statement seems a bit exaggerated. I would say that the overspread is reduced in the cyclogenesis composite. Also if I read the colors correctly, the residual difference does not reach statistical significance. Why is the overspread enhanced in the counterpart over the central/east Atlantic. Is it because cyclogenesis is shifted downstream in the counterpart cases? Is this spread reduction in cyclogenesis events also happening at other centers (see major comment above)?

We have moderated and extended the text at the end of Sect. 4.5 to say "The overall assessment of ensemble spread is seen in the residual terms (Fig. 8e and Fig. 8j). Here it is evident for the ECMWF ensemble that most of the over–spread in the region of focus, at the western end of the North Atlantic winter storm-track (Fig. 6e), is associated with the cyclogenesis composite — with statistically significant residuals in Fig. 8e and statistically insignificant residuals (indicated by the light blue and light grey colours) in Fig. 8j. Differences are shown in Fig. 8o. They are particularly strong and significant over Newfoundland. Downstream, differences have the opposite sign — possibly associated with differences in downstream cyclogenesis, and consistent with the increased spread noted above. Linking the day 2 stormtrack over–spread (in the region of focus) to cyclogenesis is a key conclusion of this study. It does appear, therefore, that ECMWF initial growth rates (Fig. 2, Fig. 3a) associated with cyclogenesis events are too strong. The next section explores the root–causes for this problem."

We have not examined the other models with this composite approach. This would require more work and a lot more explanation.

L296:

I did not understand this paragraph.

This paragraph was a note about conditional sampling. In the interests of brevity, we have removed the paragraph. The key text remains at L392-393 "(Note that the growth rate is not used within the clustering algorithm since this could potentially bias the reliability assessment)".

L313:

I notice you leave the convection scheme on at 4km resolution. Could you explain why? Usually only shallow convection is used at such high resolutions (e.g. Judt, 2018).

The 4 km experiment with the parametrization of deep convection turned off was also run. Turning off this parametrization at 4 km leads to a further enhancement in D+2 uncertainty in PV315 along the southern extreme of the cold front. It was concluded that, even at 4 km, forcing the ECMWF model to resolve convection can be unrealistic. This is an area of active research by others. We now cite Wedi et al. (2020) in the Introduction Sect. 1.

L324:

I don't understand this sentence.

We have changed the sentence 'The Z250 spread represents the temporal integral of the local tendency, and hence the effects of "material" generation and advection." to "As might be expected, the maximum Z250 spread is located downstream of the maximum 'Lagrangian' growth rates (cf. Fig. 3a)."

L333:

"attempts to resolve". This is misleading since the resolution is still 18km, right?

There is an understandable misunderstanding in our use of "resolve". We have changed this to "that would otherwise be created when the model is forced to represent this convection on its 16 km grid". Thank you. (Note that the nominal resolution is quoted now as 16 km to be consistent with the other models in the TIGGE documentation).

L338:

I am not sure about the relevance of this increased spread. It could just be a consequence of slightly displaced and explicitly resolved updrafts. Would there also be any enhanced spread in e.g. the precipitation averaged over the front?

We agree with the reviewer, and have modified our tentative explanation from "The smaller scales associated with PV315, and its sensitivity to vertical gradients in diabatic heating in the upper troposphere might help explain this sensitivity to the increase in resolution" to L487-488 "At 4 km, the model attempts to resolve more of the convection. The resolved convection can be associated with stronger updrafts, which might perturb the tropopause more vigorously, where PV gradients are particularly strong." There is likely to be enhanced spread in precipitation within the frontal region, although we have not looked at this. It is difficult to say whether the precipitation averaged over the front will be more or less uncertain.

L342:

Possibly there are now explicitly simulated updrafts which are slightly displaced among the ensemble members and generate grid-pointwise spread. Again I am not sure how relevant this is. I don't think this is the kind of uncertainty SPPT was designed to account for. So I am not surprised to see less effect from SPPT in this region.

This comment is very thought provoking. We have changed the text on "Forcing the model to explicitly resolve this convection (even if at the wrong ~18 km scale) appears to better locate the uncertainty with that generated by the ~4 km model" to L490-493 "The impact on P315 uncertainty of allowing the model to resolve more of the convection at the 4 km resolution (Fig. 11e) appears to be in closer agreement with the response to turning off the deep convection parametrisation (minus Fig. 11d), when the model is forced to represent the convection on the 16 km grid."

L379:

If a reduction of singular vectors would make forecasts more consistent then why are they still used? I suspect they do show a benefit at a different location, flow regime, lead time, etc. I think you should discuss these aspects in more detail and also possible alternatives (e.g. inflating SPPT or EDA, using SPP, higher resolution, etc). See also major comment above.

The SVs are particularly efficient at generating spread (by day 2) at synoptic scales (see the power spectra in Fig. B1). Recent spread-error spectra produced by the lead author suggest that this spread is now somewhat too large. The MU also produces spread at synoptic and planetary scales – at the planetary scales the spread-error agreement is much better. Hence this is evidence that a reduction in the magnitude of the SVs would be useful. We do now discuss the advantages of the SPP framework in the modified final sentence L569-571 "It is possible that such a focus on instabilities (rather than the effects of already-triggered instabilities) might be better explored within the future 'stochastically

perturbed parameter' (SPP) framework for model uncertainty – perturbing triggering thresholds for example."

L381:

This would only make sense if the observed increased spread with resolved convection does not mainly result from rather small displacements of individual updrafts. But this has not been investigated (see related comments above).

Please see our reply to the reviewer's comment on L342.

Minor comments:

Fig. 1:

The figure is hard to read and evaluate. I suggest to use a color for the >2PVU regions and to omit the red hatching, since it is kind of obvious and does not add any extra information. As labels of the panels I suggest "Case 1", "Case 1, +24h" or something like this for easier identification.

We only show one case now. We consider the red hatching to be important as not all readers will consider the colocation with precipitation to be obvious. The key southern PV=2 PVU contour seems clear to us.

Fig. 2 and others:

There is a lot of doubling between the figure caption and the text. I suggest to not repeat details in the text that are already included in the caption.

We have tried to eliminate duplication – this is easier now because most of the explanations are brought from the appendices into the main text.

L132:

"Single frame of animation" is not a good description of the plot.

This has been changed, thanks.

Fig. 3 caption:

I think you meant case 2.

Yes thank you.

L175:

Better "sampling from a population"?

We prefer "underlying distribution", as that is what is drawn in the figure. However, they mean the same.

L178:

Any reason why you call this "departure"? It is just a difference, isn't it?

The word departure comes from the world of data assimilation. It is used instead of "error" because the analysis (or observation) does not represent the truth. All the lines are differences.

L180:

"number of forecasts" is ambiguous. I suggest "cases".

We have better defined what we mean by "number of forecasts" now.

L182:

I suggest to remove the 2-superscript (looks like a footnote). It is clear from the context that you are considering squared quantities.

These superscripts are used to distinguish with the rooted terms. WCD discourages footnotes.

L191:

Remove (.

Done, thank you

Fig. 7 (and others):

The color bars are misleading to me. First I would appreciate if the color bars in panels a-j and k-o were identical. The main point of the figure is the comparison and identical color bars will help with that. Also it is misleading to color small positive value with saturated dark colors (e.g. panel c, it looks like a massive AnUnc). I suggest to use reddish colors for positive values, blueish colors for negative values and gray/neutral around zero (e.g. like you did in panel d).

The magnitude of these terms depends on the reliability of the forecast and the lead time. In some configurations, the errors (departures) and spread might be an order of magnitude larger than the bias and residual but, of course, the important difference between the departures and the spread (the measure of unreliability) has the same order of magnitude as (at least one of) the bias and residual. Hence, we really need to be able to see these latter terms in some detail. Moreover, the bias and residual can be of either sign, while the departures and spread are always positive. Hence, there is no clear reason to plot these terms with the same contour intervals.

Fig. 8:

The green geopotential lines are very hard to see. Please make them more prominent.

We have thickened these.

L273:

"Head of the stormtrack". This term is not clear to me. From the context I guess you mean the start/beginning (west).

We have made this clearer with L410: "western end of the stormtrack".

Fig. 11:

I find the arrows more confusing than helpful. Also: CP->DP

We will change to DCP! We have experimented and feel that the arrows help. They convey the sense (sign) of the difference. We now state L452: "Vertical arrows in Fig. 9 indicate the sign convention of the difference to be plotted".

Fig. 12:

I suggest to also revise the color bar. Panel a) shows a positive variable and should use neutral to reddish colors. Panels b)-f) should have the same color bar since this makes it much easier to assess the individual contributions. I cannot really distinguish gray from black contours.

We make it clear throughout where the contour intervals change – we feel it is important to see where a particular change has an impact, even if it is small compared to that of another change.

L328:

) missing.

We have done this, thanks.

Please note that, for old Fig. 7, a small bug was identified. The diurnal averaging aspect (discussed in the old figure caption) was implemented in a way that the errors terms in the budget were decreased slightly (error of the diurnal average rather than diurnal average of the error). This has been corrected by removing the diurnal averaging. The impact is so small that it does not affect the conclusions.

The authors would very much like to thank the reviewer for the time and insight that they have given to this review process. We feel that changes made have led to very useful improvements.

References:

Edward N. Lorenz (1969) The predictability of a flow which possesses manyscales of motion, Tellus, 21:3, 289-307

T. Palmer et al, 2014: The real butterfly effect. Nonlinearity 27 R123

Judt, F. (2018). Insights into Atmospheric Predictability through Global Convection-Permitting Model Simulations, Journal of the Atmospheric Sciences, 75(5), 1477-1497.

Zhang, F. et al, 2019: What is the predictability limit of midlatitude weather?. Journal of the Atmospheric Sciences, 76(4), 1077-1091.

---

## Referee Report (RR1)

Review of WCD-2022-6, "The Cyclogenesis Butterfly: Uncertainty growth and forecast reliability during extratropical cyclogenesis" by Rodwell and Wernli

I thank the authors for their adjustments to the manuscript and responses to my recommendations concerning the initial submission of this work. I particularly appreciate the reduction in the number of sections in the manuscript, which has helped to improve its readability. A clear statement of the objectives of the work will further help to motivate the reader and will provide a useful "point of contact" between the otherwise disparate elements of the study.

The revised text still suffers from a lack of clear organization, which will make some of the most interesting results of the study difficult for future readers to access. Centralizing data and methodological descriptions in section 2 will avoid many of the current disruptions to the flow of the manuscript. It will also help to keep readers focused on the scientific contributions contained within the impressive amount of presented work.

I hope that these notes will provide some useful suggestions for this submission.

Recommendation:  Major Revision

Reviewer:  Ron McTaggart-Cowan

**General Comments**

1.  It is unclear to me what was done to make the objectives of the study clearer in the introduction (response to General Comment #1 of the initial review). The closest thing that I can find to such a statement is the phrase that "This study focuses on uncertainty growth in the North American / North Atlantic / European region, and particularly the North Atlantic winter stormtrack (sic), with its embedded cyclogenesis events and other synoptic systems." However, this sentence does not explain what will be achieved by this "focus". Please include a clear thesis statement to help readers to understand what the intended outcome of the study is.
2.  Although I appreciate the added discussion and authors' responses, I still think that invoking the "butterfly effect" is a misnomer. Based on the title, I would expect a paper about how small-scale and small-amplitude perturbations affect cyclogenesis. A title that is more descriptive – albeit less spectacular – would serve the content better. Maybe something like, "The impact of North Atlantic winter cyclones on uncertainty growth and forecast reliability in ensemble guidance".
3.  I do not think that the strategy of "just in time" methodological description is effective or that it improves the readability of the text by motivating the reader. On the contrary, the decentralized methodology segments disrupt the flow of the text. Moreover, they are difficult to locate for readers that are not progressing linearly through the text and/or readers that wish to refer back to methodological descriptions at a later time. Please seriously consider introducing all relevant methods in section 2 of the manuscript.
4.  The comparison of spread growth rates in select TIGGE models is interesting, particularly because of the wide range of patterns shown in Fig. 3. However, the follow-up on this analysis

lacks sufficient rigor to make it as useful as possible for future readers. It would be very interesting to know the growth rates of some systems differ systematically from others, for example. Imagine adapting the anomaly correlation score using the LGR from one model at a time as the "analysis anomaly" over the North Atlantic. For example, the LGR from each TIGGE model (i.e. the "forecast anomaly") could be compared to the ECMWF patterns: an ACC would be computed for JMA, NCEP and UKMO. Then each model could be compared to the UKMO patterns for another set of scores: JMA and NCEP (the ECMWF score already being known). Et cetera. In the end, symmetric matrix of ACC scores would be obtained, and could be presented as an effective synthesis for this component of the analysis. The $95^{th}$ percentiles (or smaller, given the small number of cases) of the ACC scores could be used as a measure of the variability around the mean ACC score. Noting what the ACC score is for Fig. 3 would provide a quantification of the extent to which the case study aligns with the "typical" degree of agreement between LGR in the TIGGE systems.

5. Although much improved from the initial submission, the structure of the manuscript continues to present a challenge for readers. Aside from the need for a centralized methodology section (General Comment #3), a specific example arises at the end of section 3.3. The section was interesting, and ends with two interesting questions. If they are anything like me, the reader will be looking forward to diving into these questions. However, the section 4 introduction, and methodology introductions sections 4.1 and 4.2 mean that they will have to "hold that thought" for ~100 lines of text before they get to further discussions on these questions. By then, the reader will have forgotten the specifics of the questions or why they were interesting. If a review of reliability is required, it should appear either in the introduction or in section 2. Likewise, the complicated descriptions in sections 4.1 and 4.2 should appear in section 2. This reorganization will mean that the reader's momentum can be maintained as they progress through the results and synthesis.

6. How much of section 4.1 could be replaced by a reference to section 3 of Rodwell et al. (2015), but with "observation" (in that study) replaced by "analysis" (here)? The overlap is mentioned explicitly beginning on line 324, but a full replacement (and associated simplification of the current text) does not seem to have been considered: please consider it.

7. The term "cyclogenesis" appears to be used primarily to refer to the presence of a cyclone. This is important because the "cyclogenesis butterfly", based on a standard definition of cyclogenesis, implies uncertainty introduced by a cyclone is forming or deepening. However, the "cyclogenesis" clusters 1 and 2 (Fig. 7) only assess of the presence of a cyclone: they contain no direct information about whether the cyclone is intensifying or decaying (the westward tilt with height is not a guarantee of surface intensification). I understand that cyclones often deepen in this region; however, this makes the link to cyclogenesis anecdotal rather than data-driven. The "winding back" process (a term that should be clearly defined) appears to be an attempt to build in a cyclogenesis period. However, if I understand the procedure correctly then a cyclone moving into the defined area will be defined as "cyclogenesis", even if it has already reached its peak intensity. Alberta clippers, for example, reach peak intensity shortly after formation and slowly weaken thereafter as they move towards the region of interest for this study (Blaine and Martin 2007). Changing from "cyclogenesis" perspective to one that documents ensemble behaviour in the presence of a cyclone would not weaken the work, and

would better describe the analysis.  The recommended title (General Comment #2) reflects this change in perspective.

8. Excessive spread in the storm track during cyclone passage is labelled as a "key conclusion of this study" (line 438).  This conclusion appears to be based on Fig. 8o, which shows positive but non-significant differences between the composite residuals.  If that is correct, then assertions of cyclone-related "over-spread" should be moderated in the text.  Given the potential for type-I errors related to multiple-testing (Wilks 2016; BAMS) and an experimental design that does not sample interannual variability, the true significance of these differences is questionable.

9. Figure captions are not the appropriate place for methodological descriptions.  Although figure-specific details might be provided in captions (specific threshold values for example), complete methodological descriptions should appear in the main body of the text where it can be easily found by future readers.  Please move all methodological descriptions from captions to section 2 of the document.

10. Section 3.2 should be replaced with a brief description of the Lagrangian growth rate in section 2, including a reference to Rodwell et al. (2018).  The derivation and extensive discussion of terms that will not be employed further in the analysis does a disservice to the current study by introducing unnecessary complexity.  If the rhs of Eq. 3 will be useful in a future study, then it should be presented in the future study.  The discussion section of this work could easily refer to a hypothetical expansion of the Lagrangian growth rate rather than specific equations that disrupt the flow of the text.

11. I understand that decisions related to writing style are typically left to the author; however, the over-use of em dashes disrupts the flow of the text and reduces its readability (there are seven in the introduction alone).  Please consider rewriting the majority of phrases that currently use this form of subordination.

12. Single and double quotes are used liberally throughout the text; however it is unclear what they mean and how the authors choose between them in any given circumstance.  Please consider removing the majority of these quotation symbols and/or provide a description of what they represent.

**Specific Comments**

1. [L19] Consider rewording split infinitive.
2. [L23] It isn't "NWP" itself that develops techniques, but researchers and system developers.
3. [L26] I believe that "leadtime" is usually written as "lead time".
4. [L28] I believe that "Stormtrack" is an application while, "storm track" is the usual term for the region discussed in this study.
5. [L32] Is "propone" the word that you mean to use here?  Consider replacing with "prone" or "conducive".
6. [L35] Why is "blocking" (well-accepted terminology) enclosed in single quotes?
7. [L49] I think that a comma before the quoted question would be appropriate.
8. [L50] The term "reliability" has already been introduced with single quotes: consider removing them here for readability (the citations make it clear that this is a technical term).
9. [L53] Why does the bias problem apply only to short-range assessments of reliability as implied here?

10. [L58-59] This phrase suggests that improvements to the model and MU will not improve reliability in the presence of SV perturbations. It that guaranteed to be true? If the SV perturbations are scaled to become arbitrarily small, then they will presumably have a negligible impact on the forecast and model improvements will become dominant. This general statement might need either to be qualified or to be removed.

11. [L59] What does the term "the potential is raised" mean? Does this refer to an increase in potential, or to a subject that is raised later in the text. Please consider using clearer terminology.

12. [L63-65] This appears to be a run-on sentence: please rephrase.

13. [L69] Why is "Ensemble" capitalized here?

14. [L78] This is a highly condensed system description that is difficult to follow for those not already familiar with the ECMWF suite. Could a reference to a system description be added, either in the form of a peer-reviewed publication or an operational technical note?

15. [L90-92] Both SV and MU have already been defined. (I actually think that both acronyms should be replaced with complete terms throughout the text for readability.)

16. [L101] What does the "current EDA cycle" mean? Does that refer to the one that was operational when this paper was written? Please be more specific.

17. [L103-104] Does ERA5 use the same version and configuration of the EDA as described here? This is possibly important because a close connection might mean that systematic errors are common between the forecast and analysis.

18. [Section 2.2] The extremely brief introduction of non-ECMWF systems in section 2.2 stands in stark contrast to the preceding full page of detailed description about the ECMWF ensemble. Please provide at least a brief introduction for each system (beyond Table 1) along with relevant references.

19. [L111] For consistency with what?

20. [L121] Why is PV only conserved, "following the *horizontal* flow on an isentrope"? To my understanding the orientation of the isentrope doesn't matter for PV conservation (note that any flow across an isentropic surface is better expressed as "diabatic" rather than "vertical").

21. [L128] How is the "speed of cyclogenesis" defined? Do you mean "deepening rate" or "intensification rate"?

22. [L130] "Eastern North America" is located east of the Great Lakes. Does this mean that the cyclone initially tracked westward? I think that showing the track in Fig. 1 would be more effective than this text description.

23. [L136] Parcels with ascent midpoints at 25oN are unlikely to be ascending above the warm front in the comma cloud region. If these are not following typical WCB storm-relative trajectories, what is driving their ascent? Is this an anafront? Perhaps this is unimportant, but the WCB points are described in some detail here, as is the distribution of precipitation.

24. [L148-153] This is all standard Reynold's decomposition, is it not? If so, then that should be mentioned here. If not, then the differences should be explained and justified.

25. [L154] What is the advantage of the Eq. 2 form over that used by Baumgart and Riemer (2019)?

26. [L174] What does the "intrinsic context" mean?

27. [L176] I do not think that "ground-truth" is usually hyphenated or single-quoted.

28. [L187] How is a 24-h running mean taken for background forecasts with a range of only 12h? The preceeding methodological description should be expanded and moved to section 2.

29. [L188-193] A figure caption is not the appropriate place for methodological descriptions (the same applies for the WCB trajectory calculations described in the Fig. 1 caption). Please include this information in section 2. Lines 189-193 of the text contain the information that should appear in the Fig. 2 caption instead of the methodological description.

30. [L193-194] Please state explicitly how the location of large LGR is "consistent with Hoskins et al. (1985)", why this is important, and why further investigation would be useful (though not useful enough to be presented here).

31. [L197] Please provide a section reference rather than "above", particularly because the erosion of the trough has not been previously discussed.

32. [L198] Are the animations are for different initializing times for this case or for different cases? Please be specific about what these animations contain and why they are relevant.

33. [L199] What does it mean to "'shadow' the true synoptic evolution of the flow"? This term also appears on L226, although it remains unclear how the "true synoptic evolution" is defined, particularly given the similar amplitudes of analysis and short-range forecast uncertainty.

34. [L200] What are "large model growth rates"? Does this refer to large LGR values within model simulations? Please be specific about which synoptic features are associated with these growth rates, if they are important. If they are not, this sentence should be removed.

35. [L201-207] These events have already been listed in the introduction. Because their connection here is purely speculative (it is explicitly noted that they are "not investigated here"), these sentences should be removed. Any discussion to be retained should be included in section 6.

36. [L209-210] Rather than forcing the reader back to section 3.2 to identify the reasons, why not list them briefly here and provide a back-reference to section 3.2 for interested readers?

37. [L227-228] Does "DJF 2020/21" follow WCD date formatting conventions?

38. [L228] The phrase "the agreement can be better" is not specific enough for a scientific publication. Neither is the support of this statement with a new case study (not described in the text) sufficiently robust. Please refer to General Comment #4 for a recommended replacement.

39. [Section 4 introduction] This is a highly condensed description of reliability that is unlikely to describe the concept effectively to readers who are not already familiar with it. (I am reasonably familiar with it and have a very hard time following both this discussion and Fig. 5.) Consider moving this description to an appendix and focusing the in-text description of reliability on what it looks like to have a reliable system, or what problems are related to a lack of reliability. These concepts would be useful in the context of the current work and would help to motivate the subsequent analysis. This suggestion should be read in conjunction with General Comment #5.

40. [L243] The term "uni-modal" usually appears without a hyphen.

41. [L247-249] Has this notation not already been described in section 3.2? If so, it should not be repeated here because it appears to add complexity to this already complicated description of reliability.

42. [L255] Why is the operational status of the forecast important enough to be italicized here (or important at all for that matter)?

43. [L263] The phrase "for the interested reader" suggests that there is an alternative to reading sections 4.1 and 4.2 for the uninterested reader: is that true? If it is, then that alternative should be explicitly stated here.

44. [L271] The "as discussed above" phrase is not a useful introductory clause here:  terms 1-6 of Eq. 4 have not been explicitly "discussed above".  Please either remove it or include it in the parenthetical statement at the end of the sentence.
45. [L287] The {} symbols should be referred to as braces rather than parentheses.
46. [L289] What "later" is being referred to here?  Please be specific about where further discussion of this term appears.
47. [L315] Please be specific about where this "later" refers to in the text.
48. [L325-328] This discussion seems to be relevant only to the observation-based analysis undertaken in the Rodwell et al (2016) study.  Please consider whether it is needed here, given that it seems to add little of direct relevance to the current work.
49. [L343] Please be specific about where this "later" refers to in the text.
50. [L343] How much is "a little"?  Please provide quantification.
51. [L345] Please be specific about where this "later" refers to in the text.
52. [L346] Consider "suggests potential" rather than "reflects" because the compensation is not shown here.
53. [L347] What demonstrates the "recent deterioration in storm track reliability" claimed here?
54. [L347] It seems unlikely that the storm track itself has become unreliable.  Please rephrase to make it clear that EDA reliability has recently deteriorated in the storm track region, if that is shown to be true.
55. [L358-360] How does one pick errors, spreads and reliability from different ensembles?  My understanding is that Reliability is computed from the ensemble distribution, which involves both the $0^{th}$ and $1^{st}$ moments.  As such, the Reliability is not an independent quantity that can simply be chosen from an arbitrary ensemble.  From a more utilitarian perspective, how would picking the reliability of a given ensemble have an impact on guidance?
56. [L359] Suggest "day-2".
57. [L362] What part of this analysis demonstrates that the JMA system has the slowest initial growth rates (the ensemble has the largest spread in the second column of Fig. 6)?
58. [L363-364] Which of the two questions posed at the end of section 3.4 is being answered here? The first one (over-spread during cyclogenesis) seems the most likely referent; however, the analysis in section 4.3 does not distinguish between cyclogenesis and no-cyclogenesis events. As a result, it cannot be asserted that the ECMWF ensemble is over-dispersive "in the vicinity of cyclogenesis".  It appears to be over-dispersive in the storm track, but no more detailed statement than that would seem to be appropriate here.
59. [L364-365] This is a statement rather than a question.
60. [L376-377] Why is the K-means algorithm any better able to "cluster on structures" than other clustering approaches?  For example, EOFs could have been used and the clustering done with their PCs.  Such an approach would arguably be even more structure-aware than one adopted. There is no clear need to change the clustering strategy; however, the rationale for the methodological selection should be defensible.
61. [L392-393] It is clear from the preceding paragraph that the LGR is not used as an input for the clustering algorithm.  However, once the methodological description is moved to section 2 with the remainder of methodology information, this note will be relevant to remind readers of the independence of this field.
62. [L410] A scientific audience should not need to be old that 91:89 is "nearly 50:50".

63. [L417] The spread maximum for the cyclone cases appears to occur in the middle of the North Atlantic storm track, or even at the eastern end of its highest track density, rather than over the "western part". See for example Fig. 7a of Hoskins and Hodges (2019; JCLIM).

64. [L426] Why is there a tilde before the figure reference?

65. [L433] If this region is described as the "western end of the North Atlantic winter storm track", then it would be useful to provide a graphical description of the storm track early in this study. Cyclone tracking studies [including the recent Hoskins and Hodges (2019)] find peak cyclone density near Newfoundland, placing the western end of the storm track along the eastern seaboard. If a different definition of the storm track is used in this study, it should be clearly described to make the associated discussions easier to follow.

66. [L436] There do not appear to be any significant differences in Residual (Fig. 8o) over Newfoundland. There is a small region of significant difference over eastern Quebec and the Gulf of St. Lawrence, but this is west of the coastal storm track. The small spatial scale and multiple testing make the significance of this region questionable (using a field significance test might help in this regard). This seems inconsistent with describing the red area in Fig. 8o as "particularly strong and significant".

67. [L437] What does it mean that the opposite-signed differences "might be associated with differences in downstream cyclogenesis"? Does this refer to different realizations of downstream cyclogenesis in different members, or to different forms of downstream cyclogenesis in reality, or something else entirely?

68. [L440] "Root-cause" is not usually hyphenated.

69. [L448-451] This does not appear to be a complete sentence.

70. [L457] Suggest "day-2".

71. [Fig. 10] What is the contour interval for MSLP?

72. [L462-463] Although the use of different colour bars allows different ranges of values to be shown, it is misleading in such a figure where the panels show the results of different sensitivity tests. Please consider using the same colour bars for all panels.

73. [L482-484 and L493-494] These discussions of changes to precipitation seem somewhat tangential to the main themes of the manuscript and could be removed.

74. [L500 and L547] Reword "2 d".

75. [L505] The phrase, "indicating that the conclusions drawn in this section are robust even with only two cases" does not seem logically correct. The fact that a second case shown a similar pattern gives adds to confidence about the conclusions; however, the similarity of two cases does not provide some sort of successfully conclusive evidence as implied by this statement.

76. [L515] Suggest replacing "these aspects might be developed" with "these techniques might be modified" for clarity.

77. [L554-555] Are any modern calibration techniques state *in*dependent as implied here for machine learning techniques?

78. [L560-562] It is unclear which results are being referred to here. Fig. 10 show that SV and MU have (by far) the leading impact on Z250 spread; DCP is a distant runner-up. However, this discussion seems to imply that DCP is dominant, with SV and MU also contributing. Although the results are more uniform between the three for 315K PV, this statement could easily lead future readers to think that deep convection has more of a relatively larger impact than it actually does in this case.

79. [L564-565] The wording of this sentence seems unnecessarily vague and complex.
80. [L566] Suggest removing hyphen in "model-uncertainty".
81. [L571] This is a very abrupt ending to the manuscript. Consider adding a broader statement that is more directly related to the work undertaken in this investigation.

---

## Referee Report (RR2)

**Third Review of WCD-2022-6**

The authors have made significant changes to the submission that have improved the presentation of the work. I particularly appreciate the fact references to the "cyclogenesis butterfly" have been removed and that the number of sections has been reduced to six. The manuscript is certainly converging towards a publishable form.

Reviewer: Ron McTaggart-Cowan

Recommendation: Minor Revisions

**General Comments**

1. Description of the plotting strategy within the main body of the text disrupts the flow and will distract future readers from the main points that the figures are meant to support. Leaving details of the plot description within the caption instead of the text is a stylistic decision; however, I believe that the paper would be more effective if this convention were followed throughout.

2. Starting on L204, the growth rate of spread is used as being analogous to forecast uncertainty. But one of the key findings of this paper is that the ensemble is conditionally over-dispersed. So then the events being studied are precisely those where the ensemble spread does *not* represent uncertainty well. It seems like a more careful use of the words "spread" and "uncertainty" is warranted throughout the text. This is particularly true in relation to the LGR, which I think is formulated to represent spread growth directly, and uncertainty growth only under the assumption that the two are interchangeable (which is shown not to be entirely true during cyclogenesis events).

3. I still find the methods used in section 4 to be too complicated for the relatively simple outcome. If the goal is to demonstrate the conditional over-dispersion of the ECMWF ensemble then please reconsider dramatically simplifying the analysis. If the goal is a demonstration that a complicated spread-error decomposition can be used, then please be sure to highlight the value added by the technique over a simpler analysis.

4. The use of different shading ranges for different terms of the same equation (top two rows of Figs. 5 and 7) and equivalent plots for different sensitivity tests (Figs. 9 and 10) make these comparable panels very difficult to compare. I understand that not much will show up on the panels with small ranges, but that's useful information that the reader should be able to determine at a glance, not by looking simultaneously at the plotted structures and the colour bars simultaneously for two panels.

5. It appears that only the final paragraph of the conclusions (four lines) could really be classified as discussions or conclusions (the title of section 6). This means that ~90% of the section is actually dedicated to a thorough summary, including numerous figure and table references. As someone who read through the full manuscript I find this redundancy a missed opportunity for opening up a broader discussion of the implications of the work. Please consider either renaming section 5 to "summary and conclusion" or (better) redrafting section 5 to present a very short summary before taking a larger perspective on discussing the work.

**Specific Comments**

1. [L40-58] I suggest redrafting this paragraph (and removing Fig. 1, with any needed panel combined with Fig. 2) for two reasons. One is that it is hard to imagine any reader of this work who wouldn't already been very familiar with cyclogenesis in the storm track. The other is that this study isn't really about cyclogenesis itself, but rather about the growth of uncertainties related to cyclogenesis. I understand that the reader needs to know what cyclogenesis is to appreciate how errors may grow, but I think that a well-crafted literature review would be more effective at relaying this than the current case study. Describing baroclinic instability and diabatic contributions to cyclone deepening could be done briefly with relevant citations. This would be followed by an introduction to error growth on the waveguide, for example citing the recent work of Baumgart et al. (2019).

2. [L61] I think that the word "coordinate" implies too much intention here, and feels like an anthropomorphization as a result. Or maybe it's the word "act to": cyclogenesis doesn't really "act", it just happens. A similar construction appears in the subsequent sentence.

3. [L126-127] Many readers will probably know what it means to "warm start" VarBC and SPPT: please provide a brief explanation because this presumably impacts early spread growth in the forecast.

4. [L137] Does the WCD style guide cover web references? If so, it will hopefully cover citation format and include information about access date.

5. [L177] This introductory sentence is written as if clustering is the only way (or even the most obvious way) to accomplish the objective of identifying cyclones. Given that other methods for cyclone identification have been used in the literature, please consider rewording this sentence to provide a stronger introduction to the need for a cluster analysis in this case.

6. [L210] The use of the thin overline here (ensemble mean) is distinct from the thick overline in Eq. 1 (time mean). I'm not sure that the typesetting is going to be clear enough to allow readers to distinguish between these two. Please consider using a different operator (for example <>) for the ensemble mean.

7. [L248-252] This discussion makes it sound like Z250 is used throughout the remainder of the text, but the subsequent section moves back to P315. Please clarify here which sections are forced to use Z250 because of TIGGE database limitations.

8. [L259] "Cyclogenesis deepening" seems redundant given the adopted definition of cyclogenesis (i.e. cyclone deepening).

9. [L265-284] I don't understand the "it is tempting to speculate" concept here, especially when the paragraph goes on to say that these hypotheses could be confirmed (or refuted) by investigating the terms on the r.h.s. of the LGR equation. If it is tempting and confirmable, then why isn't it done? Then all of this theorizing could be replaced by a simple plot that shows what process is occurring. Between that simplification and the reduction in plot strategy description (General Comment #1), a solid analysis could be included without increase in manuscript length. This would also provide justification for the existence of the r.h.s. of Eq. 3, which is not otherwise used in the manuscript as noted by both reviewers in previous rounds of review.

10. [L292-294] There is an odd asymmetry in this discussion. The "forecast bust" reference (behaviour of a forecasting system that is entirely a property of model space) seems out of place with this discussion of physical features and phenomena. Maybe pulling out the "forecast bust"

phrase and making a separate statement would help, because then it is clear that the bust can have its origins in any of the listed features (or others).

11. [L310-315] It is unfortunate that there is no quantification of this difference. It seems like there is enough information contained in the TIGGE database for a systematic assessment of LGR for Z250 during cyclogenesis cases from different models.

12. [Sections 4.3 and 5] Does "day—2" refer to "day minus 2"? If so, please replace the em-dash with a space and a minus sign, to become "day -2" for readability and to distinguish it from a compound adjective (e.g. "the day-2 spread").

13. [L510-511] Is there really a need for further investigation to figure out how to reduce spread generated by SVs or SPPT? The options seem pretty obvious. Perhaps this would be better worded as how to reduce the associated spread growth during cyclogenesis without negatively affecting the overall well-balanced spread-error relationship.

14. [L564] I'm not really sure what this sentence means (even what "other" refers to or is distinct from) and how it follows logically from the current work.

15. [Data availability statement] The "data and code available on request" doesn't really live up to FAIR principles. Please consider at least uploading as much of the code used to generate the results shown here as possible to a public repository.

---

## Referee Report (RR3)

Dear authors,

thanks again for your response to my comments. I have no further concerns. Congratulation to the paper!

---

## Editor Decision (ED1)

Dear Mark, dear Heini,

I have received the second round of reviews from two highly qualified and attentive reviewers. Both reviewers acknowledge the improvements made during the first round of revision. Both reviewers, however, have further major issues that need clarification before publication.

Both reviewers reinforce their issues with your use of the term "butterfly". I agree with the reviewers that re-defining the meaning of this term creates readers' confusion, without a benefit that a would see for the reader. Your reviewer Ron McTaggart-Cowan makes a constructive suggestion for an alternative title. (Shifting the focus on cyclones instead of cyclogenesis seems helpful to me also to avoid unnecessary confusion. I further agree with the reviewer that this shift would not at all diminish the significance of your results.)

A further issue that carries over from the first round of the reviews is the presentation of the material. There may be different opinions about how to best organize the material, and there may be different approaches that may yield satisfactory results. In its current version, however, the organization of the material affects the quality of the manuscript not to a small degree. To you as authors, this issue may not become so clear, because you are well aware of the storyline of your work and the major points that you would like to communicate. Switching between discussions of results, discussions of key concepts, and technical information on methods may not seem distractive to you. For your readers, however, that is very much different. Reviewer Ron McTaggart-Cowan's comments illustrate these distractions very well. I'd like to emphasize that this is not a critique of your writing style or your writing preferences; it's a matter of the functionality of the organization. I acknowledge that a solution to the issue will most likely be more complex than simply introducing a method section. When you introduce methods, you discuss conceptual aspects of these methods also. This conceptual guidance is highly appreciated. The guidance, however, is interspersed with technical information that distracts the reader from understanding the conceptual value of the respective methods. Combined with introducing methods during a discussion of results makes the current manuscript a difficult read. Helpful comments for re-structuring are found in the reviewer's comments. My own impression is that providing the conceptual guidance when you start discussing the results obtained by the method, while putting the more technical aspects of the method into a method section (or the appendix; section 4.1 seems to be a good candidate for that) will benefit the reader. Both of you are highly experienced writers. I have no doubt that you will find a good solution to this issue once you "see the problem through the readers' eyes".

Noting these specific points, of course, does not imply that I mean to downplay any of the other points raised by the reviewers.

Below are a few minor points that I noted when I was having a look at your revised manuscript. I apologize if there is overlap with comments by the reviewers.

Kind regards, and I am looking forward to receiving your revised version.

Michael

- In the abstract, you refer explicitly to baroclinic and convective instability, just after noting the focus of your study. The role of these instabilities is hardly touched on in the

manuscript. The explicit mention could thus raise readers' expectations that your manuscript will not meet. Do you see, for the reader, a clear benefit of referring explicitly to baroclinic and convective instability in the abstract? If not, consider omitting.

- The term overbar(v dot grad P) seems to be missing in the fourth line of Eq. 2. (I do not think that this derivation needs to be shown, though, at least not during discussion of results.)

- L187: Can you clarify how a 24h running mean is applied to the 12h EDA forecast?

- Acknowledgement: As Ron McTaggart-Cowan has revealed his identity you may want to consider referring to him by name.

---

## Author Response (AR2)

Dear Michael,

Thank you for your attention to this manuscript. We had not fully appreciated the problem some readers would have with the butterfly term. This has now been removed. The idea, that it is useful to evaluate uncertainty growth in a flow-dependent way, is hopefully still conveyed. We have spent not a small amount of time addressing the reviewer's concerns, but we greatly valued their diligence and insight. We hope that the revised manuscript has satisfied their concerns. Replies to your further comments are in blue text below.

Kind regards, Mark & Heini.

Dear Mark, dear Heini,

I have received the second round of reviews from two highly qualified and attentive reviewers. Both reviewers acknowledge the improvements made during the first round of revision. Both reviewers, however, have further major issues that need clarification before publication.

Both reviewers reinforce their issues with your use of the term "butterfly". I agree with the reviewers that re-defining the meaning of this term creates readers' confusion, without a benefit that a would see for the reader. Your reviewer Ron McTaggart-Cowan makes a constructive suggestion for an alternative title. (Shifting the focus on cyclones instead of cyclogenesis seems helpful to me also to avoid unnecessary confusion. I further agree with the reviewer that this shift would not at all diminish the significance of your results.)

A further issue that carries over from the first round of the reviews is the presentation of the material. There may be different opinions about how to best organize the material, and there may be different approaches that may yield satisfactory results. In its current version, however, the organization of the material affects the quality of the manuscript not to a small degree. To you as authors, this issue may not become so clear, because you are well aware of the storyline of your work and the major points that you would like to communicate. Switching between discussions of results, discussions of key concepts, and technical information on methods may not seem distractive to you. For your readers, however, that is very much different. Reviewer Ron McTaggart-Cowan's comments illustrate these distractions very well. I'd like to emphasize that this is not a critique of your writing style or your writing preferences; it's a matter of the functionality of the organization. I acknowledge that a solution to the issue will most likely be more complex than simply introducing a method section. When you introduce methods, you discuss conceptual aspects of these methods also. This conceptual guidance is highly appreciated. The guidance, however, is interspersed with technical information that distracts the reader from understanding the conceptual value of the respective methods. Combined with introducing methods during a discussion of results makes the current manuscript a difficult read. Helpful comments for re-structuring are found in the reviewer's comments. My own impression is that providing the conceptual guidance when you start discussing the results obtained by the method, while putting the more technical aspects of the method into a method section (or the appendix; section 4.1 seems to be a good candidate for that) will benefit

the reader. Both of you are highly experienced writers. I have no doubt that you will find a good solution to this issue once you "see the problem through the readers' eyes".

Noting these specific points, of course, does not imply that I mean to downplay any of the other points raised by the reviewers.

Below are a few minor points that I noted when I was having a look at your revised manuscript. I apologize if there is overlap with comments by the reviewers.

Kind regards, and I am looking forward to receiving your revised version.

Michael

- In the abstract, you refer explicitly to baroclinic and convective instability, just after noting the focus of your study. The role of these instabilities is hardly touched on in the manuscript. The explicit mention could thus raise readers' expectations that your manuscript will not meet. Do you see, for the reader, a clear benefit of referring explicitly to baroclinic and convective instability in the abstract? If not, consider omitting.
- We hope that the revised text makes the link more clearly
- The term overbar(v dot grad P) seems to be missing in the fourth line of Eq. 2. (I do not think that this derivation needs to be shown, though, at least not during discussion of results.)
- The term associated with overbar(dP/dt) in line three disappears when multiplied by P' and averaged over the ensemble. The final equation is shown in the main text and explained comprehensively in Sect. 2.6, and then discussed in the context of cyclogenesis in Sect. 3.1. The derivation is placed in Appendix B.
- L187: Can you clarify how a 24h running mean is applied to the 12h EDA forecast?
- Yes, we have clarified this in response to Ron's specific comment 29. The 24 h running mean is made after concatenation of hourly values from each 12 h assimilation cycle.
- - Acknowledgement: As Ron McTaggart-Cowan has revealed his identity you may want to consider referring to him by name.
- Yes we do acknowledge Ron by name.

wcd-2022-6

**The Cyclogenesis Butterfly: Uncertainty growth and forecast reliability during extratropical cyclogenesis**

**by Mark J. Rodwell and Heini Wernli**

Replies to the reviewer's comments (in blue text).

The authors would like to thank Ron McTaggart-Cowan and the other anonymous reviewer again for the time and care that they put into reviewing the revised version of this manuscript, and for their additional comments, which will be addressed in detail below. The main changes compared to the previous submission are the following:

- We implemented a more classical organization of the material, as requested by reviewer 1, and now present the many methodological aspects upfront in Sect. 2.
- We further clarified the objectives of the study.
- We completely removed discussion of butterflies.

**Reviewer 1**

Review of WCD-2022-6, "The Cyclogenesis Butterfly: Uncertainty growth and forecast reliability during extratropical cyclogenesis" by Rodwell and Wernli

I thank the authors for their adjustments to the manuscript and responses to my recommendations concerning the initial submission of this work. I particularly appreciate the reduction in the number of sections in the manuscript, which has helped to improve its readability. A clear statement of the objectives of the work will further help to motivate the reader and will provide a useful "point of contact" between the otherwise disparate elements of the study.

The revised text still suffers from a lack of clear organization, which will make some of the most interesting results of the study difficult for future readers to access. Centralizing data and methodological descriptions in section 2 will avoid many of the current disruptions to the flow of the manuscript. It will also help to keep readers focused on the scientific contributions contained within the impressive amount of presented work.

I hope that these notes will provide some useful suggestions for this submission.

Recommendation: Major Revision

Reviewer: Ron McTaggart-Cowan

**General Comments**

1. It is unclear to me what was done to make the objectives of the study clearer in the introduction (response to General Comment #1 of the initial review). The closest thing that I can find to such a statement is the phrase that "This study focuses on uncertainty growth in the North American / North Atlantic / European region, and particularly the North Atlantic winter stormtrack (sic), with its embedded cyclogenesis events and other synoptic systems." However, this sentence does not explain what will be achieved by this "focus". Please include a clear thesis statement to help readers to understand what the intended outcome of the study is.

   The Abstract and Introduction have also been re-written, and we think that this makes the objective clearer.

2. Although I appreciate the added discussion and authors' responses, I still think that invoking the "butterfly effect" is a misnomer. Based on the title, I would expect a paper about how small-scale and small-amplitude perturbations affect cyclogenesis. A title that is more descriptive – albeit less spectacular – would serve the content better. Maybe something like, "The impact of North Atlantic winter cyclones on uncertainty growth and forecast reliability in ensemble guidance".

   Following the theme of our previous paper titled "Flow-Dependent Reliability: A Path to More Skillful Ensemble Forecasts", the aim of the current study was to continue (and promote) work that leads to flow-dependent improvements in ensemble reliability. The question, therefore, was "what flow aspects are most strongly associated with deteriorations in ensemble reliability at present?" Preliminary work here identifies flow aspects which have a strong impact on ensemble spread, and then we evaluate the maintenance of reliability. Hence, we believe that

the part of the current paper's title "Uncertainty growth and forecast reliability during extratropical cyclogenesis" is well justified (see response to general comment 7 about the use of the word "cyclogenesis"). The question remains over the "Cyclogenesis Butterfly" part. This was added because we foresee (or would like to see) research developing where other flow situations associated with strong uncertainty growth, other butterflies, are identified and evaluated for reliability. However, have not been able to convince the reviewers on this point, and appreciate their concerns over this phrase, so we have removed it from the title and paper.

3.  I do not think that the strategy of "just in time" methodological description is effective or that it improves the readability of the text by motivating the reader. On the contrary, the decentralized methodology segments disrupt the flow of the text. Moreover, they are difficult to locate for readers that are not progressing linearly through the text and/or readers that wish to refer back to methodological descriptions at a later time. Please seriously consider introducing all relevant methods in section 2 of the manuscript.

    There are pros and cons to both options, but we followed the reviewer's advice and centralized the methodological descriptions in Sect. 2. For the methodologies, we discuss the reliability before the growth rate, so that the text is less disjointed as we move next into the growth rate results.

4.  The comparison of spread growth rates in select TIGGE models is interesting, particularly because of the wide range of patterns shown in Fig. 3. However, the follow-up on this analysis lacks sufficient rigor to make it as useful as possible for future readers. It would be very interesting to know the growth rates of some systems differ systematically from others, for example. Imagine adapting the anomaly correlation score using the LGR from one model at a time as the "analysis anomaly" over the North Atlantic. For example, the LGR from each TIGGE model (i.e. the "forecast anomaly") could be compared to the ECMWF patterns: an ACC would be computed for JMA, NCEP and UKMO. Then each model could be compared to the UKMO patterns for another set of scores: JMA and NCEP (the ECMWF score already being known). Et cetera. In the end, symmetric matrix of ACC scores would be obtained, and could be presented as an effective synthesis for this component of the analysis. The 95$^{th}$ percentiles (or smaller, given the small number of cases) of the ACC scores could be used as a measure of the variability around the mean ACC score. Noting what the ACC score is for Fig. 3 would provide a quantification of the extent to which the case study aligns with the "typical" degree of agreement between LGR in the TIGGE systems.

    This is an interesting idea. In view of the length of the current manuscript and the diversity of the employed methods and analyses, we suggest that a more systematic comparison of LRG be left for a subsequent study. In our view, several of the diagnostics used in this study are fairly novel and we regard it therefore as positive, if the results shown trigger additional and in parts more in-depth studies on certain aspects, as the one suggested here by the reviewer.

5.  Although much improved from the initial submission, the structure of the manuscript continues to present a challenge for readers. Aside from the need for a centralized methodology section (General Comment #3), a specific example arises at the end of section 3.3. The section was interesting and ends with two interesting questions. If they are anything like me, the reader will be looking forward to diving into these questions. However, the section 4 introduction, and methodology introductions sections 4.1 and 4.2 mean that they will have to "hold that thought" for ~100 lines of text before they get to further discussions on these questions. By then, the

reader will have forgotten the specifics of the questions or why they were interesting. If a review of reliability is required, it should appear either in the introduction or in section 2. Likewise, the complicated descriptions in sections 4.1 and 4.2 should appear in section 2. This reorganization will mean that the reader's momentum can be maintained as they progress through the results and synthesis.

We have re-ordered as requested. In response to this comment and specific comment 40, we now provide a more intuitive introduction to reliability up front, in Sect. 1.

6. How much of section 4.1 could be replaced by a reference to section 3 of Rodwell et al. (2015), but with "observation" (in that study) replaced by "analysis" (here)? The overlap is mentioned explicitly beginning on line 324, but a full replacement (and associated simplification of the current text) does not seem to have been considered: please consider it.

The approach here is a little different, with departures relative to ensemble-mean rather than a single unperturbed observation. This study also goes further to evaluate the assumption of constant (flow-independent) bias. Nevertheless, we have followed the reviewer's advice. In Sect. 2.3, we appeal to the previous paper much more strongly, and simply discuss the differences. In order to discuss the impact of variations in forecast bias, the derivation and discussion of the residual term are retained, but moved to Appendix A.

7. The term "cyclogenesis" appears to be used primarily to refer to the presence of a cyclone. This is important because the "cyclogenesis butterfly", based on a standard definition of cyclogenesis, implies uncertainty introduced by a cyclone is forming or deepening. However, the "cyclogenesis" clusters 1 and 2 (Fig. 7) only assess of the presence of a cyclone: they contain no direct information about whether the cyclone is intensifying or decaying (the westward tilt with height is not a guarantee of surface intensification). I understand that cyclones often deepen in this region; however, this makes the link to cyclogenesis anecdotal rather than data driven. The "winding back" process (a term that should be clearly defined) appears to be an attempt to build in a cyclogenesis period. However, if I understand the procedure correctly then a cyclone moving into the defined area will be defined as "cyclogenesis", even if it has already reached its peak intensity. Alberta clippers, for example, reach peak intensity shortly after formation and slowly weaken thereafter as they move towards the region of interest for this study (Blaine and Martin 2007). Changing from "cyclogenesis" perspective to one that documents ensemble behaviour in the presence of a cyclone would not weaken the work, and would better describe the analysis. The recommended title (General Comment #2) reflects this change in perspective.

The attached figures for the two clustering regions quantify the deepening of cyclones in this region. Despite being a somewhat diffuse average of events, for cluster region 1, the cluster-mean deepening for the "cyclogenesis cluster" attains 14 hPa over 2 days. For cluster region 2, it is 9 hPa. We believe that this justifies definitively our use of the word "cyclogenesis" in the sense that on average, there is strong intensification of the considered cyclones in the selected regions.

PMSL change D+0 to D+2

(a) All dates (size=180)

Unit: hPa

-14    -10    -6    -2    2    6    10    14

(b) Cyclogenesis region 1 (size=32)

Unit: hPa

-18    -10    -6    -2    2    6    10    14

(c) Counterpart region 1 (size=148)

Unit: hPa

-14    -10    -6    -2    2    6    10    14

[Figure]

PMSL change D+0 to D+2

(a) All dates (size=180)             (b) Cyclogenesis region 2 (size=62)

[Figure]

(c) Counterpart region 2 (size=118)

[Figure]

8. Excessive spread in the storm track during cyclone passage is labelled as a "key conclusion of this study" (line 438). This conclusion appears to be based on Fig. 8o, which shows positive but non-significant differences between the composite residuals. If that is correct, then assertions of cyclone-related "over-spread" should be moderated in the text. Given the potential for type-I errors related to multiple-testing (Wilks 2016; BAMS) and an experimental design that does not sample interannual variability, the true significance of these differences is questionable.

Our test is a strong one – 5% significance – and this is achieved in places in the Fig. 8o (new Fig. 7o due to re-organisation). We sample synoptic variability because that is what we are interested in. It is unclear how sampling interannual variability would help. A larger sample through adding years could improve significance, but then the model cycle will have changed. We maintain that new Fig. 7e, f, and o, together are sufficient to back-up our conclusion. Further justification, however, comes from the attached plot which shows the cyclogenesis/counterpart break-down when only the first region is considered. With better confinement of cases in a single region, we see stronger significance, even in the difference (panel o).

[Figure]

9. Figure captions are not the appropriate place for methodological descriptions. Although figure specific details might be provided in captions (specific threshold values for example), complete methodological descriptions should appear in the main body of the text where it can be easily found by future readers. Please move all methodological descriptions from captions to section 2 of the document.

We have moved methodologies to Sect. 2.

10. Section 3.2 should be replaced with a brief description of the Lagrangian growth rate in section 2, including a reference to Rodwell et al. (2018). The derivation and extensive discussion of terms that will not be employed further in the analysis does a disservice to the current study by introducing unnecessary complexity. If the rhs of Eq. 3 will be useful in a future study, then it should be presented in the future study. The discussion section of this work could easily refer to a hypothetical expansion of the Lagrangian growth rate rather than specific equations that disrupt the flow of the text.

Eq. 3 (including rhs) is central to the current study. We discuss the dynamical (rhs term 2) and diabatic (rhs term 1) aspects that could be associated with cyclogenesis in Sect. 3.1. It is also important to highlight what processes can be identified by the Lagrangian growth rate.

11. I understand that decisions related to writing style are typically left to the author; however, the over-use of em dashes disrupts the flow of the text and reduces its readability (there are seven in the introduction alone). Please consider rewriting the majority of phrases that currently use this form of subordination.

Our convention here is that the short hyphen is for use in hyphenated words like "co-ordinate", the longer en dash for numerical ranges like "2--3", and the longest em dash that splits text ---

particularly when a striking conclusion or important qualification follows. We have reduced usage of em dashes. There is only one em dash in the Introduction, for example.

12. Single and double quotes are used liberally throughout the text; however it is unclear what they mean and how the authors choose between them in any given circumstance. Please consider removing the majority of these quotation symbols and/or provide a description of what they represent.

There are differences in convention (UK versus US) for use of single and double quotes, and we did not fully adopt either. We have now removed most of the quotes, and follow the "single quote inside double quote" convention throughout.

**Specific Comments**

1. [L19] Consider rewording split infinitive.
   The re-wording avoids this issue.
2. [L23] It isn't "NWP" itself that develops techniques, but researchers and system developers.
   This is now avoided.
3. [L26] I believe that "leadtime" is usually written as "lead time".
4. This has been changed throughout.
5. [L28] I believe that "Stormtrack" is an application while, "storm track" is the usual term for the region discussed in this study.
   This has been changed throughout.
6. [L32] Is "propone" the word that you mean to use here?  Consider replacing with "prone" or "conducive".
   Done.
7. [L35] Why is "blocking" (well-accepted terminology) enclosed in single quotes?
   Quotes removed.
8. [L49] I think that a comma before the quoted question would be appropriate.
   This has disappeared in the re-write.
9. [L50] The term "reliability" has already been introduced with single quotes: consider removing them here for readability (the citations make it clear that this is a technical term).
   Done.
10. [L53] Why does the bias problem apply only to short-range assessments of reliability as implied here?
    For the forecast it doesn't, for the analysis it does – we have re-worded this in Sect. 2.3, L140-142, thanks.
11. [L58-59] This phrase suggests that improvements to the model and MU will not improve reliability in the presence of SV perturbations.  It that guaranteed to be true?  If the SV perturbations are scaled to become arbitrarily small, then they will presumably have a negligible impact on the forecast and model improvements will become dominant.  This general statement might need either to be qualified or to be removed.
    This now appears in the abstract and conclusions, and is re-worded.

12. [L59] What does the term "the potential is raised" mean? Does this refer to an increase in potential, or to a subject that is raised later in the text. Please consider using clearer terminology.

    This has been re-worded in line with the previous comment.

13. [L63-65] This appears to be a run-on sentence: please rephrase.

    Done as part of the re-write of the Introduction.

14. [L69] Why is "Ensemble" capitalized here?

    Because it relates to the abbreviation which follows.

15. [L78] This is a highly condensed system description that is difficult to follow for those not already familiar with the ECMWF suite. Could a reference to a system description be added, either in the form of a peer-reviewed publication or an operational technical note?

    We cited 5 papers on various aspects of the EDA. It is difficult to find publications specifying the configuration since this changes frequently. The best we can find is an ECMWF Newsletter article "A 50-member Ensemble of Data Assimilations" which is now also cited at L96-97.

16. [L90-92] Both SV and MU have already been defined. (I actually think that both acronyms should be replaced with complete terms throughout the text for readability.)

    We have made sure that both terms are only defined once, where they are first used. The acronyms are removed except in Sect. 5 when referring to the figures and experiment names.

17. [L101] What does the "current EDA cycle" mean? Does that refer to the one that was operational when this paper was written? Please be more specific.

    It refers to the EDA cycle at the specific time under consideration, so that a single EDA cycle can be run as an experiment. In operations, this is not possible, and the scaling is based on the previous cycle, 12 hours before. This is a very small point, which was originally relegated to the appendix but brought forward to the "Models, data and methods" Sect. 2 in response to reviewers' previous comments. Getting the right balance between comprehensiveness and readability is difficult in complex diagnostic studies. We now say at L128 "singular vector perturbation scaling is based on the current EDA cycle rather than the cycle 12 h before".

18. [L103-104] Does ERA5 use the same version and configuration of the EDA as described here? This is possibly important because a close connection might mean that systematic errors are common between the forecast and analysis.

    ERA5 is based on an older model version and configuration. It is only used here for the plotting of Fig. 1.

19. [Section 2.2] The extremely brief introduction of non-ECMWF systems in section 2.2 stands in stark contrast to the preceding full page of detailed description about the ECMWF ensemble. Please provide at least a brief introduction for each system (beyond Table 1) along with relevant references.

    We have pointed the reader to the documentation available within the TIGGE archive at L137.

20. [L111] For consistency with what?

    For consistency of comparison, we need to use the same forecast start times. We now say (L137) "Here, comparisons are based on the common run times of 00 and 12 UTC".

21. [L121] Why is PV only conserved, "following the *horizontal* flow on an isentrope"? To my understanding the orientation of the isentrope doesn't matter for PV conservation (note that any flow across an isentropic surface is better expressed as "diabatic" rather than "vertical").

    The orientation of the isentrope does not matter for conservation of IPV, but the advection within the material derivative is based on the horizontal flow. If we omitted the word "horizontal" then we risk "flow on an isentrope" being interpreted as "flow along an isentrope".

22. [L128] How is the "speed of cyclogenesis" defined? Do you mean "deepening rate" or "intensification rate"?

    We now state L56-57: "However, the rate of deepening and the growth of uncertainty were not considered in the choice".

23. [L130] "Eastern North America" is located east of the Great Lakes. Does this mean that the cyclone initially tracked westward? I think that showing the track in Fig. 1 would be more effective than this text description.

    Apologies for using wrong terms; we changed "Eastern North America" to "the Midwestern U.S." (the cyclone tracked eastward). The sentence then reads L42-43: "A little earlier than shown in Figure 1, on 26 November 2019, the cyclone had begun to develop over the Midwestern U.S. A day later, it had reached the Great Lakes …".

24. [L136] Parcels with ascent midpoints at 25oN are unlikely to be ascending above the warm front in the comma cloud region. If these are not following typical WCB storm-relative trajectories, what is driving their ascent? Is this an anafront? Perhaps this is unimportant, but the WCB points are described in some detail here, as is the distribution of precipitation.

    Thanks for looking at this level of detail into this case study. Yes, to us, this looks like an anafront with slantwise ascent at the (extended) cold front (something we've seen already in very early WCB case studies with trajectories, e.g., Fig. 12 in Wernli 1997, QJ, 123, 1677-1706). For this study, however, we decided that adding such mesoscale information might be distracting.

25. [L148-153] This is all standard Reynold's decomposition, is it not? If so, then that should be mentioned here. If not, then the differences should be explained and justified.

    Yes, this is Reynold's decomposition into ensemble mean and deviations about the mean. Reynold's decomposition can also refer to (e.g.) a spatial mean and deviations about it (Reynold's stresses), and hence this could cause more confusion that help. We believe the new description (L213-214) goes some way to making things clearer.

26. [L154] What is the advantage of the Eq. 2 form over that used by Baumgart and Riemer (2019)?

    Eq. 2 (now Eq. 3) is written as the exponential growth rate (normalised by the spread). This is important, for example, when comparing TIGGE ensembles (as in Fig. 3) with different initial uncertainty. New Eq. 3 also extracts the strong advection of uncertainty by the ensemble mean, which we believe is useful for discussing the initial material growth rate when the ensemble members have similar wind fields. Baumgart and Riemer effectively extract the strong flux convergence term. We now discuss a little further these aspects in Sect. 2.6.

27. [L174] What does the "intrinsic context" mean?

    We have reduced discussion of intrinsic aspects in response to Reviewer 2's comments. We now only discuss once: L65-67: "Whether the answer hints at an intrinsic property of the atmosphere, or is dependent on the formulation of the forecast system, is explored by

comparing models within "The International Grand Global Ensemble" (TIGGE, Swinbank et al., 2016) archive".

28. [L176] I do not think that "ground-truth" is usually hyphenated or single-quoted.

We no longer mention ground truth.

29. [L187] How is a 24-h running mean taken for background forecasts with a range of only 12h? The preceeding methodological description should be expanded and moved to section 2.

We now mention the concatenation aspect ahead of the filter discussion, so it is clearer that a 24 h running mean is possible. Following the reviewer's advice, this text is now moved to Sect. 2.6

30. [L188-193] A figure caption is not the appropriate place for methodological descriptions (the same applies for the WCB trajectory calculations described in the Fig. 1 caption). Please include this information in section 2. Lines 189-193 of the text contain the information that should appear in the Fig. 2 caption instead of the methodological description.

This has been done.

31. [L193-194] Please state explicitly how the location of large LGR is "consistent with Hoskins et al. (1985)", why this is important, and why further investigation would be useful (though not useful enough to be presented here).

In Hoskins et al. (1985) Fig. 21, during cyclogenesis the equatorward flow anomaly at upper levels acts to enhance PV advection, which strengthens and slows the eastward progression of the trough. Uncertainties in this feedback process are represented in the second term on the right-hand side of Eq. 3. This is now discussed better in Sect. 3.1 para 2.

32. [L197] Please provide a section reference rather than "above", particularly because the erosion of the trough has not been previously discussed.

The development of the LGR_P equation in the old Sect. 3.2 has been moved to the enlarged "Models, data and methods" section, to Sect. 2.6. The discussion of this equation in relation to cyclogenesis has been brought together into the Sect. 3.1 "Uncertainty growth in the EDA". This avoids the need for a backwards reference.

33. [L198] Are the animations are for different initializing times for this case or for different cases? Please be specific about what these animations contain and why they are relevant.

There are EDA and TIGGE animations for the DJF 2020/21 season. There were also animations for the two original cases, but one of these cases has been dropped now and so it seems sensible to only make the full season animations available.

34. [L199] What does it mean to "'shadow' the true synoptic evolution of the flow"? This term also appears on L226, although it remains unclear how the "true synoptic evolution" is defined, particularly given the similar amplitudes of analysis and short-range forecast uncertainty.

We now state at L238-239: The resulting timeseries of fields can be used to produce animations of P315 which "shadow" (remain within the background uncertainty of) the true synoptic evolution of the flow.

35. [L200] What are "large model growth rates"? Does this refer to large LGR values within model simulations? Please be specific about which synoptic features are associated with these growth rates, if they are important. If they are not, this sentence should be removed.

> Yes, this refers to LGR_P. We are now more specific about the features associated with these growth rates in Sect. 3.1.

36. [L201-207] These events have already been listed in the introduction. Because their connection here is purely speculative (it is explicitly noted that they are "not investigated here"), these sentences should be removed. Any discussion to be retained should be included in section 6.

> These events are no longer listed in the Introduction. They are left in Sect. 3.1, where the animations are discussed.

37. [L209-210] Rather than forcing the reader back to section 3.2 to identify the reasons, why not list them briefly here and provide a back-reference to section 3.2 for interested readers?

> The reference to intrinsic growth rates is no longer included here. The only discussion is in Sect. 3.1, L272-274.

38. [L227-228] Does "DJF 2020/21" follow WCD date formatting conventions?

> The first time the season is introduced (L163) we state "December–February 2020/21 season (DJF 2020/21)".

39. [L228] The phrase "the agreement can be better" is not specific enough for a scientific publication. Neither is the support of this statement with a new case study (not described in the text) sufficiently robust. Please refer to General Comment #4 for a recommended replacement.

> Please see our reply to General Comment #4.

40. [Section 4 introduction] This is a highly condensed description of reliability that is unlikely to describe the concept effectively to readers who are not already familiar with it. (I am reasonably familiar with it and have a very hard time following both this discussion and Fig. 5.) Consider moving this description to an appendix and focusing the in-text description of reliability on what it looks like to have a reliable system, or what problems are related to a lack of reliability. These concepts would be useful in the context of the current work and would help to motivate the subsequent analysis. This suggestion should be read in conjunction with General Comment #5.

> We have followed this advice. A more intuitive introduction to the concept of reliability is now given in the first paragraph of the Introduction.

41. [L243] The term "uni-modal" usually appears without a hyphen.

> This has been changed.

42. [L247-249] Has this notation not already been described in section 3.2? If so, it should not be repeated here because it appears to add complexity to this already complicated description of reliability.

> The main difference is that an overline in the reliability evaluation relates to a mean over forecasts, while an overline in the growth rate relates to a mean over ensemble members. We appreciate that this can be confusing. With some work within LaTeX, which we hope will be acceptable to the journal, we have managed to indicate the mean over forecasts with a thick overline.

43. The inclusion of both equations in the "Models, data and methods" section, with their description hopefully improves this.

> This has been done and, yes, it does.

44. [L255] Why is the operational status of the forecast important enough to be italicized here (or important at all for that matter)?

We have removed the word operational where it is not required, and it is no longer italicized.

45. [L263] The phrase "for the interested reader" suggests that there is an alternative to reading sections 4.1 and 4.2 for the uninterested reader:  is that true?  If it is, then that alternative should be explicitly stated here.

The use of a long models, data and methods section means that this is no longer in the revised manuscript.

46. [L271] The "as discussed above" phrase is not a useful introductory clause here:  terms 1-6 of Eq. 4 have not been explicitly "discussed above".  Please either remove it or include it in the parenthetical statement at the end of the sentence.

This has been done.

47. [L287] The {} symbols should be referred to as braces rather than parentheses.

This has been changed.

48. [L289] What "later" is being referred to here?  Please be specific about where further discussion of this term appears.

We now refer to the relevant Appendix section.

49. [L315] Please be specific about where this "later" refers to in the text.

This has been removed

50. [L325-328] This discussion seems to be relevant only to the observation-based analysis undertaken in the Rodwell et al (2016) study.  Please consider whether it is needed here, given that it seems to add little of direct relevance to the current work.

Following the referral to this equation (following General comment 6), a discussion of the differences is probably more important than it originally was, and hence we retain this text.

51. [L343] Please be specific about where this "later" refers to in the text.

We now refer to Sect. 4.3.

52. [L343] How much is "a little"?  Please provide quantification.

We leave "a little" here because it is now made clear that this is quantified in Sect. 4.3 (please see answer to previous point). This seems more informative than omitting "a little".

53. [L345] Please be specific about where this "later" refers to in the text.

We again now refer to Sect. 4.3

54. [L346] Consider "suggests potential" rather than "reflects" because the compensation is not shown here.

We now state L343-345 "As part of the current study, but not shown here, this reflects compensating deficiencies elsewhere, a recent deterioration in ensemble reliability in the storm track, and the importance of accounting for bias and analysis uncertainty".

55. [L347] What demonstrates the "recent deterioration in storm track reliability" claimed here?

Please refer to the reply to the previous comment.

56. [L347] It seems unlikely that the storm track itself has become unreliable.  Please rephrase to make it clear that EDA reliability has recently deteriorated in the storm track region, if that is shown to be true.

Please refer to the reply to specific comment 54.

57. [L358-360] How does one pick errors, spreads and reliability from different ensembles? My understanding is that Reliability is computed from the ensemble distribution, which involves both the 0th and 1st moments. As such, the Reliability is not an independent quantity that can simply be chosen from an arbitrary ensemble. From a more utilitarian perspective, how would picking the reliability of a given ensemble have an impact on guidance?

We were suggesting what might be fairly easily achievable. This is a small point and has been removed.

58. [L359] Suggest "day-2".

We have improved the consistency of usage, using "day-2" and "2 day" throughout.

59. [L362] What part of this analysis demonstrates that the JMA system has the slowest initial growth rates (the ensemble has the largest spread in the second column of Fig. 6)?

We now refer to new Fig. 3 and Fig. 4.

60. [L363-364] Which of the two questions posed at the end of section 3.4 is being answered here? The first one (over-spread during cyclogenesis) seems the most likely referent; however, the analysis in section 4.3 does not distinguish between cyclogenesis and no-cyclogenesis events. As a result, it cannot be asserted that the ECMWF ensemble is over-dispersive "in the vicinity of cyclogenesis". It appears to be over-dispersive in the storm track, but no more detailed statement than that would seem to be appropriate here.

We now state L363-365 "To answer the question in Sect. 3.2, whether the ECMWF growth rates are too strong in the vicinity of cyclogenesis, it needs to be determined whether the negative Residual in Fig. 5e is associated with a general level of over-spread or whether it can be linked to cyclogenesis events per se".

61. [L364-365] This is a statement rather than a question.

This is removed by the change made in relation to specific point 60.

62. [L376-377] Why is the K-means algorithm any better able to "cluster on structures" than other clustering approaches? For example, EOFs could have been used and the clustering done with their PCs. Such an approach would arguably be even more structure-aware than one adopted. There is no clear need to change the clustering strategy; however, the rationale for the methodological selection should be defensible.

We consider the K-means approach to offer a better chance of obtaining the required structures. EOFs do not necessarily represent physical structures, but can be used as a means of reducing dimensionality. See, e.g., Corti, S., Molteni, F. & Palmer, T. Signature of recent climate change in frequencies of natural atmospheric circulation regimes. Nature 398, 799–802 (1999). https://doi.org/10.1038/19745. For brevity, we have not included this discussion in the manuscript.

63. [L392-393] It is clear from the preceding paragraph that the LGR is not used as an input for the clustering algorithm. However, once the methodological description is moved to section 2 with the remainder of methodology information, this note will be relevant to remind readers of the independence of this field.

Yes, this note is retained here.

64. [L410] A scientific audience should not need to be old that 91:89 is "nearly 50:50".

With "nearly 50:50" we are implying that DJF results lie nearly halfway between the values in the two clustered rows, and emphasizing that we have a sufficient sample on both sides.

65. [L417] The spread maximum for the cyclone cases appears to occur in the middle of the North Atlantic storm track, or even at the eastern end of its highest track density, rather than over the "western part". See for example Fig. 7a of Hoskins and Hodges (2019; JCLIM).

We have removed the words "storm track". The point being made is that the spread maximum for the cyclogenesis cases is to the west, and the spread maximum for the counterpart is to the east. Close comparison with other studies would require a lot of understanding of the differences in methodologies.

66. [L426] Why is there a tilde before the figure reference?

The figure panel indicates that there is variance in forecast bias. It may provide a rough estimate of this variance, although we have only separated into two flow regimes. There could be more variance in forecast bias associated with other flow regimes. We have changed the tilde to "implied in".

67. [L433] If this region is described as the "western end of the North Atlantic winter storm track", then it would be useful to provide a graphical description of the storm track early in this study. Cyclone tracking studies [including the recent Hoskins and Hodges (2019)] find peak cyclone density near Newfoundland, placing the western end of the storm track along the eastern seaboard. If a different definition of the storm track is used in this study, it should be clearly described to make the associated discussions easier to follow.

We refer to our reply to Specific comment 65, and have changed the text here to say L421-423 "Here it is evident for the ECMWF ensemble that most of the over-spread during DJF 2020/21 in the western North Atlantic region of focus (Fig. 5e), is associated with the cyclogenesis composite".

68. [L436] There do not appear to be any significant differences in Residual (Fig. 8o) over Newfoundland. There is a small region of significant difference over eastern Quebec and the Gulf of St. Lawrence, but this is west of the coastal storm track. The small spatial scale and multiple testing make the significance of this region questionable (using a field significance test might help in this regard). This seems inconsistent with describing the red area in Fig. 8o as "particularly strong and significant".

We refer back to our reply, with attached figure, to General Comment 8.

69. [L437] What does it mean that the opposite-signed differences "might be associated with differences in downstream cyclogenesis"? Does this refer to different realizations of downstream cyclogenesis in different members, or to different forms of downstream cyclogenesis in reality, or something else entirely?

The cluster analysis was used to separate-off cyclogenesis events over the western North Atlantic. What is left, particularly associated with cluster 3 for region 1 (new Fig. 6c) and cluster 2 for region 2 (new Fig. 6e), contains increased cyclogenesis over the eastern North Atlantic. This might be expected from knowledge of the spatial and spatial-correlation scales of cyclogenesis. We now say L427-429 "Downstream, differences have the opposite sign — possibly because the occurrences of cyclogenesis events over the western North Atlantic are likely to be anticorrelated with the occurrences immediately downstream (as seen in cluster patterns Fig. 6c,d).

70. [L440] "Root-cause" is not usually hyphenated.

We now avoid root-cause by saying L431-432 "This issue could be associated with several different aspects of the forecast system. Through sensitivity experiments, Sect. 5 explores some of the potential causes".

71. [L448-451] This does not appear to be a complete sentence.

Sorry, this was very garbled. We have changed the text to L440-444 "Firstly, singular vector perturbations to the initial conditions of the ENS are turned off globally (OP-SV) and then model uncertainty in the ENS is turned off globally (OP-SV-MU). From this point, the parametrization of deep convection in the ENS is turned off in a local box (OP-SV-MU-DCP) or the ENS model horizontal grid resolution is increased to ∼ 4 km (OP-SV-MU+4km). Finally, the assimilation of observations in the EDA is turned off in a local box (OP-Obs) and the ENS is run again in the OP-SV-MU configuration".

72. [L457] Suggest "day-2".

This has been changed. Please see response to specific point 58.

73. [Fig. 10] What is the contour interval for MSLP?

10 hPa. This has been added to the new Fig. 9 caption.

74. [L462-463] Although the use of different colour bars allows different ranges of values to be shown, it is misleading in such a figure where the panels show the results of different sensitivity tests.  Please consider using the same colour bars for all panels.

To allow the reader to see the structure of each sensitivity, it is necessary to vary the shading interval. We did state in the text  L454-456 "Note that shading intervals vary over the panels shown in these two figures, so that the structures of all impacts can be seen". We now also make this clear in the panel caption.

75. [L482-484 and L493-494] These discussions of changes to precipitation seem somewhat tangential to the main themes of the manuscript and could be removed.

We consider that these remarks are useful. Indeed, it is quite interesting that spread can change in response to DCP without any overall change in total precipitation.

76. [L500 and L547] Reword "2 d".

This is changed to day-2

77. [L505] The phrase, "indicating that the conclusions drawn in this section are robust even with only two cases" does not seem logically correct.  The fact that a second case shown a similar pattern gives adds to confidence about the conclusions; however, the similarity of two cases does not provide some sort of successfully conclusive evidence as implied by this statement.

These are spread sensitivities based on 50-member ensembles, and do not relate to errors where only a single realisation of the truth is available. We now replace with "may be" and say L499-501 "Very similar results to those above were obtained for a second set of experiments initialised at 00 UTC on 17 January 2020 — indicating that the conclusions about ensemble spread sensitivity (based on 50 members) may be robust even with only two cases (the same could not be said for error sensitivity with just two cases).

78. [L515] Suggest replacing "these aspects might be developed" with "these techniques might be modified" for clarity.

Done.

79. [L554-555] Are any modern calibration techniques state *in*dependent as implied here for machine learning techniques?

    We believe there is a lot of calibration done without knowledge of the history of the forecast. However, to avoid having to go into a lot of detail, this minor comment has been removed.

80. [L560-562] It is unclear which results are being referred to here. Fig. 10 show that SV and MU have (by far) the leading impact on Z250 spread; DCP is a distant runner-up. However, this discussion seems to imply that DCP is dominant, with SV and MU also contributing. Although the results are more uniform between the three for 315K PV, this statement could easily lead future readers to think that deep convection has more of a relatively larger impact than it actually does in this case.

    We agree. In an attempt to highlight that the chaotic growth of EDA uncertainty is itself sensitive to deterministic model formulation, we added a phrase before the SV and MU impact discussion. This clearly led to the reviewer's interpretation. This phrase has now been removed at L551-554.

81. [L564-565] The wording of this sentence seems unnecessarily vague and complex.

    This has been re-worded in the re-write of the last paragraphs.

82. [L566] Suggest removing hyphen in "model-uncertainty".

    The hyphen is added as the text would otherwise include "model and model", which sounds a bit jarring.

83. [L571] This is a very abrupt ending to the manuscript. Consider adding a broader statement that is more directly related to the work undertaken in this investigation.

    We have tried to broaden the statement in the last paragraph.

**Reviewer 2**

Review of "The Cyclogenesis Butterfly: Uncertainty growth and forecast reliability during extratropical cyclogenesis" by Mark John Rodwell and Heini Wernli

Dear authors,

Let me first apologize for being late with my review.

Thank you for carefully considering my comments! The paper has substantially been rewritten and in my opinion it has improved a lot. However, there are some crucial issues left which I suggest to reconsider.

Main points:

1. Butterfly and predictability

I am still not happy with the "butterfly" discussion. You did add some explanation what you mean with the term "Cyclogenesis butterfly", but I still think this term might be confusing and also not really relevant for the paper and the issues it discusses:

We did not manage to convince either of the reviewers on this point, and appreciate their concerns. The broader aim with the term "cyclogenesis butterfly" was to promote the identification of other flow situations associated with large ensemble uncertainty growth, and to evaluate forecast reliability in these situations. We have dropped "cyclogenesis butterfly" from the title and paper. References to intrinsic predictability are also minimal now (only in Sect. 3.1). We suspect that many of the reviewer's comments below are addressed by this change.

Since you are not investigating intrinsic predictability, I would suggest to not start the abstract with a quote from Lorenz that refers to intrinsic predictability.

This has been dropped from the abstract.

Further in the abstract and also in the introduction and conclusion you refer to decreased predictability and high sensitivity associated with the cyclogenesis. But can this really be concluded based on your investigation? Yes, the Lagrangian growth rates in the cyclone in your examples are high (mostly for ECMWF) and divers, but you show later that they are too high.

The differences with other models were/are highlighted in the abstract. The ECMWF over-spread is based on the model with SVs applied (which we consider to be the main culprit in the over-spread), while the large growth rates exist even without the SVs. The discussion is more nuanced now and, with the dropping of the cyclogenesis butterfly, we hope this satisfies the reviewer.

Second, looking at Figs. 8a) and 8f) the errors in your target area seem to be smaller in the cyclogenesis composite compared to the counterpart. Not much is said about this in the paper, but wouldn't this indicate that cyclogenesis events are actually more predictable than the rest, at least on average in this period?

This is the error of the ensemble-mean, which is a different aspect. To clarify this, we have added the text L413-414 "Note that the stronger bias along the eastern coast of North America for the counterpart composite (cf. Fig. 7d,i) explains its larger ensemble-mean error (cf. Fig. 7a,f) in that area".

So is there really a physical based high sensitivity or "butterfly"-Lorenz63 phenomenon present? Or is this just a "malfunction" sensitivity of the rather unphysical inflation methods used (SPPT, SV)?

The sensitivity studies at the end demonstrate that at least 50% of the uncertainty growth is associated with chaotic growth from initial (non-SV and non-SPPT) uncertainty.

2. Reliability

In Fig. 8e) you show a large residual in the target area and argue that the models uncertainty representation may have a problem with cyclogenesis. And I think that's fair to say. But what about the even larger residual east of the target area in the counterpart? I think this should be discussed in somewhat more detail than in the current draft (only L436-437). Does the model may have even larger problems with other flow configurations? Is this related to ridge building, cyclone decay, secondary cyclogenesis? I think this should be at least roughly addressed, otherwise the statement "… flow-type clustering demonstrates that its over-spread in the stormtrack is indeed associated with cyclogenesis events" (L13-14, also L438) is not really justified in my opinion.

This downstream area is outside the compositing region. Increased downstream residuals in the counterpart composite can be due to cyclogenesis, which is likely to be more prevalent than in the composite with upstream cyclogenesis. We have modified the text to say L427-429 "Downstream, differences have the opposite sign — possibly because the occurrences of cyclogenesis events over the western North Atlantic are likely to be anticorrelated with the occurrences immediately downstream (as seen in cluster patterns Fig. 6c,d)". Hence the downstream residuals may be explainable in exactly the same way.

Specific comments:

L42, 210: Intrinsic predictability is not really a sensitivity only to small-scale perturbations. It is rather characterized by a loss of sensitivity to the scale of the perturbation if their amplitude is sufficiently small (e.g. Sun and Zhang, 2016).

This has been removed, but there seem to be differing points of view. Reviewer 1 mentioned in their first review that "these are very big butterflies".

L43 (Although the underlying processes…): I disagree with this statement. The recent study of Selz et al. 2022 (which you cite a few lines late) clearly showed how the error driving processes change when the amplitude of the initial condition uncertainty is reduced.

This is now acknowledged in Sect. 3.1

L45-46, L92: What kind of errors SPPT represent is not entirely clear, however, I think there is substantial evidence that it is not primarily missing interactions with unresolved scales: First, the recent success of SPP indicates that the parameters of the parameterizations (hence their assumptions and approximations) are associated with model uncertainty. Second, not all parameterizations account for unresolved motions (e.g. radiation, microphysics) but are also perturbed in SPPT and SPP. Third, a stochastic convection scheme that does account for unresolved motions only has virtually no impact on

error growth when the initial condition uncertainty is operational, see Selz et al. 2022. And don't you arrive at the same conclusion in L453-454?

Any model uncertainty representation is a pragmatic approach to improving ensemble reliability. One of the aspects which they purport to account for is missing interactions with unresolved processes. We have changed the text to L119-120 "A model uncertainty parametrization, which partly aims to represent scale interactions with (missing) sub-grid-scale variations".

L527-532: This paragraph confused me. First, yes, the initial growth rate in operational systems is much smaller than in intrinsic predictability experiments, but the latter are rather insensitive to the scale of the perturbations. Also small-amplitude large-scale perturbation lead to extreme growth rates on small scales. I think this was one of Durran's main points. Since the cyclogenesis growth rates are too high and associated with "unphysical" methods like SV and SPPT (see main point 1), I don't see how you can conclude that scale interactions and diabatic processes are important here. And what do you mean with "longer than expected intrinsic limit"?

This has all been removed. The "longer than expected intrinsic limit" aspect relates directly to the Palmer et al (2014) paper.

L533-534: No reason to speculate, this has now been done (Selz et al. 2022). The paper showed that on average error growth from operational uncertainties is mainly in the dry, balanced and larger-scale part of the flow.

A potential difference may be that here we consider growth rates in a very specific flow situation. In addition, Selz et al 2022 acknowledge that results can be sensitive to the model used in perfect model studies. Fig. 11d (new Fig. 10d) highlights this point from the perspective of diabatic processes.

L533-543: Again, is there really a problem with predictability in form of error growth or is there "only" a reliability problem, caused by the rather empirical model uncertainty representation methods?

The conclusions section has been re-worded. A conclusion of the study is that we would be better able to answer this point if SVs are removed from the operational forecasts. From an operational forecasting perspective, model uncertainty will likely remain a central aspect of the model, albeit empirical.

Minor comments, typos:

L131: 18UTC vs. 12UTC in the figure?

Thanks, should read 12 UTC.

---

## Author Response (AR3)

wcd-2022-6

**Uncertainty growth and forecast reliability during extratropical cyclogenesis**

by Mark J. Rodwell and Heini Wernli

**Replies to the reviewers' comments**

The authors would like to thank again both reviewers for the time and care that they put into reviewing the 2nd revised version of the manuscript, and for their comments, which helped to further improve the clarity of the text. Below we provide point-by-point responses to the individual comments; reviewers' comments are in black, and our replies in blue.

**Reviewer 1 (**Ron McTaggart-Cowan)

The authors have made significant changes to the submission that have improved the presentation of the work. I particularly appreciate the fact references to the "cyclogenesis butterfly" have been removed and that the number of sections has been reduced to six. The manuscript is certainly converging towards a publishable form.

Many thanks for this positive assessment of our efforts to improve the paper.

Recommendation: Minor Revisions

General Comments

1. Description of the plotting strategy within the main body of the text disrupts the flow and will distract future readers from the main points that the figures are meant to support. Leaving details of the plot description within the caption instead of the text is a stylistic decision; however, I believe that the paper would be more effective if this convention were followed throughout.

The caption is used to provide a complete and rigorous description of the figure. The main body text is useful for the reader to readily access the figure – for example "The black border indicates the union of the two clustering regions". Hence, there can sometimes be a small amount of duplication, but this seems necessary to us.

2. Starting on L204, the growth rate of spread is used as being analogous to forecast uncertainty. But one of the key findings of this paper is that the ensemble is conditionally over-dispersed. So then the events being studied are precisely those where the ensemble spread does not represent uncertainty well. It seems like a more careful use of the words "spread" and "uncertainty" is warranted throughout the text. This is particularly true in relation to the LGR, which I think is formulated to represent spread growth directly, and uncertainty growth only under the assumption that the two are interchangeable (which is shown not to be entirely true during cyclogenesis events).

This is an interesting point. We do view spread and uncertainty somewhat interchangeably (spread being one facet of uncertainty). This is because we believe that uncertainty can only be quantified in the context of the given observations and model (including model uncertainty representation); both of

which are likely to improve in the future. The text does make this clear. For example, in the previous version of the manuscript we state at L65 "The possibility that there is synoptic-scale coordination of uncertainty growth is interesting. Whether this hints at an intrinsic property of the atmosphere, or is dependent on the formulation of the forecast system, is explored by comparing models…" and at L70 "The variance of the ensemble, which generally grows with lead time, is then a measure of forecast uncertainty".

3. I still find the methods used in section 4 to be too complicated for the relatively simple outcome. If the goal is to demonstrate the conditional over-dispersion of the ECMWF ensemble, then please reconsider dramatically simplifying the analysis. If the goal is a demonstration that a complicated spread-error decomposition can be used, then please be sure to highlight the value added by the technique over a simpler analysis.

Although a simple comparison of spread and error also suggests over-spread, an analysis of the extended equation is useful. Partly this is to confirm that there are no strong (negative) covariance terms which invalidate the conclusion about over-spread. More importantly, the extended equation provides the better target for future system development. We mentioned the need to reduce day-2 ensemble standard deviation from the current 18 m to 13 m, rather than to the 15 m indicated by the simple spread-error relationship. We now increase the discussion in Section 6 by stating "We would argue that balance in the extended error-spread equation provides a superior reliability target for system development. For example, in the ECMWF ensemble, the standard deviation in day–2 Z250 over the east coast of North America during DJF 2020/21 was 18 m. The standard error-spread equation suggests a target of 15 m, while the extended error-spread equation suggests a target of 13.4 m. Including an estimate of the variance of forecast bias reduces this target further to 13.1 m. Moreover, at short forecast ranges, the extended equation can provide a better target for improvements in flow-dependent reliability".

4. The use of different shading ranges for different terms of the same equation (top two rows of Figs. 5 and 7) and equivalent plots for different sensitivity tests (Figs. 9 and 10) make these comparable panels very difficult to compare. I understand that not much will show up on the panels with small ranges, but that's useful information that the reader should be able to determine at a glance, not by looking simultaneously at the plotted structures and the colour bars simultaneously for two panels.

There is often a dilemma whether shading ranges should be such that they optimize the information content within a particular figure or across figures. We decided for the first option because we think that each figure per se also contains valuable structures that can be best seen with different shading ranges between figures. For example, the information in Fig. 9f shows the extent and magnitude of the influence of the local observations – this could be useful for future reference if (for example) SV perturbations were turned off, and model uncertainty scaled down. We have now added text in the caption to Fig. 5 stating "Note that the Bias and Residual, which can take positive and negative values, are shaded with a different interval to the other terms". The caption to Fig. 9 already stated "Note that the shading interval varies across the panels".

5. It appears that only the final paragraph of the conclusions (four lines) could really be classified as discussions or conclusions (the title of section 6). This means that ~90% of the section is actually dedicated to a thorough summary, including numerous figure and table references. As someone who read through the full manuscript, I find this redundancy a missed opportunity for opening up a broader

discussion of the implications of the work. Please consider either renaming section 5 to "summary and conclusion" or (better) redrafting section 5 to present a very short summary before taking a larger perspective on discussing the work.

As noted above in response to general comment 3, we now also discuss in this section the use of the extended error-spread equation as a target for system development. Nevertheless, we have re-named this section "Summary and Conclusions".

Specific Comments

1. [L40-58] I suggest redrafting this paragraph (and removing Fig. 1, with any needed panel combined with Fig. 2) for two reasons. One is that it is hard to imagine any reader of this work who wouldn't already been very familiar with cyclogenesis in the storm track. The other is that this study isn't really about cyclogenesis itself, but rather about the growth of uncertainties related to cyclogenesis. I understand that the reader needs to know what cyclogenesis is to appreciate how errors may grow, but I think that a well-crafted literature review would be more effective at relaying this than the current case study. Describing baroclinic instability and diabatic contributions to cyclone deepening could be done briefly with relevant citations. This would be followed by an introduction to error growth on the waveguide, for example citing the recent work of Baumgart et al. (2019).

This paper addresses at least two communities: scientists interested in ensemble forecasting in general (and we claim that not all of them are fully into the synoptic-scale dynamics of cyclogenesis) and colleagues familiar with cyclones and storm tracks (who need more background, e.g., about forecast reliability). We therefore think that it would be a pity to remove Fig. 1. Also, this figure is setting the scene for the later discussion of the same cyclogenesis event.

2. [L61] I think that the word "coordinate" implies too much intention here, and feels like an anthropomorphization as a result. Or maybe it's the word "act to": cyclogenesis doesn't really "act", it just happens. A similar construction appears in the subsequent sentence.

We like the word "coordinate", which links to the cited hypothesis of Palmer et al (2014). It is quite central for our results that there is a natural theoretically-based scale for baroclinic growth. However, we agree that "act" might be not the ideal word, we changed the sentence to "… to investigate whether cyclogenesis events can coordinate strong growth of forecast uncertainty …".

3. [L126-127] Many readers will probably know what it means to "warm start" VarBC and SPPT: please provide a brief explanation because this presumably impacts early spread growth in the forecast.

The sentence was providing technical information in an overly-complex way. The details actually ensure that the experiments are started as cleanly and smoothly as possible. We have removed this over-complexity by stating simply "Prior information for the experiments comes from the operational EDA".

4. [L137] Does the WCD style guide cover web references? If so, it will hopefully cover citation format and include information about access date.

This will be sorted out during the typesetting.

5. [L177] This introductory sentence is written as if clustering is the only way (or even the most obvious way) to accomplish the objective of identifying cyclones. Given that other methods for cyclone

identification have been used in the literature, please consider rewording this sentence to provide a stronger introduction to the need for a cluster analysis in this case.

We changed "clustering" to "focusing". Further justification for the chosen method is given on lines 182-184.

6. [L210] The use of the thin overline here (ensemble mean) is distinct from the thick overline in Eq. 1 (time mean). I'm not sure that the typesetting is going to be clear enough to allow readers to distinguish between these two. Please consider using a different operator (for example <>) for the ensemble mean.

We have thought a lot about this, and experimented with different approaches. If one looks through the literature, an overbar is generally used to denote a mean, regardless of what it is a mean over. Conversely, "<..>" is often used to denote an inner product. Hence, we would prefer to use overbars throughout. To help differentiate a little, we now used thick and thin overbars. However, thick overbars are defined in (and relate to) Section 2.3 while thin overbars are defined in (and relate to) Section 2.6, so we believe there is little risk of confusion.

7. [L248-252] This discussion makes it sound like Z250 is used throughout the remainder of the text, but the subsequent section moves back to P315. Please clarify here which sections are forced to use Z250 because of TIGGE database limitations.

Thanks, we have clarified this. We now state "Since the required fields are not available in TIGGE, when comparing with other models in Section 3.2, the pragmatic decision is made…"

8. [L259] "Cyclogenesis deepening" seems redundant given the adopted definition of cyclogenesis (i.e. cyclone deepening).

Agreed. We have changed "cyclogenesis deepening period" to simply "cyclogenesis".

9. [L265-284] I don't understand the "it is tempting to speculate" concept here, especially when the paragraph goes on to say that these hypotheses could be confirmed (or refuted) by investigating the terms on the r.h.s. of the LGR equation. If it is tempting and confirmable, then why isn't it done? Then all of this theorizing could be replaced by a simple plot that shows what process is occurring. Between that simplification and the reduction in plot strategy description (General Comment #1), a solid analysis could be included without increase in manuscript length. This would also provide justification for the existence of the r.h.s. of Eq. 3, which is not otherwise used in the manuscript as noted by both reviewers in previous rounds of review.

It is not easy to verify these statements. They require work beyond quantifying the terms on the r.h.s. of the LGR equation. If we could have done it easily, we would have. The reviewer has already mentioned that this manuscript contains a lot of research, and we therefore prefer to keep our cautious formulation.

10. [L292-294] There is an odd asymmetry in this discussion. The "forecast bust" reference (behaviour of a forecasting system that is entirely a property of model space) seems out of place with this discussion of physical features and phenomena. Maybe pulling out the "forecast bust" phrase and making a separate statement would help, because then it is clear that the bust can have its origins in any of the listed features (or others).

We do not claim that the busts are entirely (or even partially) a property of the model space. Rather, we are relating the busts to the single, deterministic, sampling of the forecast. To make it clearer what is meant, we have reworded the sentence to: "All these situations can lead over Europe to extreme precipitation (Grams and Blumer, 2015), blocking events (Rodwell et al., 2013) and, in deterministic forecasts (which can be thought of a single sampling of an ensemble distribution), "busts" or "dropouts" (Lillo and Parsons, 2017)".

11. [L310-315] It is unfortunate that there is no quantification of this difference. It seems like there is enough information contained in the TIGGE database for a systematic assessment of LGR for Z250 during cyclogenesis cases from different models.

In the meantime, the TIGGE animations for the entire winter season have been uploaded to an open access repository: https://www.research-collection.ethz.ch/handle/20.500.11850/605102. We trust that the reviewer and future readers of the paper will appreciate the high variability of growth rates shown for the different events and between models.

12. [Sections 4.3 and 5] Does "day—2" refer to "day minus 2"? If so, please replace the em-dash with a space and a minus sign, to become "day -2" for readability and to distinguish it from a compound adjective (e.g. "the day-2 spread").

Apologies for this incorrect use of the Latex en-dashes. In the example, it refers to "day 2" rather than "day minus 2" (indeed it never refers to "day minus 2"). Throughout, we now retain "day—2" only when it is a compound adjective; otherwise we write, for example, "at day 2".

13. [L510-511] Is there really a need for further investigation to figure out how to reduce spread generated by SVs or SPPT? The options seem pretty obvious. Perhaps this would be better worded as how to reduce the associated spread growth during cyclogenesis without negatively affecting the overall well-balanced spread-error relationship.

The operational implementation of developments requires, in general, that they improve (or at least don't degrade) proper scores of the ensemble forecast. A well-balanced spread-error (or extended spread-error) relationship indicates reliability, which is one aspect assessed by a proper score, but refinement (or resolution) is another aspect. Hence, we rephrase this now slightly differently as "…it makes sense to investigate how these techniques might be modified to reduce the growth of spread during cyclogenesis without negatively impacting the overall performance of the ensemble."

14. [L564] I'm not really sure what this sentence means (even what "other" refers to or is distinct from) and how it follows logically from the current work.

The "other" flow types are distinct from the cyclogenesis situations discussed here. We have changed the last sentence to: "In addition to cyclogenesis situations, initial growth rates tend to support the idea that uncertainty can be concentrated in other flow situations, including those prone to mesoscale convection over North America (Palmer et al., 2014) and during the extratropical transition of tropical cyclones. Similar investigations of these initial growth rates, and how reliably the forecast predicts their evolution, could also lead to better flow-dependent reliability and improved overall forecast performance".

15. [Data availability statement] The "data and code available on request" doesn't really live up to FAIR principles. Please consider at least uploading as much of the code used to generate the results shown here as possible to a public repository.

We also stated that the TIGGE and ERA data are freely available and have made the animations available under a doi. The details of the code are carefully laid-out in the manuscript. It is not practicable to provide the code without providing help and guidance, since there are numerous settings to configure and utilities (such as Metview and spectral transforms) required.

**Reviewer 2**

Thank you for again considering my previous comments and carefully rewriting the paper. I very much like how you restructured the manuscript. The objectives of the study come across much more clearly and it is much easier and less confusing to follow the paper and also more fun to read. In my opinion the paper can be published as it is, I see however still room for some improvements. I leave it to you and the editor to decide whether or not you want to incorporate these points.

Many thanks for this positive assessment of our efforts to improve the paper.

I give my main concerns below, starting with the most important one and then a few minor points in ascending order.

L427ff: Here I am coming back to a concern I already raised in the previous round which I think is still not really addressed convincingly: If I by eye average the residual over the plotted area in Fig. 7 of the cyclogenesis cluster and the counterpart, it seems that the counterpart is equal to even more over-spread, only the location of the over-spread is shifted downstream. It is fair to assume that, since cyclogenesis in the box is excluded in the counterpart cluster, it is more frequent in the downstream region. But since it is the main point of the paper to link cyclogenesis and over-spread I am still missing a quantitative analysis of this feature, e.g. by investigating surface pressure tendencies or even another cluster analysis based on the downstream region where the counterpart residual blob occurs.

We agree that we don't discuss the structure of the residual outside of the clustering regions in detail. One reason is that, as also mentioned in the earlier reviews of the paper, the study is rather complex by combining different methods and sets of simulations. We therefore focus on the western North Atlantic, which is known from climatologies to be the hotspot of cyclone intensification and warm conveyor belt activity in the North Atlantic-European sector. A second reason is that towards the end of the storm track, processes get even more variable than in the entrance region. In the entrance region (which we focus on in our paper), days can meaningfully clustered into "cyclogenesis" and "non-cyclogenesis", whereas in the eastern North Atlantic, there can be propagation of cyclones from upstream at different latitudes, or the formation of secondary cyclones along fronts, or the formation of cutoff lows, etc. This makes it more difficult to explain the main reasons leading to the residuum field in a clear and concise way. However, we manually analysed maps of the synoptic flow in this region downstream of the clustering regions, which revealed that indeed, at many time steps that belong to the non-cyclogenesis counterpart cluster, there is cyclogenesis in the region downstream of the clustering regions, but as mentioned above with a lot of variability between cases (with cyclones either close to Iceland, over the

UK or associated with upper-level cutoffs, further south near the Azores. Together, they most likely explain the over-spread mentioned by the reviewer.

We have added the text towards the end of Section 4.2 L392-394:

"Further east, towards the end of the storm track, processes get more variable. There can be propagation of cyclones from upstream at different latitudes, the formation of secondary cyclones along fronts, and the formation of cutoff lows, for example. This makes it more difficult to meaningfully cluster this region into cyclogenesis and non-cyclogenesis cases."

To be more clear about the comparisons in Fig. 7, we also change, in Section 4.3, the text:

"Downstream, differences have the opposite sign — possibly because the occurrences of cyclogenesis events over the western North Atlantic are likely to be anticorrelated with the occurrences immediately downstream (as seen in cluster patterns Fig. 6c,d)"

to:

"Downstream, both composites individually display negative residuals. This is consistent with the above discussion since there has been no direct control for downstream cyclogenesis. Indirectly, it is likely that occurrences of cyclogenesis events over the western North Atlantic are anticorrelated with the occurrences immediately downstream (as seen in cluster patterns Fig. 6c,e) and this might explain the negative cluster differences in absolute residual for the downstream region (Fig. 7o)".

L230ff: You did not state the motivation for using the EDA system instead of the forecast system to derive the Lagrangian growth rates at this point. I infer from later discussions that you wanted to exclude the singular vector perturbations? On the other hand, you have a forecast without singular vectors of this case in your sensitivity dataset. Also, for the TIGGE-analysis you (have to) include the singular vectors. I am still a bit confused here.

It is true that the case shown in the sensitivity experiments included the experiment without singular vectors. However, the growth-rates from the EDA are provided in the animation for the entire DJF 2020/21 season. It is important to show that the concentration of initial growth rates into particular flow situations is not purely due to the SVs (which we are suggesting could be reduced in amplitude). The most appropriate place to explain this seems to be in Sect. 6 where the effects of SVs are discussed. Following the text:

"Results highlight a few flow situations over the North American / North Atlantic / European region where ensemble variance growth at synoptic scales is particularly strong and concentrated"

to:

"The supplementary animation of LGRp for the DJF 2020/21 season highlights a few flow situations over the North American / North Atlantic / European region where ensemble variance growth at synoptic scales is particularly strong and concentrated. Note that this concentration of growth-rates is not dependent on singular vectors perturbations, since these perturbations are not included in the EDA."

Appendix C: I could not find any reference to the variable and vertical level for which you show the spectrum. Is this for kinetic energy (i.e. variance of the wind)? Also, I don't understand the meaning of the dotted lines.

Apologies, we have now added T500 in the main text and figure caption.

L269/L283, second vs. first source term: The text reads as if you are speculating about this. But wouldn't it be very easy to check if this is true by comparing the contributions from those terms?

It is not so easy to verify these statements. They require quite a lot of further work, and inevitably raise further questions. The paper is dense already, and we hope the reviewer will agree to this being left for a subsequent study.

L272ff: I find these references to intrinsic predictability misplaced here (maybe that's partly my fault) and they could maybe also removed. However, I would like to point out that first the term you are referring to can reflect scale-interactions (because it is non-linear), but it could also be dominated by interactions within similar (synoptic) scales. Which one it is is speculation at this point. Second, please note that in Selz et al 2022 also large total variances (as produced by the EDA) were considered. With similar methods also Baumgart et al. 2018 ("Potential Vorticity Dynamics of Forecast Errors: A Quantitative Case Study") considered (operational) large forecast uncertainties. Both studies showed that variance ("error") growth happens predominantly in the rotational component of the flow (e.g. figure 6 in Baumgart et at.), which in my opinion disfavours large contributions from scale interactions, since from the mesoscale downward the kinetic energy is roughly equally partitioned between the rotational and divergent component.

We are largely in agreement with the reviewer here; the only difference might be in terminology. As the reviewer says, it is up for speculation at present, but much of the growth could be "dominated by interactions within similar (synoptic) scales". We were very much including this within our reference to "scale interactions" within the continuum of scales between 2000 and 100 km. We agree that interactions can also be at the same scale. Maybe it is also possible that synoptic rotational anomalies can be driven by mesoscale interactions with divergent winds (as illustrated by the "Rossby Wave Source")? We have changed the text:

"Scale interactions embodied within this source term can produce initial uncertainty growth at synoptic scales because the EDA contains considerable variance power at spatial scales between about 100 and 2000 km (illustrative power spectra are presented in Fig. C1 in Appendix C). This may not be the case in predictability studies, where initial uncertainty is restricted to grid-point noise (Judt, 2018) or has small total variance (Selz et al., 2022). See also Durran and Gingrich (2014)."

to:

"Interactions within and between scales, represented in this non-linear source term, can produce initial uncertainty growth at synoptic scales because the EDA contains considerable variance power at spatial scales between about 100 and 2000 km (illustrative power spectra of T 500 are presented in Fig. C1 in Appendix C). This may not be the case in predictability studies, where initial uncertainty is restricted to grid-point noise (Judt, 2018). See also Durran and Gingrich (2014); Selz et al. (2022) for relevant discussions.

Minor points:

L146: "forecast time". I was confused at first. Maybe replace with "forecast init time" or "case" to distinguish it from "forecast lead time".

We have changed to "forecast initial time".

L165: Should that be "...its correct form. For example, since R can be..."?

This was poorly worded. We have changed "This approach leaves the bias in its correct form, for example" to "Note that the square-root of Bias^2 is Bias (with its correct sign)".

L250-251: I am confused. Are you using centred differences in time to calculate d/dt? So like f(12h)-f(0)/12h? But this then would be centred at 06 or 18 valid time (figure 3 however says 12Z)? Could you clarify?

For the TIGGE data, centred differences are valid at times 03, 09, 15, 21 UTC. The 24h running-mean then has the effect of putting the fields at 00, 06, 12, 18 UTC. For example, the 12 UTC field shown in Fig. 3 is the mean of centred differences at times 03, 09, 15, 21 UTC. Note that the animations of the TIGGE data include a frame at each hour - these being produced using linear interpolation between the 00, 06, 12, 18 UTC fields. We had omitted this detail. We now state:

Earlier, after "The filter also includes a 24 h running-mean", "(the nominal validity time is at the centre of the running-mean window - placing the final fields back on the full hours)"

and, at the end of Sect. 2.6, we add "The resulting timeseries of fields can again be used to produce animations (hourly frames being derived using linear interpolation between the 6 hourly fields)".

L264: Should that be 0.2 h^-1 (like in the colour bar of the figure)?

As per the caption for Fig. 2, the orange contours "extend the shading scheme, with the same interval. In the largest red blob, there are two orange contours which have values of 0.14 and 0.18 $h^{-1}$. This is why we state "in excess of 0.18 $h^{-1}$". Note that the value at the end of the colour bar (0.26 $h^{-1}$) indicates that there is just a third contour at 0.22 $h^{-1}$ but this must be so small that it is not plotted/visible.

L337: I suggest to first emphasise the discrepancy between the 18^2 and the 13.4^2 (the overspread of the ECMWF system, the main point of the paper) before going into the need to account for AnUnc and Bias.

The 13.4 m is the value which does take into account the analysis uncertainty and bias; it is 15 m when they are not accounted for. Nevertheless, we appreciate the spirit of the comment. We take this discussion to Sect. 6 because this is the natural place to put it and also because, by then, an estimate of the variance in forecast bias has been obtained (in Sect. 4.3), which lowers the ensemble variance target to 13.1 m. We now state in Sect. 6 "We would argue that balance in the extended error-spread equation provides a superior reliability target for system development. For example, in the ECMWF ensemble, the standard deviation in day–2 Z250 over the east coast of North America during DJF 2020/21 was 18 m. The standard error-spread equation suggests a target of 15 m, while the extended error-spread equation suggests a target of 13.4 m. Including an estimate of the variance of forecast bias reduces this target further to 13.1 m. Moreover, at short forecast ranges, the extended equation can provide a better target for improvements in flow-dependent reliability".

L484: In my opinion this "lack of agreement" confirms the point I made in the last review that the model uncertainty representation (SPPT) is not accounting for the impact of missing sub-gridscale variability,

which is (better) resolved at 4 km, but mainly for "flow-dependent biases" in the parameterisation schemes on larger scales. Would you agree?

We say, "model uncertainty representation is thought to partly account for the impact of sub-grid-scale uncertainty". We then highlight the discrepancy with the 4 km results. Yes, the reviewer's hypothesis may well be correct. Probably few people think that the model uncertainty representation only accounts for missing sub-grid-scale uncertainty. In the end it is quite pragmatic - exemplified by the need at present at ECMWF to apply it at scales much larger than would be justified by the "missing sub-grid-scale uncertainty" argument. The modified text at the end of Sect. 5 now states "Since the motivation for the use of singular vector perturbations (Magnusson et al., 2009) and the initial reason for the development of model uncertainty representations (Buizza et al., 1999) was to increase ensemble spread, it makes sense to investigate how these techniques might be modified to reduce the growth of spread during cyclogenesis without negatively impacting the overall performance of the ensemble".

---

## Author Response (AR4)

wcd-2022-6

**Uncertainty growth and forecast reliability during extratropical cyclogenesis**

by Mark J. Rodwell and Heini Wernli

**Replies to the reviewers and the handling editor**

The authors would like to thank once again Ron McTaggart-Cowan and our other anonymous reviewer for the considerable time and care that they have both put into reviewing the three revisions of this manuscript. We also thank Michael Riemer for the appreciable attention and wisdom that he has put into handling and editing the manuscript.